# Identification of Late Pleistocene and Holocene fossil lizards from Hall's Cave (Kerr County, Texas) and a primer on morphological variation in North American lizard skulls

David T. Ledesma[1]*, Simon G. Scarpetta[2¤a], John J. Jacisin, III[1], Antonio Meza[1¤b], Melissa E. Kemp[1]

1 Department of Integrative Biology, The University of Texas, Austin, Texas, United States of America,
2 Department of Integrative Biology, Museum of Vertebrate Zoology, University of California Berkeley, Berkeley, California, United States of America

¤a Current address: Department of Environmental Science, University of San Francisco, San Francisco, California, United States of America
¤b Current address: School of Life Sciences, Arizona State University, Tempe, Arizona, United States of America
* ledesma-david@utexas.edu

## Abstract

Fossil identification practices have a profound effect on our interpretation of the past because these identifications form the basis for downstream analyses. Therefore, well-supported fossil identifications are necessary for examining the impact of past environmental changes on populations and communities. Here we apply an apomorphic identification framework in a case study identifying fossil lizard remains from Hall's Cave, a late Quaternary fossil site located in Central Texas, USA. We present images and descriptions of a broad comparative sample of North American lizard cranial elements and compile new and previously reported apomorphic characters for identifying fossil lizards. Our fossil identifications from Hall's Cave resulted in a minimum of 11 lizard taxa, including five lizard taxa previously unknown from the site. Most of the identified fossil lizard taxa inhabit the area around Hall's Cave today, but we reinforce the presence of an extirpated species complex of horned lizard. A main goal of this work is to establish a procedure for making well-supported fossil lizard identifications across North America. The data from this study will assist researchers endeavoring to identify fossil lizards, increasing the potential for novel discoveries related to North American lizards and facilitating more holistic views of ancient faunal assemblages.

## Introduction

Fossils are valuable for providing a glimpse into past life on earth, and fossil data have played critical roles in understanding range shifts [1], extinctions [2], and diversification during critical junctures in earth history [3]. Paleontological data also lend a long-term perspective on the responses of taxa and ancient communities to environmental changes and paleontological

**Data Availability Statement:** All relevant data are within the manuscript and its Supporting Information files.

**Funding:** MEK and DTL were supported in conducting this research by funding from The University of Texas at Austin Office of the Vice President for Research as part of the Planet Texas 2050 Bridging Barriers initiative (https://bridgingbarriers.utexas.edu/planet-texas-2050). Supporting funding for this research was also awarded to DTL by the University of Texas Stengl-Wyer graduate fellowship (https://cns.utexas.edu/info-graduate-students-postdocs/find-funding/stengl-wyer-graduate-fellowships). The funders had no role in study design, data collection and analysis, decision to publish, or preparation of the manuscript.

**Competing interests:** The authors have declared that no competing interests exist.

perspectives have become increasingly important in conservation [4]. One of the first and most critical steps for incorporating fossils into ecological and evolutionary analyses is identifying fossil remains. It is important to recognize that fossil identification practices have a profound effect on our interpretation of the past because these identifications form the basis for downstream analyses. Well-supported fossil identifications are paramount for examining past populations and communities and have been used to infer the impact of past environmental changes on North American Quaternary herpetofauna (non-avian reptiles and amphibians) [5].

North American fossil herpetofaunas from the Quaternary period (2.56 Ma–present) were the subjects of numerous paleontological investigations during the 20th century (see [5] and references therein). The cumulative findings of these investigations led to the formation of a hypothesis predicting relative taxonomic and biogeographic stability in North American herpetofauna throughout the Quaternary [6], specifically during the last 1.8 million years [5]. This hypothesis was derived from paleoherpetologists who published on Quaternary fossils and reported few extinct taxa, noted few to no speciation events, and reported that identified species from fossil sites were generally congruent with species found in the area today [7, 8]. It was later argued, however, that this stability hypothesis was predicated on biased fossil identification practices that resulted in a circular argument for stability [9]. Those biases include the historical practice of identifying Quaternary fossil herpetofauna using comparative extant specimens sourced from the immediate vicinity of the fossil deposit or within nearby circumscribed geopolitical or geographical regions. Using these geographically limited comparative datasets, fossils were commonly identified to the species level, generally based on phenetic similarity to nearby species [9]. Those practices effectively predetermined the taxonomic and geographic stability of Quaternary herpetofauna fossil assemblages [9]. To address the circularity of the stability hypothesis, Bell et al. [9] suggested the use of apomorphies–evolutionary derived features–as an alternative method to the historical fossil identification practices. Importantly, apomorphic identifications, when paired with an expansive comparative dataset, are less sensitive to geographic biases and provide a more objective and replicable system for identifying fossils.

Here we employ an apomorphy based identification framework to identify fossil lizard remains from a late Quaternary fossil site in Central Texas. Our study site, Hall's Cave, is an exceptional study system because there is an abundance of well-preserved fossil material, including a sizable accumulation of fossil herpetofauna that would benefit from rigorous examination [10]. Hall's Cave contains a remarkably continuous late Quaternary stratigraphic sequence in Texas, spanning the last 20,000 years [10, 11], and is located in Kerr County near the southern edge of the Edwards Plateau, an uplifted region occupying much of central Texas. Hall's Cave was first excavated beginning in the late 1960s (then under the name Klein Cave; [12]) and excavations continued in the late 80's-early 90's [10] as well as in recent years [13, 14]. Identification of fossil remains from these excavations largely focused on mammalian taxa [10, 15, 16] and plant microfossils [17], but some herpetofauna were identified, including lizards, snakes, frogs, salamanders, and turtles [10, 18]. Previous research on Hall's Cave also included bulk bone ancient DNA metabarcoding analyses, yet lizard ancient DNA was not recovered and amplified [13]. Previous investigations of the fossil lizards from Hall's Cave reported at least six different lizard taxa [10, 18]. Most of those identifications lacked a corresponding discussion of apomorphic morphological features supporting their taxonomic assignment. Here we reassess previously examined as well as new fossil lizard material from Hall's Cave using an apomorphic identification framework.

In addition to identifying fossil lizards from Hall's Cave, we also sought to establish a procedure for making well-supported fossil lizard identifications, particularly for North American lizards. We compiled previously reported apomorphic characters as well as new potential

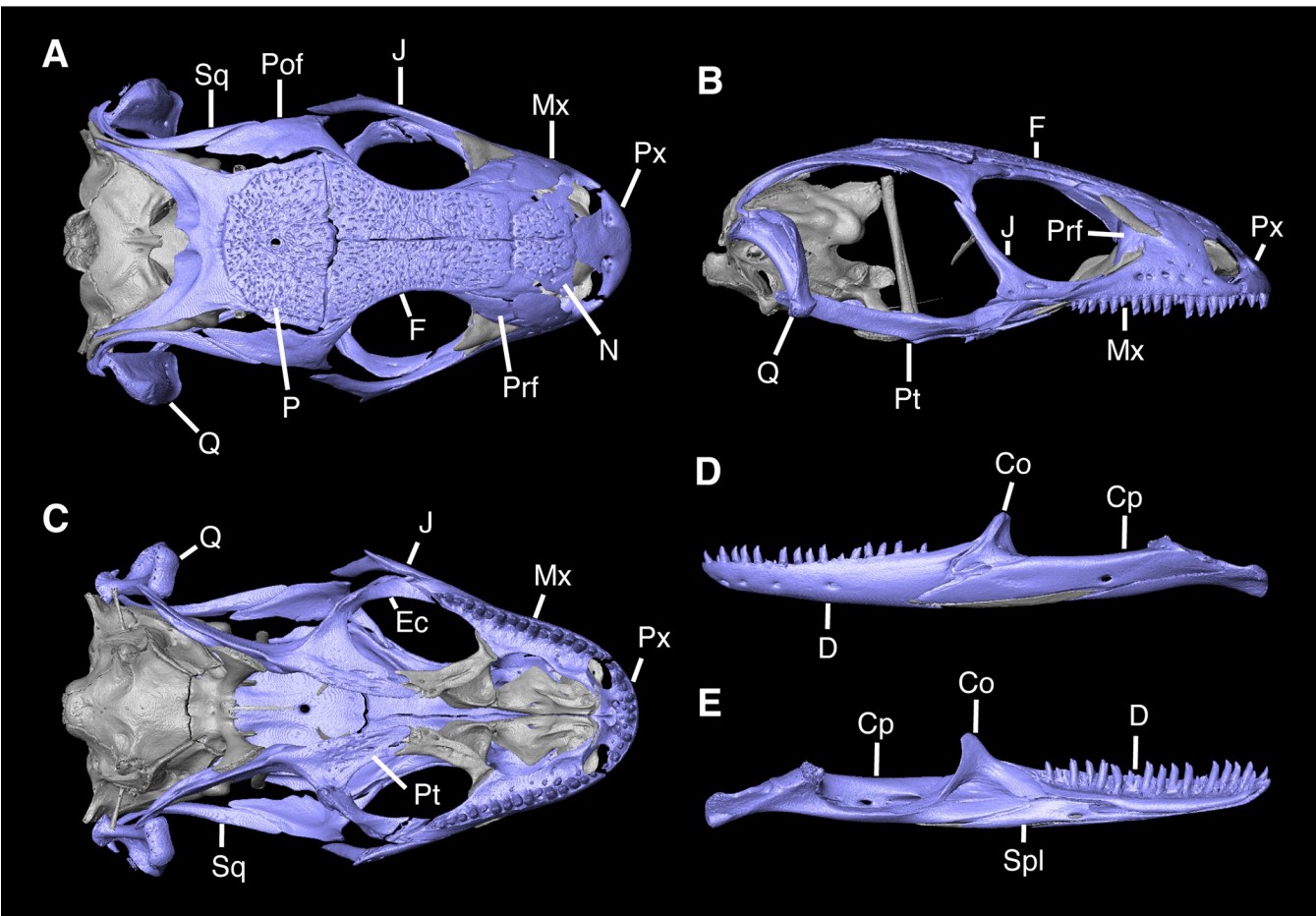

**Fig 1. Skull of *Diploglossus bilobatus* TNHC 31933 with the elements recovered as fossils and examined in the text highlighted in purple. A**. Dorsal view; **B**. Lateral view; **C**. Ventral view; **D**. Mandible lateral view; **E**. Mandible medial view. **Abbreviations**: Co, Coronoid; Cp, Compound bone; D, Dentary; Ec, Ectopterygoid; F, Frontal; J, Jugal; Mx, Maxilla; N, Nasal; P, Parietal; Pof, Postorbitofrontal; Prf, Prefrontal; Pt, Pterygoid; Px, Premaxilla; Spl, Splenial; Sq, Squamosal; Q, Quadrate.

apomorphies that can be used to identify fossil lizard remains from the Quaternary of North America. The Neogene and Quaternary fossil record for lizards is largely composed of tooth-bearing elements from the upper and lower jaws [5, 9], yet it has been shown that there remain previously undiscovered apomorphies on other skeletal elements that are useful for fossil identification [19, 20]. The authors of Bell et al. [9] posited that the abundance of fossil lizard tooth-bearing elements from the upper and lower jaws is due to a historical emphasis placed on mammal fossil collections and a general unfamiliarity of paleontologists with the broader lizard skeletal system. Therefore, for fossil skeletal elements examined here (Fig 1), we also include images from a diverse set of North American lizard taxa to showcase morphological diversity and facilitate the recognition and identification of fossil lizard remains by paleontologists who do not specialize in lizard osteology.

## Methods

Fossils in this study were previously excavated from Hall's Cave and are accessioned at the University of Texas Vertebrate Paleontology Collections (TxVP), locality 41229. No permits were required for this study, which complied with all relevant regulations. We largely

restricted our osteological comparative material to North American (NA) lizard taxa (see Table 1 of [21]) including those that today live on or north of the Isthmus of Panama, as well as those that inhabit Caribbean islands. Our comparative dataset was chiefly based on dry skeletal specimens; however, we also examined skeletons from specimens scanned using high-resolution x-ray computed tomography (CT). We aimed to examine at least one specimen for every North American lizard family for our comparative dataset, and for some families (e.g., Phrynosomatidae) we were able to sample all or nearly all genera. However, it was difficult to obtain comparative material for a few NA lizard families (e.g., Gymnophthalmidae) and so we instead relied on morphological evidence from published literature. The full list of comparative specimens and their associated metadata can be found in S1 Table. Institutional abbreviations appearing in the manuscript are as follows: CAS, California Academy of Sciences; FMNH, Field Museum of Natural History; LACM, Natural History Museum of Los Angeles County; TNHC, Texas Biodiversity Collections (Texas Natural History Collections), The University of Texas at Austin; TxVP M, Texas Vertebrate Paleontology Modern Collections, The University of Texas at Austin; UF, University of Florida; SDNHM, San Diego Natural History Museum. Our phylogenetic framework follows that of Burbrink et al. [22] for higher squamate relationships (Fig 2).

We employed an apomorphy-based fossil identification framework using previously published global-scale apomorphies for squamates. Apomorphies taken from the literature are listed in S2 Table. Although we do not treat snake fossils here, we note that the global-scale apomorphies used in this study have been evaluated in a framework that includes snakes and are used here to diagnose specific clades within Squamata. Furthermore, all fossil lizard elements described here preserve apomorphic states that provide evidence against identification of fossils to Serpentes. Apomorphy-based treatment of fossil snakes from Hall's Cave awaits an equally lengthy assessment in future work.

In some cases, our global-scale apomorphy based identifications allowed us to identify fossils to clades with NA and non-NA components. In these cases, we geographically restricted our identifications on the continental scale. We supplemented our global-scale apomorphic identification framework with new tentative apomorphies and morphological differences among taxa that we report here based on our comparative dataset of largely NA lizard taxa. We were not able to examine every NA lizard species or clade and therefore take a conservative approach to fossil identifications. Newly reported apomorphies can be found in the text and are based on our NA-restricted comparative dataset. We note that newly reported apomorphies must be examined in non-NA taxa before being used in a global-scale apomorphic diagnosis. Anatomical terminology follows that of Evans [24] unless otherwise noted and we provide labeled skeletal elements identifying many of the anatomical structures referenced in the text to orient readers (Figs 3 and 4). We were able to base our fossil descriptions on one or a few exemplar fossil specimens because we observed no substantive differences between exemplar and other referred specimens, unless otherwise noted.

## Systematic paleontology

### Iguania Cuvier, 1817 [25]

**Pleurodonta Cope, 1864 [26].** Referred specimen: Nasal, 41229–25562 right.

**Nasal.** *Description.* TxVP 41229–25562 is a right nasal (Fig 5A). There is a long anteromedial process with a lateral articulation facet for the premaxilla. There is a shorter anterolateral process (supranarial process). The posterior end is narrow compared to the anterior end. The ventral surface is concave, and there are two foramina that pierce the bone vertically.

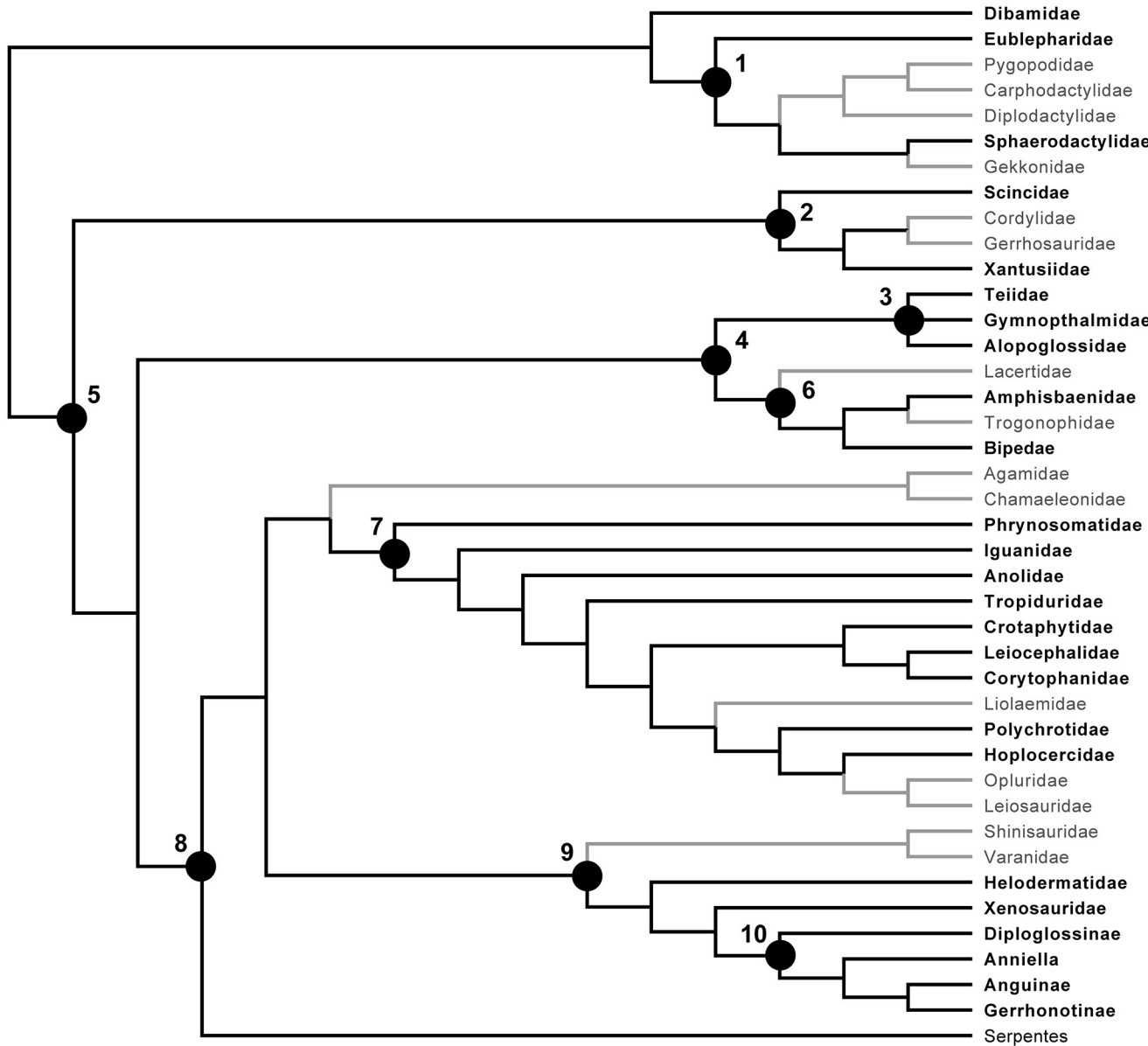

**Fig 2. Cladogram of Squamata showing evolutionary framework used for evaluating apomorphies and tentative apomorphies reported in this study.** Native North American clades are shown in black while non-North American clades are in grey. Labeled nodes correspond to the following: 1, Gekkota; 2, Scincomorpha; 3, Gymnophthalmoidea (= Teiioidea of [23]); 4, Laterata (= Lacertoidea); 5, Unidentata; 6, Amphisbaenia; 7, Pleurodonta; 8, Toxicofera; 9, Anguimorpha (= Neoanguimorpha of [22]); 10, Anguidae.

*Identification.* The fossil shares with pleurodontans a supranarial process [27], although the process is much longer in some pleurodontan taxa (e.g., *Dipsosaurus dorsalis*), and a nasal that gradually narrows posteriorly. A supranarial process is also present in some non-iguanians but it is usually small except for *Diploglossus bilobatus* (Fig 6). The fossil nasal distinctly narrows posteriorly similar to that observed in many examined North American pleurodontans (Fig 7). The fossil nasal differs from non-pleurodontan North American lizards in steadily tapering posteriorly, resulting in a sub-triangular shape. Furthermore, many non-pleurodontan North

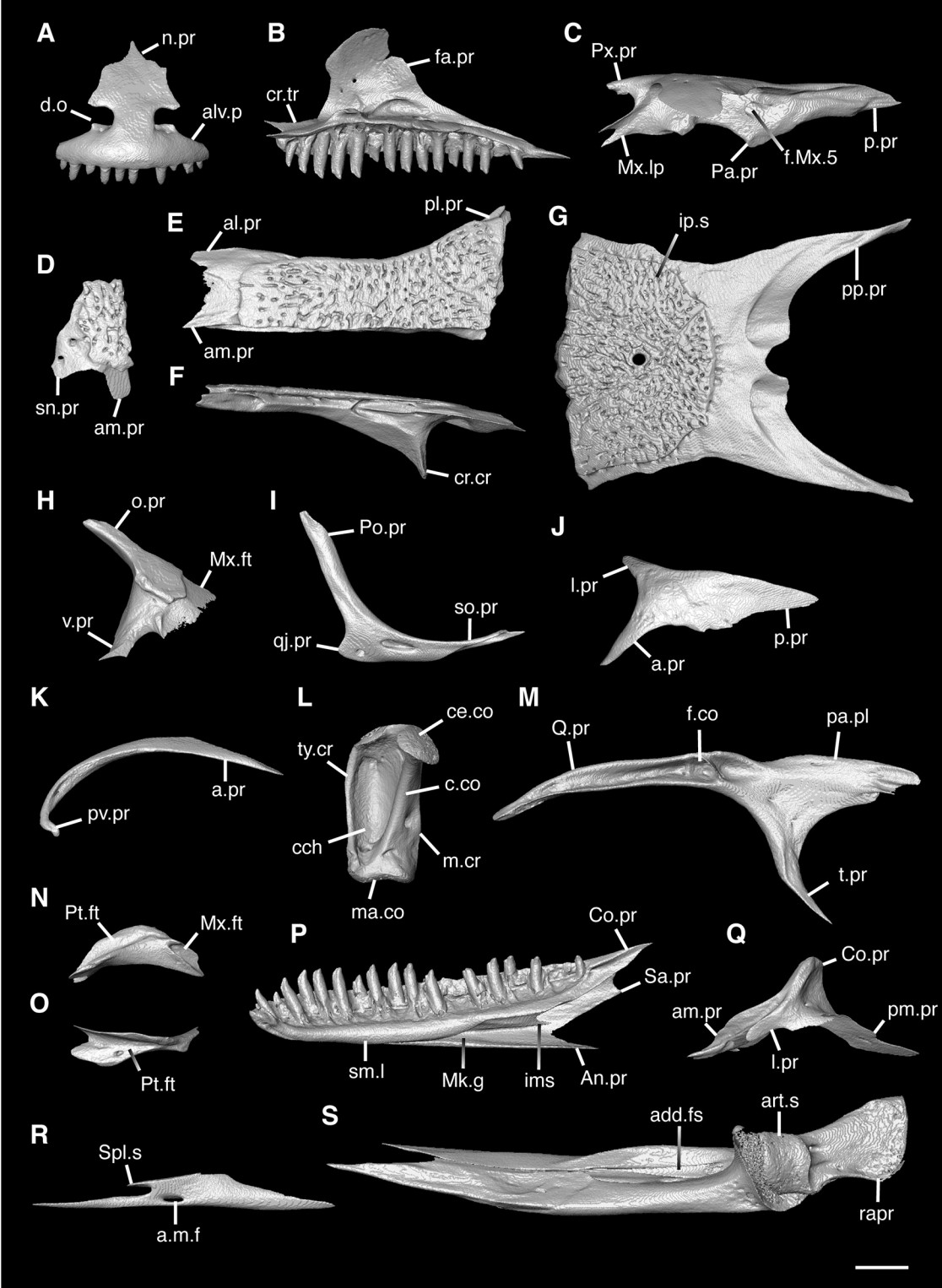

**Fig 3. Cranial elements of *Diploglossus bilobatus* TNHC 31933 with relevant anatomical structures labeled. A**. Premaxilla in anterior view; **B-C**. Right maxilla in medial and dorsal views; **D**. Left nasal dorsal view; **E-F**. Right frontal dorsal and lateral views; **G**. Parietal in dorsal view; **H**. Right prefrontal lateral view; **I**. Right jugal lateral view; **J**. Right postorbitofrontal dorsal view; **K**. Right squamosal lateral view; **L**. Left quadrate posterior view; **M**. Right pterygoid dorsal view; **N-O**. Right ectopterygoid ventral and lateral views; **P**. Right dentary medial view; **Q**. Left coronoid lateral view; **R**. Right splenial medial view; **S**. Left compound bone dorsal view.

Scale bar = 1 mm. **Abbreviations**: add.fo, adductor fossa; al.pr, anterolateral process; alv.p, alveolar plate; a.m.f, anterior mylohyoid foramen; am.pr, anteromedial process; An.pr, Angular process; a.pr, anterior process; art.s, articular surface; cch, conch; c.co, central column; ce.co, cephalic condyle; Co.pr, Coronoid process; cr.cr, cristae cranii; cr.tr, crista transversalis; d.o, dorsal ossification; fa.pr, facial process; f.co, fossa columella; f.Mx.5, maxillary trigeminal foramina; ims, intramandibular septum; ip.s, interparietal shield; l. pr, lateral process; ma.co, mandibular condyle; m.cr, medial crest; Mk.g, Meckelian groove; Mx.ft, maxilla facet; Mx.lp, maxillary lappet; n.pr, nasal process; o.pr, orbital process; pa.pl, palatal plate; Pa.pr, palatine process; pl.pr, posterolateral process; pm.pr, posteromedial process; pp.pr, postparietal process; p.pr, posterior process; Pt.ft, Pterygoid facet; pv.pr, posteroventral process; Px.pr, premaxillary process; qj.pr, quadratojugal process; Q.pr, Quadrate process; rapr, retroarticular process; Sa.pr, surangular process; sm.l, suprameckelian lip; sn.pr, supranarial process; so.pr, suborbital process; Spl.s, Splenial spine; t.pr, transverse process; ty.cr, tympanic crest; v.pr, ventral process.

American lizards, and anguimorphs in particular, differ from the fossil in often having a distinct rugose dorsal surface of the nasal.

## Crotaphytidae Smith & Brodie, 1982 [28]

Illustrated specimens referenced in the text: Compound bone, 41229–25853 left; Coronoid, 41229–25578 right; Dentary, 41229–27575 left; Dermarticular, 41229–25613 left; Ectopterygoid, 41229–25592 right; Frontal, 41229–28800; Jugal, 41229–25945 left; Maxilla, 41229–27039 right; Parietal, 41229–25836, 41229–28505; Postorbital, 41229–26975, 41229–27010 left, 41229–28984; Premaxilla, 41229–29095; Pterygoid, 41229–27144 right, 41229–28513 left; Quadrate, 41229–25579 left; Squamosal, 41229–28355 left; See S3 Table for complete list of specimens assigned to Crotaphytidae.

**Premaxilla.** *Description*.TxVP 41229–29095 is a premaxilla with five tooth positions filled with widely spaced unicuspid teeth (Fig 5B). It has a relatively flat anterior rostral surface and a long nasal process with distinct nasal articulation facets visible on the anterior and lateral surfaces. There are small foramina just lateral to the base of the nasal process and there are no anterior foramina. There are shallow maxillary facets laterally on the alveolar plate. The palatal plate is narrow and steeply slanted posteriorly. The short, rounded incisive process is slightly bilobed.

*Identification*. The fossil premaxilla is assigned to Pleurodonta based on being fused [23] with fewer than seven tooth positions [29]. Pleurodontans such as *Anolis*, *Polychrus*, corytophanids, and *Enyaliodes* differ from the fossil in having seven or more tooth positions [20, 30] (Fig 8) and examined specimens of *Anolis* have an incised posterior edge of the palatal process not found in the fossil. Members of Iguanidae differ in often having multicuspid teeth on the premaxilla [31]. Unicuspid teeth are sometimes found in *Ctenosaura*, *Cyclura*, and *Sauromalus* [32] but those taxa differ in that the anterior rostral face of the premaxilla in *Ctenosaura* is distinctly rounded [32] and the nasal process of *Ctenosaura*, *Cyclura*, and *Sauromalus* curves far posteriorly [20, 32]. The nasals were reported to overlap the nasal process of the premaxilla in *Crotaphytus collaris* [30], and the fossil shares with many examined *Crotaphytus* distinct facets visible on the nasal process in anterior view. The nasals of phrynosomatids do not overlap the nasal process of the premaxilla [30] and no examined phrynosomatids have distinct nasal facets on the anterior face of the nasal process. The fossil premaxilla differs from corytophanids and phrynosomatines in lacking anterior premaxillary foramina, which are also absent in most sceloporines and crotaphytids [29, 32]. We found that *Crotaphytus collaris* TxVP M-9255 has one small anterior premaxilla foramen. *Leiocephalus* also lacks anterior premaxillary foramina [29]. Examined *Leiocephalus* differ from the fossil in having at least seven tooth positions (see also [33]), but *Leiocephalus personatus* was reported to have six [29]. Additionally, the fossil and many examined *Crotaphytus collaris* have a palatal shelf that does not extend as far posteriorly compared to examined *Sceloporus* and *Leiocephalus*. On that basis, the fossil was assigned to Crotaphytidae.

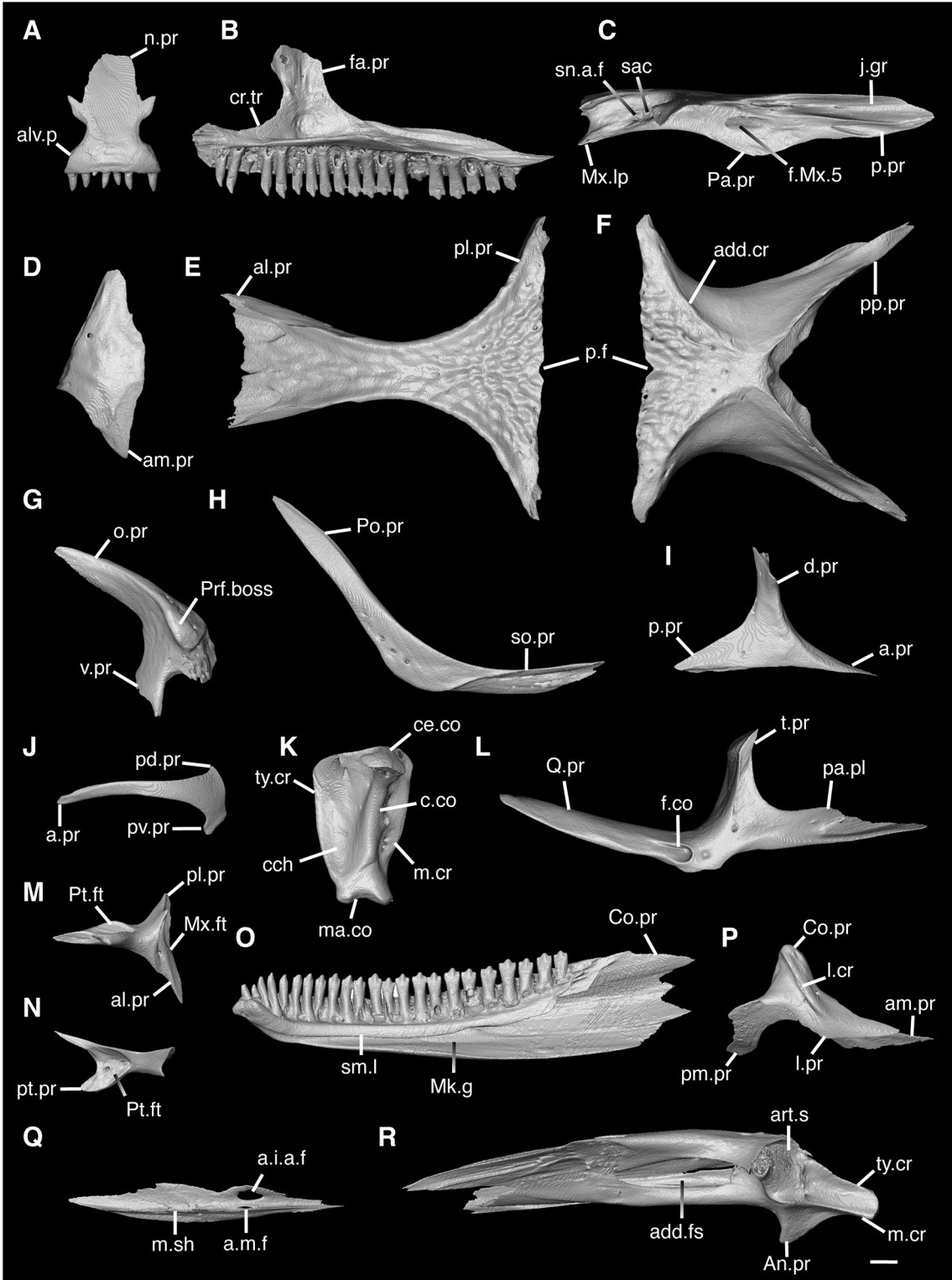

**Fig 4. Cranial elements of *Enyalioides heterolepis* UF 68015 with relevant anatomical structures labeled. A**. Premaxilla in anterior view; **B-C**. Right maxilla in medial and dorsal views; **D**. Right nasal dorsal view; **E**. Frontal dorsal view; **F**. Parietal in dorsal view; **G**. Right prefrontal lateral view; **H**. Right jugal lateral view; **I**. Right postorbitofrontal in dorsolateral view; **J**. Left squamosal lateral view; **K**. Left quadrate posterior view; **L**. Left pterygoid dorsal view; **M-N**. Right ectopterygoid ventral and anterior views; **O**. Right dentary medial view; **P**. Right coronoid lateral view; **Q**. Right splenial medial view; **R**. Left compound bone dorsal view. Scale bar = 1 mm.

**Abbreviations**: add.cr, adductor crest; add.fs, adductor fossa; al.pr, anterolateral process; alv.p, alveolar plate; a.m.f, anterior mylohyoid foramen; am.pr, anteromedial process; An.pr, Angular process; a.pr, anterior process; art.s, articular surface; cch, conch; c.co, central column; ce.co, cephalic condyle; Co.pr, Coronoid process; cr.tr, crista transversalis; d.pr, dorsal process; fa.pr, facial process; f.co, fossa columella; f.Mx.5, maxillary trigeminal foramina; j.gr, jugal groove; l.cr, lateral crest; l.pr, lateral process; ma.co, mandibular condyle; m.cr, medial crest; Mk.g, Meckelian groove; m.sh, medial shelf; Mx.ft, maxilla facet; Mx.lp, maxillary lappet; n.pr, nasal process; o.pr, orbital process; pa.pl, palatal plate; Pa.pr, palatine process; pd.pr, posterodorsal process; p.f, parietal forman; pl.pr, posterolateral process; pm.pr, posteromedial process; pp.pr, postparietal process; p.pr, posterior process; Pft.boss, prefrontal boss; Pt.ft, Pterygoid facet; pt.pr, pterygoid process; pv.pr, posteroventral process; Q.pr, Quadrate process; sac, opening for the superior alveolar canal; sm.l, suprameckelian lip; sn.a.f, subnarial arterial foramen; so.pr, suborbital process; t.pr, transverse process; ty.cr, tympanic crest; v.pr, ventral process.

**Maxilla.** *Description*. TxVP 41229–27039 serves as the basis for our description (Fig 5C). TxVP 41229–27039 is a right maxilla with 15 tooth positions. The facial process is tall and narrow, and gently curves anteromedially where it diminishes and merges with the crista transversalis. There is an opening for the superior alveolar canal (identified as the anterior inferior alveolar foramen by [20]) anterior to the facial process, and an elongate opening for the subnarial artery. There is a sub-triangular, symmetrical palatine process and an anteroposteriorly elongate depression (gutter) on the palatal shelf housing the superior alveolar nerve and maxillary artery. The lateral wall of the posterior orbital process (suborbital process) is short. The posterior orbital process is narrow, and elongated with a deep, narrow jugal groove. Teeth are tricuspid and the distal teeth are widened such that the base of the tooth is substantially wider than the crown. There are five nutrient foramina along the lateral surface of the bone, and a few foramina scattered on the lateral face of the facial process.

*Identification*. Maxillae were referred to Pleurodonta based on the presence of an elongate depression on the palatal shelf encompassing the superior alveolar nerve and maxillary artery [29] (Fig 9). A well-defined depression is absent in non-iguanian lizards [34]. Furthermore, maxillae can be assigned to Pleurodonta based on the presence of pleurodont teeth with tricuspid crowns, and two foramina on the dorsal surface of the premaxillary process [29]. Maxillae were identified to Crotaphytidae based on having teeth with bases much wider than the crowns, and a deep jugal groove beginning lateral to the palatine process and running posteriorly along the posterior orbital process [29, 35]. A deep jugal groove is absent in phrynosomatids but is present in other North American pleurodontans, including Corytophanidae, Iguanidae, and Leiocephalidae [29, 35]. Among those taxa, only Crotaphytidae, some members of Corytophanidae, and *Leiocephalus* have a large palatine process [35]. Many iguanids also differ from crotaphytids in having multicuspid teeth that are flared at the crown [31]. *Leiocephalus* is precluded because species within that genus have tightly packed teeth with flared crowns without a widened base [29]. Corytophanids differ in having a medially widened posterior orbital process of the maxilla, an anteroposteriorly long facial process, a tall lateral edge of the maxilla posterior to the facial process, and lacking teeth with bases much wider than the crowns [35].

**Frontal.** *Description*. TxVP 41229–28800 lacks only the end of the right posterolateral corner (Fig 5D). The bone has a narrow, waisted interorbital region with posterolateral margins that flare laterally. There is a large, deep midline notch on the posterior edge. The dorsal surface has a slightly rugose texture. The anterior end is triradiate with large, well defined nasal facets separated by an anteromedial process. There are distinct prefrontal facets on the anterolateral ends of the bone. TxVP 41229–28800 is slightly concave ventrally and has short cristae cranii that approach one another in the interorbital region and bound an indistinct groove for attachment of the solium supraseptale. The cristae cranii diverge posteriorly and extend along the lateral margins of the ventral surface.

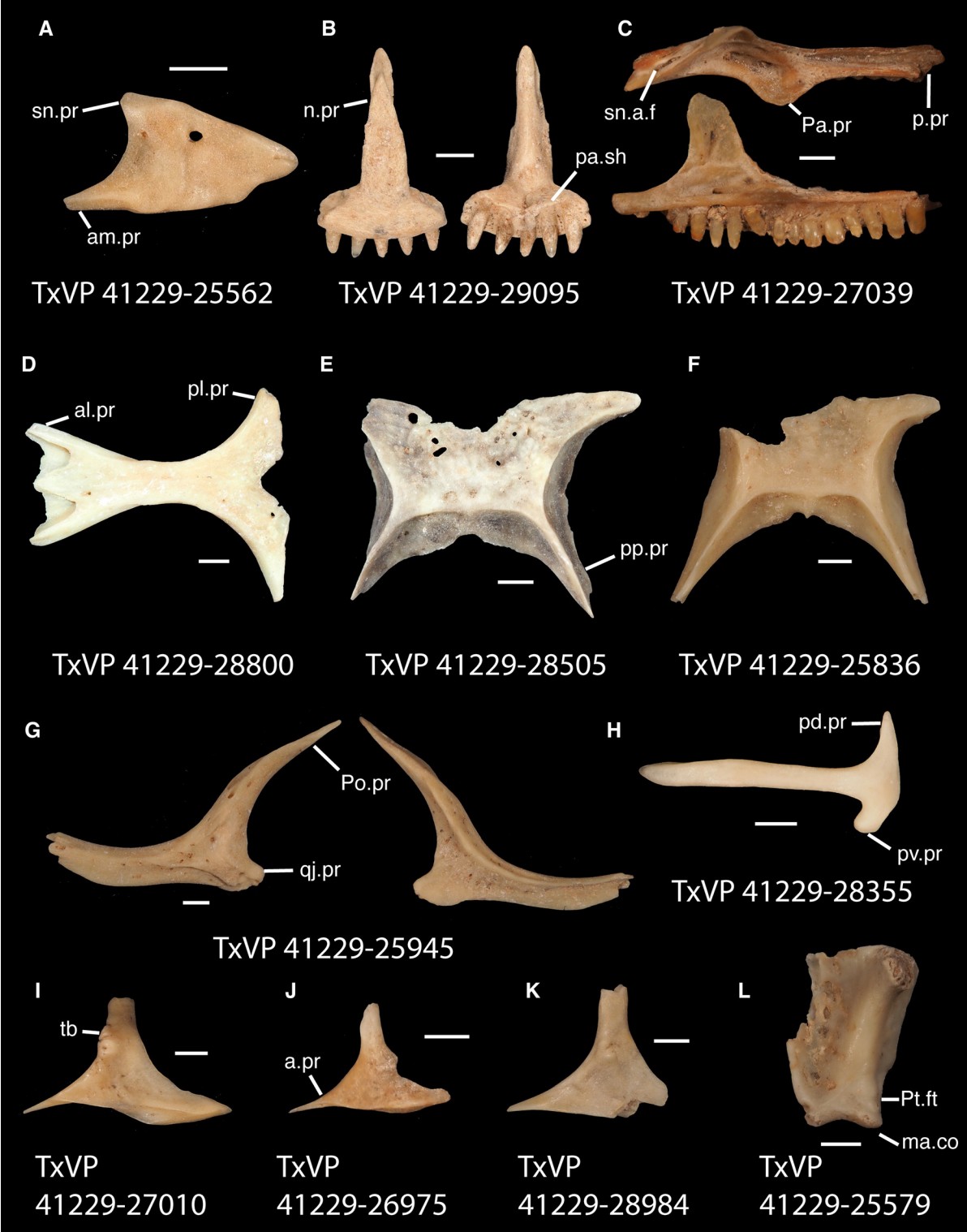

**Fig 5.** Fossil pleurodontans, A: Pleurodonta, B–L: Crotaphytidae. **A**. TxVP 41229–25562 Dorsal view of nasal; **B**. TxVP 41229–29095 Anterior view of premaxilla; **C**. TxVP 41229–27039 Dorsal and medial view of right maxilla; **D**. TxVP 41229–28800 Dorsal view of frontal; **E**. TxVP 41229–28505 Dorsal view of parietal; **F**. TxVP 41229–25836 Dorsal view of parietal; **G**. TxVP 41229–25945 Lateral and medial view of left jugal; **H**. TxVP 41229–28355 Lateral view of left squamosal; **I**. TxVP 41229–27010 Dorsolateral view of left postorbital; **J**. TxVP 41229–26975 Dorsolateral view of left postorbital; **K**. TxVP 41229–28984 Dorsolateral view of left postorbital; **L**. TxVP 41229–25579 Posterior view

of left quadrate. Scale bars = 1 mm. **Abbreviations**: al.pr, anterolateral process; am.pr, anteromedial process; a.pr, anterior process; ma.co, mandibular condyle; n.pr, nasal process; Pa.pr, palatine process; pa.sh, palatal shelf; pd.pr, posterodorsal process; pl.pr, posterolateral process; pp.pr, postparietal process; p.pr, posterior process; Pt.ft, pterygoid facet; pv.pr, posteroventral process; sn.a.f, subnarial arterial foramen; sn.pr, supranarial process; tb, tubercle.

*Identification*. TxVP 41229–28800 is assigned to Pleurodonta based on the presence of a fused frontal with reduced cristae cranii [23]. Teiids, gymnophthalmids, alopoglossids, geckos, some anguids, and some skins also have a fused frontal [23, 36, 37]. Teiids were reported to

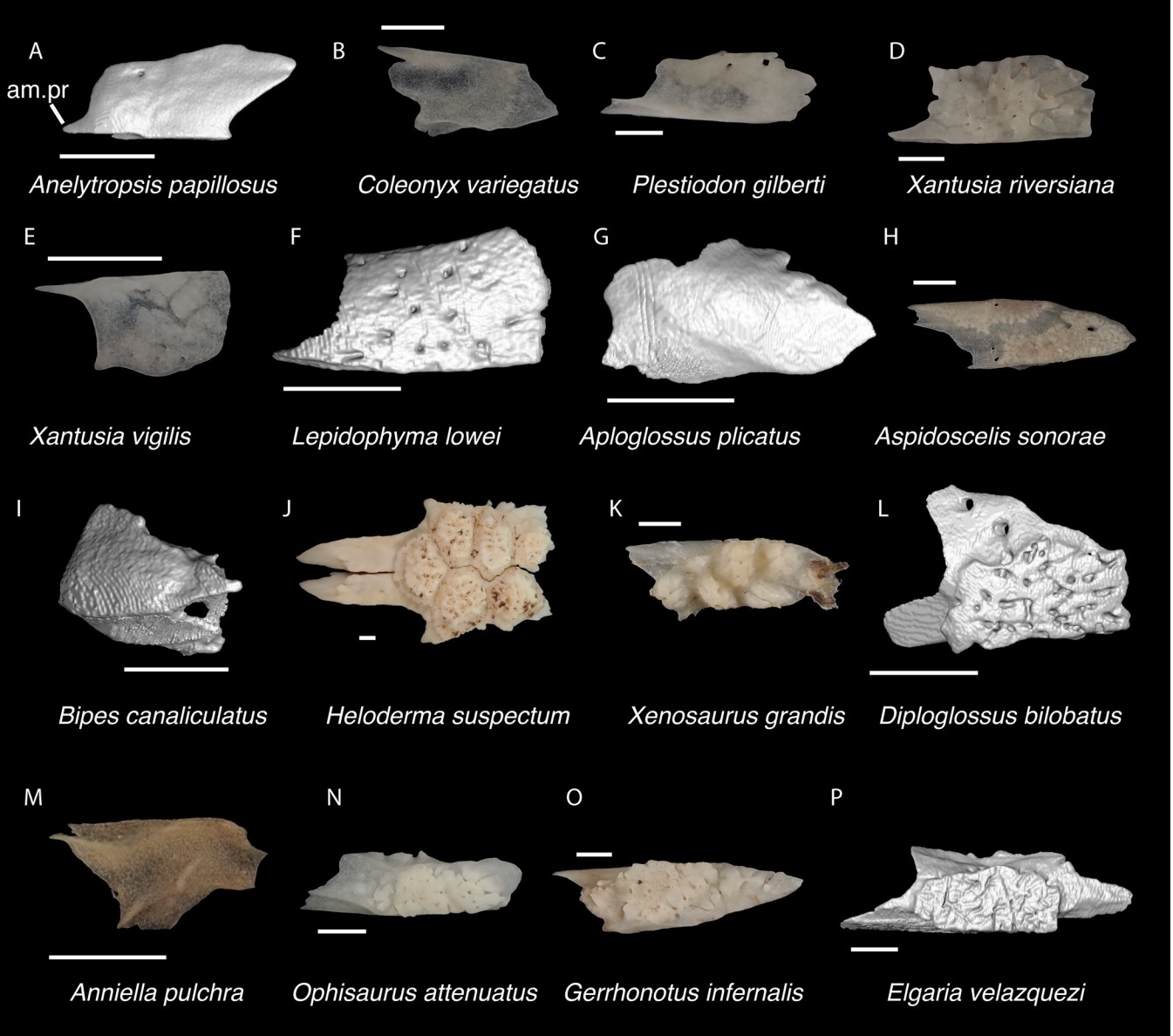

**Fig 6. Non-pleurodontan nasals.** Nasals in dorsal view–**A**. *Anelytropsis papillosus* UF 86708 right nasal; **B**. *Coleonyx variegatus* TxVP M-12109 left nasal; **C**. *Plestiodon gilberti* TxVP M-8587 right nasal; **D**. *Xantusia riversiana* TxVP M-8505 right nasal; **E**. *Xantusia vigilis* TxVP M-12130 left nasal; **F**. *Lepidophyma lowei* LACM 143367 right nasal; **G**. *Aploglossus plicatus* TNHC 34481 right nasal; **H**. *Aspidoscelis sonorae* TxVP M-15670 left nasal; **I**. *Bipes canaliculatus* CAS 134753 right nasal; **J**. *Heloderma suspectum* TxVP M-9001 left and right nasals; **K**. *Xenosaurus grandis* TxVP M-8960 left nasal; **L**. *Diploglossus bilobatus* TNHC 31933 right nasal; **M**. *Anniella pulchra* TxVP M-8678 left nasal; **N**. *Ophisaurus attenuatus* TxVP M-8979 right nasal; **O**. *Gerrhonotus infernalis* TxVP M-13441 left nasal; **P**. *Elgaria velazquezi* SDNHM 68677 left nasal. Scale bars = 1 mm. **Abbreviations**: am.pr, anteromedial process.

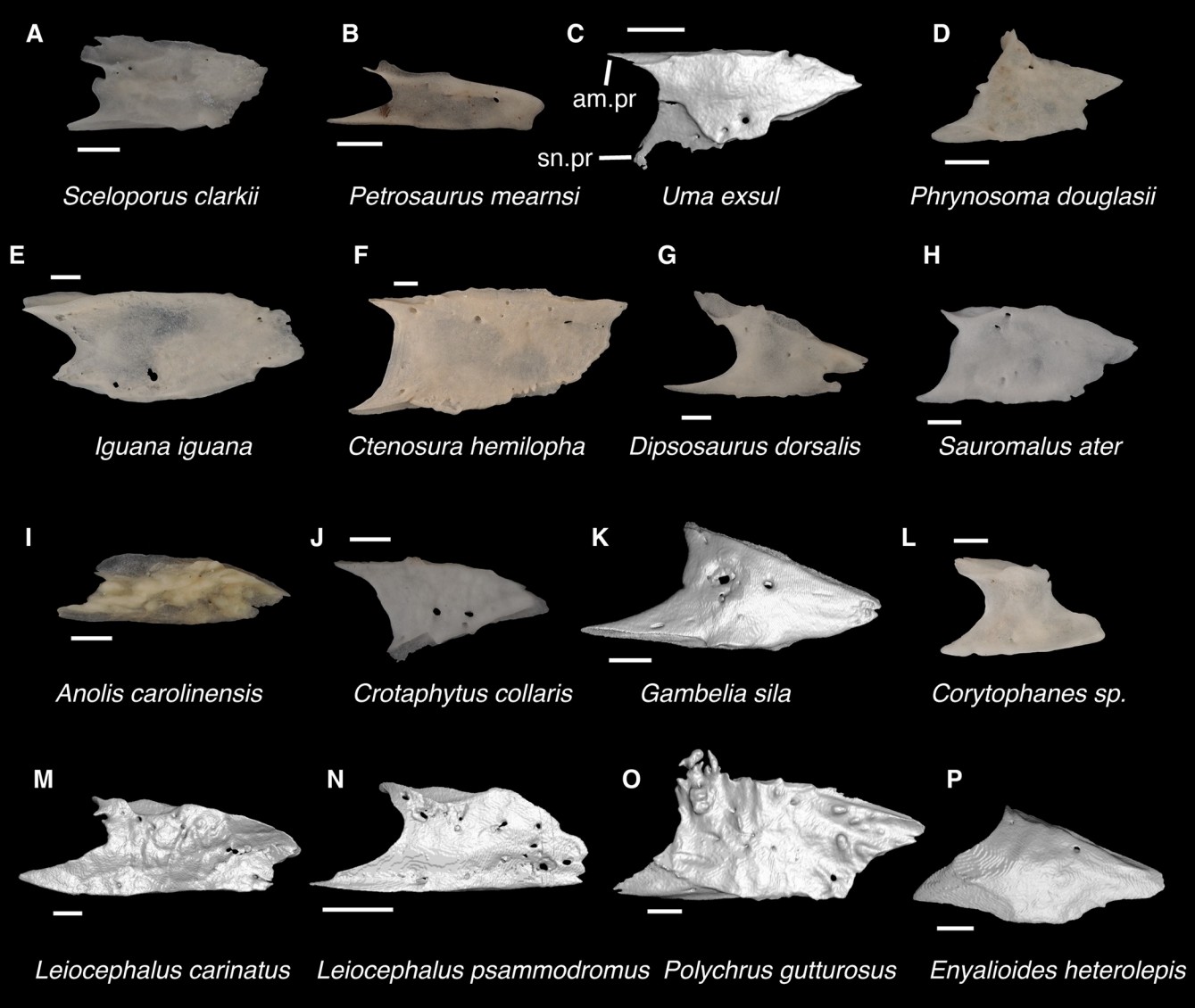

**Fig 7. Pleurodontan nasals.** Nasals in dorsal view–**A**. *Sceloporus clarkii* TxVP M-12157 right nasal; **B**. *Petrosaurus mearnsi* TxVP M-14910 right nasal; **C**. *Uma exsul* TNHC 30247 left nasal; **D**. *Phrynosoma douglasii* TxVP M-8526 right nasal; **E**. *Iguana iguana* TxVP M-8454 left nasal; **F**. *Ctenosura hemilopha* TxVP M-8616 right nasal; **G**. *Dipsosaurus dorsalis* TxVP M-13086 right nasal; **H**. *Sauromalus ater* TxVP M-11599 right nasal; **I**. *Anolis carolinensis* TxVP M-9042 right nasal; **J**. *Crotaphytus collaris* TxVP M-12468 left nasal; **K**. *Gambelia sila* TNHC 95261 right nasal; **L**. *Corytophanes* sp. TxVP M-16765 left nasal; **M**. *Leiocephalus carinatus* TNHC 89274 right nasal; **N**. *Leiocephalus psammodromus* TNHC 103220 right nasal; **O**. *Polychrus gutturosus* TNHC 24152 right nasal; **P**. *Enyalioides heterolepis* UF 68015 right nasal. Scale bars = 1 mm. **Abbreviations**: am.pr, anteromedial process; sn.pr, supranarial process.

differ from iguanians in lacking strongly constricted interorbital margins of the frontal [23]; however, we observed some specimens of *Aspidoscelis* that have similarly constricted interorbital margins and the degree of interorbital constriction may be ontogenetically related in *Pholidoscelis* (Bochaton et al. 2019). Examined *Aspidoscelis*, mabuyines, and *Scincella* differ from the fossil and from other pleurodontans in having an interorbital edge that weakly curves posterolaterally. Gymnophthalmids and alopoglossids possess frontal tabs not found in iguanians [23, 37]. Geckos differ from iguanians in having the crista cranii meet to form an enclosed olfactory canal [23, 38]. Anguimorphs have well-developed cristae cranii and usually have co-ossified osteoderms [24]. The fossil and Crotaphytidae share a deep, narrow notch in the

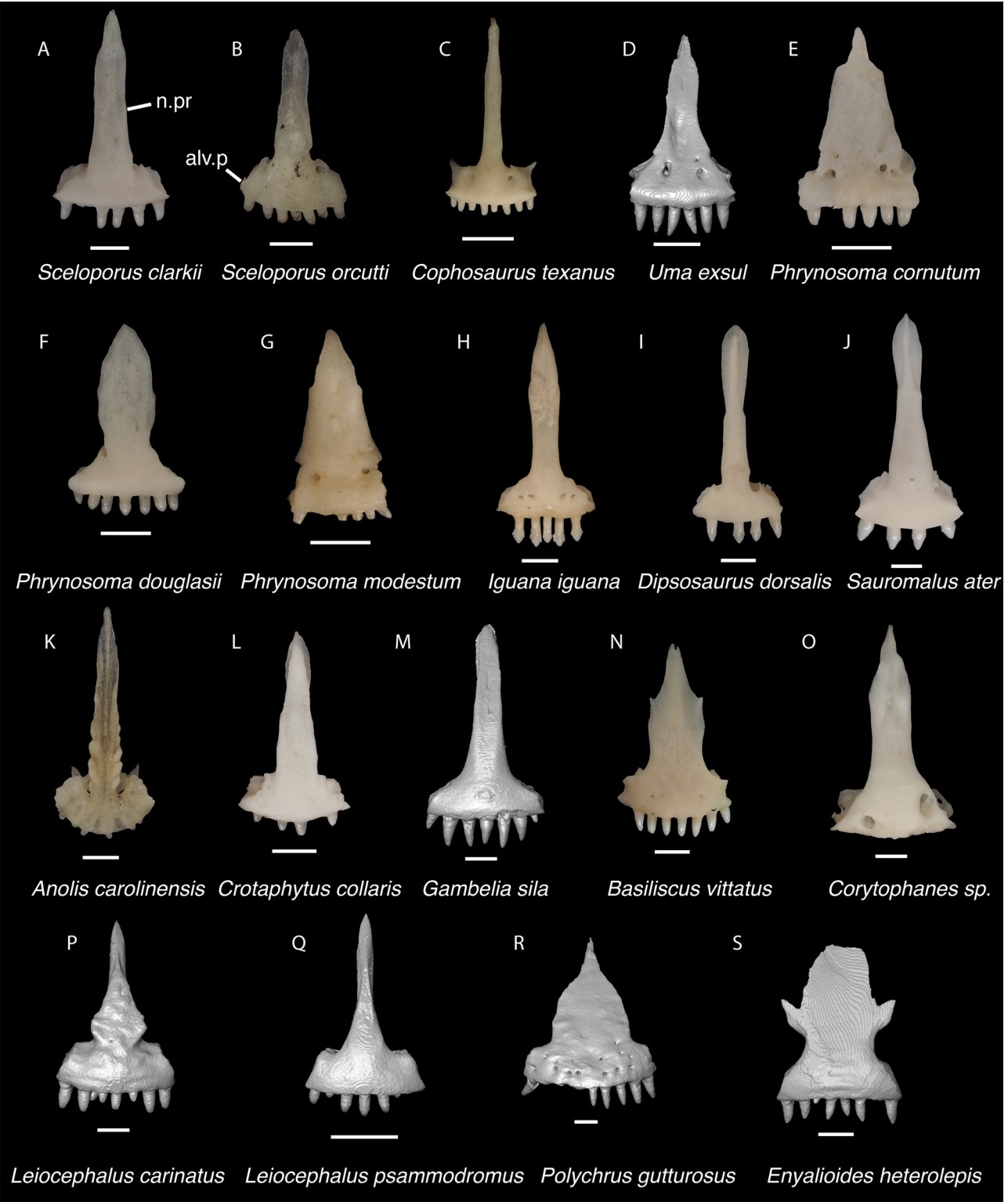

**Fig 8. Pleurodontan premaxillae.** Premaxillae in anterior view–**A**. *Sceloporus clarkii* TxVP M-12157; **B**. *Sceloporus orcutti* TxVP M-12155; **C**. *Cophosaurus texanus* TxVP M-8527; **D**. *Uma exsul* TNHC 30247; **E**. *Phrynosoma cornutum* TxVP M-9621; **F**. *Phrynosoma douglasii* TxVP M-8526; **G**. *Phrynosoma modestum* TNHC 95921; **H**. *Iguana iguana* TxVP M-13054; **I**. *Dipsosaurus dorsalis* TxVP M-13086; **J**. *Sauromalus ater* TxVP M-11599; **K**. *Anolis carolinensis* TxVP M-9042; **L**. *Crotaphytus collaris* TxVP M-12468; **M**. *Gambelia sila* TNHC 95261; **N**. *Basiliscus vittatus* TxVP M-8556; **O**. *Corytophanes* sp. TxVP M-16765; **P**. *Leiocephalus carinatus* TNHC 89274; **Q**. *Leiocephalus psammodromus* TNHC 103220; **R**. *Polychrus gutturosus* TNHC 24152; **S**. *Enyalioides heterolepis* UF 68015. Scale bars = 1 mm. **Abbreviations**: alv.p, alveolar plate; n.pr, nasal process.

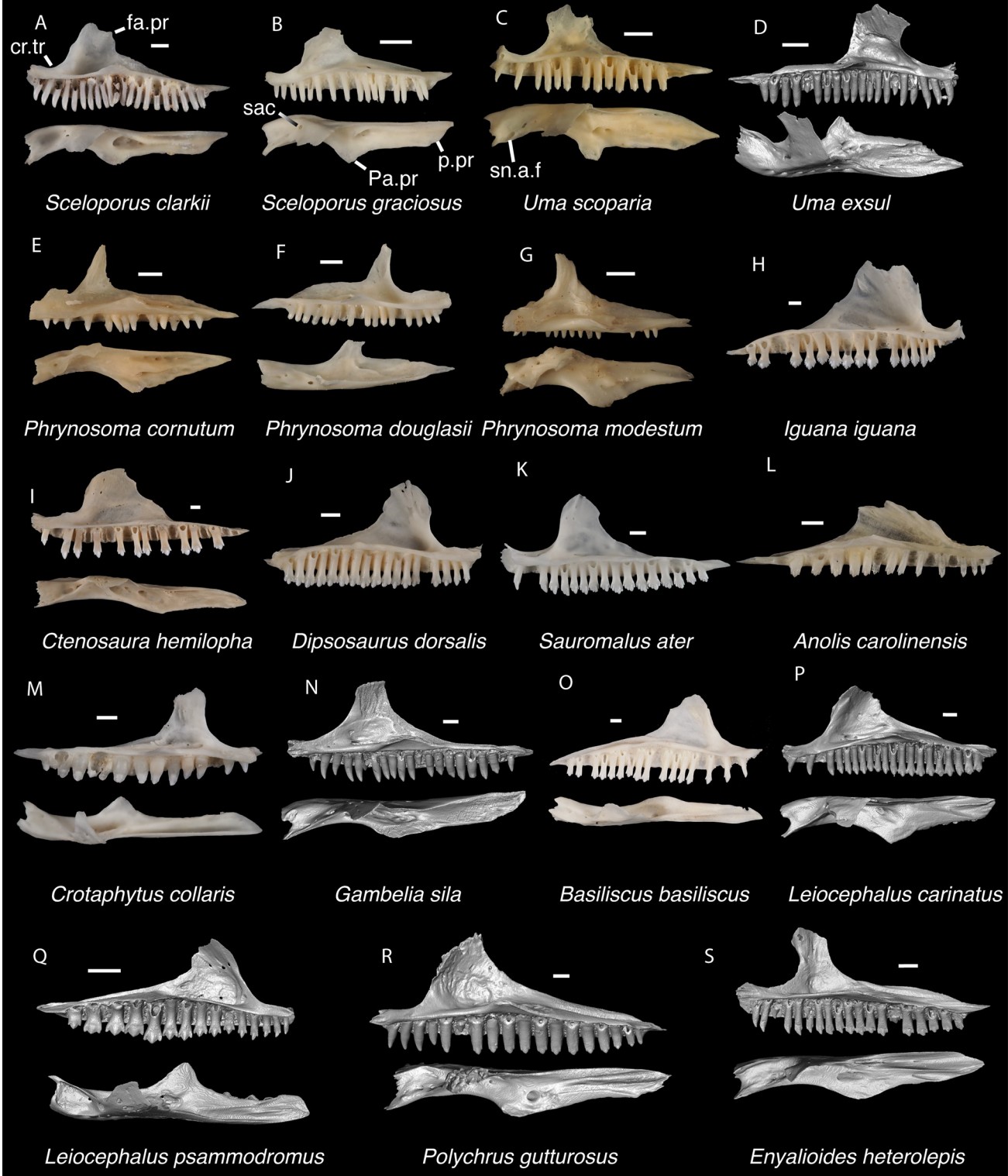

**Fig 9. Pleurodontan maxillae.** Maxillae in medial and dorsal views–**A**. *Sceloporus clarkii* TxVP M-12157 right maxilla; **B**. *Sceloporus graciosus* TxVP M-14879 right maxilla; **C**. *Uma scoparia* TxVP M-8529 right maxilla; **D**. *Uma exsul* TNHC 30247 left maxilla; **E**. *Phrynosoma cornutum* TxVP M-6405 right maxilla; **F**. *Phrynosoma douglasii* TxVP M-8526 left maxilla; **G**. *Phrynosoma modestum* TNHC 95921 right maxilla; **H**. *Iguana iguana* TxVP M-8454 left maxilla; **I**. *Ctenosaura hemilopha* TxVP M-8616 right maxilla; **J**. *Dipsosaurus dorsalis* TxVP M-13086 left maxilla; **K**. *Sauromalus ater* TxVP M-11599 right maxilla; **L**. *Anolis carolinensis* TxVP M-9042 left maxilla; **M**. *Crotaphytus collaris* TxVP M-12468 left maxilla; **N**. *Gambelia sila* TNHC 95261 right

maxilla; **O**. *Basiliscus basiliscus* TxVP M-11907 left maxilla; **P**. *Leiocephalus carinatus* TNHC 89274 right maxilla; **Q**. *Leiocephalus psammodromus* TNHC 103220 left maxilla; **R**. *Polychrus gutturosus* TNHC 24152 right maxilla; **S**. *Enyalioides heterolepis* UF 68015 right maxilla. Scale bars = 1 mm.
**Abbreviations**: cr.tr, crtista transversalis; fa.pr, facial process; Pa.pr, palatine process; p.pr, posterior process; sn.a.f, subnarial arterial foramen.

posterior edge of the frontal for the parietal foramen, constricted interorbital margins of the frontal, and a distinct anteromedial process on the anterior end of the frontal (Fig 10). The frontal of *Anolis* generally does not contribute to the parietal foramen [20, 39] and in *Anolis* and examined *Polychrus* the interorbital margins are considerably widened [20, 40]. The interorbital margins of the frontal are also considerably widened in *Ctenosaura similis* [20], *Iguana iguana*, *Iguana delicatissima*, and some *Leiocephalus* [40]; however, that morphology varies ontogenetically [31, 41]. *Anolis*, *Polychrus*, *Enyalioides heterolepis*, and some *Leiocephalus* differ from crotaphytids and the fossil in having a distinct rugose texture on the dorsal surface of the frontal [40, 42]. The parietal foramen is located entirely within the frontal in *Basiliscus*, *Corytophanes*, some *Laemanctus*, some *Sauromalus*, and in many *Dipsosaurus dorsalis* [31]. Among examined specimens of *Crotaphytus collaris*, the position of the parietal foramen is variably at the frontoparietal suture, largely within the frontal as demarcated by a deep, narrow notch in the posterior edge, or completely within the frontal (*C. collaris* TxVP M-8615). One examined specimen of *Gambelia* (*G. wislizenii* TxVP M-9974) also has the parietal foramen largely within the frontal. Among phrynosomatids, the parietal foramen is either located at the frontoparietal suture (e.g., some *Phrynosoma* [43]) or largely within the parietal [44]. However, most phrynosomatids lack the deep, narrow notch in the posterior edge of the frontal that is present in crotaphytids. Furthermore, many examined phrynosomatids do not have a distinct anteromedial process on the frontal, and larger species of *Sceloporus* that are similar in size to *Crotaphytus* have relatively wider interorbital margins of the frontal. On this basis, the fossil was assigned to Crotaphytidae.

**Parietal.** *Description*. TxVP 41229–25836 is a parietal that is missing the anterolateral corner and the ventral tips of the postparietal processes (Fig 5F). There is a distinct anterolateral process that strongly curves laterally from the parietal table. The adductor crests do not approach one another posteriorly, giving the parietal table a trapezoidal shape. The ventrolateral crests are low without distinct epipterygoid processes and are easily visible in dorsal view. The posterior edge between the postparietal processes is characterized by two distinct depressions (nuchal fossae) separated by a small ridge above a small notch. The postparietal processes have dorsal crests that slant medially. On the ventral surface there are shallow depressions (cerebral vault) divided by a low ridge. There is a deep pit for the processus ascendens just anterior to the posterior edge. TxVP 41229–28505 is similar to TxVP 41229–25836 but has a larger notch at the anterior edge (Fig 5E). This notch may be at least partially caused by erosion of the bone indicated by holes in the parietal table.

*Identification*. Parietals are assigned to Pleurodonta based on the presence of a fused parietal [23], the absence of co-ossified osteoderms [23, 24, 45], the absence of a parietal foramen fully enclosed by the parietal [23], and the absence of distinct ventrolateral crests (parietal downgrowths of [23]). The parietal of xantusiids is unfused except for *Cricosaura* and some *Xantusia riversiana* [23, 46]. The parietal in *Cricosaura* and some *Xantusia riversiana* differs from pleurodontans in being relatively rectangular without long postparietal processes [46]. Cnemidophorines, gymnophthalmids, alopoglossids, and skinks generally have well-developed, tall ventrolateral crests or projections on the ventral surface of the parietal [23, 47] and anguimorphs often have co-ossified osteoderms on the dorsal surface of the parietal table [19, 35]. Fossil parietals are assigned to Pleurodonta based on have a fused parietal without co-ossified osteoderms nor distinct ventrolateral crests. Among pleurodontans, the adductor crests

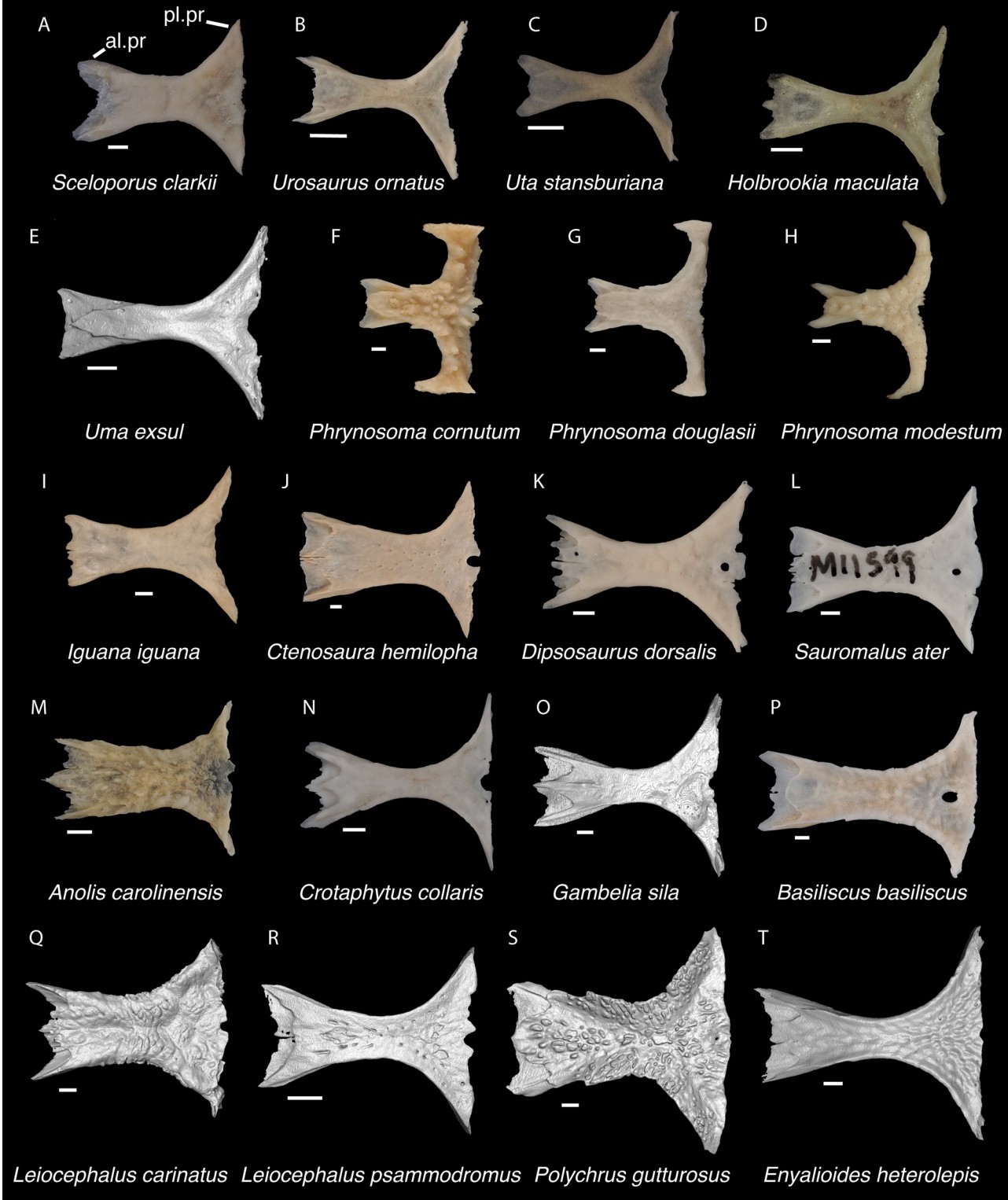

**Fig 10. Pleurodontan frontals.** Frontals in dorsal view–**A**. *Sceloporus clarkii* TxVP M-12157; **B**. *Urosaurus ornatus* TxVP M-8638; **C**. *Uta stansburiana* TxVP M-14935; **D**. *Holbrookia maculata* TxVP M-14322; **E**. *Uma exsul* TNHC 30247; **F**. *Phrynosoma cornutum* TxVP M-6405; **G**. *Phrynosoma douglasii* TxVP M-8526; **H**. *Phrynosoma modestum* TNHC 95921; **I**. *Iguana iguana* TxVP M-13054; **J**. *Ctenosaura hemilopha* TxVP M-8616; **K**. *Dipsosaurus dorsalis* TxVP M-13086; **L**. *Sauromalus ater* TxVP M-11599; **M**. *Anolis carolinensis* TxVP M-9042; **N**. *Crotaphytus collaris* TxVP M-12468; **O**. *Gambelia sila* TNHC 95261; **P**. *Basiliscus basiliscus* TxVP M-11907; **Q**. *Leiocephalus carinatus* TNHC 89274; **R**. *Leiocephalus psammodromus* TNHC

103220; **S**. *Polychrus gutturosus* TNHC 24152; **T**. *Enyalioides heterolepis* UF 68015. Scale bars = 1 mm. **Abbreviations**: al.pr, anterolateral process; pl.pr, posteolateral process.

form a "Y" shape of the parietal table in corytophanids, many anoles, and some iguanids [20, 48]. The adductor crests form a "V" shape in *Iguana*, *Ctenosaura*, *Cyclura*, some hoplocercids, some *Leiocephalus*, and some anoles [39, 40, 42] although there is substantial ontogenetic variation in this morphology [31, 39–41]. *Polychrus*, some *Anolis*, and some *Leiocephalus* differ from crotaphytids in having distinct rugosities on the dorsal surface of the parietal [33, 40, 42]. The pit for the processus ascendens in *Sauromalus* is reduced compared to other iguanids [20].

In the remaining NA pleurodontans, there is substantial morphological variation of the parietal (Fig 11), and it is possible additional variation not captured in our sample may make fossil identification to the family level more difficult. Here we list some tentative differences observed among examined specimens. In *Dipsosaurus dorsalis*, the postparietal processes gradually taper to a tip whereas in *Crotaphytus*, the postparietal processes are often widened until the posterior end. The dorsal crests on the postparietal processes in *Dipsosaurus dorsalis* are slanted laterally or directly lie midline along the process so that the medial surface of the postparietal process is visible in dorsal view. In crotaphytids, the crests on the postparietal processes are slanted medially, even just slightly, so that the medial surface of the postparietal process is obscured in dorsal view. The ventrolateral crests in crotaphytids are oriented more laterally compared to *Dipsosaurus dorsalis* and phrynosomatids, such that more of the supratemporal fossa is visible in dorsal view. The hoplocercid *Enyalioides heterolepis* has a large portion of the supratemporal fossa visible in dorsal view but has adductor crests that form a V shape. Parietals are assigned to Crotaphytidae based on the presence of a trapezoidal shaped parietal table, a distinct pit for the processus ascendens, medially slanted crests on the postparietal processes, and a supratemporal fossa that is broadly visible in dorsal view. The posterior portion of the parietal table is reportedly more constricted in *Crotaphytus* compared to *Gambelia*, but it was noted that there is substantial ontogenetic and perhaps sexual variation [49], so we refrain from making generic identifications.

**Jugal.** *Description*. TxVP 41229–25945 serves as the basis for our description (Fig 5G). TxVP 41229–25945 is a left jugal. Laterally, there is a distinct maxillary facet on the ventral half of the suborbital process and a depression near the inflection point. There is a distinct, wide quadratojugal process. The postorbital process is long, curves posterodorsally, and has a distinct postorbital facet on the anterodorsal surface. On the medial surface, there is a distinct ridge that is located at the dorsal margin of the suborbital process and that runs posteriorly on the anterior edge of the postorbital process. There are several foramina below the medial ridge and on the lateral surface including within the depression, on the postorbital process, and on the quadratojugal process.

*Identification*. Jugals were assigned to Pleurodonta based on the presence of a posteriorly deflected postorbital process (present in many but not all pleurodontan taxa, see S2 Table) and a medial ridge that is located anteriorly on the postorbital process and medially on the suborbital process [50]. Geckos have a small, reduced jugal, and dibamids lack a jugal [24, 51]. In examined anguids and scincids, the postorbital process is oriented dorsally and the jugal is more angulated [45]. Xantusiids differ in having a short suborbital process and *Xantusia riversiana* and *Lepidophyma* have an anteroposteriorly widened postorbital process [46]. Gymnophthalmoids differ in having a distinct medial ectopterygoid process, which is also present, albeit smaller, in some gymnophthalmids and alopoglossiids [24, 37]. The position of the medial ridge varies among North American lizards, but the presence of a medial ridge that is

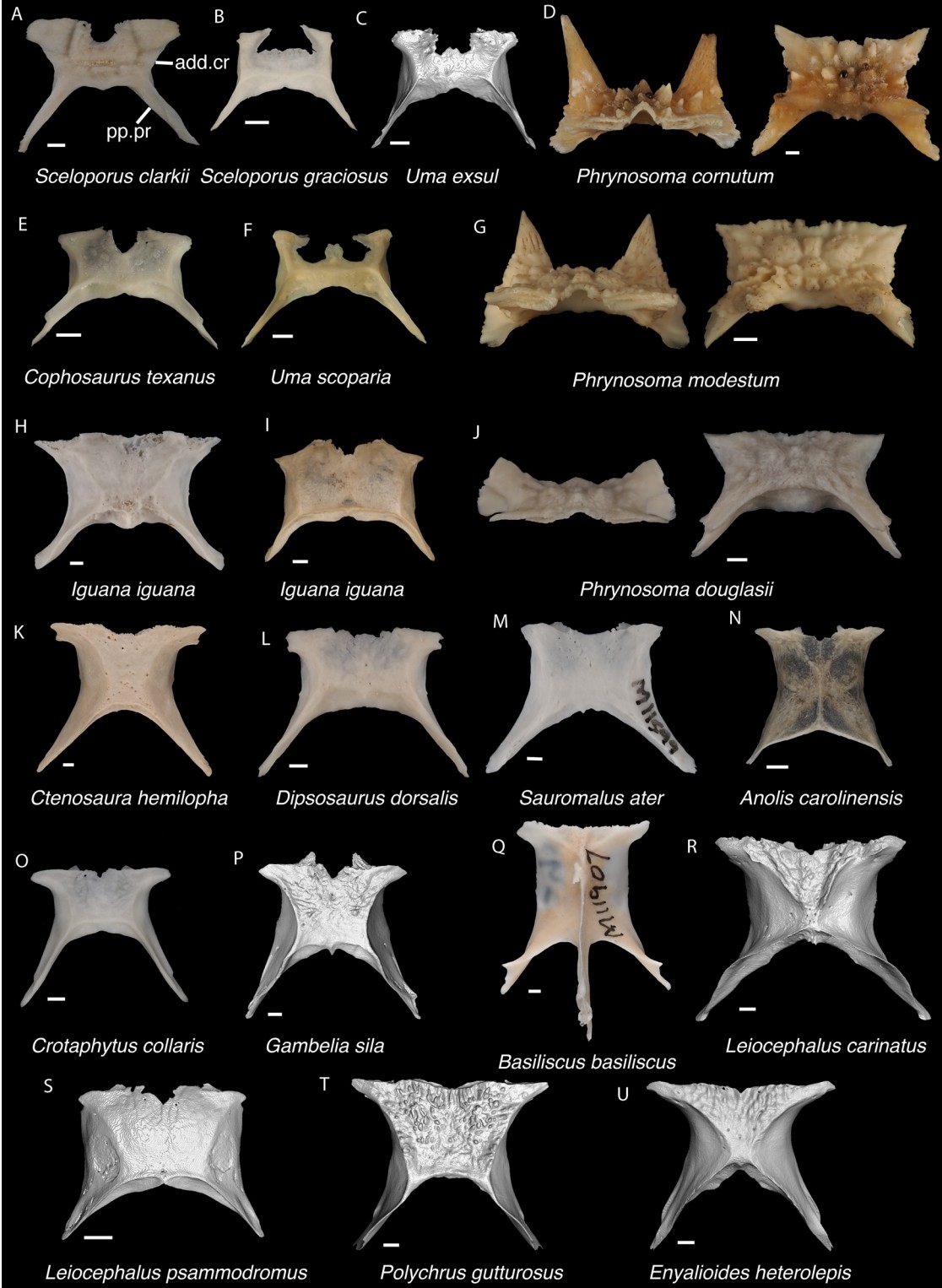

**Fig 11. Pleurodontan parietals.** Parietals in dorsal view–**A**. *Sceloporus clarkii* TxVP M-12157; **B**. *Sceloporus graciosus* TxVP M-14879; **C**. *Uma exsul* TNHC 30247; **D**. *Phrynosoma cornutum* TxVP M-6405; **E**. *Cophosaurus texanus* TxVP M-8527; **F**. *Uma scoparia* TxVP M-8529; **G**. *Phrynosoma modestum* TNHC 95921; **H**. *Iguana iguana* TxVP M-8454; **I**. *Iguana iguana* TxVP M-13054; **J**. *Phrynosoma douglasii* TxVP M-8526; **K**. *Ctenosaura hemilopha* TxVP M-8616; **L**. *Dipsosaurus dorsalis* TxVP M-13086; **M**. *Sauromalus ater* TxVP M-11599; **N**. *Anolis carolinensis* TxVP M-9042; **O**. *Crotaphytus collaris* TxVP M-12468; **P**. *Gambelia sila*

TNHC 95261; **Q**. *Basiliscus basiliscus* TxVP M-11907; **R**. *Leiocephalus carinatus* TNHC 89274; **S**. *Leiocephalus psammodromus* TNHC 103220; **T**. *Polychrus gutturosus* TNHC 24152; **U**. *Enyalioides heterolepis* UF 68015. Scale bars = 1 mm. **Abbreviations**: add. cr, adductor crest; pp.pr, postparietal process.

located anteriorly on the postorbital process and medially on the suborbital process is consistent with several pleurodontan clades [50]. Examined phrynosomatids, *Leiocephalus*, *Enyalioides heterolepis*, *Polychrus* (except *P. femoralis*; [20]), and iguanids differ from crotaphytids and the fossils in lacking a posteroventral process (quadratojugal process) [20] (Fig 12). Additionally, the anterior suborbital process in *Iguana iguana* and *Dipsosaurus dorsalis* is shortened relative to crotaphytids and the fossils. Examined corytophanids also lack a posteroventral process and the postorbital process is directed only dorsally, whereas in *Crotaphytus* the process is posteriorly deflected [29]. Examined *Corytophanes* differ in having a posteriorly widened postorbital process. Examined *Anolis* also have a posteroventral process [20], albeit somewhat less distinct than in the fossil and in crotaphytids. Examined anoles differ from crotaphytids and the fossils in having a thinner anterior suborbital process and a dorsal margin of the suborbital process that is everted laterally to a lesser degree. Additionally, examined *Anolis* differ in having a posteriorly widened postorbital process that tapers dorsally. Based on these differences with other NA pleurodontans listed above, fossils were assigned to Crotaphytidae.

**Postorbital.** *Description*. TxVP 41229–27010 is a well preserved left postorbital (Fig 5I). It is triradiate with a thin anterior process and thicker dorsal and posterior processes. There are distinct jugal and squamosal facets along the ventrolateral edge. The posterior edge is concave and there is a distinct tubercle near the middle of the anterior surface of the dorsal process. The tip of the dorsal process is squared off. In TxVP 41229–26975 the squamosal facet is less visible in dorsolateral view compared to TxVP 41229–27010, and the tubercle is on the posterior edge of the dorsal process (Fig 5J). The posterior process and the tip of the dorsal process are broken in TxVP 41229–28984. TxVP 41229–28984 differs from the other specimens in having a smaller tubercle on the dorsal process (Fig 5K).

*Identification*. Postorbitals were identified to Pleurodonta based on a sub-triangular shape (Fig 13) with a distinct ventral process [23]. Many anguimorphs, skinks, and gymnophthalmids differ in having a long posterior process of the postorbital. Teiids differ in having a quadradiate postorbitofrontal and alopoglossids differ in having a long posterior process of the postorbitofrontal. Among pleurodontans, crotaphytids and the fossils differ from phrynosomatids, *Enyalioides*, *Anolis*, and some *Leiocephalus* in having a large, round knob laterally on the postorbital [29, 52]. *Leiocephalus carinatus* has a knob but differs from the fossils in having a much taller posterior process and a posteriorly deflected dorsal process with an anterior articulation facet for the frontoparietal corner. *Polychrus* has a knob but differs from the fossils in having a distinct longitudinal canthal crest [20]. Examined juveniles of *Iguana iguana* lack a large round knob, but large skeletally mature *Iguana iguana* have a large knob. In large *Iguana* the bone is wider closer to the articulation with the frontal. In iguanids and corytophanids, the knob is located closer to the articulation with the frontal compared to the fossils and crotaphytids [29]. In addition, corytophanids differ in having a convex dorsal margin of the posterior process where it borders the supratemporal fenestra [29]. Based on these differences with other NA pleurodontans listed above, fossils were assigned to Crotaphytidae.

**Squamosal.** *Description*. TxVP 41229–28355 is a left squamosal (Fig 5H). The main shaft of the bone is concave ventrally and there are distinct posterodorsal and posteroventral processes. The posterodorsal process is elongated and pointed and the posteroventral process is rounded and points anteroventrally. There is an elongate facet of the posterodorsal process for articulation with the postparietal process.

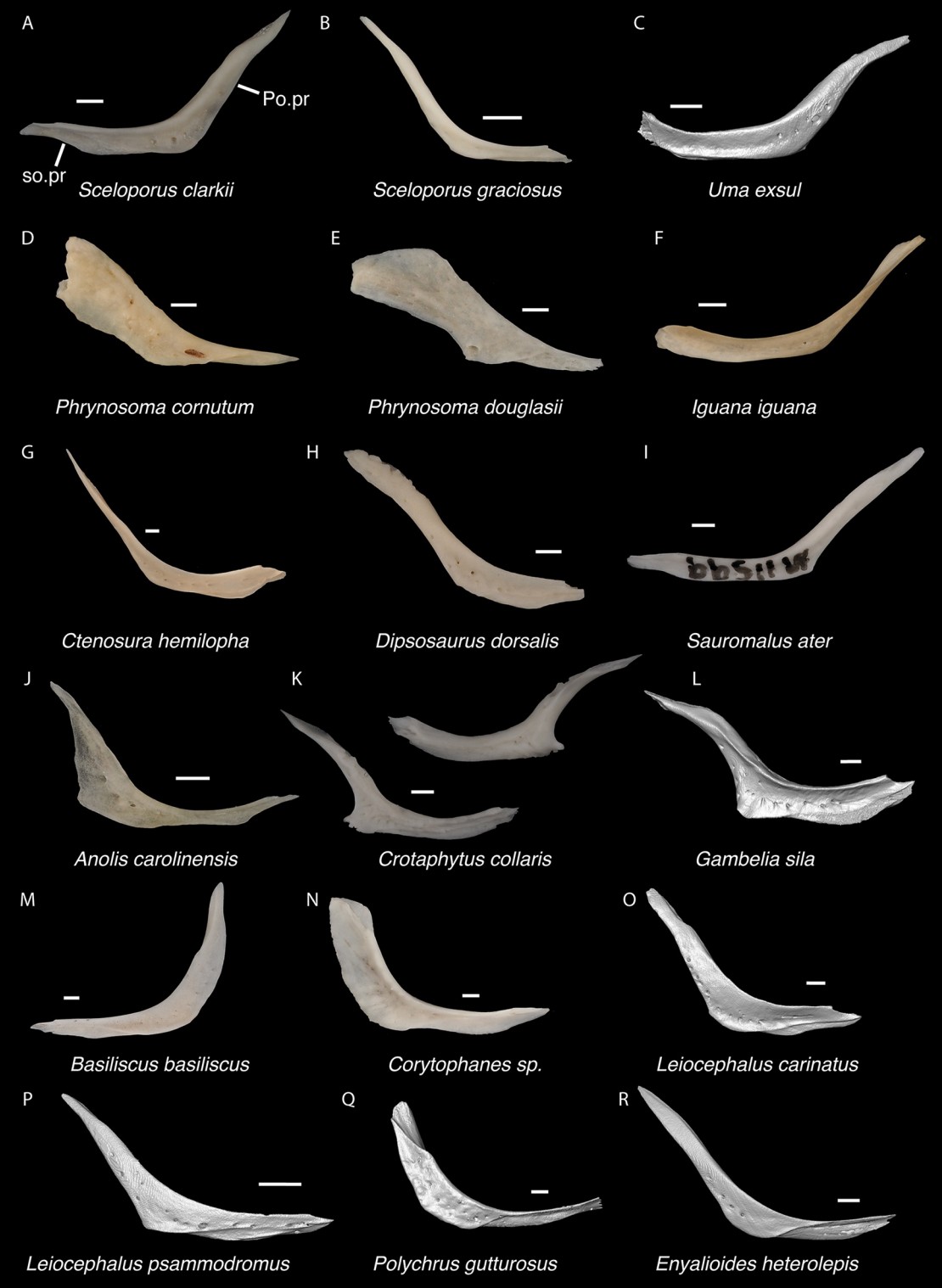

**Fig 12. Pleurodontan jugals.** Jugals in lateral view–**A**. *Sceloporus clarkii* TxVP M-12157 left jugal; **B**. *Sceloporus graciosus* TxVP M-14879 right jugal; **C**. *Uma exsul* TNHC 30247 left jugal; **D**. *Phrynosoma cornutum* TxVP M-6405 right jugal; **E**. *Phrynosoma douglasii* TxVP M-8526 right jugal; **F**. *Iguana iguana* TxVP M-13054 left jugal; **G**. *Ctenosaura hemilopha* TxVP M-8616 right jugal; **H**. *Dipsosaurus dorsalis* TxVP M-13086 right jugal; **I**. *Sauromalus ater* TxVP M-11599 left jugal; **J**. *Anolis carolinensis* TxVP M-9042 right jugal; **K**. *Crotaphytus collaris* TxVP M-12468 right jugal; **L**. *Gambelia sila* TNHC 95261 right jugal; **M**. *Basiliscus basiliscus*

TxVP M-11907 left jugal; **N**. *Corytophanes* sp. TxVP M-16765 right jugal; **O**. *Leiocephalus carinatus* TNHC 89274 right jugal; **P**. *Leiocephalus psammodromus* TNHC 103220 right jugal; **Q**. *Polychrus gutturosus* TNHC 24152 right jugal; **R**. *Enyalioides heterolepis* UF 68015 right jugal. Scale bars = 1 mm. **Abbreviations**: Po.pr, postorbital process, so.pr, suborbital process.

*Identification*. TxVP 41229–28355 shares with Pleurodonta, some teiids, and xenosaurids a posterodorsal process [23] (Fig 14). Some examined *Aspidoscelis* (e.g., *A. tigris* TxVP M-13877) and other teiids (e.g., *Ameiva sp.* TxVP M-8459) also have a distinct dorsal and ventral projection on the posterior end of the squamosal [24, 53]. However, the posteroventral process projects ventrally or posteriorly in examined *Ameiva* [24] and *Aspidoscelis*, and the postero-ventral process is often shorter and less distinct in examined *Aspidoscelis* (Fig 15). The dorsal process of *Xenosaurus* differs in that it is developed into a broad sheet and more anteriorly located along the squamosal [19]. The fossil was assigned to Pleurodonta based on the presence of a posterodorsal process that differs from teiids and xenosaurids in the ways listed above. The fossil and crotaphytids differ from iguanids, *Enyalioides heterolepis*, and phrynosomatids in having a longer and more distinct posterodorsal process of the squamosal [31]. In some crotaphytids (e.g., *C. bicinctores* TxVP M-8612, M-8947) the posterodorsal process is less distinct than in other *Crotaphytus* because it is connected with the main rod by a broad sheet of bone, but no fossils were found with that morphology. Most corytophanids, except *Laemanctus longipes*, have a relatively shorter posterodorsal process [20], and *Laemanctus longipes* differs in having a distinctly downturned main rod of the squamosal (see Fig 66D–F of [20]). *Anolis* and *Leiocephalus* also have a shorter posterodorsal process compared to the fossil and crotaphytids [20, 40]. The posterodorsal process is long among *Polychrus* [20]; however, *Polychrus* differs in having a relatively shorter main rod of the squamosal and may have lateral protuberances [20]. Based on these differences with other NA pleurodontans listed above, the fossil was assigned to Crotaphytidae.

**Quadrate.** *Description*. TxVP 41229–25579 is a left quadrate that is well preserved but is missing the dorsolateral portion of the bone (Fig 5L). The central column is wide, and the bone narrows ventrally. There is no pterygoid lappet, but there is a well-developed and antero-medially directed medial crest. The conch is deep, gradually slants laterally from the central column, and narrows ventrally. The cephalic condyle projects posteriorly and there is no extensive ossification dorsally. The dorsal margin is straight past the lateral extent of the mandibular condyle. There is a foramen medial to the central columnar at mid height.

*Identification*. TxVP 41229–25579 shares with geckos, some scincids, xantusiids, alopoglossids, some pleurodontans, and anguimorphs (besides *Heloderma*) the absence of a distinct pterygoid lappet [23, 24, 37]. The fossil quadrate differs from that of geckos and xantusiids in being relatively wide overall and having a wide central column [24, 46, 54]. NA scincids, except *Scincella*, differ in having a pterygoid lappet [23] and examined *Scincella* differ in having a mediolaterally thin quadrate with a strongly curved tympanic crest. The fossil and pleurodontans differ from anguimorphs and alopoglossids in having a quadrate that is much wider along the dorsal surface compared to the mandibular condyle [37, 55]. The fossil was identified to Pleurodonta based on having a wide central column, a dorsal margin that much wider compared to the mandibular condyle, and lacking a distinct pterygoid lappet.

Compared to the fossil and examined crotaphytids, the quadrate of *Dipsosaurus dorsalis* is much wider dorsally relative to the articular surface (Fig 16L), the conch is deeper, and the pterygoid lamina does not extend as far anteriorly. The quadrate of a small *Iguana iguana* (TxVP M-13054; SVL = 90 mm) is much slenderer and has a reduced pterygoid lamina compared to the fossil and crotaphytids. Larger *Iguana iguana* differ from the fossil and crotaphytids in that the pterygoid lamina is extended medially such that the central column is vertically

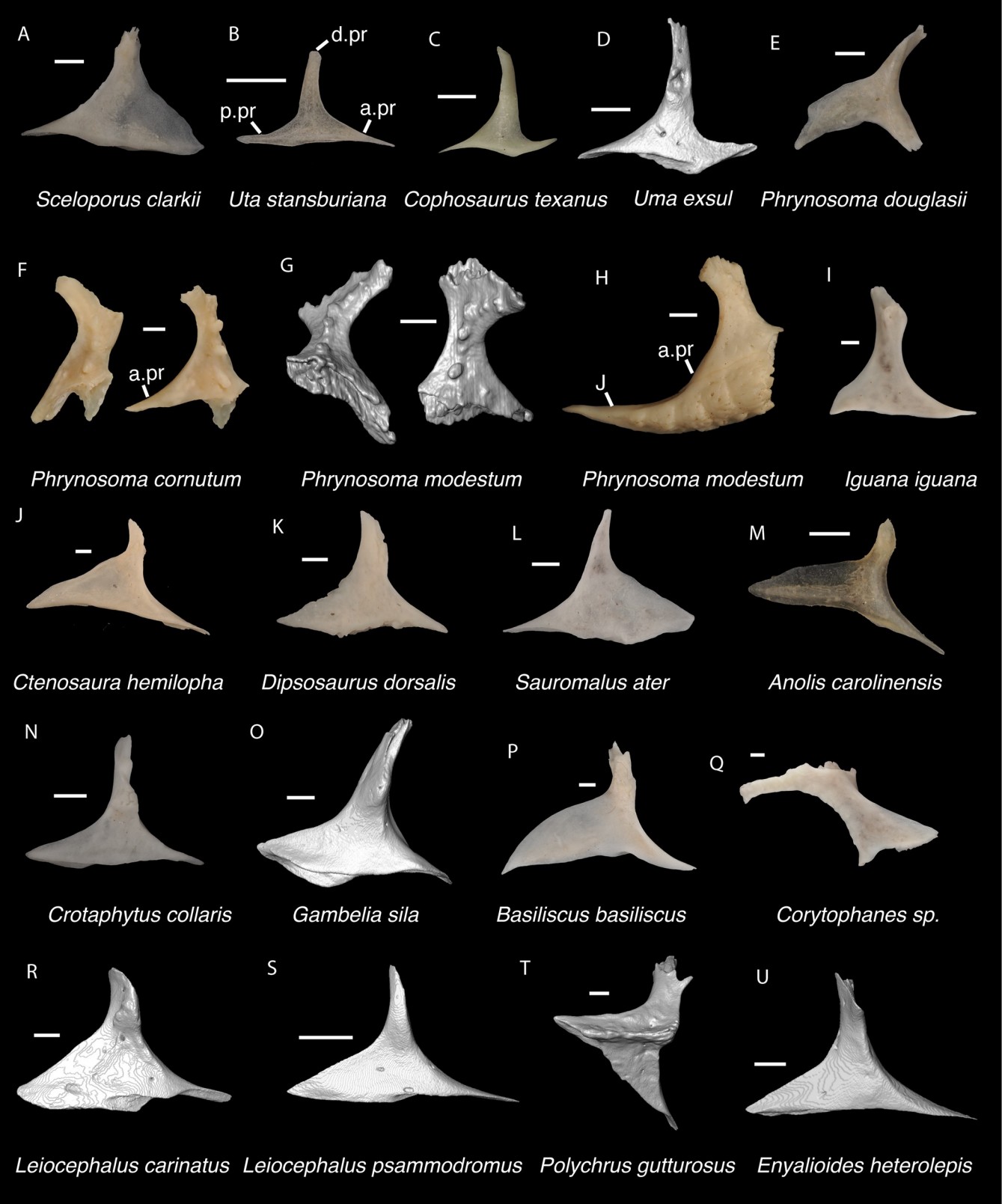

**Fig 13. Pleurodontan postorbitals.** Postorbitals in dorsolateral and dorsal views–**A**. *Sceloporus clarkii* TxVP M-12157 left postorbital; **B**. *Uta stansburiana* TxVP M-14935 right postorbital; **C**. *Cophosaurus texanus* TxVP M-8527 left postorbital; **D**. *Uma exsul* TNHC 30247 left postorbital; **E**. *Phrynosoma douglasii*

TxVP M-8526 right postorbital; **F**. *Phrynosoma cornutum* TxVP M-6405 left postorbital; **G**. *Phrynosoma modestum* TNHC 95921 right postorbital; **H**. *Phrynosoma modestum* TNHC 95921 left postorbital and jugal; **I**. *Iguana iguana* TxVP M-8454 right postorbital; **J**. *Ctenosaura hemilopha* TxVP M-8616 right postorbital; **K**. *Dipsosaurus dorsalis* TxVP M-13086 left postorbital; **L**. *Sauromalus ater* TxVP M-11599 left postorbital; **M**. *Anolis carolinensis* TxVP M-9042 right postorbital; **N**. *Crotaphytus collaris* TxVP M-12468 right postorbital; **O**. *Gambelia sila* TNHC 95261 right postorbital; **P**. *Basiliscus basiliscus* TxVP M-11907 right postorbital; **Q**. *Corytophanes* sp. TxVP M-16765 right postorbital; **R**. *Leiocephalus carinatus* TNHC 89274 right postorbital; **S**. *Leiocephalus psammodromus* TNHC 103220 right postorbital; **T**. *Polychrus gutturosus* TNHC 24152 right postorbital; **U**. *Enyalioides heterolepis* UF 68015 right postorbital. Scale bars = 1 mm. **Abbreviations**: a.pr, anterior process; d.pr, dorsal process; J, jugal; p.pr, posterior process.

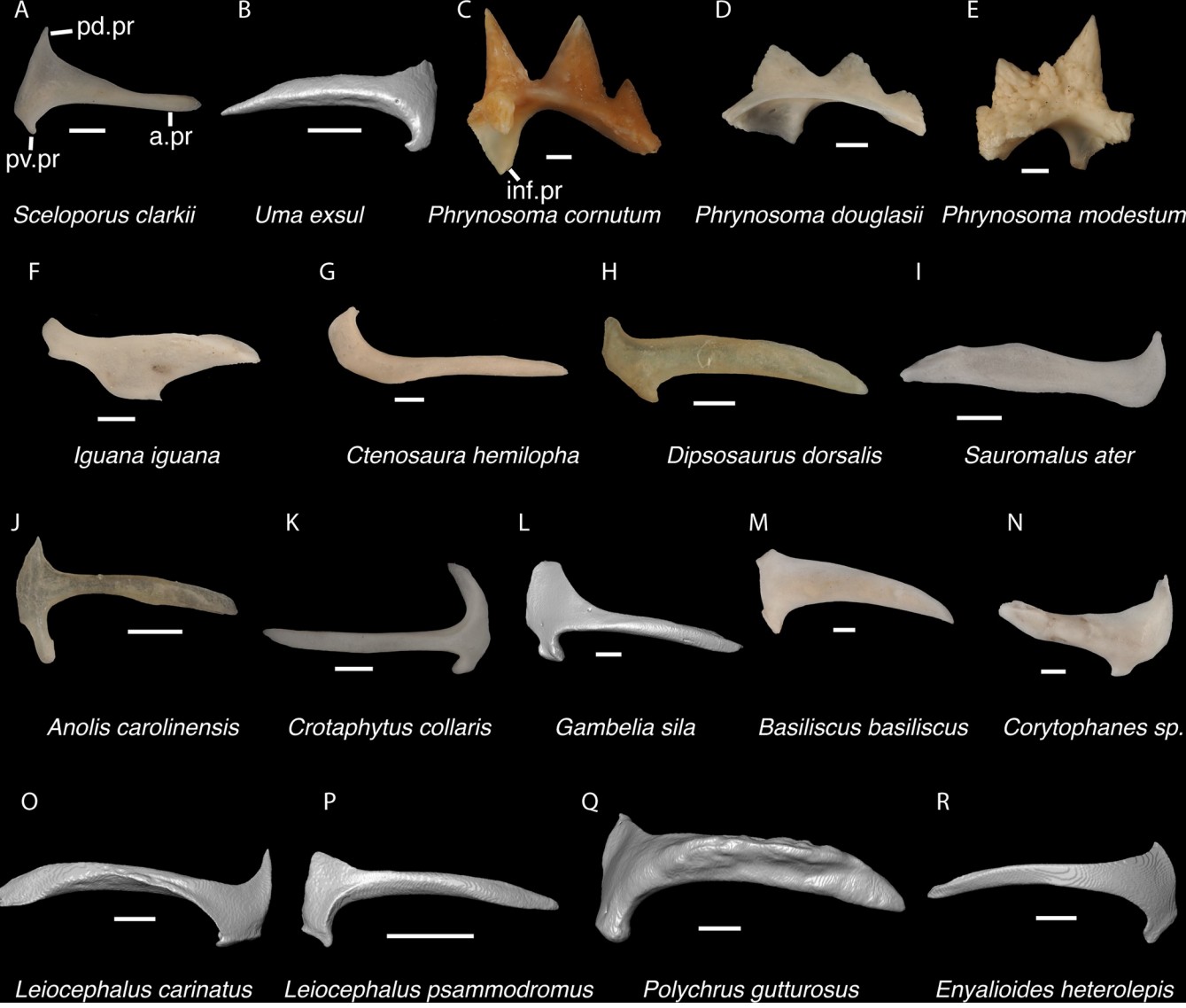

**Fig 14. Pleurodontan squamosals.** Squamosals in lateral and posterior views–**A**. *Sceloporus clarkii* TxVP M-12157 right squamosal in lateral view; **B**. *Uma exsul* TNHC 30247 left squamosal in lateral view; **C**. *Phrynosoma cornutum* TxVP M-6405 right squamosal in anterior view; **D**. *Phrynosoma douglasii* TxVP M-8526 right squamosal in anterior view; **E**. *Phrynosoma modestum* TNHC 95921 left squamosal in anterior view; **F**. *Iguana iguana* TxVP M-8454 right squamosal in lateral view; **G**. *Ctenosaura hemilopha* TxVP M-8616 right squamosal in lateral view; **H**. *Dipsosaurus dorsalis* TxVP M-9285 right squamosal in lateral view; **I**. *Sauromalus ater* TxVP M-11599 left squamosal in lateral view; **J**. *Anolis carolinensis* TxVP M-9042 right squamosal in lateral view; **K**. *Crotaphytus collaris* TxVP M-12468 left squamosal in lateral view; **L**. *Gambelia sila* TNHC 95261 right squamosal in lateral view; **M**. *Basiliscus basiliscus* TxVP M-11907 right squamosal in lateral view; **N**. *Corytophanes* sp. TxVP M-16765 left squamosal in lateral view; **O**. *Leiocephalus carinatus* TNHC 89274 left squamosal in lateral view; **P**. *Leiocephalus psammodromus* TNHC 103220 right squamosal in lateral view; **Q**. *Polychrus gutturosus* TNHC 24152 right squamosal in lateral view; **R**. *Enyalioides heterolepis* UF 68015 left squamosal in lateral view. Scale bars = 1 mm. **Abbreviations**: a.pr, anterior process; inf.pr, inferior process; pd.pr, posterodorsal process; pv.pr, posteroventral process.

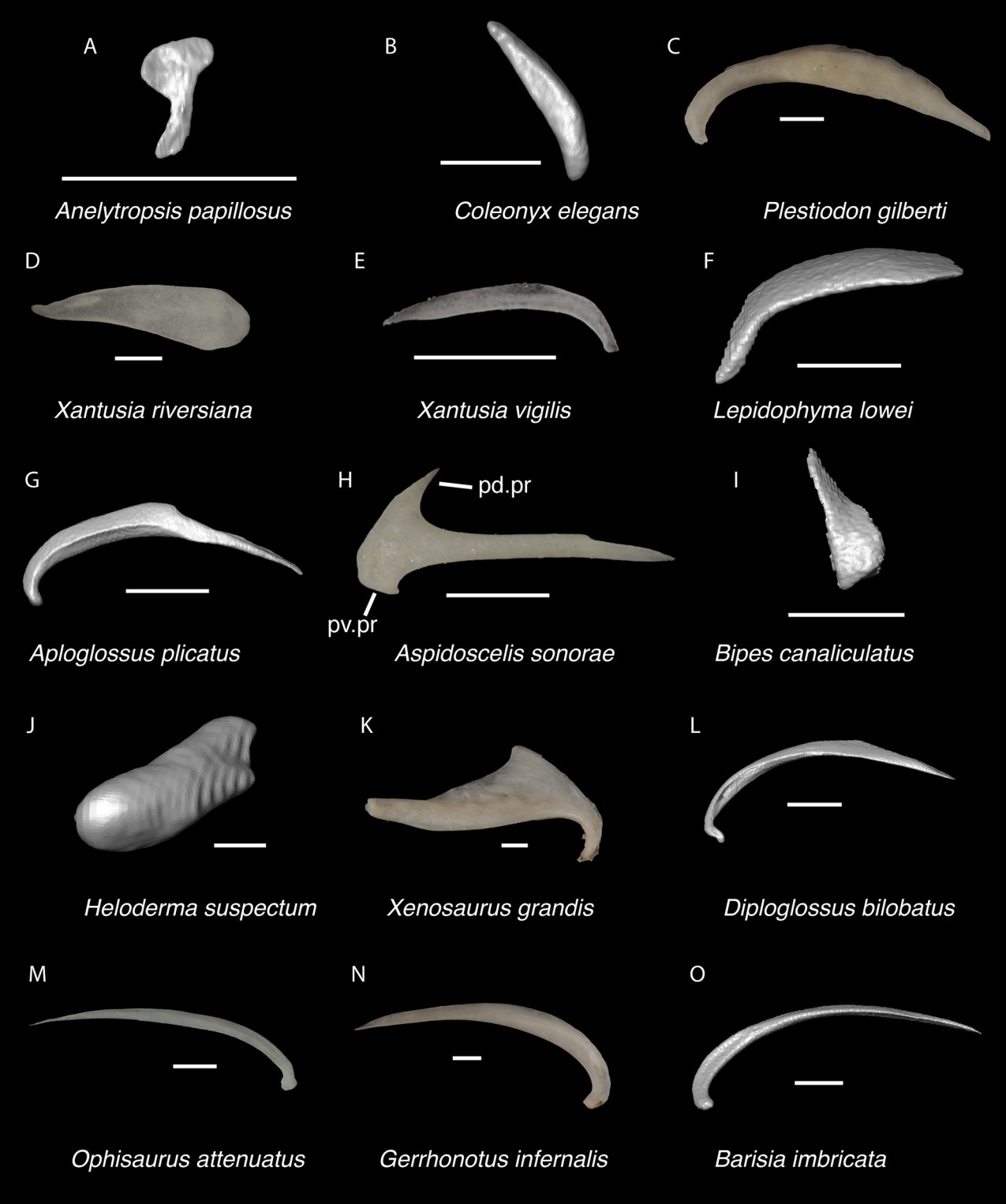

**Fig 15. Non-pleurodontan squamosals.** Squamosals in lateral view–**A**. *Anelytropsis papillosus* UF 86708 left squamosal; **B**. *Coleonyx elegans* UF 11258 left squamosal; **C**. *Plestiodon gilberti* TxVP M-8587 right squamosal; **D**. *Xantusia riversiana* TxVP M-8505 left squamosal; **E**. *Xantusia vigilis* TxVP M-12130 left squamosal; **F**. *Lepidophyma lowei* LACM 143367 right squamosal; **G**. *Aploglossus plicatus* TNHC 34481 right squamosal; **H**. *Aspidoscelis sonorae* TxVP M-15670 right squamosal; **I**. *Bipes canaliculatus* CAS 134753 right squamosal; **J**. *Heloderma suspectum* TNHC 62766 right squamosal; **K**. *Xenosaurus grandis* TxVP M-8960 left squamosal; **L**. *Diploglossus bilobatus* TNHC 31933 right squamosal; **M**. *Ophisaurus attenuatus* TxVP M-8979 left

squamosal; **N**. *Gerrhonotus infernalis* TxVP M-13441 left squamosal; **O**. *Barisia imbricata* TNHC 76984 right squamosal. Scale bars = 1 mm.
**Abbreviations**: pd.pr, posterodorsal process; pv.pr, posteroventral process.

oriented and near the midline of the bone (*Iguana iguana* OUVC 10612). The quadrate of most examined *Sauromalus* (except *Sauromalus ater* TNHC 18483) differ in having a low dorsolateral portion, such that the dorsal margin of the bone is sloped ventrolaterally. Additionally, the quadrate of *Sauromalus* has a dorsomedial expansion that contacts the paroccipital process [24] and which is especially prominent in large specimens. *Ctenosaura* differs in having a more distinct and curved tympanic crest and a deeper conch. In *Anolis* there is a distinct boss at the ventromedial margin of the bone below the articulation with the pterygoid. In *Anolis* and *Polychrus*, the lateral and medial margins are more parallel compared to the fossil and crotaphytids. *Leiocephalus barahonensis* (USNM 260564) and *L. carinatus* differ from crotaphytids in having a medially expanded pterygoid lamina that extends medially beyond the dorsomedial corner of the quadrate (see also Fig 6 of [33]). *Leiocephalus psammodromus* differs from crotaphytids in having a quadrate that is not as distinctly widened dorsally and having a distinct medial notch where the pterygoid articulates. The quadrate conch of *Enyalioides heterolepis* is shallower compared to the fossil and crotaphytids. The quadrate of *Basiliscus basiliscus* is proportionally taller and the central column slenderer. The quadrate is exceptionally slender in examined *Corytophanes* and *Laemanctus* [56]. The quadrate of phrynosomatids besides *Phrynosoma* differs from crotaphytids in having a more curved lateral margin (tympanic crest) [57]. Additionally, the central column is slenderer near the base and the conch is deeper in examined large-bodied *Sceloporus* (e.g., *S. clarkii*, Fig 16A). The quadrate of *Phrynosoma* differs from the fossil and crotaphytids in having a reduced pterygoid lamina, a dorsal portion of the bone that is much wider than the ventral portion, and a conch that remains deep lateral to the central column. Based on these differences with other NA pleurodontans listed above, quadrates were assigned to Crotaphytidae.

**Pterygoid.**  *Description*. TxVP 41229–28513 is a pterygoid that is missing only the distal end of the palatine process (Fig 17B). The palatine process is thin near its base and the transverse process extends laterally at nearly a 90-degree angle with the quadrate process. The transverse process is dorsoventrally tall, and the distal end is oriented near vertical. There is a distinct ridge on the dorsal surface for insertion of the superficial pseudotemporal muscle and a distinct ectopterygoid facet just anterior to it. There is a large ridge on the ventral surface for insertion of the pterygomandibular muscle and a small, deep fossa columella that is confluent posteriorly with a short, narrow groove. The quadrate process is elongated, and the medial surface has an elongate groove that serves for insertion of the pterygoideus muscle. There is a ventromedial projection at the floor of the basipterygoid fossa. There is a long patch of 17 tooth positions filled with 16 pterygoid teeth. There are four small foramina on the dorsal surface in the area from the palatine process to just anterior to the fossa columella. There are two foramina just anterior to the ridge for insertion of the pterygomandibular muscle. TxVP 41229–27144 is missing portions of the palatine and quadrate processes (Fig 17A) and does not differ substantively from TxVP 41229–28513.

Identification: Pterygoids are assigned to Pleurodonta based on the presence of a well-developed ventromedial projection at the floor of the basipterygoid fossa [24, 52] (Fig 18). A ventromedial projection at the floor of the basipterygoid fossa is also in some non-pleurodontans (e.g., *Ophisaurus*), but it is less distinct in those taxa compared to the fossil and in many pleurodontans (e.g., *Crotaphytus*). Among pleurodontans, pterygoid teeth are absent in phrynosomatids and are variably present in *Anolis*, *Polychrus*, *Dipsosaurus*, and *Leiocephalus* [42]. The palatal plate in *Anolis* and phrynosomatids is broad relative to the fossils. Compared with

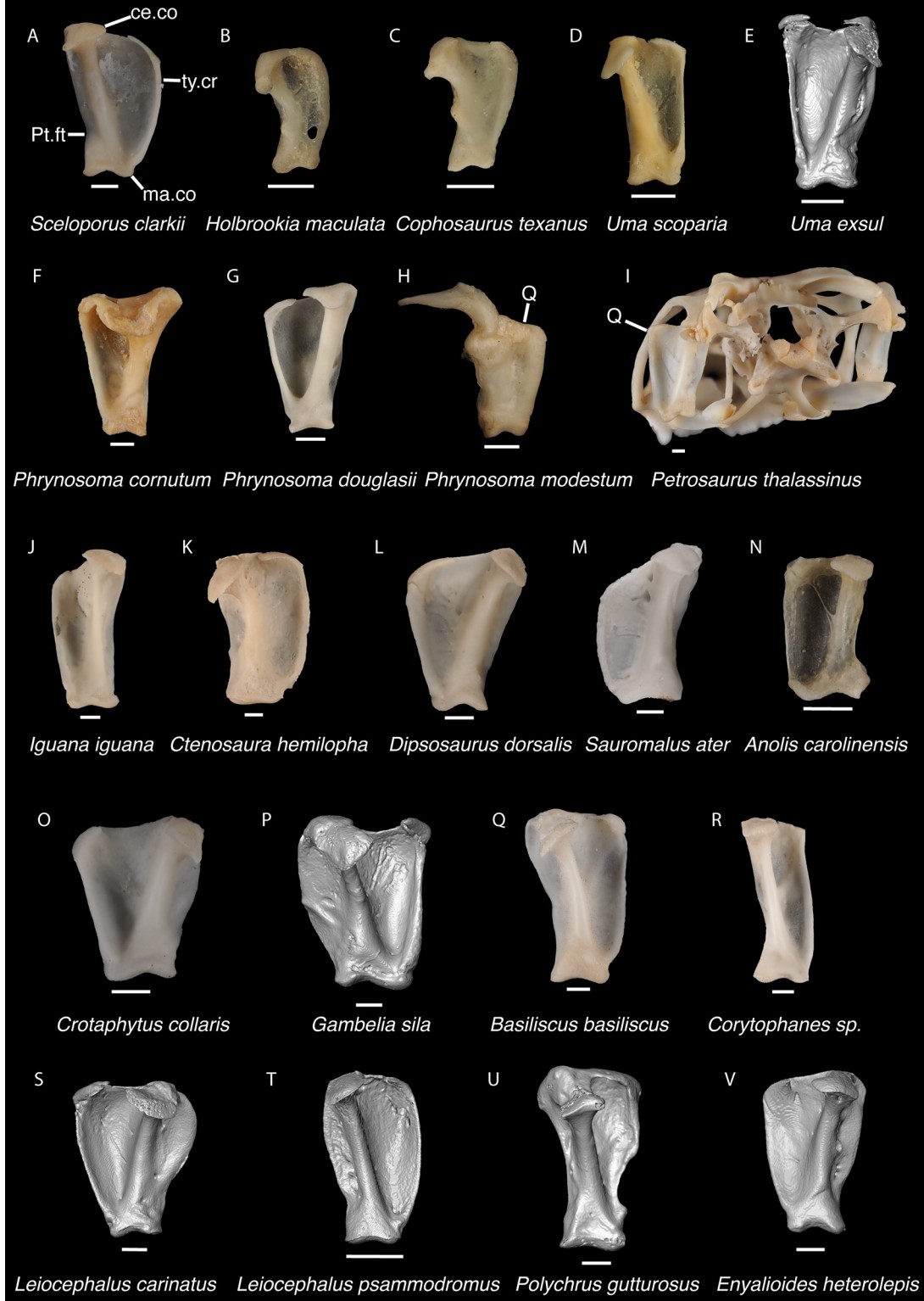

**Fig 16. Pleurodontan quadrates.** Quadrates in posterior view–**A**. *Sceloporus clarkii* TxVP M-12157 right quadrate; **B**. *Holbrookia maculata* TxVP M-14322 right quadrate; **C**. *Cophosaurus texanus* TxVP M-8527 right quadrate; **D**. *Uma scoparia* TxVP M-8529 right quadrate; **E**. *Uma exsul* TNHC 30247 left quadrate; **F**. *Phrynosoma cornutum* TxVP M-6405 left quadrate; **G**. *Phrynosoma douglasii* TxVP M-8526 left quadrate; **H**. *Phrynosoma modestum* TNHC 95921 right quadrate and squamosal; **I**. *Petrosaurus thalassinus* TxVP M-9612 skull in posterolateral view; **J**. *Iguana iguana* TxVP M-8454 left quadrate; **K**. *Ctenosaura*

*hemilopha* TxVP M-8616 right quadrate; **L**. *Dipsosaurus dorsalis* TxVP M-13086 left quadrate; **M**. *Sauromalus ater* TxVP M-11599 left quadrate; **N**. *Anolis carolinensis* TxVP M-9042 left quadrate; **O**. *Crotaphytus collaris* TxVP M-12468 left quadrate; **P**. *Gambelia sila* TNHC 95261 right quadrate; **Q**. *Basiliscus basiliscus* TxVP M-11907 right quadrate; **R**. *Corytophanes* sp. TxVP M-16765 right quadrate; **S**. *Leiocephalus carinatus* TNHC 89274 left quadrate; **T**. *Leiocephalus psammodromus* TNHC 103220 right quadrate; **U**. *Polychrus gutturosus* TNHC 24152 right quadrate; **V**. *Enyalioides heterolepis* UF 68015 left quadrate. Scale bars = 1 mm. **Abbreviations**: ce.co, cephalic condyle; ma.co, mandibular condyle; Pt.ft, pterygoid facet; Q, quadrate; ty.cr, tympanic crest.

the fossils, *Sauromalus* has a shorter transverse process and a shorter quadrate process. The quadrate process in *Iguana iguana* is more distinctly short and the fossa columella elongated. Pterygoid teeth of examined *Ctenosaura* are oriented more obliquely [58], and the palatal plate is stepped along the anterior edge [59]. An examined *Ctenosaura clarki* (TxVP M-14824) has a unique ridge on the ventral surface of the pterygoid along which the teeth are positioned. *Dipsosaurus dorsalis* has a narrow linear notch on the transverse process not seen on the fossils. The transverse process is shorter and less robust in *Enyalioides heterolepis*. *Basiliscus* and *Corytophanes* have a taller quadrate process and tend to have longer pterygoid teeth which may be slightly recurved in *Basiliscus* [56]. *Basiliscus* and *Laemanctus* generally have fewer pterygoid teeth, ranging from three to five in number [56]. The shelf under the basipterygoid fossa is less extensive in *Basiliscus basiliscus*. The quadrate process is anteroposterioly shorter, and the transverse process is exceptionally tall in *Corytophanes*. Among North American pleurodontans, the largest number of pterygoid teeth is found in crotaphytids and *Ctenosaura*, which can have a similar number of tooth positions to the fossils [60]. However, there is a large amount of variation in the number of pterygoid teeth among crotaphytids [60]. The transverse process of the fossils and of *Crotaphytus* is more dorsoventrally elongated and less pointed in dorsal view compared to examined *Leiocephalus*. Fossil pterygoids are assigned to Crotaphytidae based on the differences from other NA pleurodontans described above.

**Ectopterygoid.** *Description*. TxVP 41229–25592 is a right ectopterygoid (Fig 17C). It has distinct anterolateral and posterolateral processes. The ventral corner of the posterolateral process is narrow and extends far ventrally. There is a groove for articulation with the maxilla on the anterolateral edge of the bone below the dorsal corner. The dorsal surface is slightly concave at the lateral end and the pterygoid process extends far ventrally. The pterygoid facet is deep with a distinct overhanging corner of bone dorsally. The main shaft of the ectopterygoid is orthogonal to the lateral edge of the bone and there are two small foramina within the jugal facet.

*Identification*. TxVP 41229–25592 shares with iguanians, xantusiids, and xenosaurids the presence of an elongate posterolateral process on the ectopterygoid [52]. The ectopterygoid (os transversum of [61]) of *Xantusia* can be distinguished from that of pleurodontans in that it is a broad triangular plate [61] (Fig 19). The ectopterygoid of xenosaurids differs from pleurodontans in having the main shaft of the bone oriented more anteroposteriorly in the skull and in having a much shorter posterolateral process. Among NA pleurodontans, *Anolis* differs from the fossil and Crotaphytidae in lacking a flange on the lateral head that articulates ventrally or ventromedially with the maxilla, and in having a main shaft that is oblique relative to the lateral face of the bone [52] (Fig 20). The main shaft of the ectopterygoid is slightly oblique relative to the lateral face of the bone in some corytophanids, including *Corytophanes hernandesii* and *Laemanctus longipes* [20]. In phrynosomatids, *Leiocephalus*, and *Enyalioides heterolepis*, the main shaft of the ectopterygoid is also slightly oblique relative to the lateral face of the bone, while in the fossil and *Crotaphytus*, the main shaft is orthogonal to the lateral face of the bone. Additionally, the shaft of the bone is much more slender in non-*Phrynosoma* phrynosomatids compared to the fossil and *Crotaphytus*. The ventral corner of the posterolateral process of the

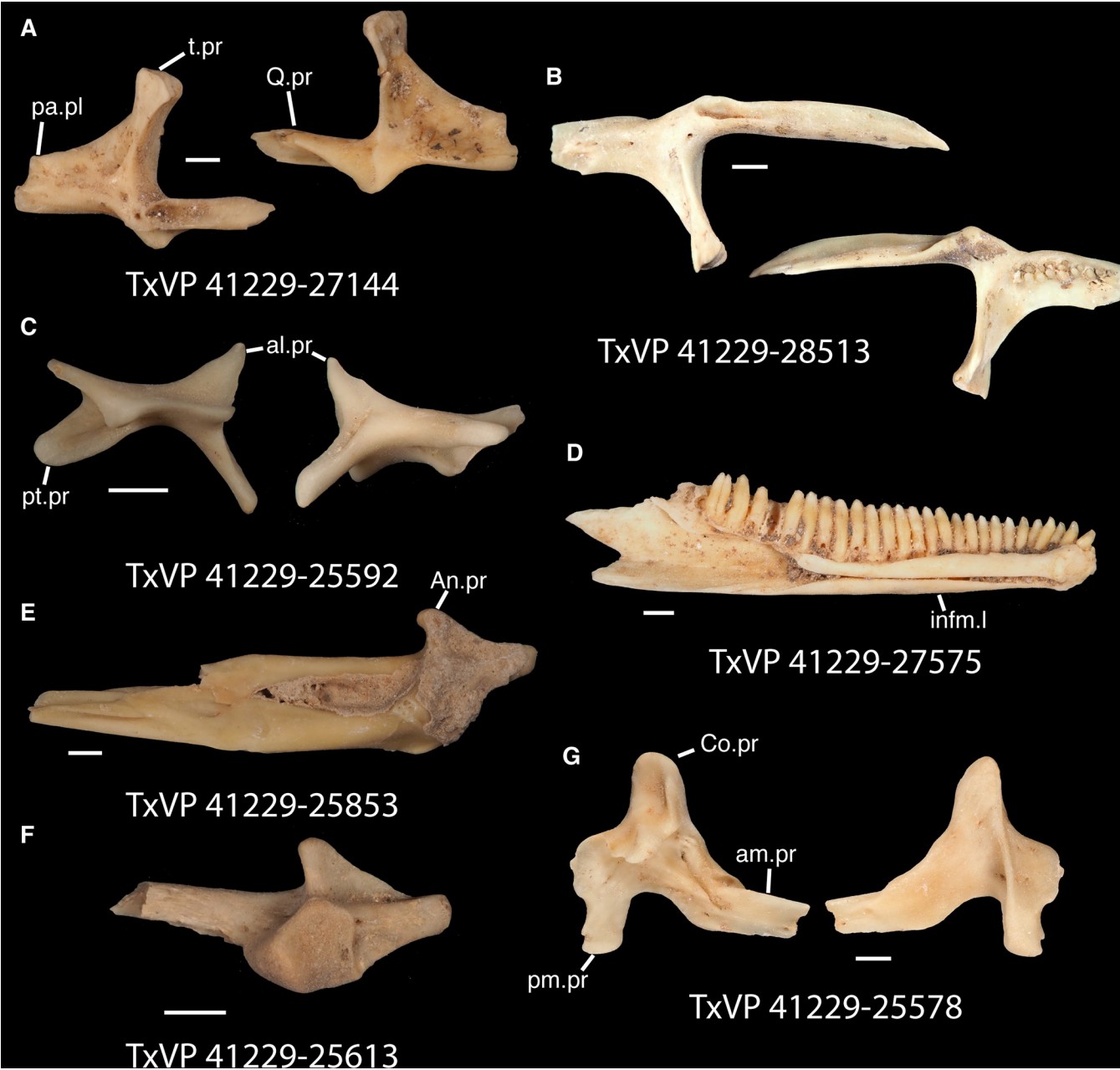

**Fig 17. Fossil crotaphytids. A**. TxVP 41229–27144 Dorsal and ventral view of right pterygoid; **B**. TxVP 41229–28513 Dorsal and ventral view of left pterygoid; **C**. TxVP 41229–25592 Posterior and ventral view of right ectopterygoid; **D**. TxVP 41229–27575 Medial view of left dentary; **E**. TxVP 41229–25853 Dorsal view of left compound bone; **F**. TxVP 41229–25613 Dorsal view of left dermarticular; **G**. TxVP 41229–25578 Lateral and medial view of right coronoid. Scale bars = 1 mm. **Abbreviations**: al.pr, anterolateral process; am.pr, anteromedial process; An.pr, angular process; Co.pr, coronoid process; imfm.l, inframeckelian lip; pa.pl, palatal plate; pm.pr, posteromedial process; pt.pr, pterygoid process; Q.pr, quadrate process; t.pr, transverse process.

ectopterygoid in the fossil and *Crotaphytus* is developed into a distinct wedge that projects ventrally, similar to that seen in corytophanids [20]. The ventral corner of the posterolateral process is well-developed in crotaphytids, corytophanids, and *Dipsosaurus dorsalis*, semi-developed in *Ctenosaura hemilopha*, and undeveloped in *Sauromalus ater* and *Ctenosaura similis* [20]. Although the ventral corner of the posterolateral process is well-developed in

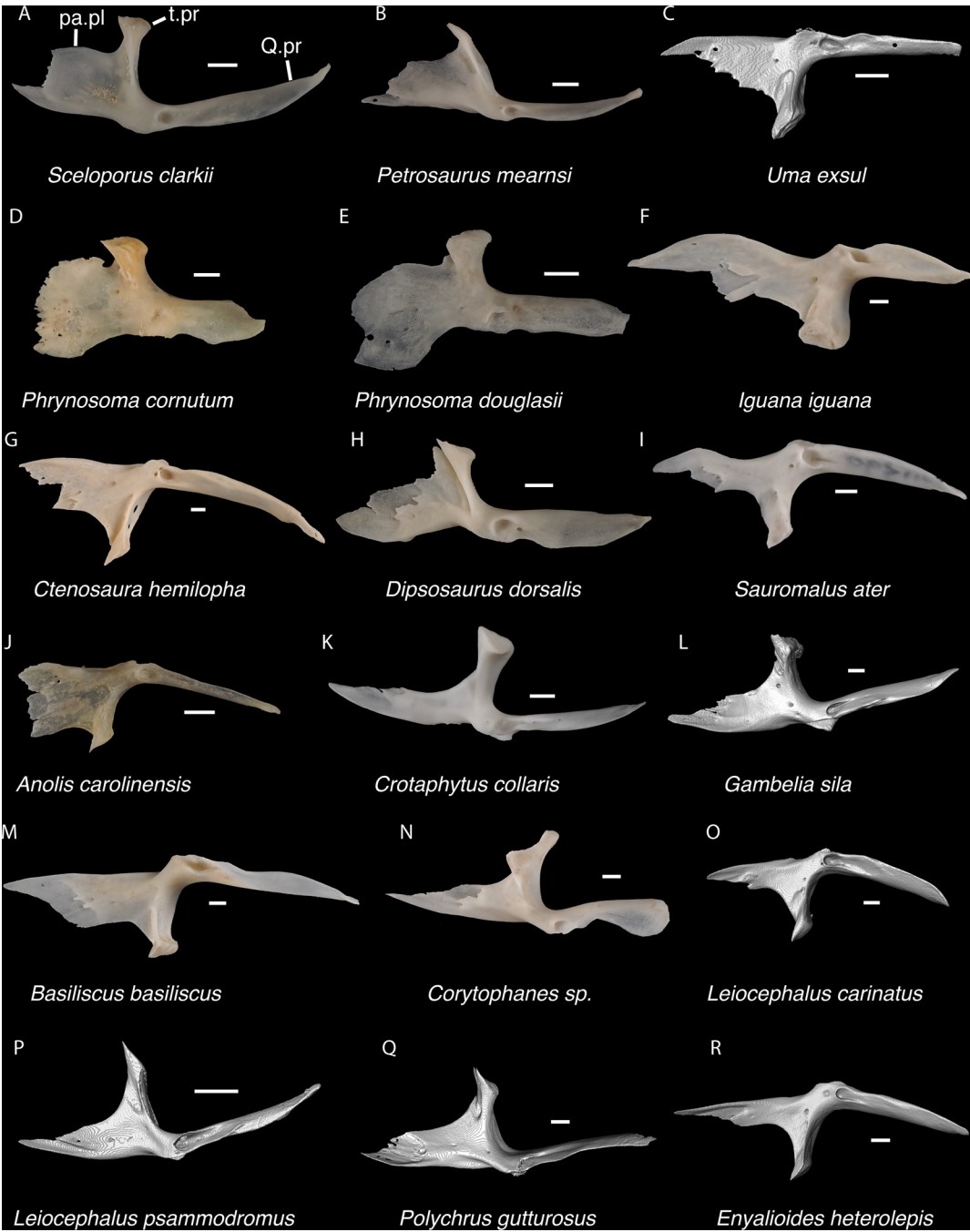

**Fig 18. Pleurodontan pterygoids.** Pterygoids in dorsal view–**A**. *Sceloporus clarkii* TxVP M-12157 right pterygoid; **B**. *Petrosaurus mearnsi* TxVP M-14910 right pterygoid; **C**. *Uma exsul* TNHC 30247 left pterygoid; **D**. *Phrynosoma cornutum* TxVP M-6405 right pterygoid; **E**. *Phrynosoma douglasii* TxVP M-8526 right pterygoid; **F**. *Iguana iguana* TxVP M-8454 left pterygoid; **G**. *Ctenosaura hemilopha* TxVP M-8616 left pterygoid; **H**. *Dipsosaurus dorsalis* TxVP M-13086 right pterygoid; **I**. *Sauromalus ater* TxVP M-11599 left pterygoid; **J**. *Anolis carolinensis* TxVP M-9042 left pterygoid; **K**. *Crotaphytus collaris* TxVP M-12468 right pterygoid; **L**. *Gambelia sila* TNHC 95261 right pterygoid; **M**. *Basiliscus basiliscus* TxVP M-11907 left pterygoid; **N**. *Corytophanes* sp. TxVP M-16765 right pterygoid; **O**. *Leiocephalus carinatus* TNHC 89274 left pterygoid; **P**. *Leiocephalus psammodromus* TNHC 103220 right pterygoid; **Q**. *Polychrus gutturosus* TNHC 24152 right pterygoid; **R**. *Enyalioides heterolepis* UF 68015 left pterygoid. Scale bars = 1 mm. **Abbreviations**: pa.pl, palatal plate; Q.pr, quadrate process; t.pr, transverse process.

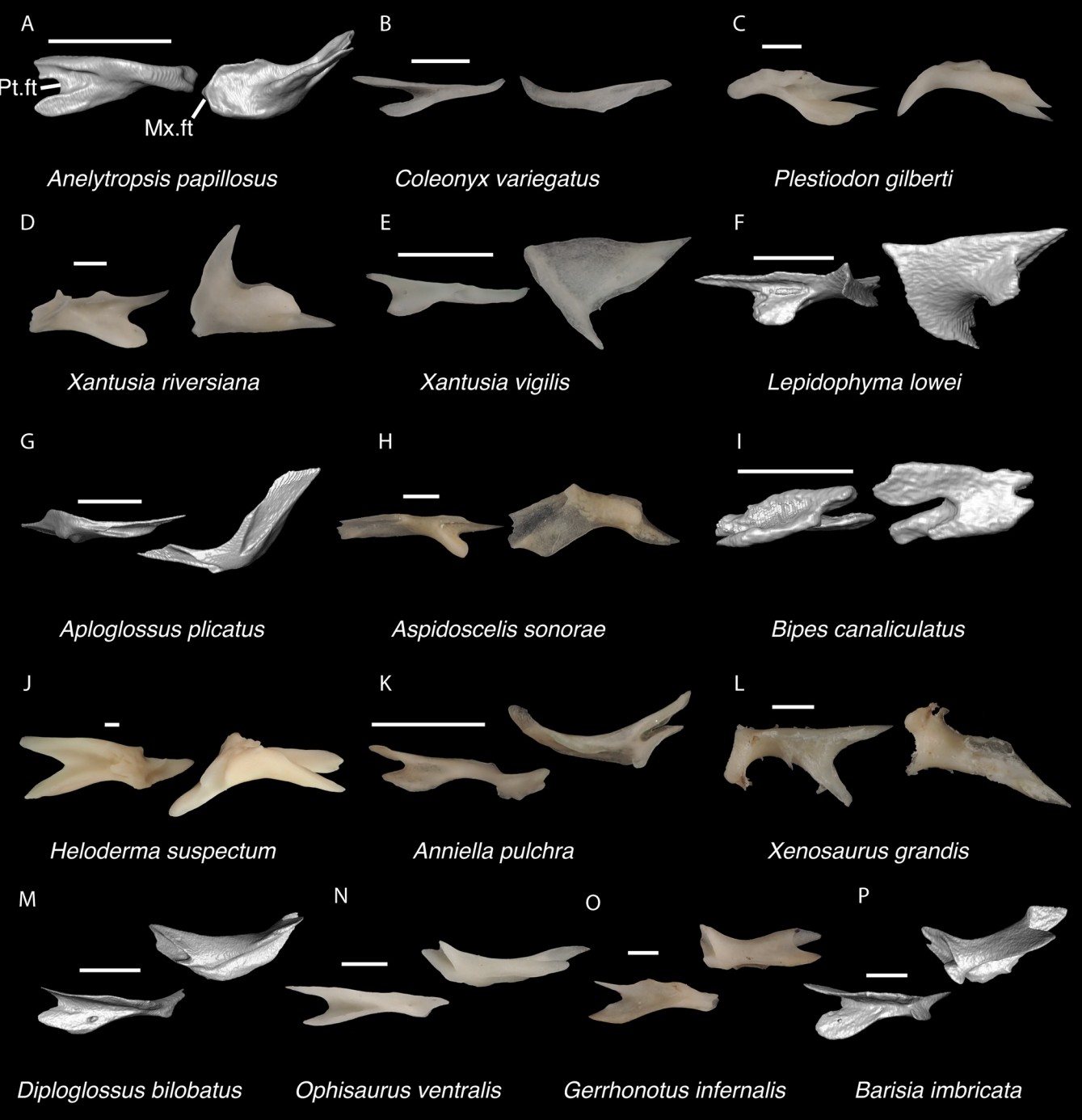

**Fig 19. Non-pleurodontan ectopterygoids.** Ectopterygoids in lateral and ventral views–**A**. *Anelytropsis papillosus* UF 86708 right ectopterygoid; **B**. *Coleonyx variegatus* TxVP M-12109 right ectopterygoid; **C**. *Plestiodon gilberti* TxVP M-8587 left ectopterygoid; **D**. *Xantusia riversiana* TxVP M-8505 left ectopterygoid; **E**. *Xantusia vigilis* TxVP M-12130 right ectopterygoid; **F**. *Lepidophyma lowei* LACM 143367 right ectopterygoid; **G**. *Aploglossus plicatus* TNHC 34481 left ectopterygoid; **H**. *Aspidoscelis sonorae* TxVP M-15670 left ectopterygoid; **I**. *Bipes canaliculatus* CAS 134753 right ectopterygoid; **J**. *Heloderma suspectum* TxVP M-9001 right ectopterygoid; **K**. *Anniella pulchra* TxVP M-8678 right ectopterygoid; **L**. *Xenosaurus grandis* TxVP M-8960 left ectopterygoid; **M**. *Diploglossus bilobatus* TNHC 31933 right ectopterygoid; **N**. *Ophisaurus ventralis* TxVP M-8585 right ectopterygoid; **O**. *Gerrhonotus infernalis* TxVP M-13441 right ectopterygoid; **P**. *Barisia imbricata* TNHC 76984 right ectopterygoid. Scale bars = 1 mm. **Abbreviations**: Mx.ft, maxilla facet; Pt.ft, pterygoid facet.

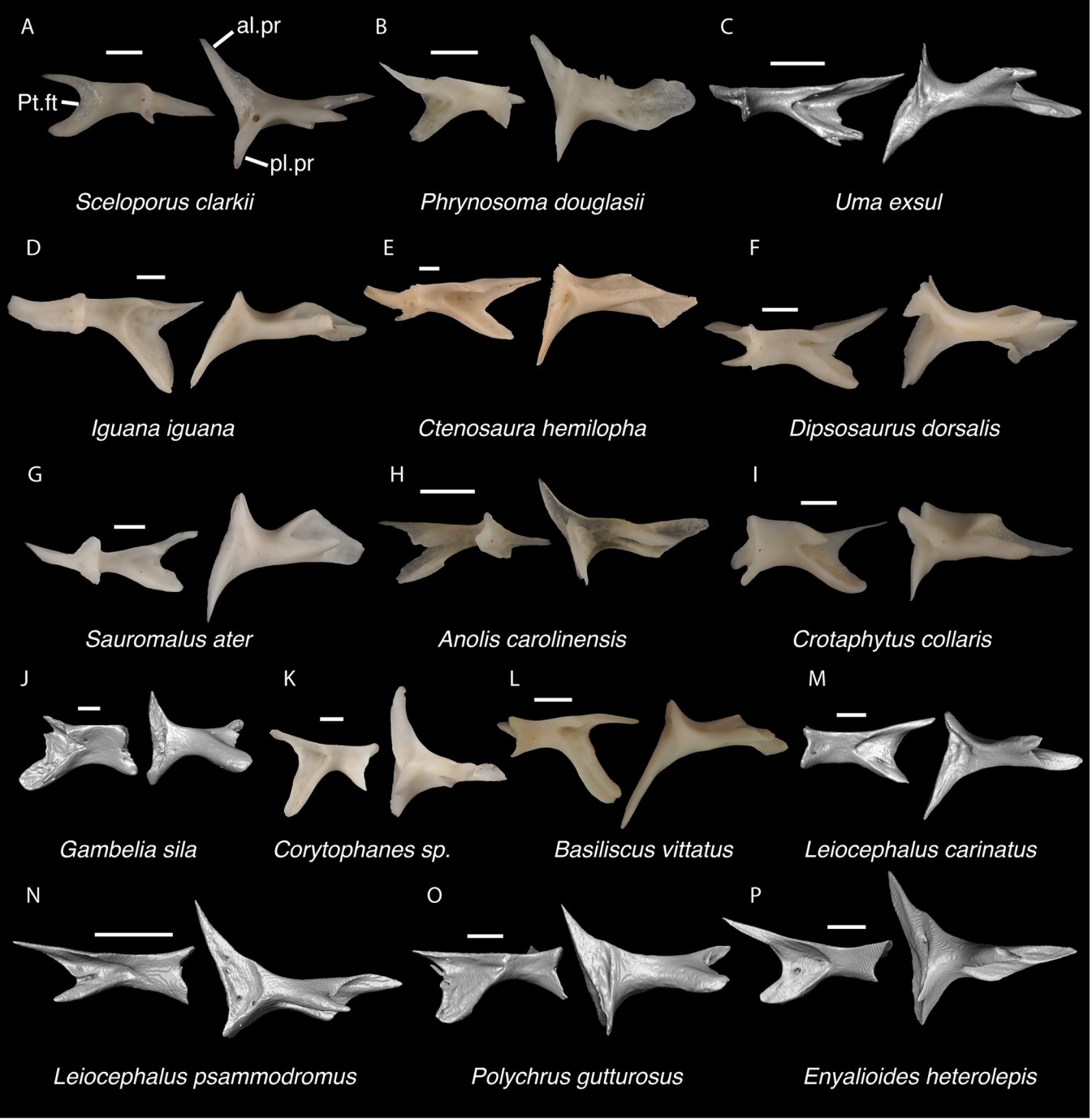

**Fig 20. Pleurodontan ectopterygoids.** Ectopterygoids in posterior and ventral views–**A**. *Sceloporus clarkii* TxVP M-12157 right ectopterygoid; **B**. *Phrynosoma douglasii* TxVP M-8526 right ectopterygoid; **C**. *Uma exsul* TNHC 30247 left ectopterygoid; **D**. *Iguana iguana* TxVP M-8454 left ectopterygoid; **E**. *Ctenosaura hemilopha* TxVP M-8616 left ectopterygoid; **F**. *Dipsosaurus dorsalis* TxVP M-13086 left ectopterygoid; **G**. *Sauromalus ater* TxVP M-11599 left ectopterygoid; **H**. *Anolis carolinensis* TxVP M-9042 right ectopterygoid; **I**. *Crotaphytus collaris* TxVP M-12468 left ectopterygoid; **J**. *Gambelia sila* TNHC 95261 right ectopterygoid; **K**. *Corytophanes* sp. TxVP M-16765 right ectopterygoid; **L**. *Basiliscus vittatus* TxVP M-8556 left ectopterygoid; **M**. *Leiocephalus carinatus* TNHC 89274 left ectopterygoid; **N**. *Leiocephalus psammodromus* TNHC 103220 right ectopterygoid; **O**. *Polychrus gutturosus* TNHC 24152 right ectopterygoid; **P**. *Enyalioides heterolepis* UF 68015 right ectopterygoid. Scale bars = 1 mm. **Abbreviations**: al.pr, anterolateral process; pl.pr, posterolateral process; Pt.ft, pterygoid facet.

*Dipsosaurus dorsalis*, it does not project as far ventrally as in corytophanids and *Crotaphytus collaris*. In *Crotaphytus collaris* there is a corner of bone dorsal to the posterior pterygoid facet that is downturned. This downturned corner is also present in *Laemanctus longipes*, some *Anolis*, and *Enyalioides laticeps*, although it is absent in other hoplocercids [20]. A downturned corner is absent in *Polychrus* and *Gambelia wislizenii* [20]. The fossil ectopterygoid can be referred to Crotaphytidae based on possessing a main shaft that is nearly orthogonal to the lateral face of the bone, a wide shaft of the ectopterygoid, a ventral corner of the posterolateral process of the ectopterygoid developed into a distinct wedge that projects ventrally, and a downturned corner of bone dorsal to the posterior pterygoid facet. A downturned corner of bone dorsal to the posterior pterygoid facet is absent in *Gambelia wislizenii* [20]. A slightly downturned corner is present in *Gambelia sila* (TxVP M-95261), although not to the extent seen in *Crotaphytus*. More information on variation in this feature among species of *Gambelia* is needed before a confident generic assignment can be made.

**Dentary.** *Description.* TxVP 41229–27575 serves as the basis for our description (Fig 17D). TxVP 41229–27575 is a left dentary with 27 tooth positions. The posterior end is bifurcated and the Meckelian canal is open, with tall suprameckelian and inframeckelian lips. There is a distinct intramandibular lamella and the intramandibular septum is restricted to the anterior portion of the dentary. There is a narrow dental shelf that becomes wider closer to the symphysis. The teeth are slightly eroded at the crowns, but some distal teeth display weakly tricuspid morphology. The tooth bases are much wider than the crown for the more distal teeth. There are six nutrient foramina arranged in a row on the lateral surface near the anterior end of the dentary.

*Identification.* Dentaries were placed in Pleurodonta based on the presence of pleurodont tricuspid teeth and a closed or partially closed Meckelian groove bounded by the suprameckelian and inframeckelian lips [35] (Fig 21). Other non-pleurodontan lizards with tricuspid dentary teeth include *Xantusia riversiana* [46], some teiids [53], and some gymnophthalmids [62]. *Xantusia* differs from pleurodontans in having a fused spleniodentary [63]. Teiids differ in having a tall and open Meckelian groove and substantial cementum deposits at the tooth bases [23, 64]. In gymnophthalmids with tricuspid teeth, the Meckelian groove is completely fused to almost the level of the posterior-most tooth position [62]. Dentaries were identified to Crotaphytidae based on mesiodistally expanded tooth bases, a narrow subdental shelf, and an anteriorly tall suprameckelian lip [35]. Among pleurodontans, the Meckelian groove is fused in iguanids, anolids, extant *Leiocephalus*, tropidurids, polychrotids, leiosaurids, and some liolaemids and oplurids [20, 35]. Corytophanids differ from crotaphytids in lacking mesiodistally expanded tooth bases, and in *Basiliscus*, *Corytophanes*, some *Laemanctus*, and *Enyalioides* the tooth crowns are flared. Examined hoplocercids besides *Enyalioides laticeps* (FMNH 206132) have an open Meckelian groove that is not bounded by suprameckelian and inframeckelian lips [35, 48]. Phrynosomatids have relatively more slender teeth that do not widen at the base [65], often have a larger subdental shelf, and have an anteriorly short suprameckelian lip [35].

**Coronoid.** *Description.* TxVP 41229–25578 is a right coronoid (Fig 17G). The coronoid process is tall and rounded, the anteromedial process is elongated, and the tip is missing. The posteromedial process is oriented ventrally with an expanded lamina of bone posteriorly to articulate with the surangular medially and dorsally. There is a distinct medial crest that extends from the coronoid process onto the posteromedial process. There is a small, rounded lateral process. There is a distinct vertically oriented lateral crest that ends at the anterior margin of the lateral process. There is a relatively broad facet for dorsal articulation with the surangular that has a narrow groove medial to the lateral process.

*Identification.* The fossil coronoid shares with several pleurodontans and xantusiids the lack of an anteriorly projecting lateral process that overlaps the dentary [23] (Fig 22). Although

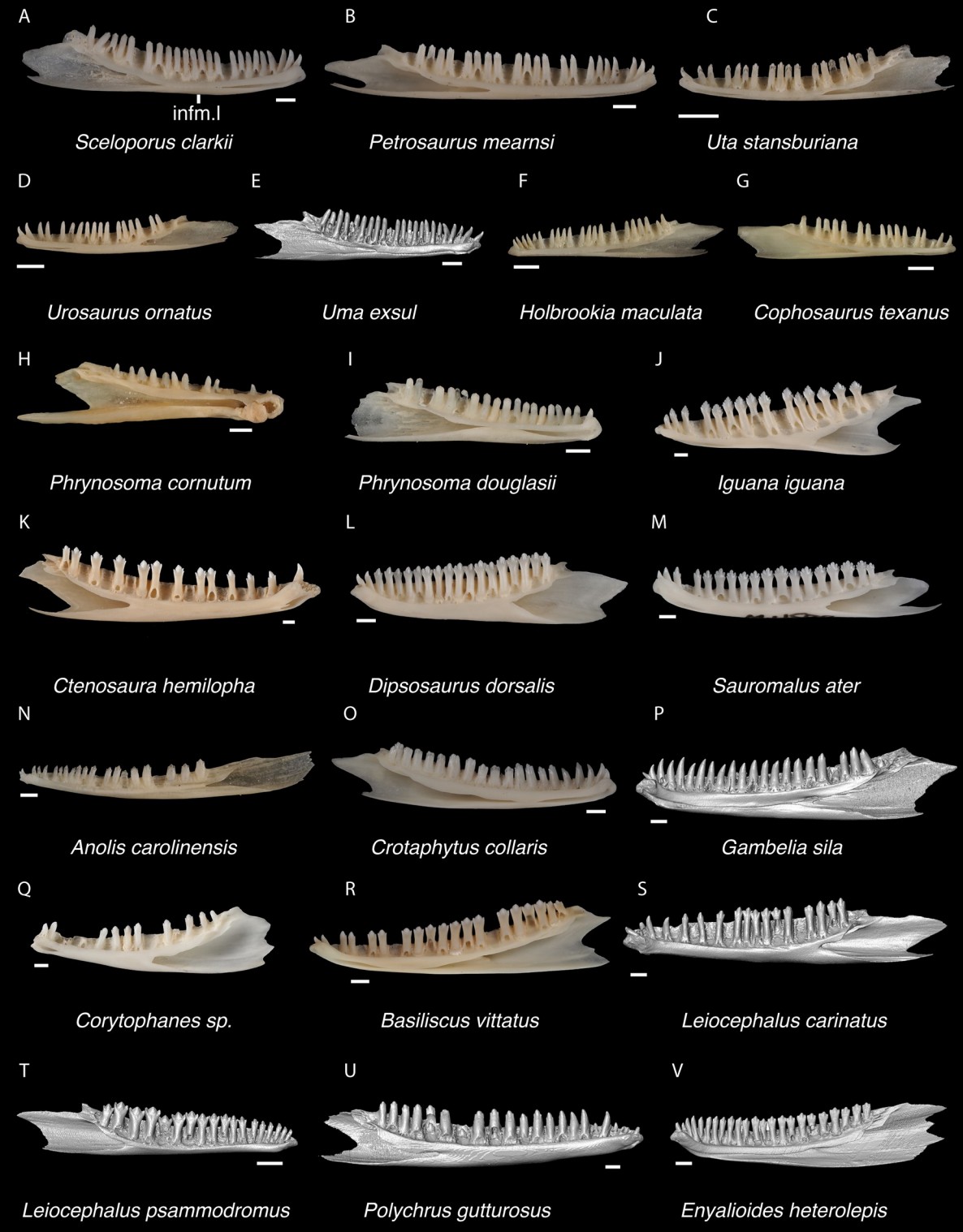

**Fig 21. Pleurodontan dentaries.** Dentaries in medial view–**A**. *Sceloporus clarkii* TxVP M-12157 right dentary; **B**. *Petrosaurus mearnsi* TxVP M-14910 right dentary; **C**. *Uta stansburiana* TxVP M-14935 left dentary; **D**. *Urosaurus ornatus* TxVP M-14330 left dentary; **E**. *Uma exsul* TNHC 30247 right dentary; **F**. *Holbrookia maculata* TxVP M-14322 left dentary; **G**. *Cophosaurus texanus* TxVP M-8527 right dentary; **H**. *Phrynosoma cornutum* TxVP M-6405 right dentary; **I**. *Phrynosoma douglasii* TxVP M-8526 right dentary; **J**. *Iguana iguana* TxVP M-8454 left dentary; **K**. *Ctenosaura hemilopha* TxVP M-8616 right dentary; **L**. *Dipsosaurus dorsalis* TxVP M-13086 right dentary; **M**. *Sauromalus ater*

TxVP M-11599 left dentary; **N**. *Anolis carolinensis* TxVP M-9042 right dentary; **O**. *Crotaphytus collaris* TxVP M-8395 right dentary; **P**. *Gambelia sila* TNHC 95261 right dentary; **Q**. *Corytophanes* sp. TxVP M-16765 right dentary; **R**. *Basiliscus vittatus* TxVP M-8556 right dentary; **S**. *Leiocephalus carinatus* TNHC 89274 right dentary; **T**. *Leiocephalus psammodromus* TNHC 103220 left dentary; **U**. *Polychrus gutturosus* TNHC 24152 left dentary; **V**. *Enyalioides heterolepis* UF 68015 right dentary. Scale bars = 1 mm. **Abbreviations**: infm.l, inframeckelian lip.

xantusiids also lack an anteriorly projecting lateral process [23], they differ from pleurodontans in having an anterior groove extending onto the coronoid process for articulation with the coronoid process of the spleniodentary [46]. Additionally, xantusiids differ from most pleurodontans and the fossil in having a relatively short and blunt anteromedial process and, except for *Xantusia riversiana*, in having a wide and low dorsal coronoid process [46]. Crotaphytids lack an anterolateral process of the coronoid, while iguanids, *Enyalioides heterolepis*, *Anolis*, and *Leiocephalus* have an anteriorly projecting anterolateral process that strongly articulates with the lateral portion of the dentary [20, 33]. A smaller specimen of *Iguana iguana* (TxVP M-13054), similar in size to adult *Crotaphytus*, has a tall, pointed coronoid process, although this morphology varies ontogenetically in *Iguana* [41]. The fossil and crotaphytids are further distinguished from *Dipsosaurus* in that the posteromedial process in *Dipsosaurus dorsalis* does not extend far ventrally below the notch formed between the posteromedial and anteromedial processes. Furthermore, the anteromedial process is short in *Dipsosaurus dorsalis* relative to examined *Crotaphytus*. It was previously reported that *Dipsosaurus dorsalis* lacks a notch (concavity of [20]) on the posterior margin of the posterolateral process that borders the adductor fossa anteriorly [20]; however, that feature is variable because a notch is present on at least one examined specimen of *Dipsosaurus dorsalis* (Digimorph.org specimen YPM 14376). In corytophanids, the lateral crest on the coronoid process continues ventrally onto the medial or posterior margin of the lateral process (see Fig 69 of [20]), whereas crotaphytids and the fossil have a lateral crest that merges with the anterior margin of the lateral process. Examined *Polychrus*, *Basiliscus basiliscus*, and *Corytophanes* differ from the fossil in lacking an expanded lamina of bone posterior to the coronoid process to articulate with the surangular dorsally. Phrynosomatids differ in having a relatively narrow facet on the ventral surface for articulation with the dorsal surface of the surangular. Additionally, in large species of *Sceloporus* (e.g., *S. clarkii*), the coronoid process is tall relative to the body of the bone. The posteromedial process in *Crotaphytus* is directed nearly ventrally, whereas the process is directed posteromedially (~45 degrees from vertical) in *Gambelia wislizenii* and *Gambelia copei* [49, 66]. The more vertically oriented posteromedial process on the fossil coronoid makes it likely that the fossils represent *Crotaphytus*; however, this is tentative given that a similar morphology also occurs in *Gambelia sila* [49].

**Compound bone.** *Description*. TxVP 41229–25853 is a left compound bone (fused prearticular, articular, and surangular), with only the anterior portion of the prearticular missing (Fig 17E). The adductor fossa is elongate. The dorsal face of the articular surface and the retroarticular process is coated in precipitate. The retroarticular process is narrow and elongate. There is an anteromedially-oriented angular process with a small lamina of bone connecting the retroarticular and angular processes. There is a medial crest that extends along the retroarticular process, and a distinct ventral ridge on the ventral surface of the process. The margin just lateral to the articular surface is devolved into a small boss. There is no lateral process nor a shelf connecting the body of the surangular to the tubercle (medial process of [49]) anterior to the articular surface. There is a distinct dentary articulation facet on the lateral surface. There is a foramen in the surangular just anterior to the adductor fossa and the surangular foramen is near the dorsal edge of the bone on the lateral surface. TxVP 41229–25613

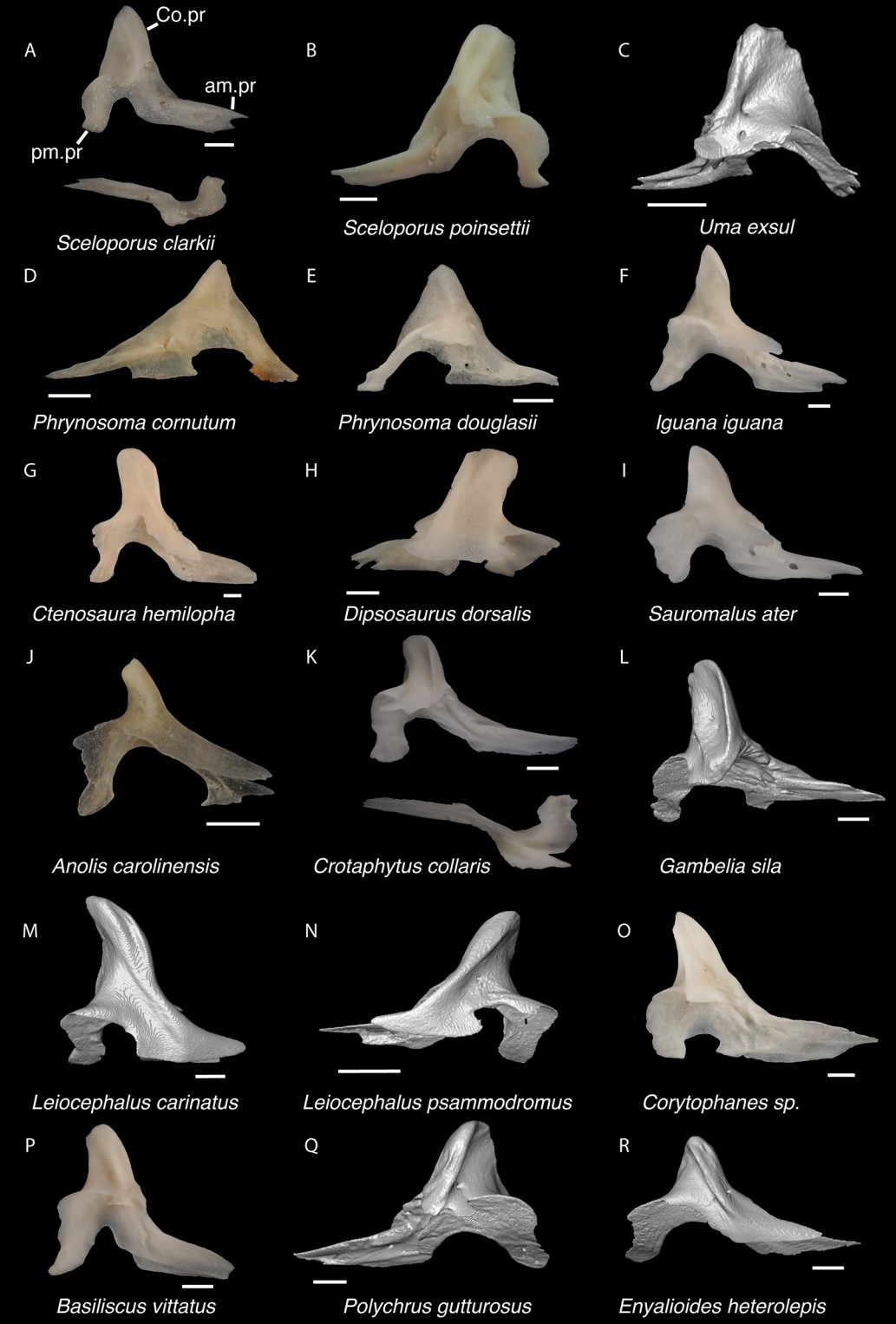

**Fig 22. Pleurodontan coronoids.** Coronoids in lateral and ventral views–**A**. *Sceloporus clarkii* TxVP M-12157 left coronoid in lateral and ventral views; **B**. *Sceloporus poinsettii* TxVP M-8373 left coronoid; **C**. *Uma exsul* TNHC 30247 left coronoid; **D**. *Phrynosoma cornutum* TxVP M-6405 left coronoid; **E**. *Phrynosoma douglasii* TxVP M-8526 right coronoid; **F**. *Iguana iguana* TxVP M-8454 right coronoid; **G**. *Ctenosaura hemilopha* TxVP M-8616 right coronoid; **H**. *Dipsosaurus dorsalis* TxVP M-13086 left coronoid; **I**. *Sauromalus ater* TxVP M-11599 right coronoid; **J**. *Anolis carolinensis* TxVP M-

9042 right coronoid; **K**. *Crotaphytus collaris* TxVP M-12468 right coronoid in lateral and ventral views; **L**. *Gambelia sila* TNHC 95261 right coronoid; **M**. *Leiocephalus carinatus* TNHC 89274 right coronoid; **N**. *Leiocephalus psammodromus* TNHC 103220 left coronoid; **O**. *Corytophanes* sp. TxVP M-16765 right coronoid; **P**. *Basiliscus vittatus* TxVP M-8556 right coronoid; **Q**. *Polychrus gutturosus* TNHC 24152 left coronoid; **R**. *Enyalioides heterolepis* UF 68015 right coronoid. Scale bars = 1 mm. **Abbreviations**: am.pr, anteromedial process; Co.pr, coronoid process; pm.pr, posteromedial process.

(Fig 17F) is a dermarticular (fused articular and prearticular) that does not differ substantially from TxVP 41229–25853 in the features preserved.

*Identification*. Compound bones are referred to Pleurodonta based on the presence of an angular process and a posteriorly directed retroarticular process that is not broadened posteriorly [23] (Fig 23). Teiids also have a posteriorly directed retroarticular process that is not broadened posteriorly; however, the adductor fossa in teiids is greatly expanded for insertion of the m. adductor mandibulae posterior into Meckel's canal [23] and the angular process is deflected further ventrally compared to pleurodontans [37, 53]. Among NA pleurodontans, the retroarticular process is reduced in *Polychrus* and the angular process is more horizontally oriented in *Dipsosaurus dorsalis* [20] compared to crotaphytids. The medial crest is more distinct in *Dipsosaurus dorsalis*, *Iguana iguana*, *Ctenosaura similis*, and *Enyalioides* compared to crotaphytids [20]. *Sauromalus* differs from the fossil and crotaphytids in having a short triangular angular process [20]. In *Laemanctus* and *Corytophanes*, the angular process is short, and the angular process in *Basiliscus basiliscus* does not project as far medially compared to *Crotaphytus collaris* (see Fig 71 of [20]). *Leiocephalus* differs from the fossil and crotaphytids in having a wider and rounded retroarticular process [33]. In phrynosomatids and in *Anolis*, the angular process is connected to the retroarticular process by a sheet of bone that is more extensive than seen in examined crotaphytid specimens; however, one large specimen of *Sceloporus clarkii* (TxVP M-12202) has a reduced connection between the angular and retroarticular processes. That specimen lacks a medial crest and has a relatively wide retroarticular process, distinguishing it from examined crotaphytids. Fossils are assigned to Crotaphytidae based on an elongate, thin retroarticular process, an elongate and ventrally slanted angular process, a short medial crest on the retroarticular process, and a reduced lamina of bone connecting the angular process and retroarticular process. *Crotaphytus* differs from *Gambelia* in having a distinct knob-like process just anterolateral to the articular surface (lateral process of [49]) and having a more reduced shelf of bone connecting the body of the surangular to the dorsal tubercle anterior to the articular surface [49]. A distinct knob-like process just anterolateral to the articular surface was not observed on some examined specimens of *Crotaphytus collaris*, so this feature likely varies intra- or interspecifically within *Crotaphytus*. A shelf of bone anterior to the dorsal tubercle is absent in most specimens of *Crotaphytus*, and a small shelf is present in specimens of *Gambelia wislizenii*. A small shelf is also reported in other species of *Crotaphytus* [49] so the presence of a shelf alone cannot be used to diagnose a fossil to *Gambelia*; however, only *Crotaphytus* lack the shelf [49]. A distinct ridge on the dorsolateral surface of the surangular reportedly occurs in *Crotaphytus* and is reduced or absent in *Gambelia* [49]. We could not identify a distinct ridge on the dorsolateral surface of the surangular in examined specimens of *Crotaphytus*, so this feature likely varies intra- or interspecifically within *Crotaphytus* and may be related to the development of the m. adductor mandibularis externus [49].

## Phrynosomatidae Fitzinger, 1843 [67]

Illustrated specimens referenced in the text: Compound bone, Morphotype A: 41229–26996 right, 41229–27307 right, Morphotype B: 41229–26293 left; Coronoid, 41229–26792 right, 41229–26887 left; Dentary, 41229–27590 left, 41229–8195 left; Frontal, 41229–26798, 41229–

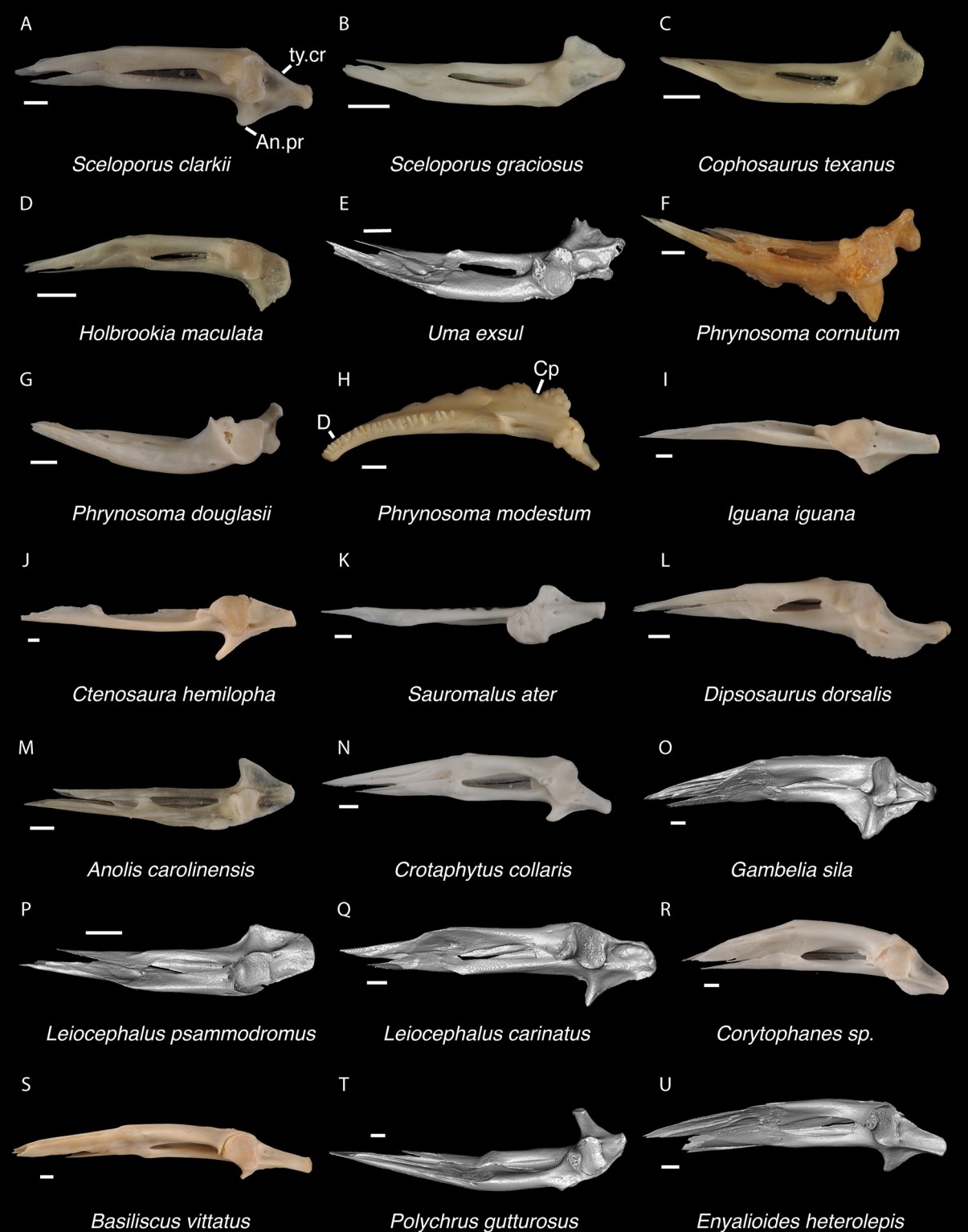

**Fig 23. Pleurodontan compound bones.** Compound bones in dorsal view–**A**. *Sceloporus clarkii* TxVP M-12157 right compound bone; **B**. *Sceloporus graciosus* TxVP M-14879 left compound bone; **C**. *Cophosaurus texanus* TxVP M-8527 left compound bone; **D**. *Holbrookia maculata* TxVP M-14322 right compound bone; **E**. *Uma exsul* TNHC 30247 left compound bone; **F**. *Phrynosoma cornutum* TxVP M-6405 left compound bone; **G**. *Phrynosoma douglasii* TxVP M-8526 left compound bone; **H**. *Phrynosoma modestum* TNHC 95921 right mandible; **I**. *Iguana iguana* TxVP M-8454 right articular; **J**. *Ctenosaura hemilopha* TxVP M-8616 right articular; **K**. *Sauromalus ater* TxVP M-11599 left

articular; **L**. *Dipsosaurus dorsalis* TxVP M-13086 right compound bone; **M**. *Anolis carolinensis* TxVP M-9042 left compound bone; **N**. *Crotaphytus collaris* TxVP M-12468 right compound bone; **O**. *Gambelia sila* TNHC 95261 right compound bone; **P**. *Leiocephalus psammodromus* TNHC 103220 left compound bone; **Q**. *Leiocephalus carinatus* TNHC 89274 right compound bone; **R**. *Corytophanes* sp. TxVP M-16765 right compound bone; **S**. *Basiliscus vittatus* TxVP M-8554 right compound bone; **T**. *Polychrus gutturosus* TNHC 24152 left compound bone; **U**. *Enyalioides heterolepis* UF 68015 right compound bone. Scale bars = 1 mm. **Abbreviations**: An.pr, angular process; D, dentary; CP, compound bone; ty.cr, tympanic crest.

27359; Jugal, 41229–27734 left; Maxilla, 41229–27044 left, 41229–27461 right; Parietal, 41229–27120, 41229–27364; Postorbital, Morphotype A: 41229–25584, Morphotype B: 41229–27522; Prefrontal, 41229–26976 right; Pterygoid, 41229–27011 left, 41229–27488 left; Splenial, 41229–26883; See S3 Table for complete list of specimens assigned to Phrynosomatidae.

**Maxilla.** *Description*. TxVP 41229–27461 and TxVP 41229–27044 serve as the basis for our description (Fig 24A and 24B). TxVP 41229–27461 is a right maxilla with 22 tooth positions. Teeth are weakly tricuspid except for the mesial teeth, and teeth are slender throughout the tooth row with bases near equal to the crowns in width. The facial process is broken dorsally and is curved medially with a distinct canthal crest. The facial process diminishes and merges with the crista transversalis, which extends anteromedially. There is a depression dorsally on the premaxillary process. The palatine process is present but broken. Dorsally the postorbital process has a groove that widens posteriorly for articulation with the jugal. Laterally there is a longitudinal ridge on the postorbital process. There is a superior alveolar foramen lateral to the palatine process. There are three foramina on the premaxillary process consisting of two openings for the superior alveolar canal (one anterior to the facial process and one on the crista transversalis) and a subnarial arterial foramen located anteromedially on the premaxillary process. There are eight lateral nutrient foramina. TxVP 41229–27044 differs in having 17 tooth positions, a less defined dorsal depression on the premaxillary process, no longitudinal ridge on the postorbital process, a flat dorsal surface of the postorbital process, a distinct asymmetric palatine process, two superior alveolar foramina, and four lateral nutrient foramina.

*Identification*. Maxillae were assigned to Pleurodonta based on the presence of pleurodont tricuspid teeth, having two foramina on the premaxillary process, and having a medially folded facial process with an anteroventrally trending canthal crest [29]. Phrynosomatids and the fossil maxillae differ from crotaphytids in having slender teeth throughout the tooth row with bases near equal in width compared to the crowns [65]. Furthermore, the fossils differ from crotaphytids in having a facial process that curves anteromedially and reaches the medial edge of the premaxillary process, as opposed to a facial process that is more weakly folded and diminishes into a low ridge far from the medial edge of the premaxillary process [29]. Fossil maxillae share with phrynosomatids, *Anolis*, and *Polychrus* a relatively flat dorsal surface of the palatal plate [29]. *Anolis* differs from phrynosomatids and the fossils in having an anteroposteriorly extended facial process with a canthal crest closer to horizontal, and *Polychrus* differs in having a facial process medially folded only at the dorsal tip [29].

**Frontal.** *Description*. TxVP 41229–27359 and TxVP 41229–26798 serve as the basis for our description (Fig 24C and 24D). There is a small amount of sculpting on the posterodorsal surface, and the bone is slightly concave ventrally. Anteriorly there are lateral prefrontal facets and two distinct dorsal nasal facets defined by distinct anterolateral processes and separated by a smaller anteromedial process. The interorbital margins are waisted and the posterolateral processes flare laterally. The posterior edge is wavy and has a wide midline notch. There are small postfrontal facets laterally on the posterolateral processes and distinct parietal facets posteriorly. The cristae cranii are short, but approach one another in the interorbital region, and bound an indistinct groove for attachment of the solium supraseptale. The cristae cranii

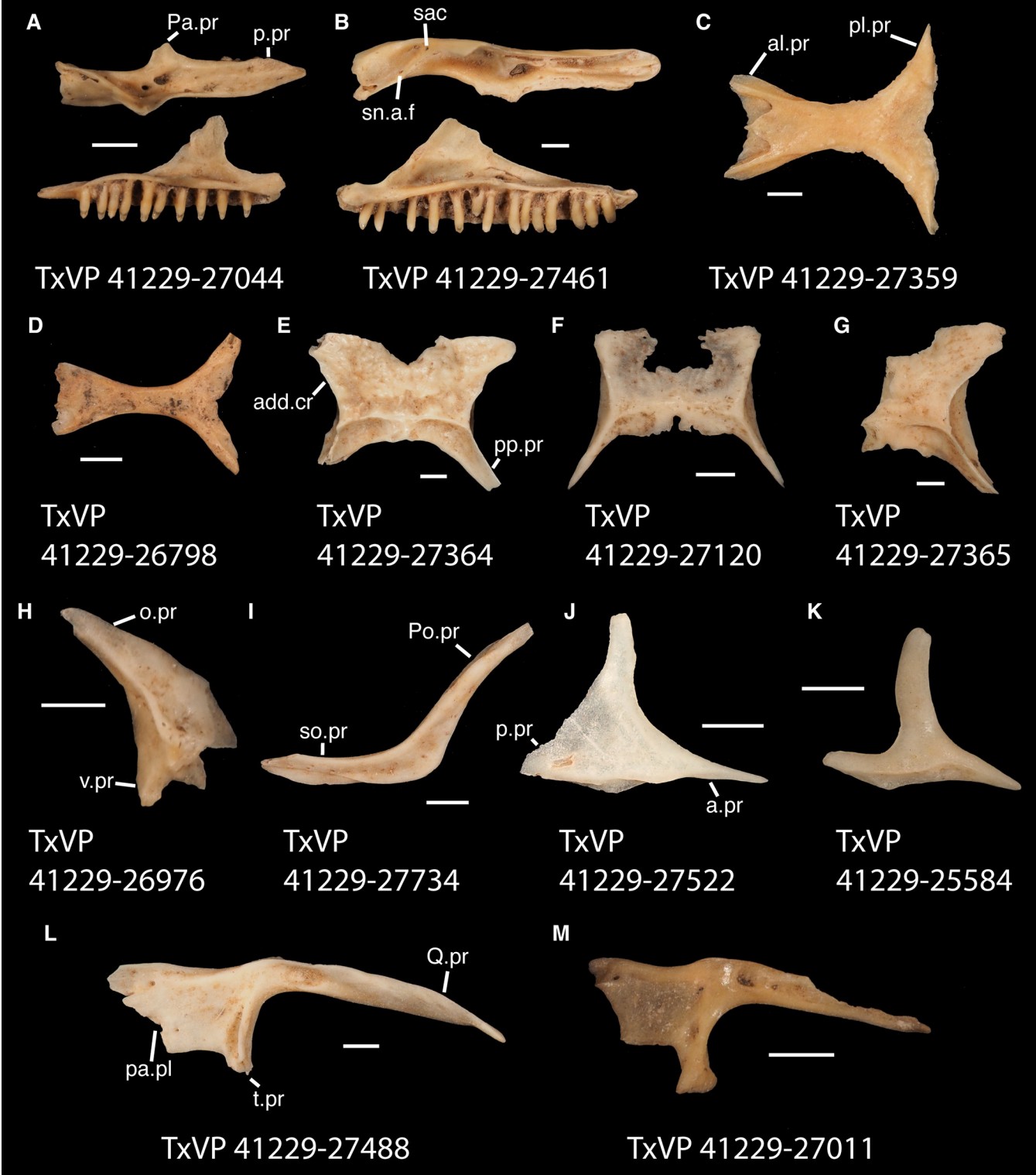

**Fig 24. Fossil phrynosomatids. A**. TxVP 41229–27044 Dorsal and medial view of left maxilla; **B**. TxVP 41229–27461 Dorsal and medial view of right maxilla; **C**. TxVP 41229–27359 Dorsal view of frontal; **D**. TxVP 41229–26798 Dorsal view of frontal; **E**. TxVP 41229–27364 Dorsal view of parietal; **F**. TxVP 41229–27120 Dorsal view of parietal; **G**. TxVP 41229–27365 Dorsal view of parietal; **H**. TxVP 41229–26976 Lateral view of right prefrontal; **I**. TxVP 41229–27734 Lateral view of left jugal; **J**. TxVP 41229–27522 Dorsolateral view of right postorbital; **K**. TxVP 41229–25584 Dorsolateral view of right postorbital; **L**. TxVP 41229–27488 Dorsal view of left pterygoid; **M**. TxVP 41229–27011 Dorsal view of left pterygoid. Scale bars = 1 mm. **Abbreviations**: add.cr, adductor crest; al.

pr, anterolateral process; a.pr, anterior process; o.pr, orbital process; pa.pl, palatal plate; Pa.pr, palatine process; pl.pr, posterolateral process; Po.pr, postorbital process; pp.pr, postparietal process; p.pr, posterior process; Q.pr, quadrate process; so.pr, suborbital process; sac, opening for the superior alveolar canal; sn.a.f, subnarial arterial foramen; t.pr, transverse process; v.pr, ventral process.

diverge posteriorly and extend along the lateral margins of the ventral surface. TxVP 41229–26798 differs in having less distinct nasal facets, minute cristae cranii, a larger notch in the posterior edge, and an especially thin interorbital region.

*Identification.* Fossil frontals share with Pleurodonta and Gymnophthalmoidea a fused frontal with reduced cristae cranii [23]. Fossil frontals differ from teiids in having strongly constricted interorbital margins of the frontal that strongly curve posterolaterally [23]. Fossils differ from gymnophthalmids and alopoglossids in lacking frontal tabs [23, 37]. *Mabuya* and *Scincella* also have a fused frontal with reduced cristae cranii [36, 68] but differ from the fossils in having a well-developed anteromedial process on the frontal and a relatively wider anterior end. The fossils and phrynosomatids differ from many iguanids (see above) in lacking a parietal foramen enclosed within the frontal, and differ from *Anolis*, *Polychrus*, and some *Leiocephalus* (e.g., *L. carinatus*) in having more slender interorbital margins [20, 40, 69]. Examined *Leiocephalus* have a small, narrow notch at the posterior margin of the frontal (see also [33]) that is not found in the fossils or phrynosomatids. *Enyalioides heterolepis* differs from the fossil in having a bumpy, knob-like texture on the dorsal surface of the frontal. *Ctenosaura* and adult *Iguana* have wider interorbital margins compared to phrynosomatids, but the interorbital margins of juvenile *Iguana iguana* are comparable to that of larger *Sceloporus* and only differ in having more slender posterolateral processes [41]. Fossil frontals differ from crotaphytids in lacking a distinctly triradiate anterior end of the frontal. Additionally, smaller fossil frontals differ from crotaphytids in having a wider concave margin in the posterior edge, and larger fossils more closely resemble *Sceloporus* in having wider interorbital margins of the frontal compared to similar-sized crotaphytids.

**Parietal.**   *Description.* TxVP 41229–27364, TxVP 41229–27365, TxVP 41229–27120 serve as the basis for our description (Fig 24E–24G). TxVP 41229–27364 is a parietal with some sculpturing on the dorsal surface. The left anterolateral process and distal portions of the postparietal processes are broken. There is a large midline notch on the anterior edge. The adductor crests do not meet posteriorly, and the parietal table has a trapezoidal appearance. The ventrolateral crests are short and obscured in dorsal view by the adductor crests. The anterolateral process flares laterally. The posterior edge between the postparietal processes is characterized by two distinct depressions (nuchal fossae). The postparietal process has a dorsal crest that slants laterally. The ventral surface has shallow depressions (cerebral vault) divided by a low ridge. There is a deep pit for the processus ascendens just anterior to the posterior edge. TxVP 41229–27365 is missing the left half but has a large midline notch on the anterior edge. TxVP 41229–27365 has adductor crests that partially cover the ventrolateral crests and a postparietal process dorsal crest that directed dorsally. TxVP 41229–27120 differs from TxVP 41229–27364 and TxVP 41229–27365 in lacking laterally flared anterolateral processes, having a large fontanelle on the anteromedial portion of the parietal, having adductor crests that do not cover the ventrolateral crests in dorsal view, lacking dorsal crests on the postparietal processes, and having a shallow pit for the processus ascendens. TxVP 41229–27120 has a notch on the posterior edge, but this may be due to erosion of the bone because there are many pitted areas.

Identification: Parietals are assigned to Pleurodonta based on the presence of a fused parietal [23], the absence of co-ossified osteoderms [23, 24], the absence of a parietal foramen that is fully enclosed by the parietal [23], and the absence of distinct ventrolateral crests (parietal

downgrowths of [23]). Parietals are assigned to Phrynosomatidae based on the presence of a large unossified anteromedial portion of the parietal around the location of the parietal foramen [24], which is largely ossified in other NA pleurodontans. A few *Crotaphytus* (e.g., *C. collaris* TxVP M-8354) do have a large unossified anteromedial portion of the parietal, but in *Crotaphytus*, the ventrolateral crests are slanted more medially. A large unossified anteromedial portion of the parietal was noted to occur in smaller-bodied phrynosomatid genera, such as many phrynosomatines [24], but we observed this feature in smaller species of *Sceloporus* (e.g., *S. occidentalis*) and *Uta* (see also [70]) as well.

**Prefrontal.** *Description.* TxVP 41229–26976 is a right prefrontal (Fig 24H). It has a long and pointed orbital process, a short ventral process, and an anterior main body developed into a sheet of bone. The anterior sheet is slightly broken and has a broad articulation facet for the facial process of the maxilla. There is a distinct ridge on the lateral surface near the base of the orbital process that is continuous with a large lateral boss. The ventral process is narrow and squared-off. There is a distinct notch for the lacrimal foramen, and the ventral process forms the posterior border of the foramen. Medially, the boundary of the olfactory chamber is a smooth, rounded, and concave surface. Dorsal to the olfactory chamber is a shallow groove for articulation with the frontal. The orbitonasal flange is narrow.

*Identification.* The fossil shares with pleurodontans and teiids the presence of a distinct prefrontal boss [27] (Fig 25). The fossil and prefrontals of many pleurodontans differ from teiids in having a boss that widens ventrally and is semicircular or tear-dropped shaped [29]. Additionally, teiids differ in having a thin, laterally projecting lamina with a distinct articulation facet for the facial process of the maxilla [53]. *Basiliscus*, *Polychrus*, and *Anolis* differ from the fossil in often having a more prominent boss or a strong lateral canthal ridge [20]. The prefrontal boss is larger in adult *Cyclura* and *Iguana* compared to the fossil [20], and juvenile *Iguana* differ in having an exceptionally thin ventral process. *Corytophanes* and *Laemanctus* differ from the fossil in having rugosities on the dorsal surface [20]. Furthermore, *Corytophanes* differs from the fossil in having a long supraorbital process, and *Laemanctus* differs in having a posterior process that is directed more posteriorly [20]. *Crotaphytus* differs from the fossil in having the prefrontal boss slightly obscure the lacrimal notch in lateral view. Examined *Gambelia* differ in having a more distinct boss that projects farther laterally. The ridge dorsal to the prefrontal boss is wider in examined *Leiocephalus* compared to the fossil. *Enyalioides heterolepis* differs from the fossil in having a more bulbous prefrontal boss. The fossil differs from other NA pleurodontans and shares several features with phrynosomatids excluding *Phrynosoma*, which has a long supraorbital process, and phrynosomatines that have a prefrontal boss developed into a laterally projecting thin lamina (e.g., *Cophosaurus* and *Holbrookia*).

**Jugal.** *Description.* TxVP 41229–27734 is a left jugal (Fig 24I). There is a maxillary facet laterally and ventrally on the suborbital process. The dorsal margin of the suborbital process is everted laterally and there is a groove on the medial surface. There is no quadratojugal process. There is a ridge on the anteromedial edge of the postorbital process and a postorbital facet anteriorly. The postorbital process is posteriorly directed and concave posteriorly. There is a foramen medially near the inflection point, two foramina anteriorly on the postorbital process, and many small foramina along the lateral surface.

*Identification.* Jugals were assigned to Pleurodonta based on the absence of a quadratojugal process [23, 29]. *Xantusia* also lack a quadratojugal process but differ from pleurodontans in having an exceptionally short and thin suborbital process [61]. Among NA pleurodontans, the fossil shares with Phrynosomatidae and Crotaphytidae a posteriorly deflected distal end of the postorbital process [29]; however, crotaphytids differ in having a quadratojugal process.

**Postorbital.** *Description.* Morphotype A: TxVP 41229–27522 is a right postorbital (Fig 24J). The fossil is triradiate with thin anterior and dorsal processes and a wider posterior

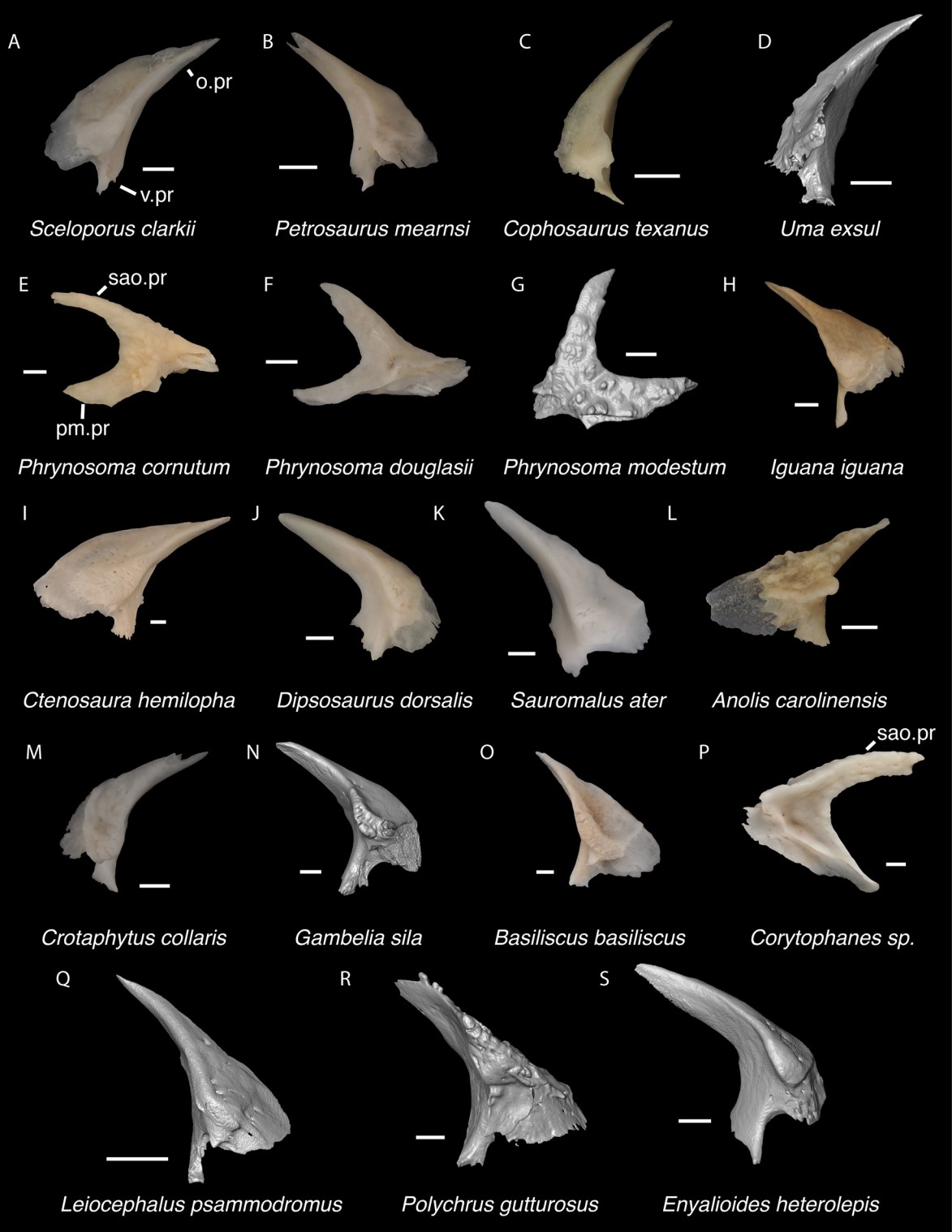

**Fig 25. Pleurodontan prefrontals.** Prefrontals in lateral and dorsal views–**A**. *Sceloporus clarkii* TxVP M-12157 left prefrontal in lateral view; **B**. *Petrosaurus mearnsi* TxVP M-14910 right prefrontal in lateral view; **C**. *Cophosaurus texanus* TxVP M-8527 left prefrontal in lateral view; **D**. *Uma exsul* TNHC 30247 left prefrontal in lateral view; **E**. *Phrynosoma cornutum* TxVP M-6405 left prefrontal in dorsal view; **F**. *Phrynosoma douglasii* TxVP M-8526 left prefrontal in dorsal view; **G**. *Phrynosoma modestum* TNHC 95921 right prefrontal in dorsal view; **H**. *Iguana iguana* TxVP M-13054 right prefrontal in lateral view; **I**. *Ctenosaura hemilopha* TxVP M-8616 left prefrontal in lateral view; **J**.

*Dipsosaurus dorsalis* TxVP M-13086 right prefrontal in lateral view; **K**. *Sauromalus ater* TxVP M-11599 right prefrontal in lateral view; **L**. *Anolis carolinensis* TxVP M-9042 left prefrontal in lateral view; **M**. *Crotaphytus collaris* TxVP M-12468 left prefrontal in lateral view; **N**. *Gambelia sila* TNHC 95261 right prefrontal in lateral view; **O**. *Basiliscus basiliscus* TxVP M-11907 right prefrontal in lateral view; **P**. *Corytophanes* sp. TxVP M-16765 left prefrontal in lateral view; **Q**. *Leiocephalus psammodromus* TNHC 103220 right prefrontal in lateral view; **R**. *Polychrus gutturosus* TNHC 24152 right prefrontal in lateral view; **S**. *Enyalioides heterolepis* UF 68015 right prefrontal in lateral view. Scale bars = 1 mm. **Abbreviations**: o.pr, orbital process; pm.pr, posteromedial process; sao.pr, supraorbital process; v.pr, ventral process.

process. The anterior and dorsal processes are relatively straight. There is a small articulation facet anteromedially on the orbital process and there are distinct jugal and squamosal facets along the ventrolateral edge.

Morphotype B: TxVP 41229–25584 is a right postorbital (Fig 24K). The bone is triradiate with thin anterior, dorsal, and posterior processes. The ventrolateral margin is concave below the anterior portion and the posterior process is slightly curved dorsally. The dorsal process is concave anteriorly without an anterior articulation facet. There are distinct jugal and squamosal facets that are visible in dorsolateral view.

*Identification*. Postorbitals were identified to Pleurodonta based on a sub-triangular morphology with a distinct ventral process [23]. The fossils share with phrynosomatids, *Anolis*, some iguanids, and some *Leiocephalus* a smooth lateral face of the dorsal process [20, 29, 52]. The posterior process in *Ctenosaura* (see also Fig 40G of [20]) is proportionally more elongated compared to the fossils. *Dipsosaurus dorsalis* differs in often having a flangelike expansion on the orbital process [20]. *Iguana* generally has a wider orbital process compared to the fossil and phrynosomatids excluding *Phrynosoma*. The postorbital of *Sauromalus* and *Dipsosaurus dorsalis* is more straight in anterior view compared to the fossils and phrynosomatids [20]. *Anolis* differs from the fossils and phrynosomatids in having an articulation facet visible on the lateral or anterolateral surface of the orbital process as it underlaps the frontoparietal corner [29]. The postorbital of *Leiocephalus* differs from the fossils in lacking a distinct squamosal facet along the ventrolateral edge (see also Fig 3F of [33]). The postorbitals of examined *Uta*, *Urosaurus*, *Cophosaurus*, and *Callisaurus* have a slender build relative to *Sceloporus* and are more similar to Morphotype B. Sand lizards differ from sceloporines in lacking a postfrontal [71] and therefore also lack a postfrontal articulation surface on the postorbital. Fossils identified to morphotype A likely represent *Sceloporus* or *Petrosaurus* and fossils identified to morphotype B likely represent sand lizards.

**Pterygoid.**   *Description*. TxVP 41229–27488 and TxVP 41229–27011 serve as the basis for our description (Fig 24L and 24M). TxVP 41229–27488 is a left pterygoid that is missing the distal end of the transverse process. The palatine process is broad at the base and has a palatine facet anteriorly. The transverse process extends laterally at nearly a 90-degree angle with the quadrate process. The transverse process is dorsoventrally tall and bears a distinct ectopterygoid facet. There is a distinct ridge on the dorsal surface for insertion of the superficial pseudotemporal muscle. There is a large ridge on the ventral surface for insertion of the pterygomandibular muscle and a deep fossa columella without a pterygoid groove. The quadrate process is elongated, and the medial surface has a groove that serves for insertion of the pterygoideus muscle. There is a medial groove dorsal to a shelf-like projection at the floor of the basipterygoid fossa. There are no pterygoid teeth, but there is a large hole on the ventral surface of the palatal plate along with several smaller foramina on both the dorsal and ventral surface. TxVP 41229–27011 is smaller than TxVP 41229–27488 and the distal end of the transverse process is pointed and tall vertically.

*Identification*. Among examined NA lizards, only in pleurodontans is the transverse process oriented medially, creating a near right angle with the quadrate process. In other NA lizards,

the transverse process is oriented anteromedially. This distinction in transverse process orientation likely only applies to NA lizards because at least some non-pleurodontans (e.g., scincids [72]) from other geographic regions share a medially oriented transverse process. Fossil pterygoids have medially oriented transverse process and were identified to Pleurodonta. Among NA pleurodontans, pterygoid teeth are absent in phrynosomatids and are variably present in *Anolis*, *Polychrus*, *Dipsosaurus*, and *Leiocephalus* [29, 42]. Pterygoid teeth previously were reported as present in *Sauromalus* [42], but we observed one specimen (*Sauromalus ater* TxVP M- 9782) without pterygoid teeth. *Dipsosaurus dorsalis* differs from phrynosomatids in having a more narrow palatal plate and a narrow linear notch on the transverse process. Compared with the fossils, *Sauromalus* has a narrower palatal plate and a taller quadrate process. *Leiocephalus* differs from the fossils in having a narrower palatal plate and a more distinct ventromedial projection at the floor of the basipterygoid fossa. Examined *Anolis* and *Polychrus* differ from the fossils and examined phrynosomatids in having a taller ridge for the insertion of the pterygomandibular muscle [73] on the ventral surface of the pterygoid extending from the transverse process to the basipterygoid fossa. In addition, examined *A. carolinensis* have a ventral projection near the basipterygoid fossa not seen in the fossils or phrynosomatids. Based on the differences from other NA pleurodontans, fossils were assigned to Phrynosomatidae.

**Dentary.** *Description*. TxVP 41229–27590 and TxVP 41229–8195 serve as the basis for our description (Fig 26A and 26B). TxVP 41229–27590 is a left dentary with 26 tooth positions. Distal teeth are tricuspid and relatively slender. The suprameckelian and inframeckelian lips approach one another midway along the tooth row, but the Meckelian groove is open for its entire length. The suprameckelian lip is relatively short anteriorly. The dental shelf is narrow but widens slightly anteriorly. There is a distinct intramandibular lamella, and the intramandibular septum reaches to the level of the fourth most distal tooth position. There is a small posterior projection from the intramandibular septum. The posterior end is broken, but there is a coronoid facet with a projecting corner of bone. There are four nutrient foramina on the anterolateral surface of the bone. TxVP 41229–27590 differs from TxVP 41229–8195 in having an intramandibular suptum that extends farther posteriorly. TxVP 41229–8195 differs further from TxVP 41229–27590 in lacking the small posterior projection from the intramandibular septum, having a bifurcated posterior end, and having five nutrient foramina on the anterolateral surface.

*Identification*. Dentaries share with Pleurodonta and some teiids pleurodont tricuspid teeth and an inframeckelian lip that curls dorsolingually, producing a medial exposure of the Meckelian groove along the mid-length of the dentary [2, 27, 35]. Fossil dentaries differ from teiids in lacking a broad subdental shelf [23], lacking asymmetric bicuspid teeth, and lacking large amounts of cementum deposits at base of teeth [23, 64, 74]. Fossils are assigned to Pleurodonta. Among NA pleurodontans, the fossils differ from iguanids, *Anolis*, extant *Leiocephalus*, and *Polychrus*, which all have a fused Meckelian groove [20, 35]. *Basiliscus*, *Corytophanes*, and some *Laemanctus* differ from the fossils in having flared tooth crowns [29, 35]. Some *Laemanctus* have a fused Meckelian groove [20, 35], but it is unfused in *Laemanctus serratus* [29, 56]. Hoplocercids, except besides *Enyalioides laticeps* (FMNH 206132), differ in having an open Meckelian groove that is not bounded by suprameckelian and inframeckelian lips [35, 48]. Crotaphytids differ from the fossils in having a relatively tall suprameckelian lip on the anterior half of the dentary [35]. Furthermore, *Crotaphytus* differs in having teeth that widen towards the base [65] and *Gambelia* differs in having sharper and more recurved mesial teeth [35]. Based on these differences with other NA pleurodontans, fossils were assigned to Phrynosomatidae.

**Coronoid.** *Description*. TxVP 41229–26792 and TxVP 41229–26887 serve as the basis for our description (Fig 26D and 26E). TxVP 41229–26792 is a right coronoid. The coronoid

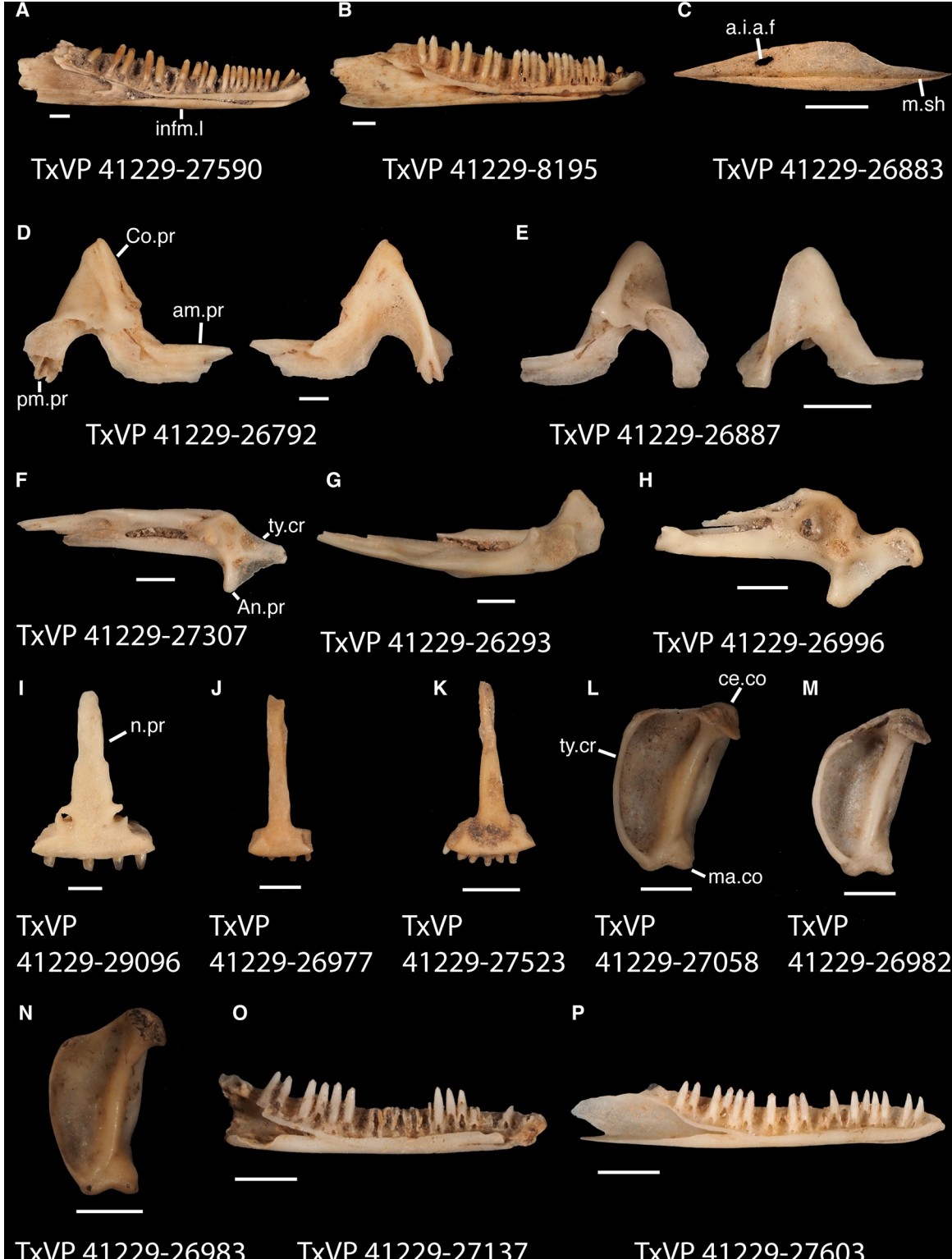

**Fig 26.** Fossil phrynosomatids, A–H: Phrynosomatidae, I–N: Sceloporinae, O–P: *Urosaurus*. A. TxVP 41229–27590 Lateral view of left dentary; **B**. TxVP 41229–8195 Lateral view of left dentary; **C**. TxVP 41229–26883 Lateral view of left splenial; **D**. TxVP 41229–26792 Lateral and medial view of right coronoid; **E**. TxVP 41229–26887 Lateral and medial view of left coronoid; **F**. TxVP 41229–27307 Dorsal view of right compound bone; **G**. TxVP 41229–26293 Dorsal view of left compound bone; **H**. TxVP 41229–26996 Dorsal view of right compound bone; **I**. TxVP 41229–29096 Anterior view of premaxilla; **J**. TxVP 41229–26977 Anterior view of premaxilla; **K**. TxVP 41229–

27523 Anterior view of premaxilla; **L**. TxVP 41229–27058 Posterior view of left quadrate; **M**. TxVP 41229–26982 Posterior view of left quadrate; **N**. TxVP 41229–26983 Posterior view of left quadrate; **O**. TxVP 41229–27137 Lateral view of left dentary; **P**. TxVP 41229–27603 Lateral view of left dentary. Scale bars = 1 mm. **Abbreviations**: a.i.a.f, anterior inferior alveolar foramen; am.pr, anteromedial process; An.pr, angular process; ce.co, cephalic condyle; Co.pr, coronoid process; infm.l, inframeckelian lip; m.sh, medial shelf; n.pr, nasal process; pm.pr, posteromedial process; ty.cr, tympanic crest.

process is tall and pointed and the anteromedial process is elongated. The posteromedial process is missing the distal end, but the remaining portion is ventrally oriented. There is a distinct medial crest that extends from the coronoid process onto the posteromedial process. There is a small, rounded lateral process. There is a vertically oriented lateral crest that ends at the anterior margin of the lateral process. The ventral surface is characterized by a narrow concave facet for articulation with the surangular dorsally. The ventral surface also bears a small lateral groove. TxVP 41229–26887 differs from TxVP 41229–26792 in being smaller, having a more rounded coronoid process, and having a more distinct lateral crest.

*Identification*. Fossil coronoids share with some members of Pleurodonta and xantusiids the absence of a distinct anterolateral process [23]. Xantusiids differ from the fossils and pleurodontans in having an anterior groove extending onto the coronoid process for articulation with the coronoid process of the spleniodentary. Fossils differ from iguanids, *Enyalioides heterolepis*, *Anolis*, and *Leiocephalus*, which all have an anterolateral process [20, 33]. The fossils differ from corytophanids (see also Fig 69 of [20]) but share with phrynosomatids and crotaphytids a lateral crest that merges with the anterior margin of the lateral process. Additionally, corytophanids differ in having the apex of the coronoid process posterodorsally deflected to a greater degree compared to the fossils (see also Fig 69 of [20]). Fossils differ from crotaphytids in having a narrow facet on the ventral surface of the coronoid for articulation with the dorsal surface of the surangular. Moreover, *Crotaphytus* differs from the fossils in having more extensive lamina of bone posterior to the coronoid process to articulate with the surangular dorsally and *Gambelia*, except for *G. sila*, differs in having a posteromedial process directed more posteriorly [49, 66]. Based on these differences with respect to other NA pleurodontans, fossils were assigned to Phrynosomatidae.

**Splenial.** *Description*. TxVP 41229–26883 is a left splenial (Fig 26C). It is slender with elongate anterior and posterior processes. There is a large anterior inferior alveolar foramen that is positioned anterodorsal to the smaller anterior mylohyoid foramen. There is a medial shelf that curls dorsally and obscures the anterior mylohyoid foramen in medial view.

*Identification*. The fossil shares an enclosed anterior inferior alveolar foramen with pleurodontans, teiids, some gymnophthalmids, alopoglossids, and some anguids [24, 27, 37] (Fig 27). Teiinae differ from the fossil and other North American lizards in having an anterior inferior alveolar foramen that is larger and posterodorsally located relative to the anterior mylohyoid foramen [27, 53, 75] (Fig 28). Anguids usually have an anterior inferior alveolar foramen that is not entirely enclosed by the splenial, but some individuals have been observed with a completely enclosed foramen [55, 76]. Many anguids differ from the fossil and pleurodontans in having the posterior end of the splenial bifurcated into a distinct dorsal and ventral process [77]. There is substantial variation in splenial morphology among gymnophthalmids. Some gymnophthalmids differ from the fossil in having an anteroposteriorly short splenial [24, 78, 79]. Other gymnophthalmids (e.g., *Gymnophthalmus speciosus*) differ from the fossil in lacking an anterior inferior alveolar foramen fully enclosed within the splenial [24]. Some alopoglossids (e.g., *Ptychoglossus vallensis*, see Fig 7 of [37]) resemble the fossil, but are excluded based on geography. The fossil splenial is identified to Pleurodonta.

The splenial of crotaphytids is much slenderer and elongate compared to that of the fossil. The prearticular crest in examined *Iguana iguana*, *Ctenosaura*, *Dipsosaurus dorsalis*,

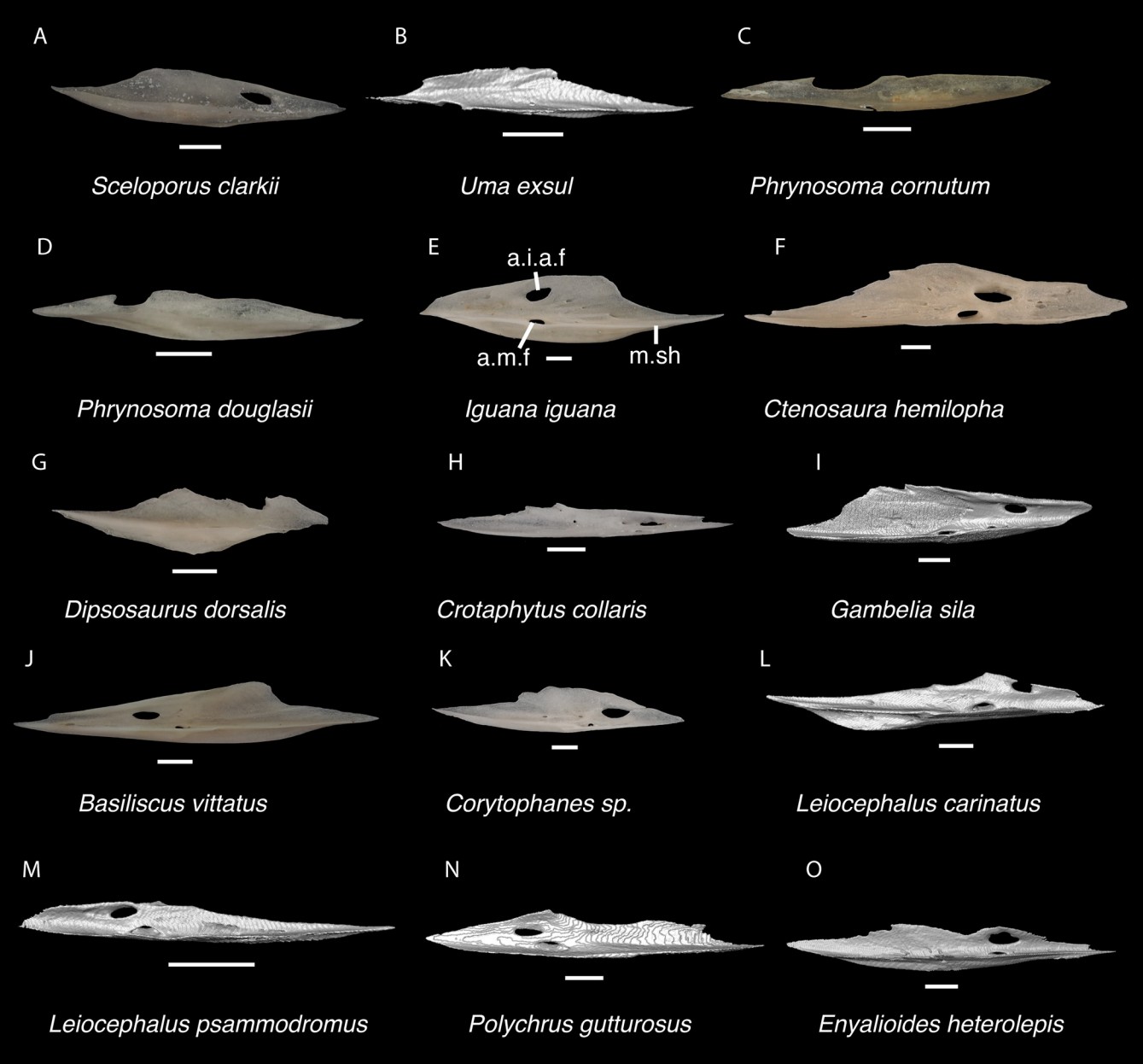

**Fig 27. Pleurodontan splenials.** Splenials in lateral view–**A**. *Sceloporus clarkii* TxVP M-12157 right splenial; **B**. *Uma exsul* TNHC 30247 left splenial; **C**. *Phrynosoma cornutum* TxVP M-6405 left splenial; **D**. *Phrynosoma douglasii* TxVP M-8526 left splenial; **E**. *Iguana iguana* TxVP M-8454 left splenial; **F**. *Ctenosaura hemilopha* TxVP M-8616 right splenial; **G**. *Dipsosaurus dorsalis* TxVP M-13086 right splenial; **H**. *Crotaphytus collaris* TxVP M-12468 right splenial; **I**. *Gambelia sila* TNHC 95261 right splenial; **J**. *Basiliscus vittatus* TxVP M-8556 left splenial; **K**. *Corytophanes* sp. TxVP M-16765 right splenial; **L**. *Leiocephalus carinatus* TNHC 89274 right splenial; **M**. *Leiocephalus psammodromus* TNHC 103220 left splenial; **N**. *Polychrus gutturosus* TNHC 24152 left splenial; **O**. *Enyalioides heterolepis* UF 68015 right splenial. Scale bars = 1 mm. **Abbreviations**: a.i.a.f, anterior inferior alveolar foramen; a.m.f, anterior mylohyoid foramen; m.sh, medial shelf.

*Sauromalus*, *Corytophanes*, and *Basiliscus vittatus* is flat, whereas it curves dorsally in the fossil and in most examined *Sceloporus* (except for some *Sceloporus jarrovii*) such that the anterior mylohyoid foramen is obscured in lateral view. *Polychrus marmoratus* UF 65135, *Iguana iguana*, and *Iguana delicatissima* differ from the fossil in having the anterior inferior alveolar foramen positioned directly dorsal to the anterior mylohyoid foramen [80]. *Polychrus*

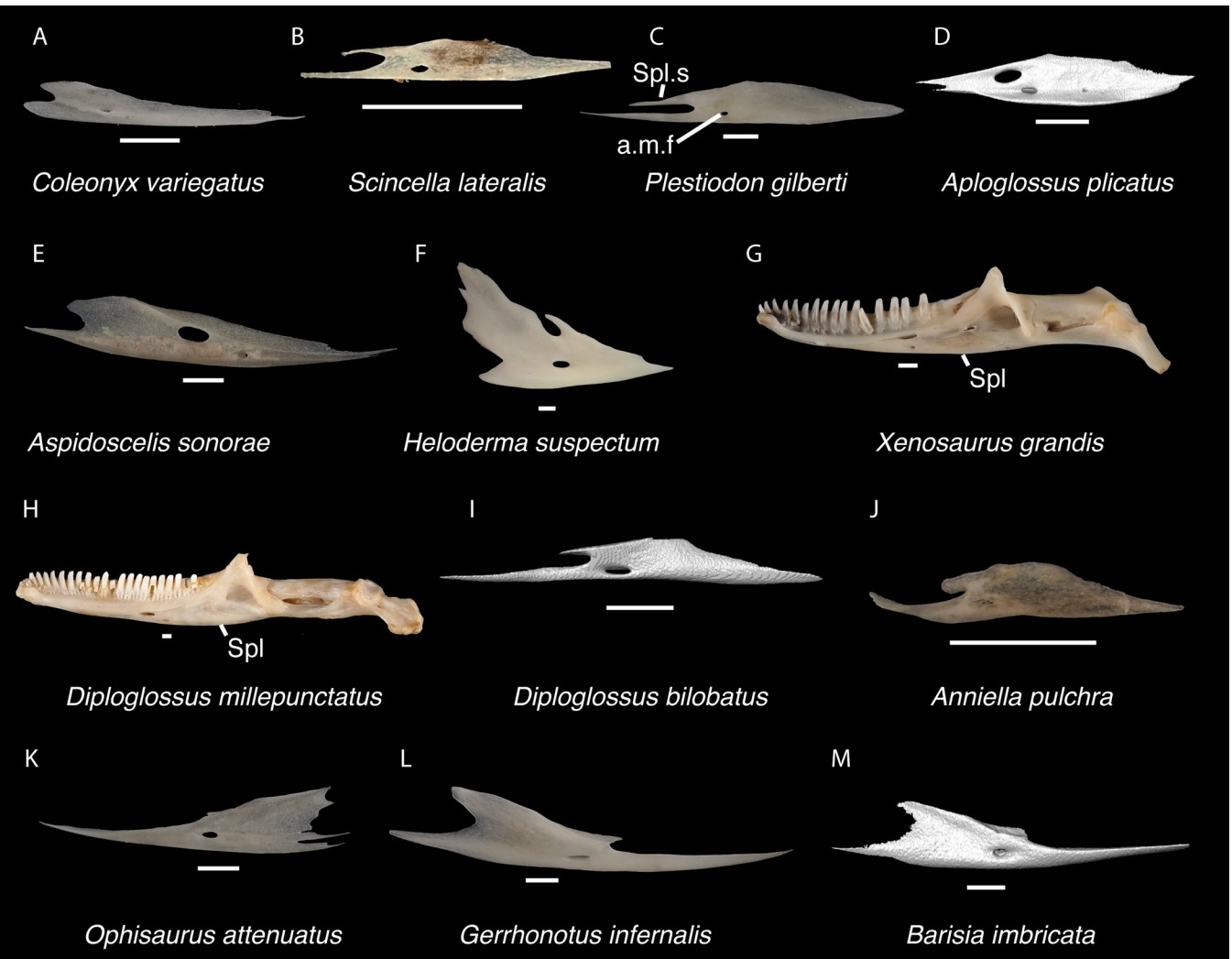

**Fig 28. Non-pleurodontan splenials.** Splenials in medial view–**A**. *Coleonyx variegatus* TxVP M-12109 left splenial; **B**. *Scincella lateralis* TxVP M-5531 right splenial; **C**. *Plestiodon gilberti* TxVP M-8587 right splenial; **D**. *Aploglossus plicatus* 34481 right splenial; **E**. *Aspidoscelis sonorae* TxVP M-15670 left splenial; **F**. *Heloderma suspectum* TxVP M-9001 left splenial; **G**. *Xenosaurus grandis* TxVP M-8960 right mandible in medial view; **H**. *Diploglossus millepunctatus* TxVP M-9010 right mandible in medial view; **I**. *Diploglossus bilobatus* TNHC 31933 right splenial; **J**. *Anniella pulchra* TxVP M-8678 right splenial; **K**. *Ophisaurus attenuatus* TxVP M-8979 right splenial; **L**. *Gerrhonotus infernalis* TxVP M-13441 left splenial; **M**. *Barisia imbricata* TNHC 76984 left splenial. Scale bars = 1 mm. **Abbreviations**: a.m.f, anterior mylohyoid foramen; Spl, splenial; Spl.s, splen0/0/00 0:00:00 AMial spine.

*gutturosus* differs in having a relatively shorter splenial along the posterior half of the bone. The splenials of *Leiocephalus* and *Enyalioides heterolepis* differ from the fossil in being relatively longer and slenderer. *Anolis* differs in either lacking a separate and distinct splenial bone or having a short splenial [42]. Based on these differences with other NA pleurodontans, fossils were assigned to Phrynosomatidae.

**Compound bone.** *Description*. Morphotype A: TxVP 41229–27307 serves as the basis for our description and is a right compound bone that is missing the distal end of the prearticular (Fig 26F). The adductor fossa is elongate and open. The retroarticular process narrows posteriorly and is squared off at the posterior end. There is a medially oriented angular process with a lamina of bone connecting the retroarticular and angular processes. There is a medial crest (tympanic crest of [49]) that extends longitudinally along the dorsal surface of the

retroarticular process, and a longitudinal ridge on the ventral surface of the process. The medial process is tall. There is a small crest dorsally on the surangular and a distinct dentary articulation facet on the lateral surface. There are two posterior surangular foramina, one lateral to the medial process and the other slightly ventral, and one anterior surangular foramen lateral to the crest on the surangular. TxVP 41229–26996 is an unusual dermarticular with a distinct lateral notch on the retroarticular process (Fig 26H). The margin just lateral to the articular surface is developed into a small boss.

Morphotype B: TxVP 41229–26293 serves as the basis for our description (Fig 26G). TxVP 41229–26293 is a left compound bone that is missing the distal end of the prearticular. The adductor fossa is elongate and open. The retroarticular process is broad and flat. There is a posteromedially oriented angular process with a low dorsal ridge. There is no distinct medial crest and a ventral ridge on the retroarticular process. The medial process is tall. There is a small crest dorsally on the surangular and a distinct dentary articulation facet on the lateral surface.

*Identification*. Fossils are referred to Pleurodonta based on having a posteriorly directed retroarticular process that is not broadened posteriorly and that is lacking a widely opened adductor fossa [23]. *Polychrus* differs from the fossils in having the retroarticular process reduced [20]. The angular process does not project as far medially in *Dipsosaurus dorsalis*, *Sauromalus*, and *Iguana iguana* compared to the fossils (see also Fig 48 of [20]). The medial crest tends to be more distinct in *Dipsosaurus dorsalis*, *Iguana iguana*, *Ctenosaura similis*, and *Enyalioides* compared to crotaphytids [20]. In *Laemanctus* and *Corytophanes*, the angular process is shorter compared to the fossils and in *Basiliscus basiliscus* the medial crest is more distinct and the retroarticular process is thinner (see Fig 71 of [20]). Examined *Anolis* resemble fossils from Morphotype A in having a distinct angular process connected to the retroarticular process by a sheet of bone but differ in having a horizontally oriented posterior terminus of the retroarticular process. The posterior terminus of the retroarticular process is slightly angled in the fossils, similar to many phrynosomatids [43]. Additionally, examined *Anolis carolinensis* (also in *A. garmani*, see Fig 25 of [20]) have a broad depression on the dorsal surface of the surangular just anterior to the articular surface not seen in the fossils. Based on these differences with other NA pleurodontans, fossils were assigned to Phrynosomatidae.

## Sceloporinae Savage, 1958 [81]

Referred specimens: <u>Premaxilla</u>, 41229–26977, 41229–27523, 41229–28002, 41229–28207, 41229–28608, 41229–29096; <u>Quadrate</u>, 41229–26982 left, 41229–26983 left, 41229–26984 left, 41229–27058 left, 41229–27059 right, 41229–27525 left.

**Premaxilla.** *Description*. TxVP 41229–29096, TxVP 41229–26977, and TxVP 41229–27523 serve as the basis for our description (Fig 26I–26K). TxVP 41229–29096 is a premaxilla with six tooth positions with four preserved unicuspid teeth. It has a slightly rounded rostral surface and a long nasal process with distinct lateral nasal articulation facets. There is an ossified bridge extending laterally from the nasal process and enclosing the medial ethmoidal foramen on the right side and an incomplete bridge of bone on the left side. There are small foramina just lateral to the base of the nasal process that pierce ventrally. There are shallow maxillary facets laterally on the alveolar plate. Posteriorly, the palatal plate is steeply slanted with distinct vomer articulation facets. The incisive process is short, squared-off, and slightly bilobed. TxVP 41229–27523 differs from TxVP 41229–29096 in having seven tooth positions, a narrow nasal process with a small anterior nasal facet on the left side, and in lacking an ossified bridge. TxVP 41229–26977 has six tooth positions and differs from TxVP 41229–27523 and TxVP 41229–29096 in having a more narrow, rectangular alveolar plate.

*Identification*. TxVP 41229–29096 and TxVP 41229–26977 are assigned to Pleurodonta based on having a fused premaxilla [23] with less than seven tooth positions [29]. The seven tooth positions of TxVP 41229–27523 overlaps with other NA lizards including scincids, teiids, xantusiids, and *Anniella* [24, 29]. Scincids differ in having an unfused premaxilla [82], teiids differ in lacking an incisive process [37, 45, 83], xantusiids differ in having a much thinner nasal process and a broad, flat palatal plate [46], and *Anniella* have a more posteriorly directed nasal process [24, 84]. TxVP 41229–27523 is assigned to Pleurodonta. Among NA pleurodon-tans, *Anolis* and *Polychrus* differ from the fossils in having greater than seven tooth positions [20, 30]. Many members of Iguanidae differ in having multicuspid teeth on the premaxilla [31]. Unicuspid teeth are sometimes found in *Ctenosaura*, *Cyclura*, and *Sauromalus* [32], but these taxa differ in that the anterior rostral face of the premaxilla in *Ctenosaura* is distinctly rounded [32] and the nasal processes of *Ctenosaura*, *Cyclura*, and *Sauromalus* curve far poste-riorly [20, 32]. *Basiliscus*, *Corytophanes*, and *Enyaliodes* differ from the fossils in having a broader nasal process [29]. *Laemanctus* differs from the fossils in having a strongly tapering nasal process [20]. TxVP 41229–29096 differs from examined *Crotaphytus* in lacking facets on the nasal process in anterior view; however, facets may be absent in some species of *Gambelia* [30]. TxVP 41229–27523 has a small anterior nasal facet on the left side but differs from many *Crotaphytus* in having a more slender nasal process [49]. *Gambelia* differ from the fossils in having longer and sharper teeth and a vertical ridge on the posterior edge of the palatal plate [49]. Examined *Leiocephalus* differ from the fossil in having at least seven tooth positions (see also [33]), but *Leiocephalus personatus* was reported to have six tooth positions [29]. Further-more, examined *Leiocephalus* differ from the fossils in having a more triangular-shaped nasal process that tapers posterodorsally (see also [33]). Based on these differences with other NA pleurodontans, fossils were assigned to Phrynosomatidae.

Among phrynosomatids, anterior premaxillary foramina are lacking in *Sceloporus*, *Uro-saurus*, and *Petrosaurus*, are present in *Uma* and *Phrynosoma*, and are variably present in *Cal-lisaurus*, *Cophosaurus*, *Holbrookia*, and *Uta stansburiana* [32]. Some *Sceloporus* (e.g., *S. orcutii*) have an ossified bridge extending laterally from the nasal process and enclosing the medial eth-moidal foramen, but these foramina, when present, are positioned farther dorsolaterally com-pared to the anterior premaxillary foramina in other phrynosomatids. Sand-lizards (*Callisaurus*, *Cophosaurus*, *Holbrookia*, and *Uma*) typically have a flat rostral face of the pre-maxilla, while sceloporines (*Sceloporus*, *Urosaurus*, *Petrosaurus*, and *Uta*) have a more rounded rostral face [32]. Fossil premaxillae are assigned to Sceloporinae on that basis.

**Quadrate.** *Description*. TxVP 41229–27058, TxVP 41229–26982, and TxVP 41229–26983 serve as the basis for our description (Fig 26L and 26M). TxVP 41229–27058 is a left quadrate. There is a thin, medially slanted central column, and the bone narrows ventrally. There is a moderately developed medial crest but no pterygoid lappet. The conch is deep and gradually slants laterally from the central column. The cephalic condyle projects posteriorly without extensive dorsal ossification and the condyle has a small dorsal tubercle. The dorsolateral mar-gin of the tympanic crest is rounded. There is a foramen on the anteroventral surface (quadrate foramen of [73]), and a small hole on the dorsal margin of the conch that may be a taphonomic alteration. TxVP 41229–26983 and TxVP 41229–26982 differ from TxVP 41229–27058 in hav-ing a dorsal margin that slopes ventrolaterally. In TxVP 41229–26983, the dorsal margin slopes ventrolaterally in a step-like fashion.

*Identification*. Fossils share the absence of a distinct pterygoid lappet with geckos, *Scincella*, xantusiids, alopoglossids, some pleurodontans, and anguimorphs beside *Heloderma* [23, 24]. Geckos, xantusiids, *Scincella*, alopoglossids, and anguimorphs differ from the fossils in lacking a quadrate that has a dorsal surface much wider compared to the mandibular column. Among NA pleurodontans, in *Anolis* and *Polychrus* the lateral and medial margins are more parallel

compared to the fossils (Fig 16). Additionally, in *Anolis* there is a distinct boss at the ventrome-dial margin of the quadrate not seen in the fossils. The quadrate of *Basiliscus basiliscus*, *Coryto-phanes*, and *Laemanctus* is proportionally taller relative to its width compared to the fossils. The quadrate of *Enyalioides heterolepis* has a shallower conch and straighter lateral margin (tympanic crest) compared to the fossils. *Ctenosaura*, large *Iguana*, and *Leiocephalus* differ from the fossils in having a pterygoid lamina that extends farther medially. The quadrate of small *Iguana* is much more slender than the fossils. The quadrates of most *Sauromalus* have a ventrolaterally slanted dorsolateral margin, resembling TxVP 41229–26982 and TxVP 41229–26983, but differ in having a distinct dorsomedial expansion at the dorsal margin of the pterygoid lamina [24]. Some examined *Sceloporus* (*S. olivaceus* TxVP M-8375 and *S. orcutti* TxVP M-12155) also have a ventrolaterally slanted dorsolateral margin. Crotaphytids, *Uma*, and *Phrynososma* differ from the fossils in having a straight lateral margin (tympanic crest) [57]. Based on these differences with other NA pleurodontans, quadrates were assigned to Phrynosomatidae. The quadrate of phrynosomatines (except for *Uma* and *Phrynosoma*) differs from the fossils in having a nearly flat conch [24] and a strongly curved medial edge, including a medially curved central column. Based on these differences, fossils were assigned to Sceloporinae.

### *Urosaurus* Hallowell, 1854 [85]

Referred specimens: Dentary 41229–27137 left, 41229–27603 left.

   **Dentary.**   *Description*. TxVP 41229–27603 is a left dentary with 25 tooth positions (Fig 26P). Distal teeth are weakly tricuspid and slender. The Meckelian groove is fused for about 12 tooth positions and opens at the anterior end of the dentary. The dental shelf is narrow but widens slightly anteriorly. There is a small intramandibular lamella. The posterior end bears a distinct surangular and angular process and there is a coronoid facet within a dorsally projecting corner of bone (coronoid process). There are four nutrient foramina on the antero-lateral surface. TxVP 41229–27137 is missing the anterior and posterior ends (Fig 26O). In TxVP 41229–27137, the Meckelian groove is fused for nine tooth positions.

   *Identification*. Fossil dentaries share with Pleurodonta, Xantusiidae, and some gym-nophthalmids pleurodont tricuspid teeth and a fused Meckelian groove [24, 35]. Xantusiids differ in having a fused spleniodentary [24, 46, 63]. In gymnophthalmids with tricuspid teeth and a fused Meckelian groove, the groove is completely fused to almost the level of the poste-rior-most tooth position [62]. Among NA pleurodontans, the fossils share with some phryno-somatids, iguanids, anolids, extant *Leiocephalus*, polychrotids, and some corytophanids a fused Meckelian groove [20, 35]. Fossils differ from these taxa, with the exception of some phrynosomatids, in having weakly tricuspid teeth. Additionally, iguanids, *Leiocephalus*, *Basilis-cus*, *Corytophanes*, and some *Laemanctus* differ from fossils in having flared tooth crowns [29, 35] and *Polychrus* differ in having labial and lingual striations on the teeth [29]. On this basis, fossils were identified to Phrynosomatidae. Among phrynosomatids, having a fused Meckelian groove and weakly tricuspid teeth is a combination of features only found among *Urosaurus* [35, 86]. A fused Meckelian groove is occasionally present in other phrynosomatids (e.g., *Pet-rosaurus mearnsi* TxVP M-14910); but the teeth have more prominent accessory cusps.

### Phrynosomatinae Wiens et al. 2010 [87]

### Callisaurini "sand lizards" (*Callisaurus*, *Cophosaurus*, *Holbrookia*, and *Uma*) Wiens et al. 2010 [87]

Referred specimens: Premaxilla, 41229–25618, 41229–26027, 41229–26071, 41229–28865, 41229–29179; Quadrate, 41229–25586 right, 41229–26026 left.

**Premaxilla.**    *Description*. TxVP 41229–29179 and has seven tooth positions and unicuspid teeth (Fig 29A). It has a flat rostral surface with two anterior premaxillary foramina. The long, thin nasal process is missing the distal tip. There is a ventral keel separating the nasal facets. There are small foramina posteriorly on the palatal shelf. There are indistinct maxillary facets laterally on the alveolar plate. Posteriorly, the palatal plate is short and steeply slanted. The incisive process is minute and narrow. TxVP 41229–25618 differs from TxVP 41229–29179 in having a slightly narrower nasal process and distinct posterolaterally directed flanges on the palatal plate (Fig 29B). TxVP 41229–28865 differs in having eight tooth positions (Fig 29C).

*Identification*. Fossil premaxillae share with some pleurodontans, some amphisbaenians, some anguimorphs (*Ophisaurus*, gerrhonotines, and xenosaurids) anterior premaxillary foramina [19, 32, 52]. Examined amphisbaenians differ in having an enlarged median tooth [52]. Anguimorphs differ from the fossils in having a more distinctly bilobed and elongate incisive process [24]. Furthermore, *Xenosaurus* differ in having a rugose rostral surface of the premaxilla [19]. Fossils are assigned to Pleurodonta.

Among NA pleurodontans, crotaphytids and *Leiocephalus* differ in lacking anterior premaxillary foramina [29]. The South American hoplocercid *Enyalioides oshaughnessyi* was reported to have anterior premaxillary foramina [29], but they are absent in examined *E. heterolepis*. Many iguanids differ from the fossils in having multicuspid teeth on the premaxilla [31], and the remaining iguanids differ in having a rounded anterior rostral face of the premaxilla [32]. *Anolis*, *Polychrus*, and corytophanids also differ from the fossils in having a more rounded anterior rostral face of the premaxilla (see also [20]). Furthermore, *Basiliscus*, *Corytophanes*, and *Enyalioides* differ from the fossils in having a broader and parallel-sided nasal process [29, 52]. Based on differences with other NA pleurodontans, fossils were identified to Phrynosomatidae.

Among phrynosomatids anterior premaxillary foramina are known in *Uma*, *Phrynosoma*, *Callisaurus*, *Cophosaurus*, *Holbrookia*, and *Uta stansburiana* [32]. *Uta* differs from phrynosomatines in having a rounded rostral face of the premaxilla [32]. Fossil premaxillae are assigned to Phrynosomatinae based on the presence of anterior premaxillary foramina and a flat anterior rostral face of the premaxilla. Among phrynosomatines, *Phrynosoma* differs from sandlizards in having a nasal process that is directed dorsally and a nasal process that at its base is as wide as the body of the premaxilla [32, 88]. Premaxillae are assigned to the sand lizard clade based on having a nasal process that is directed posterodorsally and a nasal process that is narrower than the body of the premaxilla. The premaxilla of *Uma* differs from other sand-lizards in having a nasal process that resembles an isosceles triangle. Fossil premaxillae lack that morphology and so fossils likely represent *Callisaurus*, *Cophosaurus*, or *Holbrookia*.

**Quadrate.**    *Description*. TxVP 41229–26026 is a left quadrate (Fig 29D). The central column is indistinct, curves medially, and has a distinct medial groove. The bone narrows ventrally and there is no pterygoid lappet nor medial crest. The conch is shallow. The cephalic condyle projects posteriorly without extensive ossification dorsally. The dorsolateral margin of the tympanic crest is rounded. There is an anteriorly slanted dorsal tubercle. TxVP 41229–25586 differs in having an ossified squamosal foramen and having a dorsal surface with two depressions separated by a slanted medial ridge (Fig 29E).

*Identification*. Fossils share the absence of a distinct pterygoid lappet with geckos, *Scincella*, xantusiids, alopoglossids, some pleurodontans, and anguimorphs beside *Heloderma* [23, 24]. Geckos, xantusiids, *Scincella*, alopoglossids, and anguimorphs differ from the fossils in lacking a quadrate that has a dorsal surface much wider compared to the mandibular column. On this basis the fossil was identified to Pleurodonta. Among NA pleurodontans, the fossil shares the absence of a well-developed pterygoid lamina (medial concha of [29]) with *Anolis*, *Polychrus*, *Corytophanes*, *Laemanctus*, and some phrynosomatines [29, 56]. Corytophanids and *Polychrus*

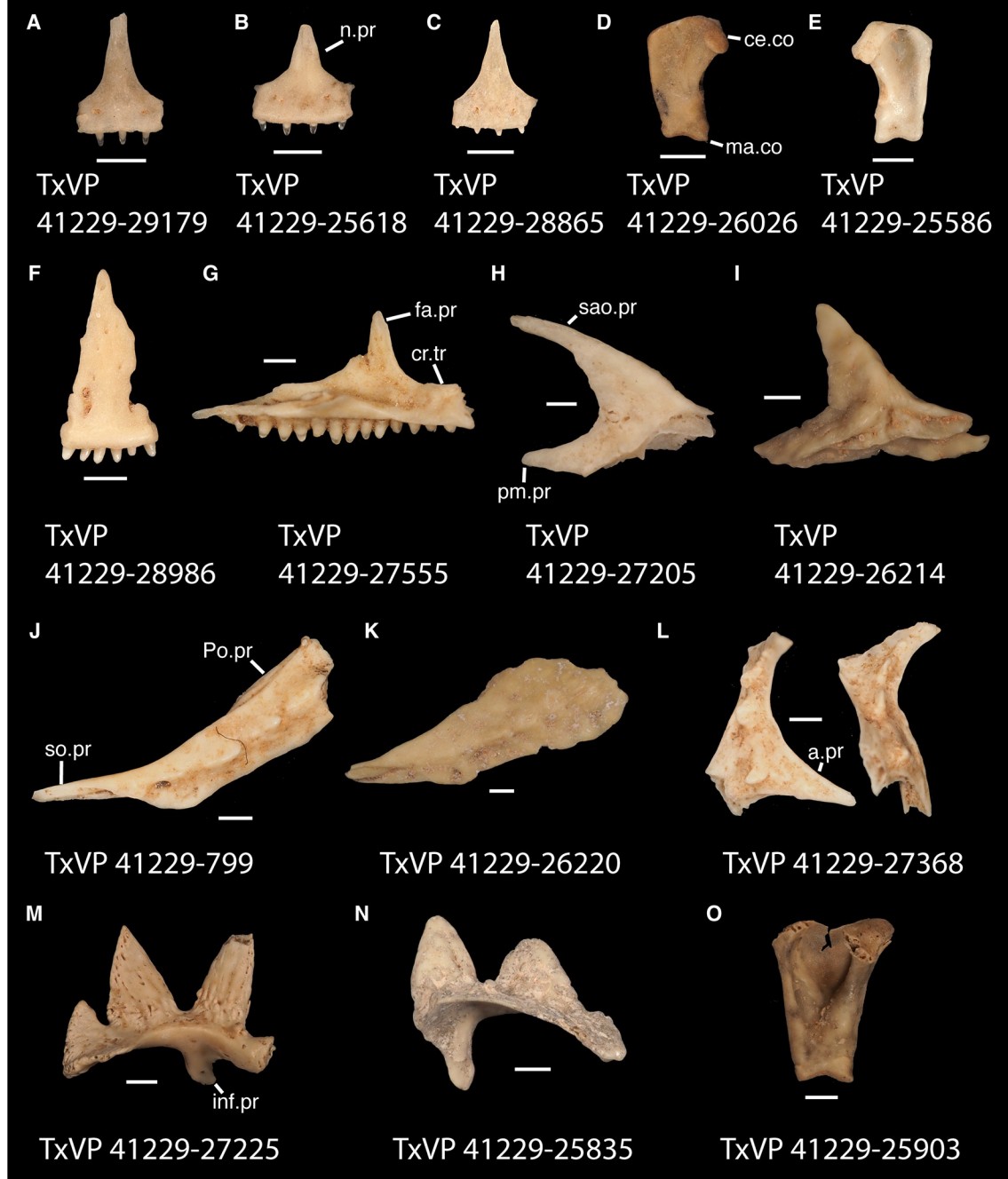

**Fig 29.** Fossil phrynosomatines, A–E: Phrynosomatinae, F–M & O: *Phrynosoma*, K & N: *Phrynosoma douglasii* species complex. **A**. TxVP 41229–29179 Anterior view of premaxilla; **B**. TxVP 41229–25618 Anterior view of premaxilla; **C**. TxVP 41229–28865 Anterior view of premaxilla; **D**. TxVP 41229–26026 Posterior view of left quadrate; **E**. TxVP 41229–25586 Posterior view of right quadrate; **F**. TxVP 41229–28986 Anterior view of premaxilla; **G**. TxVP 41229–27555 Medial view of left maxilla; **H**. TxVP 41229–27205 Dorsal view of left prefrontal; **I**. TxVP 41229–26214 Dorsal view of left prefrontal; **J**. TxVP 41229–799 Lateral view of left jugal; **K**. TxVP 41229–26220 Lateral view of left jugal; **L**. TxVP 41229–27368 Lateral and posterior view of postorbital; **M**. TxVP 41229–27225 Anterior view of squamosal; **N**. TxVP 41229–25835 Anterior view of squamosal; **O**. TxVP 41229–25903 Posterior view quadrate. Scale bars = 1 mm. **Abbreviations**: a.pr, anterior process; ce.co, cephalic condyle; cr.tr, crista transversalis; fa.pr, facial process; inf.pr, inferior process; ma.co, mandibular condyle; n.pr, nasal process; pm.pr, posteromedial process; Po.pr, postorbital process; sao.pr, supraorbital process; so.pr, suborbital process.

differ in having a low ridge within the conch (lateral concha of [29]). *Anolis* differs in having a distinct boss at the ventromedial margin of the bone below the articulation with the pterygoid. The fossils share with phrynosomatines (except for *Uma* and *Phrynosoma*) a nearly flat conch [24] and a strongly curved medial edge, including a medially curved central column. This combination of features is unique to sand lizards among NA pleurodontans. Fossils are assigned to the sand lizard clade on this basis. The quadrate of *Xenosaurus grandis* also has a strongly curved medial edge and a shallow conch but differs from phrynosomatines in having an expanded medial pterygoid lamina and a proportionally wider mandibular condyle.

## Phrynosomatidae Fitzinger, 1843 [67]

### Phrynosomatinae Wiens et al. 2010 [87]

*Phrynosoma* **Wiegmann, 1828 [89].** Illustrated specimens referenced in the text: <u>Compound bone</u>, 41229–26299 right; <u>Coronoid</u>, 41229–25848 left, 41229–26885 left, 41229–27341 left; <u>Dentary</u>, Morphotype A: 41229–27134 left, Morphotype B: 41229–28744 right; <u>Jugal</u>, 41229–799 left; <u>Maxilla</u>, 41229–27555 left; <u>Postorbital</u>, 41229–27368 right; <u>Prefrontal</u>, 41229–26214 left, 41229–27205 left; <u>Premaxilla</u>, 41229–28986; <u>Quadrate</u>, 41229–25903 left; <u>Squamosal</u>, 41229–27225 right; See S3 Table for complete list of specimens assigned to *Phrynosoma*.

**Premaxilla.** *Description.* TxVP 41229–28986 is a premaxilla with six tooth positions and unicuspid teeth (Fig 29F). It has a flat rostral surface with two anterior premaxillary foramina on the right side. The base of the nasal process is notched on the left side without enclosed anterior foramina. There is a long, wide nasal process with small, irregularly spaced anterior foramina. The nasal process has a ventral keel separating the nasal facets. There are shallow maxillary facets laterally on the alveolar plate. Posteriorly, the palatal plate is short and steeply slanted. The incisive process is missing.

*Identification.* Premaxillae are assigned to Pleurodonta based on having a fused premaxilla [23] with less than seven tooth positions [29]. Premaxillae are assigned to Phrynosomatinae based on the presence of less than seven tooth positions, unicuspid teeth, anterior premaxillary foramina, and a flat anterior rostral face of the premaxilla (see discussion above) [32]. Premaxillae can be referred to *Phrynosoma* based on a nasal process that is directly dorsally and a nasal process that at its base is as wide as the body of the premaxilla [32, 88].

**Maxilla.** *Description.* TxVP 41229–27555 serves as the basis for our description (Fig 29G). TxVP 41229–27555 is a left maxilla with 13 tooth positions. Teeth are unicuspid and slender throughout the tooth row. The facial process is a narrow projection that diminishes anteriorly to merge with the tall crista transversalis. The palatal shelf is flat, and the palatine process is asymmetrical. The superior alveolar foramen pierces a raised area lateral to the palatine process. There is a deep depression on the premaxillary process, housing two foramina, one anterior (subnarial arterial foramen) and one slightly posterior (opening for the superior alveolar canal). There are four lateral nutrient foramina.

*Identification.* Fossils were assigned to Pleurodonta based on the presence of two foramina on the premaxillary process [29]. The fossils share with *Phrynosoma* a facial process that is narrow anteroposteriorly and triangular to sub-triangular in shape [35, 90]. This morphology of the facial process is unlike that of any other NA lizard and fossils were assigned to *Phrynosoma* on that basis. Additionally, maxillae of most *Phrynosoma* differ from other pleurodontans in lacking multicuspid teeth and differ from many pleurodontans in having an asymmetrical palatine process in dorsal and ventral view [29]. The crista transversalis in many species of *Phrynosoma* is tall compared to other phrynosomatids [35].

**Prefrontal.** *Description.* Morphotype A: TxVP 41229–27205 is a left prefrontal that serves as the basis for our description (Fig 29H). It is triradiate with a small anterior orbital process, a

long posteromedial process, and a long supraorbital process. The supraorbital process thins distally and curves medially. The medial surface is characterized by deep articulation facets for the frontal posteriorly and the maxilla anteriorly. Dorsal to the frontal facet, there is a small groove for the nasal articulation. The dorsal surface has several distinct tubercles. The ventral surface is smooth, concave, and has a small anterior foramen.

Morphotype B: TxVP 41229–26214 is also a left prefrontal that serves as the basis for our description (Fig 29I). It is triradiate and the processes are nearly the same length. The anterior orbital process is notched at the anterior end for passage of the lacrimal foramen. The supraorbital process is directed posterolaterally and is relatively straight. The dorsal surface has a few low, round tubercles. The ventral surface is smooth, concave, and has a small foramen medially.

*Identification.* The prefrontal of *Phrynosoma* is distinct from nearly all NA lizards (except for *Corytophanes*) in having a long supraorbital process that extends posteriorly to enclose the supraorbital fenestra partially or fully [43, 56]. The prefrontal of *Corytophanes* differs from that of *Phrynosoma* in having a deep depression on the lateral surface of the bone. Fossils were identified to *Phrynosoma* on that basis.

Morphotype A: Fossils of morphotype A have a long supraorbital process that curves medially. A long supraorbital process also occurs in *P. cornutum*, *P. mcallii*, *P. solare*, *P. taurus*, *P. asio*, and somewhat in *P. braconnieri* [88, 91]. The supraorbital process of *P. solare* appears to be relatively longer compared to the fossils and other species of *Phrynosoma* [88, 91].

Morphotype B: Fossils of morphotype B have a shorter supraorbital process that is straighter. A short supraorbital process also occurs in *P. coronatum*, *P. cerroense*, *P. blainvilli*, *P. ditmarsi*, *P. modestum*, *P. obiculare*, *P. platyrhinos*, and species in the *P. douglasii* species complex [88, 91]. *P. modestum* differs from the fossil in having a more distinctly rugose surface and having a large anterior notch.

**Jugal.** *Description.* TxVP 41229–799 is a left jugal (Fig 29J). The anterior orbital process is thin with a ventral maxillary articulation facet. The postorbital process is broad and is directed posterodorsally and has a distinct anterior postorbital articulation facet. The lateral surface of the postorbital process has a row of three long tubercles. The medial surface is smooth, and there is a depression on the postorbital process. There is a foramen on the lateral surface near the inflection point.

*Identification.* Fossil jugals share with *Phrynosoma*, *Anolis*, *Corytophanes*, *Lepidophyma*, and *Xantusia riversiana* an expanded postorbital process [20, 46, 56, 88]. *Anolis* differs in having a relatively less expanded postorbital process [20]. Fossil jugals differ from *Anolis*, *Corytophanes*, *Xantusia riversiana*, and *Lepidophyma* in having a more posteriorly oriented postorbital process [20, 46, 56]. Fossils were identified to *Phrynosoma*.

Fossils have a row of long tubercles on the lateral surface. This is similar to the condition described for *P. ditmarsi*, *P. coronatum*, *P. solare*, *P. cornutum*, *P. modestum*, *P. mcallii*, and *P. platyrhinos* [88]. *Phrynosoma braconnieri*, *P. taurus*, *P. obiculare*, *P. douglassi*, and *P. hernandesi* have lower and more rounded tubercles compared to the fossils [88]. We were unable to examine this feature in all species of *Phrynosoma* and do not make species-level identifications.

**Postorbital.** *Description.* TxVP 41229–27368 serves as the basis for our description and is a right postorbital (Fig 29L). The dorsal process widens mediolaterally, and projects anteriorly with a lateral facet for articulation with the frontal and parietal. Laterally, the postorbital is twisted with a long, ventrally pointed process. Posterolaterally, there is a squamosal articulation facet, and the lateral margin has a groove for articulation with the jugal. The dorsal surface has several long tubercles. Ventrally there is a ridge that runs from the posteromedial corner onto the ventrolateral process.

*Identification*. Postorbitals were identified to Pleurodonta based on a sub-triangular morphology with a distinct ventral process [23]. Among NA pleurodontans, a broadened dorsal process is found in *Phrynosoma*, *Iguana iguana*, and *Corytophanes* [24, 43, 56]. The dorsal process in examined *Iguana iguana* is not expanded to the extent seen in the fossils. The postorbital of *Corytophanes* differs in having an anteriorly projecting spine extending from the dorsal process that contacts the prefrontal [56]. On that basis, fossils were identified to *Phrynosoma*. The fossils share with *P. solare*, *P. braconnieri*, *P. cornutum*, *P. coronatum*, *P. ditmarsi*, *P. mcallii*, *P. modestum*, *P. obiculare*, *P. platyrhinos*, and *P. taurus* a rugose lateral surface with distinct tubercles [88]. *Phrynosoma mcallii* differs in having a posteriorly expanded postorbital that restricts the supratemporal fossa [92].

**Squamosal.**  *Description*. TxVP 41229–27225 is a right squamosal that serves as the basis for our description (Fig 29M). The anterior edge is concave with a posteromedial process that is broken at the distal end. The dorsal surface has two long posterior horns and one short anterior horn. There is a short posteroventrally projecting inferior process (*sensu* [91]) with a lateral facet for the supratemporal.

*Identification*. The squamosal of *Phrynosoma* is unique among North American lizards in having horns along the dorsal surface. Fossils were identified to *Phrynosoma* on that basis. The length of the horns relative to the overall size of the squamosal as well as the number of horns varies among extant *Phrynosoma*; however, there is considerable overlap in both relative length and number of horns among extant species [43, 66, 93]. Fossil squamosals have relatively long horns similar to *P. cornutum*, *P. modestum*, *P. asio*, *P. coronatum*, *P. mcallii*, *P. platyrhinos*, and *P. solare* [94]. *Phrynosoma asio* differs in only having two horns and *P. solare* has four [88, 93]. *Phrynosoma modestum* and *P. platyrhinos* differ in having a long posterior horn and shorter anterior horns [66, 94]. Based on those features, fossils are most similar to *P. cornutum*, *P. coronatum*, *P. mcallii*, and *P. platyrhinos*. Species of *Phrynosoma* reportedly differ in the spacing between horns; however, ontogenetic variation in that feature was noted in at least *P. coronatum* [66]. More information on intraspecific variation in that feature among species of *Phrynosoma* is needed before a more specific identification can be made.

**Quadrate.**  *Description*. TxVP 41229–25903 is a left quadrate that serves as the basis for our description (Fig 29O). There is a wide, medially slanted central column, and the bone narrows ventrally. There is no pterygoid lappet, and the medial crest is minute. The conch is deep and is obscured from view at its dorsomedial margin by the central column. The cephalic condyle projects posteriorly without extensive dorsal ossification. The dorsolateral margin of the tympanic crest is slightly rounded. The anterior surface is convex. There is a foramen on the anteroventral surface (quadrate foramen of [73]) and a foramen medial to the central column.

*Identification*. Fossils share the absence of a distinct pterygoid lappet with geckos, *Scincella*, xantusiids, alopoglossids, some pleurodontans, and anguimorphs besides *Heloderma* [23, 24]. Geckos, xantusiids, *Scincella*, alopoglossids, and anguimorphs differ from the fossils in lacking a quadrate that has a dorsal surface much wider compared to the mandibular column. The quadrate of *Phrynosoma* (except *P. modestum* and *P. mcallii*) differs from other pleurodontans in having a conch that is deep and obscured from view at its dorsomedial margin by the central column. In examined *P. modestum* and *P. mcallii* the conch is shallow. The fossil shares with *P. cornutum*, *P. coronatum*, *P. mcallii*, *P. modestum*, *P. platyrhinos*, and *P. solare* a minute medial crest.

**Dentary.**  *Description*. Morphotype A: TxVP 41229–27134 serves as the basis for our description and is a left dentary with 19 tooth positions (Fig 30A). Teeth are unicuspid and slender. The Meckelian groove is open medially for its entire length, and the dentary is tall posteriorly. The suprameckelian lip is short anteriorly. The dental shelf is narrow and there is an intramandibular lamella. The posterior end of the dentary is bifurcated. The angular process is

broken at the distal end but is flat ventrally and curves far medially. There is a small lateral tubercle at the base of the angular process. There are seven nutrient foramina on the anterolateral surface.

Morphotype B: TxVP 41229–28744 serves as the basis for our description and is a right dentary with 20 tooth positions (Fig 30B). Teeth are unicuspid and slender. The suprameckelian and inframeckelian lips meet midway along the tooth row, and the dentary is tall posteriorly. The suprameckelian lip is short anteriorly. The dental shelf is narrow. The angular process is flat ventrally and curves far medially. The lateral surface is smooth and lacks tubercles, and there are five nutrient foramina anteriorly.

*Identification*. Morphotype A: Dentaries of Morphotype A share with Pleurodonta and some teiids pleurodont teeth and an inframeckelian lip that curls dorsolingually, producing a medial exposure of the Meckelian groove along the mid-length of the dentary [2, 27]. Fossil dentaries differ from teiids in lacking a broad subdental shelf [23], lacking asymmetric bicuspid teeth, and lacking large amounts of cementum deposits at base of teeth [23, 64, 74]. Fossils are assigned to Pleurodonta. *Phrynosoma* differ from other NA pleurodontans in having unicuspid teeth (except for some slightly tricuspid posterior teeth in *P. asio*, *P. coronatum*, and *P. mcallii*), a relatively flat posteroventral surface, and a posteroventral lamina of bone that is curved far medially [95]. Fossils are assigned to *Phrynosoma* on that basis. Fossils of morphotype A share with *P. asio*, *P. cornutum*, *P. mcallii*, *P. modestum*, and *P. platyrhinos* an open Meckelian groove [43]. Of those species, *P. asio* differs in having a smooth lateral surface [43, 95]. *Phrynosoma modestum* was reported to have a smooth lateral surface by Presch; however, other authors noted that *P. modestum* has a rugose lateral surface [88, 94, 95], which is supported by our observations.

Morphotype B: Dentaries of Morphotype B share with Pleurodonta, some teiids, and some scincids pleurodont teeth and suprameckelian and inframeckelian lips that meet to close the Meckelian groove [2, 27, 35, 96]. Fossil dentaries differ from teiids and scincids in lacking a broad subdental shelf [23]. Fossils further differ from teiids in lacking asymmetric bicuspid teeth and lacking large amounts of cementum deposits at base of teeth [23, 64, 74]. Fossils are assigned to Pleurodonta. Fossils of morphotype B share a closed Meckelian groove with *P. braconnieri*, *P. coronatum*, *P. orbiculare*, *P. solare*, and species in the *P. douglasii* species complex [43]. We also observed one specimen of *P. modestum* (TxVP M-14818) with a closed Meckelian groove. *Phrynosoma solare* and examined *P. modestum* differ in having a rugose lateral surface of the dentary [43, 88].

**Coronoid.** *Description*. TxVP 41229–27341 is a left coronoid (Fig 30C). The coronoid process is tall and wide, and gradually declines anteriorly, giving the coronoid a triangular appearance. The anteromedial process is elongated, with a medial articulation facet for the dentary, and a small ventrally-projecting lamina of bone. The posteromedial process is thin and directed posteroventrally. There is a distinct medial crest that extends from the coronoid process onto the posteromedial process. There is a small, rounded lateral process. There is a vertically-oriented lateral crest that terminates at the posterior margin of the lateral process. The dorsal articulation facet for the surangular is narrow. TxVP 41229–26885 and TxVP 41229–25848 differ in having lateral crests that end anteriorly on the lateral process.

*Identification*. Fossil coronoids share with several pleurodontans and xantusiids the absence of a distinct anterolateral process [23]. Xantusiids differ in having an anterior groove extending onto the coronoid process [46]. Fossils differ from examined NA pleurodontans, except for some *Phrynosoma* (see Fig 4 of [97]), in having a triangular-shaped coronoid process that is sloped at a low angle anteriorly. Furthermore, fossils differ from examined NA pleurodontans, except for some *Phrynosoma* (e.g., *P. cornutum* TxVP M-6405 and *P. douglasii* TxVP M-8526), in having a thin, posteriorly directed posteromedial process. Fossils were assigned to

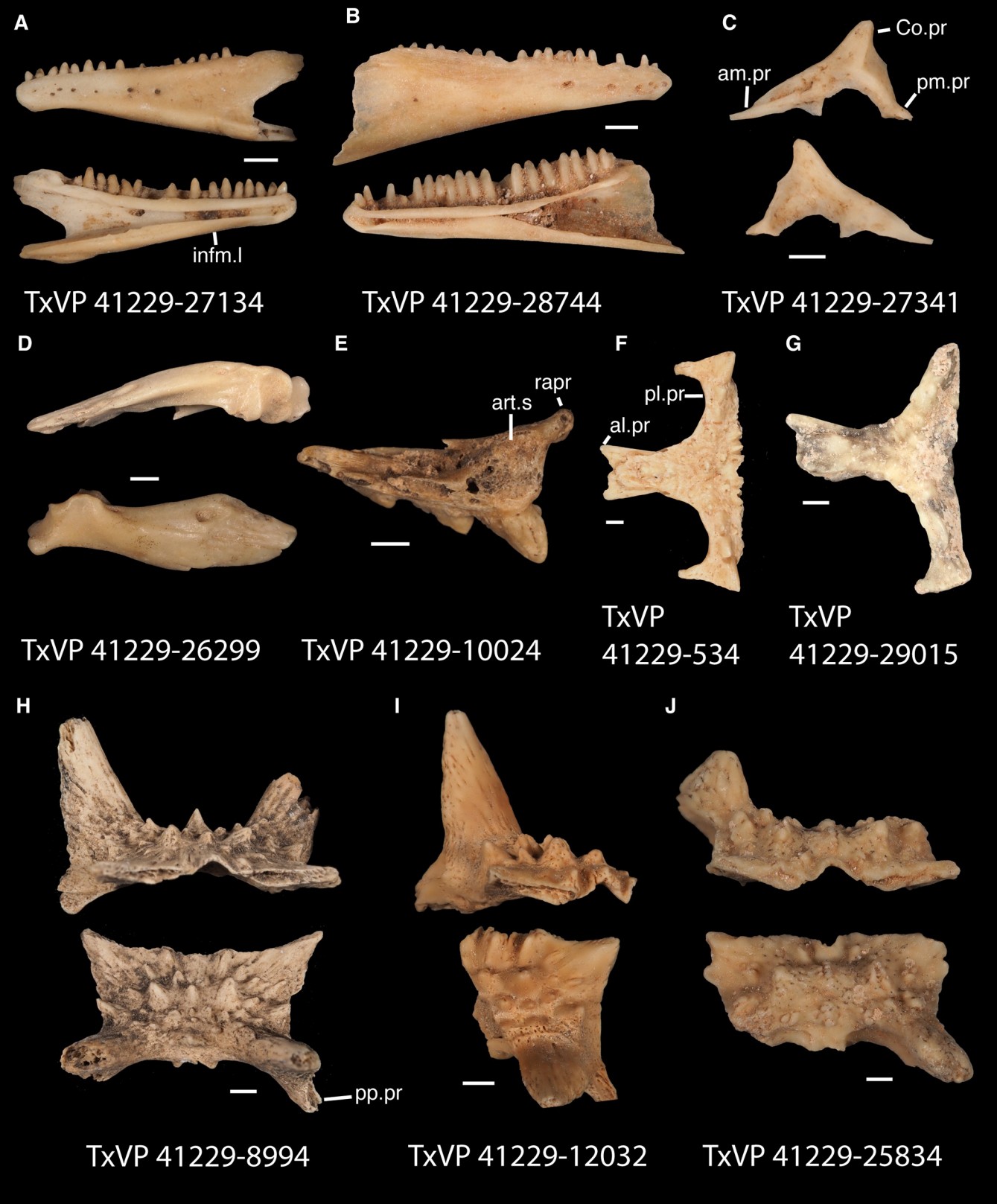

**Fig 30.** Fossil phrynosomatines, A–D: Phrynosoma, E–F & H: *Phrynosoma cornutum*, G & J, *Phrynosoma douglasii* species complex. **A**. TxVP 41229–27134 Lateral and medial view of dentary; **B**. TxVP 41229–28744 Lateral and medial view of dentary; **C**. TxVP 41229–27341 Lateral and medial view of coronoid; **D**.

TxVP 41229–26299 Dorsal and lateral of compound bone; **E**. TxVP 41229–10024 Dorsal view of compound bone; **F**. TxVP 41229–534 Dorsal view of frontal; **G**. TxVP 41229–29015 Dorsal view of frontal; **H**. TxVP 41229–8994 Anterior and dorsal view of parietal; **I**. TxVP 41229–12032 Anterior and dorsal view of parietal; **J**. TxVP 41229–25834 Anterior and dorsal view of parietal. Scale bars = 1 mm. **Abbreviations**: al.pr, anterolateral process; am.pr, anteromedial process; art.s, articular surface; Co.pr, coronoid process; infm.l, inframeckelian lip; pl.pr, posterolateral process; pm.pr, posteromedial process; pp.pr, postparietal process; rapr, retroarticular process.

*Phrynosoma* based on those differences listed above. Fossils differ from *P. braconnieri*, *P. mcallii*, and *P. solare* in having a lateral process to overlap the dentary and surangular. Examined *P. coronatum*, *P. ditmarsi*, *P. orbiculare*, and *P. hernandesi* differ in having a more steeply sloped anterior margin of the coronoid process (see Fig 4 of [97]).

**Compound bone.** *Description*. TxVP 41229–26299 serves as the basis for our description and is a right compound bone that is missing the prearticular (Fig 30D). There is a broad articular surface and a short medial process. The retroarticular process is mediolaterally flat and slanted ventrolaterally with a rounded lateral end. The lateral surface of the surangular is smooth, the dorsal margin has a dorsally expanded crest where it articulates with the coronoid, and the ventral margin is slightly convex. There is a posterior surangular foramen and an anterior surangular foramen on the lateral surface.

*Identification*. The compound bone of *Phrynosoma* is unlike that of other North American lizards in having a retroarticular process that is twisted in a near vertical plane with dorsal and ventral tubercles on the posterior end [43]. Fossils were assigned to *Phrynosoma* on that basis. The fossil has a flat lateral surface without horns, similar to that in *P. asio*, *P. cerroense*, *P. coronatum*, *P. orbiculare*, *P. taurus*, *P. braconnieri*, and species in the *P. douglasii* species complex [43].

### *Phrynosoma cornutum* [98]

Illustrated specimens referenced in the text: Compound bone, 41229–10024 left; Frontal, 41229–534; Parietal, 41229–8994; See S3 Table for complete list of specimens assigned to *Phrynosoma cornutum*.

**Frontal.** *Description*. TxVP 41229–534 serves as the basis for our description (Fig 30F). TxVP 41229–534 is a frontal with two posterolateral processes on either side, each with a small anteriorly projecting anterior superciliary process (*sensu* [88]). The frontal is dorsoventrally tall near the base of the posterolateral processes. There are many tall, peaked tubercles on the dorsal surface beginning midway at the interorbital region and extending to the posterior edge. There are long superciliary horns on the posterolateral process. Anteriorly, there are bilateral facets for articulation with the nasal and prefrontal. The cristae cranii do not project far ventrally. There is a small midline notch on the posterior edge of the bone, and posterolaterally there are deep parietal facets.

*Identification*. Fossils share with Pleurodonta and Gymnophthalmoidea a fused frontal with reduced cristae cranii [23]. Fossils differ from all other NA lizards, except for *Phrynosoma*, in having a supraorbital process extending anteriorly from each posterolateral portion of the frontal [43] Fossils are assigned to *Phrynosoma* on that basis. A major difference between frontals of morphotype A and B is the morphology of the ornamentation. Fossils of both morphotypes differ from *Phrynosoma asio*, which has a smooth dorsal surface, and from *P. mcallii* which has dorsal tubercles over nearly the entire length of the frontal [88].

Fossils most closely resemble species including *P. cornutum*, *P. modestum*, *P. platyrhinos*, *P. goodei*, *P. coronatum* (intraspecifically variable [88]), *P. blainvillii*, *P. cerroense*, *P. ditmarsi*, *P. taurus*, *P. braconnieri*, and *P. obiculare* in having peaked tubercles on the posterolateral process [88]. *Phrynosoma modestum*, *P. platyrhinos*, and *P. goodei* differ from the fossils in having relatively shorter superciliary horns (see Fig 1 of [91]). Furthermore, *P. modestum* differs in having

an anterolaterally-oriented supraorbital process. *Phrynosoma coronatum*, *P. blainvillii*, *P. cerroense*, *P. ditmarsi*, *P. modestum*, *P. platyrhinos*, and *P. obiculare* differ from the fossils in having relatively shorter tubercles and peaked tubercles largely restricted to the posterolateral process [88, 91]. The peaked tubercles appear to be shorter in *P. braconnieri* compared to the fossils, and *P. taurus* has longer superciliary horns [91]. The fossils are referred *P. cornutum* based on those differences; however, we did not examine several species of *Phrynosoma* (e.g., *P. sherbrookei*), so this identification is tentative.

**Parietal.** *Description.* TxVP 41229–8994 serves as the basis for our description (Fig 30H). The fossil is nearly complete, only missing the distal end of the left lateral horn and the left postparietal process. The parietal table is rectangular with distinct anterolateral processes. The anterior edge has deep, bilateral facets for the frontal. The dorsal surface is covered in long tubercles. The right posterolateral parietal horn is long and there is a small horn between the main posterolateral horns. The posterior edge between the postparietal processes is characterized by two depressions (nuchal fossae) separated by a small midline posterior projection. The right postparietal process is short and projects posteroventrally and bears a lateral facet for the squamosal and a medial facet for the paroccipital process of the exoccipital. The ventrolateral crests are low without distinct epipterygoid processes. On the ventral surface, there are shallow depressions (cerebral vault) divided by a low ridge. There is no pit for the processus ascendens.

*Identification.* The broad, rectangular shape of the parietal and the presence of posterolateral dorsal horns differentiate *Phrynosoma* from those of all other North American lizards [43] Fossils were identified to *Phrynosoma* on that basis.

The length of the parietal horns relative to the size of parietal table varies among extant *Phrynosoma*. Fossil parietals have relatively long lateral parietal horns, a small medial horn, and long dorsal bony tubercles. The relative length of the lateral horns of fossils is similar to *P. cornutum*, *P. solare*, *P. mcallii*, *P. platyrhinos*, *P. goodei*, *P. blainvillii*, *P. cerroense*, and *P. coronatum* [66, 94] (see also Fig 1 of [91]). It was previously suggested that a few fossil parietals (e.g., TxVP 41229–12032, Fig 30I) may represent *P. modestum* [10]; however, *P. modestum*, *P. obiculare*, *P. braconnieri*, and *P. asio* have slightly shorter lateral horns, and *P. modestum* and *P. asio* lack the long dorsal tubercles present in the fossil [86, 88, 94]. Additionally, *P. modestum* lacks a small medial horn, as do *P. asio* and some *P. platyrhinos* [43, 66]. *Phrynosoma mcallii* differ from the fossils in having short dorsal tubercles, and *P. solare* differs in having four well developed parietal horns [88]. The relatively long dorsal tubercles present in fossil parietals are similar to *P. cornutum*, *P. coronatum*, *P. solare*, *P. blainvillii*, *P. cerroense*, *P. braconnieri*, and *P. orbiculare*, but the latter two species have relatively shorter parietal horns compared to the fossils. Based on those features, fossils are most similar to *P. cornutum*, *P. coronatum*, *P. blainvillii*, and *P. cerroense*. The lateral horns of *P. coronatum* differ from the fossil in slightly curving posteriorly, and *P. coronatum* have a relatively longer medial horn. *Phrynosoma blainvillii* differs in having the anterolateral processes not extend as far laterally compared to the fossil and *P. cerroense* has fewer bony tubercles on the dorsal surface [99]. The fossils are identified to *P. cornutum* based on those differences.

**Compound bone.** *Description.* TxVP 41229–10024 is a left compound bone missing the prearticular and the anterior end of the surangular that serves as the basis for our description (Fig 30E). There is a broad articular surface. The retroarticular process (post-condylar process of [43]) is mediolaterally flat, tall, and oriented in a near vertical plane. The posteroventral end of the retroarticular process is rounded and extends posterior to the dorsal portion. On the surangular there are three lateral horns. The posterior-most horn is long and narrow, and the other horns are short and broad.

*Identification.* The compound bone of *Phrynosoma* is unlike that of other North American lizards in having a retroarticular process that is twisted in a near vertical plane with dorsal and

ventral tubercles on the posterior end [43] Fossils were assigned to *Phrynosoma* on that basis. The fossil share with *P. cornutum*, *P. ditmarsi*, *P. modestum*, *P. mcallii*, *P. platyrhinos*, *P. goodei*, and *P. solare* horns on the lateral surface [43, 88]. *Phrynosoma ditmarsi* differs in having the horns oriented more ventrally [88] (see also Fig 4 of [97]). *Phrynosoma modestum* differs in having shorter horns relative to the fossils, and *P. platyrhinos* and *P. solare* differ in only having two lateral horns [43] The lateral horns of *P. mcallii* are relatively thinner compared to the fossil and the posterior two horns in *P. mcallii* are not as distinctly different in length as in the fossil. Fossils have a comparable horn morphology to *P. cornutum*. Fossils are assigned to *P. cornutum* based on those differences; however, we did not examine several species of *Phrynosoma* (e.g., *P. sherbrookei*), so this identification is tentative.

### *Phrynosoma douglasii* species complex

Illustrated specimens referenced in the text: Frontal, 41229–29015; Jugal, 41229–26220 left; Parietal, 41229–25834; Squamosal, 41229–25835 left; See S3 Table for complete list of specimens assigned to the *Phrynosoma douglasii* species complex.

**Frontal.** *Description*. TxVP 41229–29015 serves as the basis for our description (Fig 30G). The fossil is mostly complete but is missing the distal end of the right posterolateral process and the left prefrontal and nasal facets. The bone is tall near the base of the posterolateral processes. The left posterolateral process has a small anteriorly projecting anterior superciliary process. There are low rounded tubercles posteriorly on the dorsal surface, and the superciliary horns on the posterolateral process are short. Anteriorly, there is a facet for articulation with the nasal and prefrontal. The cristae cranii are short and anteriorly border a deep depression on the ventral surface.

*Identification*. Fossils share with Pleurodonta and Gymnophthalmoidea a fused frontal with reduced cristae cranii [23]. Fossils differ from all other NA lizards, except for *Phrynosoma*, in having a supraorbital process extending anteriorly from each posterolateral portion of the frontal [43] Fossils are assigned to *Phrynosoma* on that basis. Fossils differ from *Phrynosoma asio*, which has a smooth dorsal surface, and from *P. mcallii* which has dorsal tubercles over nearly the entire length of the frontal [88].

Fossils share with species in the *P. douglasii* species complex low rounded tubercles posteriorly on the dorsal surface [88]. It was previously suggested that several fossil frontals may represent *P. modestum* [10]. However, *P. modestum* has shorter superciliary horns and more distinctively rugose tubercles compared to the fossils and examined species in the *P. douglasii* species complex. The fossils are identified to the *P. douglasii* species complex based on those differences, but we were unable to examine all species of *Phrynosoma* making this identification tentative.

**Parietal.** *Description*. TxVP 41229–25834 serves as the basis for our description (Fig 30J). TxVP 41229–25834 is a parietal missing the left posterolateral portion and the right postparietal process. The parietal table is rectangular. The anterior edge has a small midline notch between deep, bilateral facets for the frontal. The dorsal surface is covered in rounded tubercles. The right posterolateral parietal horn is short. The lateral surface has a deep depression for articulation with the squamosal. The posterior edge between the postparietal processes is characterized by two depressions (nuchal fossae) separated by a small midline posterior projection. The ventrolateral crests are low without distinct epipterygoid processes. On the ventral surface there are shallow depressions (cerebral vault) divided by a low ridge. There is no pit for the processus ascendens.

*Identification*. The broad, rectangular shape of the parietal and the presence of posterolateral dorsal horns differentiate *Phrynosoma* from those of all other North American lizards [43] Fossils were identified to *Phrynosoma* on that basis.

Fossil parietals share with *P. taurus* and species in the *P. douglasii* species complex short lateral horns relative to the size of the parietal table with no medial horn and rounded tubercles on the dorsal surface [66, 88, 94]. The tubercles in *P. taurus* are more numerous and pointed compared to the fossils (see Fig 1 of [91]) and the closely related *P. sherbrookei* has relatively longer parietal horns [100]. The fossils are identified to the *P. douglasii* species complex based on those differences.

**Jugal.** *Description.* TxVP 41229–26220 serves as the basis for our description and is a well preserved left jugal (Fig 29K). TxVP 41229–26220 differs from morphotype A in having the anterior orbital process widened relative to the postorbital process, a broad, flat posterodorsal surface in lateral view, and the lateral surface of the postorbital process has a row of several low, rounded tubercles.

*Identification.* Fossil jugals share with *Phrynosoma*, *Anolis*, *Corytophanes*, *Lepidophyma*, and *Xantusia riversiana* an expanded postorbital process [20, 46, 56, 88]. *Anolis* differs in having a relatively less expanded postorbital process [20]. Fossil jugals differ from *Anolis*, *Corytophanes*, *Xantusia riversiana*, and *Lepidophyma* in having a more posteriorly oriented postorbital process [20, 46, 56]. Fossils were identified to *Phrynosoma*.

Fossils share with *P. orbiculare* and species in the *P. douglasii* species complex a broad, flat posterodorsal surface in lateral view [88]. Fossils have a row of several low, rounded tubercles similar to the condition described for *P. taurus*, *P. braconnieri*, and species in the *P. douglasii* species complex. *Phrynosoma taurus* and *P. braconnieri* differ in having more triangular-shaped jugals [88]. *Phrynosoma orbiculare* also has low, rounded tubercles anteriorly, but may have a more distinct posterior tubercle [88]. The fossils are identified to the *P. douglasii* species complex based on those differences, but we were unable to examine features on the jugal for all species of *Phrynosoma* making this identification tentative.

**Squamosal.** *Description.* TxVP 41229–25835 is a left squamosal missing the posteromedial process and a portion of the anterior edge (Fig 29N). The dorsal surface has two short posterior horns and one minute anterior horn. Posteriorly, there is a short posteroventrally projecting inferior process (*sensu* [91]) with a lateral facet for the supratemporal.

*Identification.* The squamosal of *Phrynosoma* is unique among North American lizards in having horns along the dorsal surface [43] Fossils were identified to *Phrynosoma* on that basis. On the fossil, the lengths of the posterior horns relative to the overall size of the element are similar to *P. braconnieri*, *P. ditmarsi*, and species in the *P. douglasii* species complex [91, 94]. *Phrynosoma braconnieri* and *P. ditmarsi* differ in having a relatively broader squamosal in dorsal view [91]. The fossils are identified to the *P. douglasii* species complex based on those differences, but we were unable to examine all species of *Phrynosoma* making this identification tentative.

## Anguimorpha Fürbringer, 1900 [101]

### Anguidae Gray, 1825 [102]

Referred specimens: <u>Compound bone</u>, 41229–27094 right; <u>Coronoid</u>, 41229–27599 left; <u>Dentary</u>, 41229–27980 right, 41229–28249 right.

**Coronoid.** *Description.* TxVP 41229–27599 is a left coronoid (Fig 31A). The coronoid process is tall, broad, and rounded. The anteromedial process is elongate, but the distal end is missing. The anteromedial process has a medial splenial facet that, with the anteriorly projecting lateral process, forms a narrow facet for the coronoid process of the dentary. The posteromedial process is missing the distal end, but the remaining portion is posteroventrally directed with an expanded posterior lamina of bone to articulate medially with the surangular. The medial crest extends from the coronoid process onto the posteromedial process. There is a

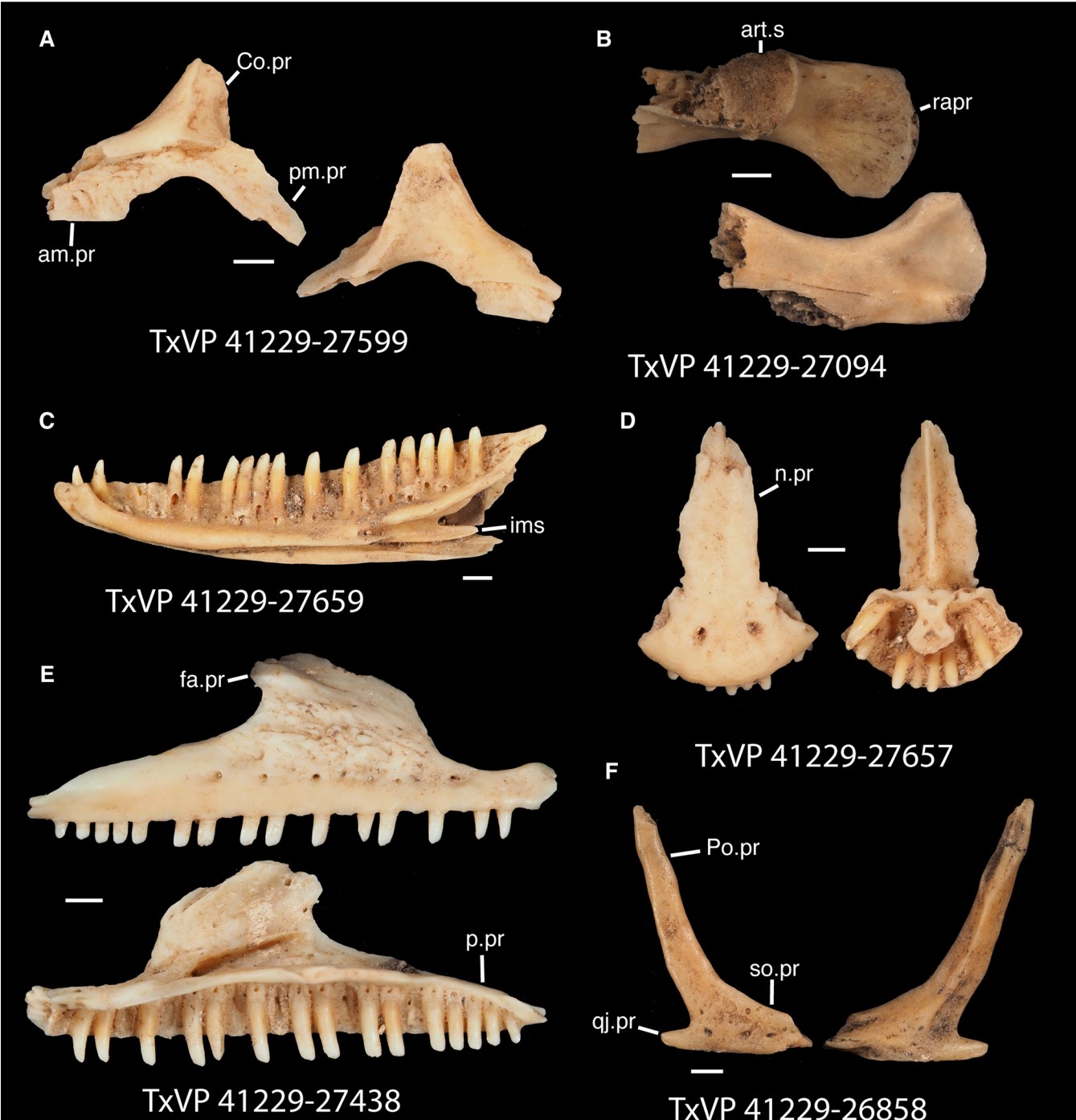

**Fig 31.** Fossil anguids, A–B: Anguidae, C: Gerrhonotinae, D & F: (*Gerrhonotus*, (*Barisia*, *Abronia*)), E: *Gerrhonotus*. **A**. TxVP 41229–27599 Lateral and medial view of left coronoid; **B**. TxVP 41229–27094 Dorsal and ventral view of right compound bone; **C**. TxVP 41229–27659 Medial view of right dentary; **D**. TxVP 41229–27657 Dorsal and ventral view of premaxilla; **E**. TxVP 41229–26858 Lateral and medial view of right jugal; **F**. TxVP 41229–27438 Lateral and medial view of right maxilla. Scale bars = 1 mm. **Abbreviations**: am.pr, anteromedial process; art.s, articular surface; Co.pr, coronoid process; fa.pr, facial process; ims, intramandibular septum; n.pr, nasal process; Po.pr, postorbital process; p.pr, posterior process; qj.pr, quadratojugal process; rapr, retroarticular process; so.pr, suborbital process.

vertically oriented lateral crest that ends at the posterior margin of the lateral process. The anteromedial process has a notch, which may convey a foramen.

*Identification*. The fossil coronoid differs from snakes in having distinct anteromedial and anterolateral processes that serve to clasp the dentary [23]. The fossil is further differentiated from snakes in having an expansive ventral articulation surface that is curved to articulate with both the dorsal and lateral surfaces of the surangular [27]. The fossil coronoid differs from xantusiids and some pleurodontans in having an anterolateral process [23] (Fig 32). The fossil differs from examined NA pleurodontans with an anterolateral process (see crotaphytid coronoid section above) in having a more posteriorly oriented posteromedial process. TxVP 41229–27599 differs from examined *Aspidoscelis* and *Ameiva* in lacking a deeply notched posterior edge, forming dorsal and ventral rami [53]. Furthermore, *Aspidoscelis*, *Ameiva*, and *Pholidoscelis* differ in having a distinct lateral crest running from the apex of the coronoid process anteroventrally onto the anterolateral process [2, 53]. Fossils differ from gymnophthalmids and alopoglossids in having a relatively shorter anterolateral process [37, 62, 103]. North American geckos such as *Coleonyx variegatus*, *C. brevis*, *Sphaerodactylus roosevelti*, and *Thecadactylus rapicauda* differ from the fossil in having a thinner posteromedial process [54, 104, 105]. The posteromedial process is slightly wider in *C. elegans* and *C. mitratus* compared to other species [54] but examined *Coleonyx*, including *C. elegans*, have a coronoid process with a posterior margin that is sloped at a lower angle compared to the fossil. The coronoid of *Coleonyx variegatus* and *C. brevis* also differ from the fossil in having a dorsoventrally expanded anterolateral process [54]. The lateral crest on the coronoid process is almost vertical in oriented in the fossil, but in examined *Plestiodon* and *Scincella* the crest is more obliquely oriented. Furthermore, the anterolateral process is more anteriorly oriented in the fossil but more ventrally oriented in *Plestiodon* (see also Fig 4 of [47]). Based on differences from other NA lizards, the fossil coronoid is referable to Anguimorpha. *Xenosaurus*, except for *X. rackhami*, differs from the fossil in having a foramen on the anterolateral process [19, 24]. *Heloderma* differs in having a dorsoventrally expanded anterolateral process and *Anniella* differs in having a shorter coronoid process [24]. Based on these differences with other anguimorphs, the fossil was identified to Anguidae.

**Compound bone.**   *Description*. TxVP 41229–27094 is a right compound bone missing much of the portion anterior to the articular surface (Fig 31B). The retroarticular process is broadened and medially oriented. The dorsal surface of the retroarticular process is slightly concave, and the ventral surface bears a distinct sub-triangular depression. There is a ventral angular articulation facet. The articular surface is broad and saddle-shaped. There are two small foramina posterior to the articular surface.

*Identification*. The fossil shares with anguimorphs, geckos, and scincids a medially directed and broadened retroarticular process [23]. Some snakes have a medially directed retroarticular process but differ from the fossils in having relatively narrower process that lacks a concavity on the dorsal surface [23]. Geckos differ from the fossil in having a distinct notch on the medial margin of the retroarticular process [23]. Scincids differ in having a tubercle or flange on the medial margin of the retroarticular process [23]; however, this feature was not obvious in all examined specimens, particularly for *Scincella* (Fig 33C). The fossil differs from examined scincids in having a distinct sub-triangular depression on the ventral surface of the retroarticular process. Based on differences from other NA lizards, the fossil is referable to Anguimorpha. Examined *Anniella* differ in having a more medially slanted retroarticular process and examined specimens of *Xenosaurus* and *Heloderma* have a more slender retroarticular process [24]. The fossil is referable to Anguidae. A distinct sub-triangular depression on the ventral surface of the retroarticular process was observed in gerrhonotines and diploglossines but was absent in examined *Ophisaurus*. We refrain from making a more refined identification pending examination of additional skeletal material of diploglossines.

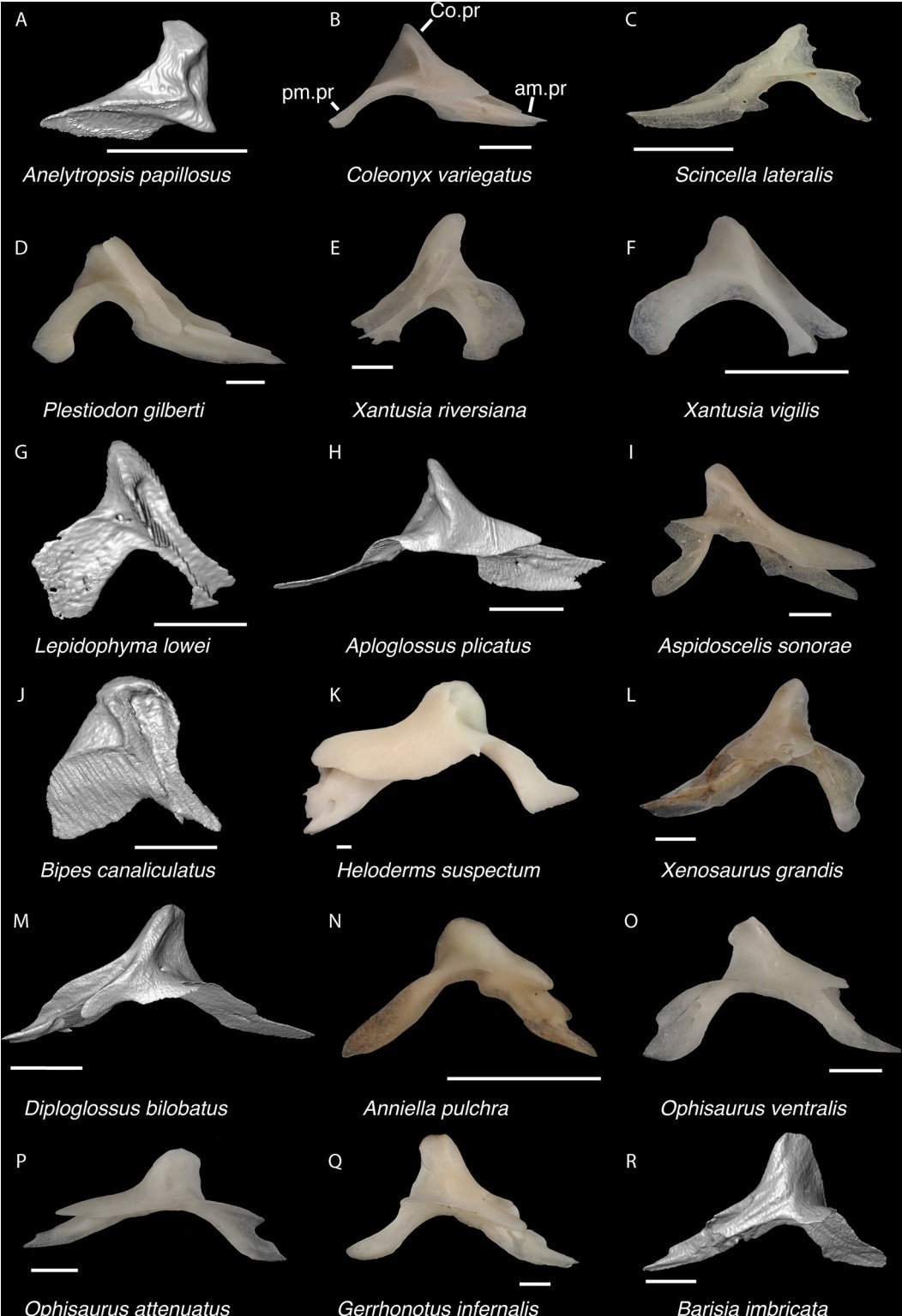

**Fig 32. Non-pleurodontan coronoid bones.** Coronoids in lateral view–**A**. *Anelytropsis papillosus* UF 86708 right coronoid; **B**. *Coleonyx varegatus* TxVP M-13892 right coronoid; **C**. *Scincella lateralis* TxVP M-4489 left coronoid; **D**. *Plestiodon gilberti* TxVP M-8587 right coronoid; **E**. *Xantusia riversiana* TxVP M-8505 left coronoid; **F**. *Xantusia vigilis* TxVP M-12130 right coronoid; **G**. *Lepidophyma lowei* LACM 143367 right coronoid; **H**. *Aploglossus plicatus* TNHC 34481 right coronoid; **I**. *Aspidoscelis sonorae* TxVP M-15670 right coronoid; **J**. *Bipes canaliculatus* CAS 134753 right coronoid; **K**. *Heloderma*

*suspectum* TxVP M-9001 left coronoid; **L**. *Xenosaurus grandis* TxVP M-8960 left coronoid; **M**. *Diploglossus bilobatus* TNHC 31933 left coronoid; **N**. *Anniella pulchra* TxVP M-8678 right coronoid; **O**. *Ophisaurus ventralis* TxVP M-8585 right coronoid; **P**. *Ophisaurus attenuatus* TxVP M-8979 left coronoid; **Q**. *Gerrhonotus infernalis* TxVP M-13441 right coronoid; **R**. *Barisia imbricata* TNHC 76984 left coronoid. Scale bars = 1 mm. **Abbreviations**: am.pr, anteromedial process; Co.pr, coronoid process; pm.pr, posteromedial process.

## Gerrhonotinae Tihen, 1949 [106]

Referred specimens: <u>Dentary</u>, 41229–27659 right, 41229–27614 left.

**Dentary.** *Description*. TxVP 41229–27659 is a right dentary with 26 tooth positions (Fig 31C). Teeth bear medial striations. Mesial teeth are unicuspid, pointed, and slightly recurved while distal teeth are near-bicuspid and more squared-off. The coronoid process is well-developed and pointed, and the angular process is present but broken posteriorly. There is no surangular process. The Meckelian canal is open ventrally. The intramandibular septum has a free posteroventral margin. The dental shelf is broad and is slanted ventromedially. There is a posteriorly projecting splenial spine. There are six nutrient foramina arranged in a row on the lateral surface.

*Identification*. The fossil is assigned to Anguimorpha based on the presence of a posteriorly projecting splenial spine [45, 107]. The fossil is assigned to Anguidae based on the presence of an intramandibular septum with a free posteroventral margin [108–111]. Diploglossines differ from the fossil in lacking a splenial spine and instead have the anterior inferior alveolar foramen contained within the splenial [112]. *Ophisaurus* differs from the fossil in having a long surangular process (Fig 34) and having a surangular spine [109, 111]. The fossil is assigned to Gerrhonotinae.

## Clade composed of (*Gerrhonotus*, (*Barisia*, *Abronia*))

Referred specimens: <u>Jugal</u>, 41229–26858 right; <u>Premaxilla</u>, 41229–27657.

**Premaxilla.** *Description*. TxVP 41229–27657 is a premaxilla with ten tooth positions (Fig 31D). Teeth are unicuspid. The anterior rostral surface is rounded, and the nasal process is curved posteriorly and broken at the posterior end. The nasal process is wide and has lateral notches near the base. The nasal process tapers posteriorly and has a well-developed posterior keel. On the alveolar plate, there are lateral maxillary facets and dorsal ossifications that form an ossified bridge with the nasal process. Posteriorly, the palatal plate is slightly broken and has a wavy posterior edge. The incisive process is large, round, and slightly bilobed. There are large foramina posterolateral to the base of the nasal process and an anterior foramen between the ossified bridge, the nasal process, and the alveolar plate on either side.

*Identification*. The fossil premaxilla is assigned to anguimorpha based on having a large, round, and bilobed incisive process [24]. The incisive process is comparatively smaller and less distinctly bilobed in pleurodontans, scincids (when present), and xantusiids [24]. Among NA anguimorphs, the fossil shares with anguids and *Xenosaurus* nine or greater tooth positions [19, 110, 113]. The fossil differs from *Xenosaurus* in lacking a rugose rostral surface of the premaxilla [19]. Anguinae and Diploglossinae differ from the fossil in having a forked palatal process [24, 110, 113]. The fossil is assigned to Gerrhonotinae.

The fossil is identifiable to the clade (*Desertum*, (*Gerrhonotus*, (*Barisia*, *Abronia*))) based on the presence of an ossified bridge (Fig 35) between the nasal process and the body of the premaxilla [111]. An ossified bridge also occurs in some species of *Elgaria*, but the bridge is much thinner [55]. Furthermore, most *Elgaria* have a midline anterior foramen, which is absent on the fossil [111]. Examined *Desertum lugoi* differ in having a thinner nasal process that narrows

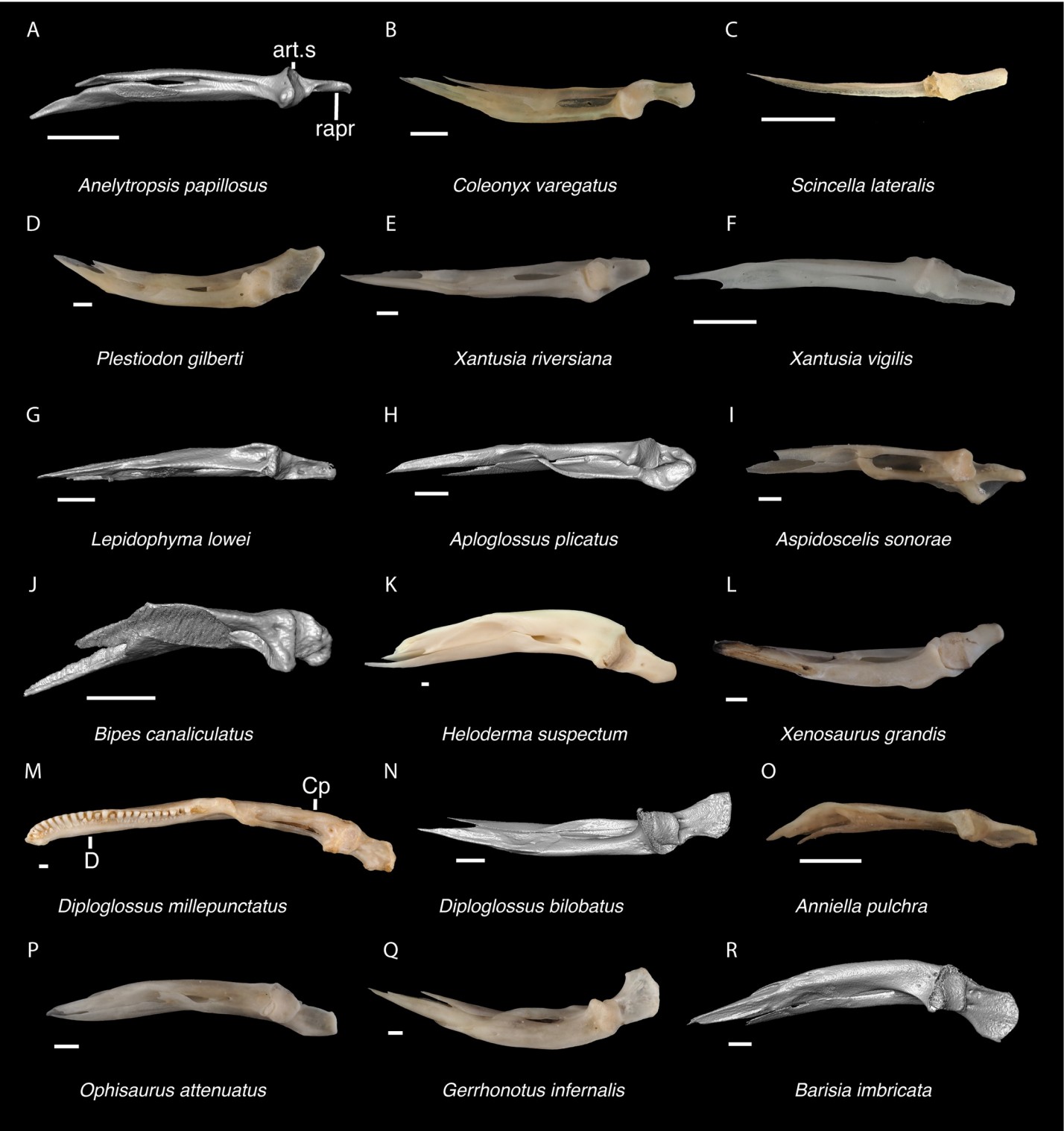

**Fig 33. Non-pleurodontan compound bones.** Compound bones in dorsal view–**A**. *Anelytropsis papillosus* UF 86708 right compound bone; **B**. *Coleonyx varegatus* TxVP M-13892 left compound bone; **C**. *Scincella lateralis* TxVP M-5531 left articular; **D**. *Plestiodon gilberti* TxVP M-8587 left compound bone; **E**. *Xantusia riversiana* TxVP M-8505 left compound bone; **F**. *Xantusia vigilis* TxVP M-12130 right compound bone; **G**. *Lepidophyma lowei* LACM 143367 right compound bone; **H**. *Aploglossus plicatus* TNHC 34481 right compound bone; **I**. *Aspidoscelis sonorae* TxVP M-15670 right compound bone; **J**. *Bipes canaliculatus* CAS 134753 right compound bone; **K**. *Heloderma suspectum* TxVP M-9001 right compound bone; **L**. *Xenosaurus grandis* TxVP M-8960 left compound bone; **M**. *Diploglossus millepunctatus* TxVP M-9010

right mandible; **N**. *Diploglossus bilobatus* TNHC 31933 left compound bone; **O**. *Anniella pulchra* TxVP M-8678 right compound bone; **P**. *Ophisaurus attenuatus* TxVP M-8979 right compound bone; **Q**. *Gerrhonotus infernalis* TxVP M-13441 left compound bone; **R**. *Barisia imbricata* TNHC 76984 right compound bone. Scale bars = 1 mm. **Abbreviations**: art.s, articular surface; Cp, compound bone; D, dentary, rapr, retroarticular process.

towards the base [55]. On that basis, the fossil is assigned to the clade (*Gerrhonotus*, (*Barisia*, *Abronia*)).

**Jugal.** *Description*. TxVP 41229–26858 is a right jugal that is missing much of the suborbital process (Fig 31E). There is a distinct ventral maxillary facet on the suborbital process. The quadratojugal process (jugal spur) is long and pointed. The postorbital process is long and is directed dorsally and has a postorbital facet on the anteromedial surface. On the medial surface, there is a medial ridge that is located at the midline of the suborbital and postorbital processes. The medial ridge defines the anterior border of a small depression on the medial surface anterior to the quadratojugal process. There are two medial and four lateral foramina near the inflection point.

*Identification*. TxVP 41229–26858 is assigned to Anguimorpha based on the presence of a medial ridge located at the midline of the suborbital and postorbital processes [50]. The fossil differs from NA pleurodontans in having a quadratojugal process and a dorsally directed postorbital process [29, 82] (Figs 12 and 36). Geckos have a highly reduced jugal, and dibamids lack a jugal [24, 51]. Examined scincids differ in having a long and thin suborbital process with a distinct notch on the ventral margin and a large depression (coronoid recess of [20]) on the medial surface near the inflection point. Gymnophthalmoids differ in having a distinct medial ectopterygoid process (smaller in some gymnophthalmids and alopoglossids compared to teiids) [24, 37]. Xantusiids differ in having a short suborbital process, and *Xantusia riversiana* and *Lepidophyma* have an anteroposteriorly widened postorbital process [46].

Among NA anguimorphs, *Anniella* differs in having a reduced jugal [84] and *Heloderma* differs in that the anterior and posterior processes form a right angle [24]. *Xenosaurus* has co-ossified osteoderms or sculpturing on the lateral surface of the jugal [19]. The fossil was assigned to Anguidae. A quadratojugal process is usually present in anguids [24, 111, 114]. The length of the quadratojugal process in the fossil is longer than that of examined *Ophisaurus* but is similar to that present in some species of *Gerrhonotus*, *Barisia*, and *Abronia* [55, 111, 114]. An examined *Celestus enneagrammus* (FMHN 108860) and *Panolopus costatus* (UF 59382) also have a long quadratojugal process but differ from the fossil in having a relatively shorter orbital process. Examined *Desertum lugoi* have a relatively shorter quadratojugal process. Jugals are assigned to the clade (*Gerrhonotus*, (*Barisia*, *Abronia*)) on this basis. *Barisia* differs in having sculpturing on the lateral surface of the jugal [111] so the fossil likely represents either *Gerrhonotus* or *Abronia*.

### *Gerrhonotus* Wiegmann, 1828 [89]

Referred specimens: <u>Maxilla</u>, 41229–27090 right, 41229–27452 left, 41229–27438 right, 41229–9917 left.

**Maxilla.** *Description*. TxVP 41229–27438 is a right maxilla (Fig 31F). There are 22 tooth positions. The teeth are unicuspid with medial striations on the crowns, and most teeth are recurved. The lateral surface of the maxilla is highly sculptured. The facial process is tall and broad, and gently curves dorsomedially. The medial margin of the facial process has a distinct nasal facet, and the posterior margin has a large notch. The premaxillary process is long and bifurcated with a longer lateral projection and a shorter medial lappet. The crista transversalis is low and trends anteromedially from the facial process, defining the medial border of a shallow depression on the premaxillary process. There is a narrow palatal shelf without a distinctly

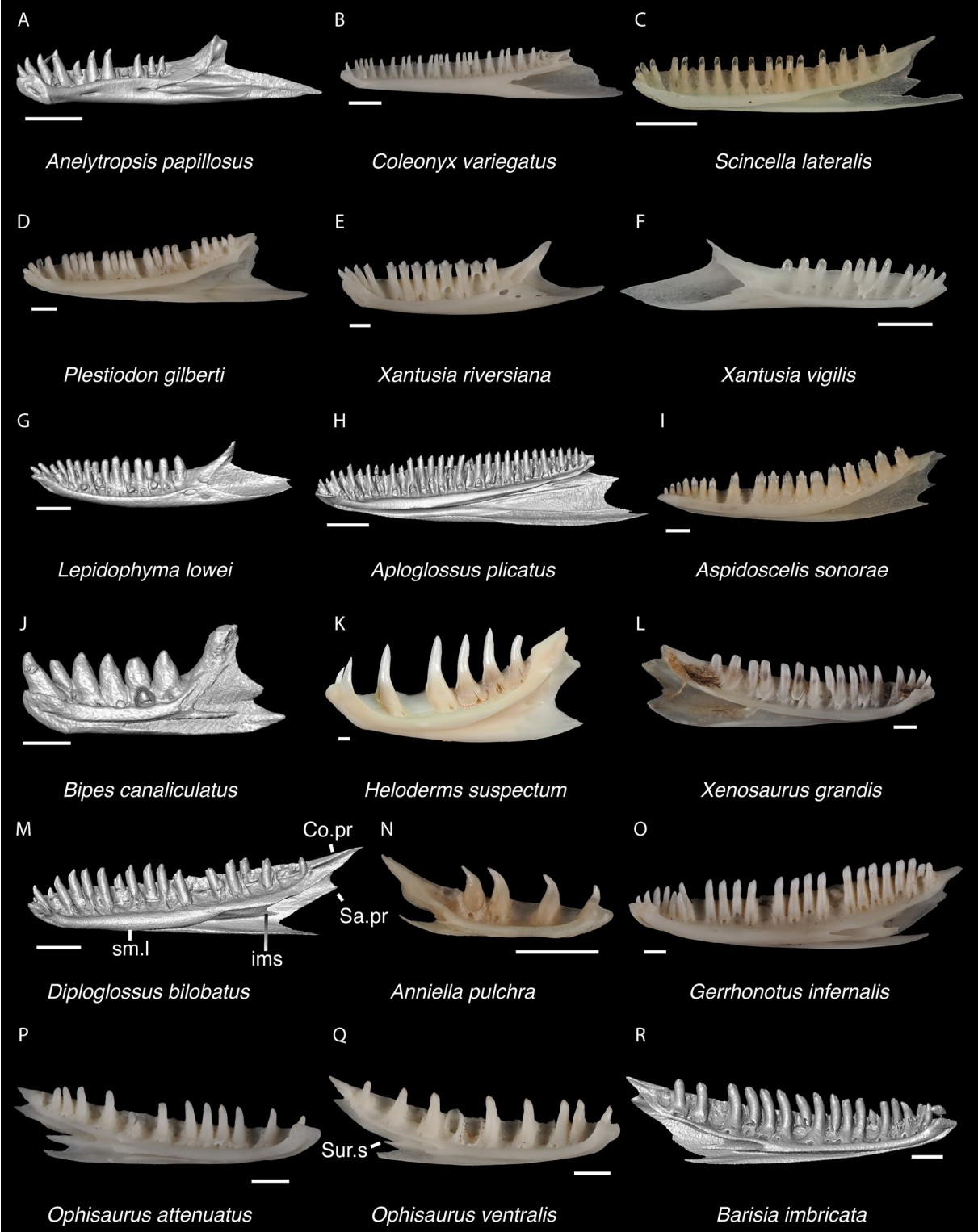

**Fig 34. Non-pleurodontan dentaries.** Dentaries in medial view–**A**. *Anelytropsis papillosus* UF 86708 right dentary; **B**. *Coleonyx variegatus* TxVP M-12109 right dentary; **C**. *Scincella lateralis* TxVP M-4489 right dentary; **D**. *Plestiodon gilberti* TxVP M-8587 right dentary; **E**. *Xantusia riversiana* TxVP M-8505 right spleniodentary; **F**. *Xantusia vigilis* TxVP M-12130 right spleniodentary; **G**. *Lepidophyma lowei* LACM 143367 right spleniodentary; **H**. *Aploglossus plicatus* TNHC 34481 right dentary; **I**. *Aspidoscelis sonorae* TxVP M-15670 right dentary; **J**. *Bipes canaliculatus* CAS 134753 right dentary; **K**. *Heloderma suspectum* TxVP M-9001 right dentary; **L**. *Xenosaurus grandis* TxVP M-8960 left

dentary; **M**. *Diploglossus bilobatus* TNHC 31933 right dentary; **N**. *Anniella pulchra* TxVP M-8678 left dentary; **O**. *Gerrhonotus infernalis* TxVP M-13441 right dentary; **P**. *Ophisaurus attenuatus* TxVP M-8979 left dentary; **Q**. *Ophisaurus ventralis* TxVP M-8585 left dentary; **R**. *Barisia imbricata* TNHC 76984 left dentary. Scale bars = 1 mm. **Abbreviations**: Co.pr, coronoid process; ims, intramandibular septum; Sa.pr, surangular process; sm.l, suprameckelian lip; Sur.s, surangular spine.

projecting palatine process. There is a deep, elongate medial recess on the medial surface of the facial process. The lateral wall of the posterior orbital process is short. The dorsal surface of the postorbital process has an elongate, deep jugal groove and a small ectopterygoid facet at the posterior end. There is an opening for the superior alveolar canal anterior to the facial process. There are three large superior alveolar foramina (maxillary trigeminal foramina of [24]) on the palatal shelf medial to the palatine process and eight lateral nutrient foramina.

*Identification*. Fossil maxillae share with some scincids and anguimorphs unicuspid teeth with striated crowns [52, 115]. Fossils differ from examined scincids in lacking a large notch at the end of the posterior orbital process (see also Fig 5 of [116]). Furthermore, *Scincella* differs in lacking striations on the crowns [117] and examined *Plestiodon* differ in that the crista transversalis abruptly trends medially anterior to the facial process and the palatal shelf is wider. Based on these differences, the fossils are referred to Anguimorpha.

*Anniella* and *Heloderma* differ from the fossils in having a greater number of pointed and recurved teeth as well as fewer tooth positions (up to seven in *Anniella* and up to ten in *Heloderma* [24, 84]). *Xenosaurus* differs in having fused osteoderms laterally on the facial process [19]. Fossils are assigned to Anguidae. The fossil is assigned to Gerrhonotinae and differs from anguines and diploglossines in having an elongated premaxillary process that lacks a deep anterior notch [118]. Some gerrhonotines, a relatively large notch where the lacrimal articulates with the posterior edge of the facial process occurs only in species of *Gerrhonotus* [55]. The distinctiveness and the presence of the notch varies intra- and interspecifically among *Gerrhonotus*, however, the relatively large notch as seen on the fossil most closely resembles that observed among specimens of *Gerrhonotus* and serves as our basis for our identification of the fossil to that genus.

## Anguinae Gray, 1825 [102]

### *Ophisaurus* Daudin, 1803 [119]

Fossils assigned to Anguinae are identified as *Ophisaurus* on the basis that *Ophisaurus* is the only genus within Anguinae known to inhabit North America during the Quaternary.

Referred specimens: <u>Compound bone</u>, 41229–29001 right; <u>Coronoid</u>, 41229–28445 left; <u>Dentary</u>, 41229–27759 left, 41229–27981 right, 41229–28388 left, 41229–28485 left, 41229–28486 left, 41229–28487 left, 41229–28553 left, 41229–28584 left, 41229–28622 right, 41229–28887 right, 41229–29059 right; <u>Frontal</u>, 41229–28737 right; <u>Maxilla</u>, 41229–28371 left, 41229–28372 left, 41229–28413 right, 41229–28435 left, 41229–29315 left, 41229–28149 left, 41229–28500 left, 41229–28502 right, 41229–28504 right, 41229–28544 left, 41229–28581 left, 41229–28679 left, 41229–28728 right, 41229–29089 right; <u>Parietal</u>, 41229–28153, 41229–28384, 41229–29026; <u>Postfrontal</u>, 41229–28467; <u>Prefrontal</u>, 41229–25597 left; <u>Premaxilla</u>, 41229–28734; <u>Pterygoid</u>, 41229–25602 right, 41229–28356 left; <u>Quadrate</u>, 41229–25582 right, 41229–25585 right.

**Premaxilla.** *Description*. TxVP 41229–28734 is a premaxilla with nine tooth positions (Fig 37A). Teeth are unicuspid. The rostral surface is rounded, and the nasal process is strongly curved posteriorly. The nasal process is thin and slightly waisted at the base and has a short posterior keel. There are lateral maxillary facets but no dorsal ossifications on the alveolar plate. The palatal plate has short posterior projections. The incisive process is large, round, and

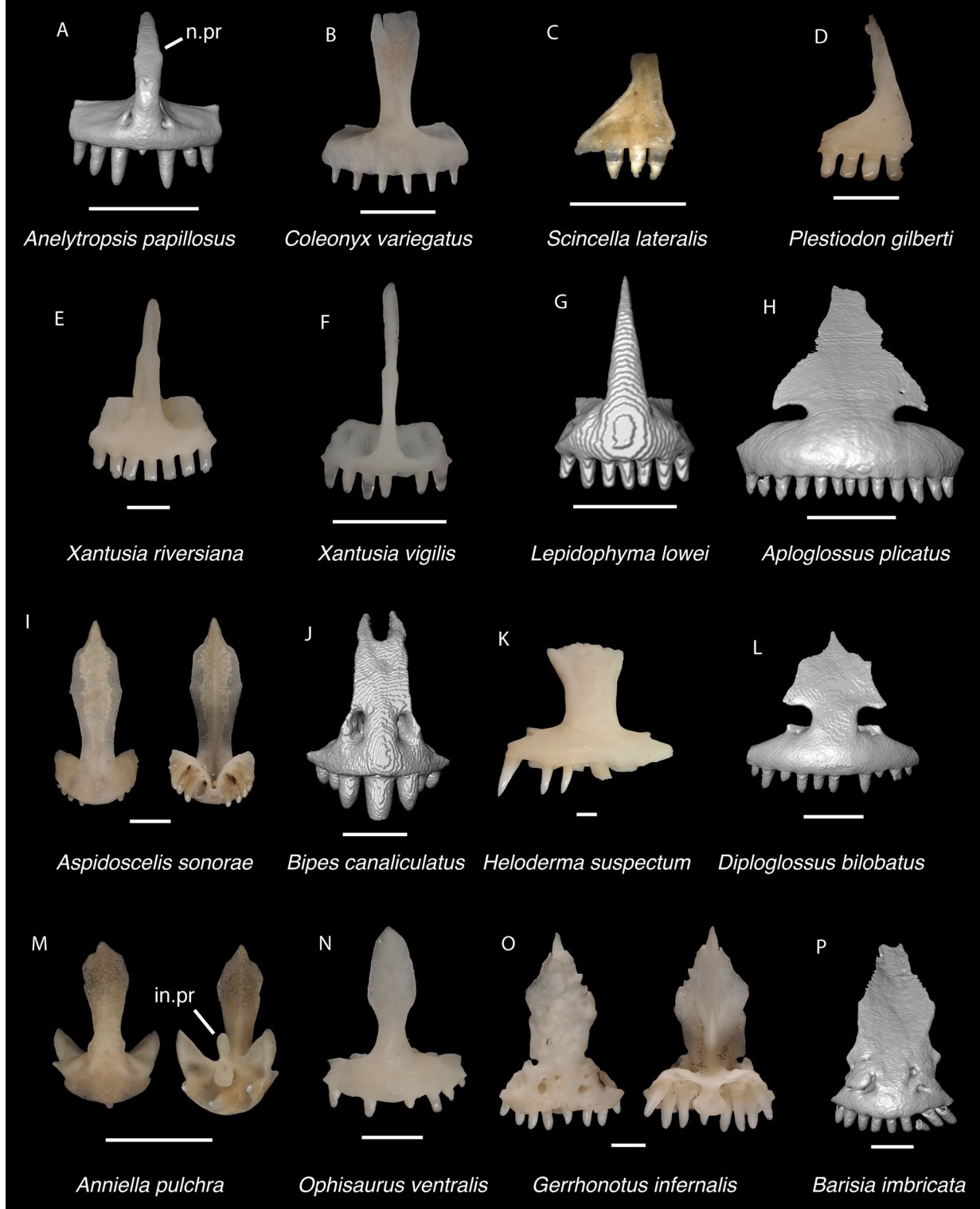

**Fig 35. Non-pleurodontan premaxillae.** Premaxillae in anterior, posterior, dorsal, and ventral views–**A**. *Anelytropsis papillosus* UF 86708 premaxilla in anterior view; **B**. *Coleonyx variegatus* TxVP M-12109 premaxilla in anterior view; **C**. *Scincella lateralis* TxVP M-4489 right premaxilla in anterior view; **D**. *Plestiodon gilberti* TxVP M-8587 right premaxilla in anterior view; **E**. *Xantusia riversiana* TxVP M-8505 premaxilla in anterior view; **F**. *Xantusia vigilis* TxVP M-12130 premaxilla in anterior view; **G**. *Lepidophyma lowei* LACM 143367 premaxilla in anterior view; **H**. *Aploglossus plicatus* TNHC 34481 premaxilla in anterior view; **I**. *Aspidoscelis sonorae* TxVP M-15670 premaxilla in dorsal and

ventral views; **J**. *Bipes canaliculatus* CAS 134753 premaxilla in anterior view; **K**. *Heloderma suspectum* TxVP M-9001 premaxilla in anterior view; **L**. *Diploglossus bilobatus* TNHC 31933 premaxilla in anterior view; **M**. *Anniella pulchra* TxVP M-8678 premaxilla in dorsal and ventral views; **N**. *Ophisaurus ventralis* TxVP M-8585 premaxilla in anterior view; **O**. *Gerrhonotus infernalis* TxVP M-13441 premaxilla in anterior and posterior views; **P**. *Barisia imbricata* TNHC 76984 premaxilla in anterior view. Scale bars = 1 mm. **Abbreviations**: in.pr, incisive process; n.pr, nasal process.

bilobed. There are small foramina posterolateral to the base of the nasal process and there is a single midline anterior foramen.

*Identification*. The fossil premaxilla is assigned to Anguimorpha based on having a large, round, and bilobed incisive process [24] (Fig 35). The incisive process is relatively smaller and less distinctly bilobed in pleurodontans, scincids (when present), and xantusiids [24]. The fossil shares with anguids and *Xenosaurus* nine tooth positions [19, 110, 113]. The fossil differs from *Xenosaurus* in lacking a rugose rostral surface of the premaxilla [19]. The fossil shares with Anguinae and Diploglossinae a forked palatal process [24, 110, 113]. The fossil differs from Diploglossinae and is assigned to Anguinae based on the absence of a dorsal ossification on the palatal plate posterior to the medial ethmoidal foramen [110].

**Maxilla.**   *Description*. TxVP 41229–28413 serves as the basis for our description and is a right maxilla (Fig 37B) with 17 tooth positions. Teeth are unicuspid with medial striations. The dorsal portion of the facial process is broken, but it is broad. The anterior face of the facial process gently curves medially and has an anterodorsal projection. The premaxillary process is strongly bifurcated with a longer, pointed lateral projection and a shorter medial lappet. The crista transversalis trends anteromedially from the facial process. There is a narrow palatal shelf with a rounded palatine process. The palatal shelf becomes especially narrow anterior to the palatine process. There is a deep recess on the medial surface of the facial process. The lateral wall of the posterior orbital process is short with a small notch posteriorly. The dorsal surface of the postorbital process has a shallow jugal groove. There is a large superior alveolar foramen on the palatal shelf lateral to the palatine process, and five lateral nutrient foramina.

*Identification*. Fossil maxillae share with some scincids and anguimorphs unicuspid teeth with striated crowns [52, 115] (Fig 38). Fossils differ from examined scincids in lacking a large notch at the end of the posterior orbital process (see also Fig 5 in [116]). *Scincella* differs in lacking striations on the crowns [117]. Examined *Plestiodon* differ in having the crista transversalis abruptly trend medially anterior to the facial process and having a much wider palatal shelf. Based on these differences, the fossils are referred to Anguimorpha.

*Anniella* and *Heloderma* differ from the fossils in having more pointed and recurved teeth as well as fewer tooth positions (up to seven in *Anniella* and up to ten in *Heloderma* [24, 84]). *Xenosaurus* differs in having fused osteoderms on the lateral surface of the facial process [19]. Fossils are assigned to Anguidae. Fossils share with anguines and diploglossines a deeply notched premaxillary process [118]. Fossil maxillae share with Anguinae sharp and widely spaced teeth [118]. Some species of *Abronia* (e.g., *Abronia mixteca* [35]) and perhaps some diploglossines (e.g., *Diploglossus fasciatus*; see Fig 2 of [112]) also have sharp and widely-spaced teeth, but in *Abronia* and diploglossines, the palatal shelf anterior to the palatine process does not narrow to the same degree as in fossils and examined anguines.

**Frontal.**   *Description*. TxVP 41229–28737 is an unfused right frontal (Fig 37C). There are osteoderms with a pitted texture fused to the frontal much of its dorsal surface. The lateral frontal sulcus separates the larger frontal shield from the smaller posterolateral frontoparietal shield, and the medial frontal sulcus separates a minute interfrontal shield (*sensu* [114]). Anterolaterally, there is a prefrontal facet, and anterodorsally, there is a nasal facet. The anterior end is elongated and pointed without a distinct anterolateral process. The medial margin and interorbital margins are straight, and the posterolateral processes gently curve laterally. The

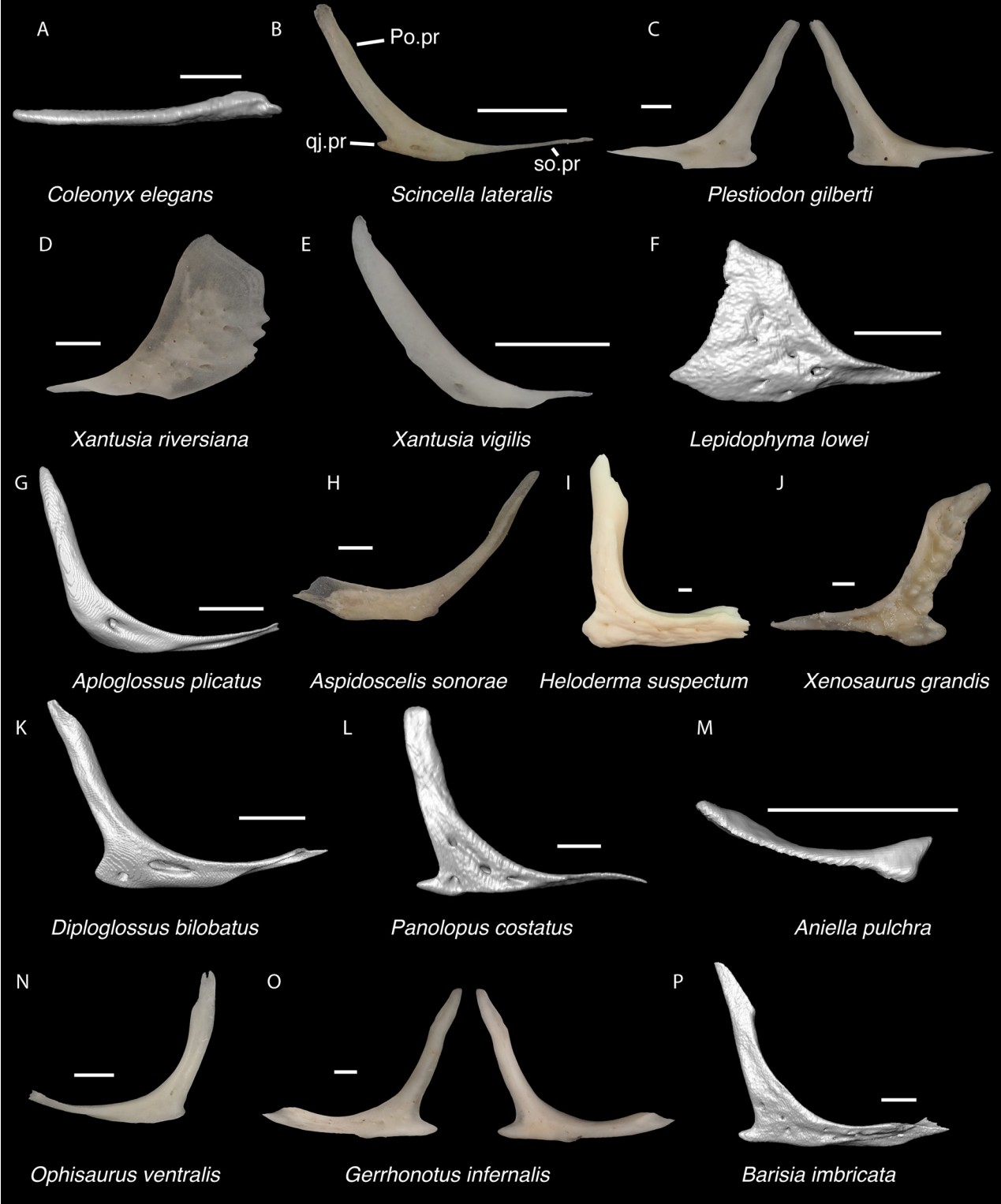

**Fig 36. Non-pleurodontan jugals.** Jugals in lateral and medial views–**A**. *Coleonyx elegans* UF 11258 right jugal in lateral view; **B**. *Scincella lateralis* TxVP M-4489 right jugal in lateral view; **C**. *Plestiodon gilberti* TxVP M-8587 left jugal in lateral and medial views; **D**. *Xantusia riversiana* TxVP M-8505 left jugal in lateral view; **E**. *Xantusia vigilis* TxVP M-12130 right jugal in lateral view; **F**. *Lepidophyma lowei* LACM 143367 right jugal in lateral view; **G**. *Aploglossus plicatus* TNHC 34481 right jugal in lateral view; **H**. *Aspidoscelis sonorae* TxVP M-15670 left jugal in lateral view; **I**. *Heloderma suspectum* TxVP M-8593 right jugal in lateral view; **J**. *Xenosaurus grandis* TxVP M-8960 left jugal in lateral view; **K**. *Diploglossus bilobatus* TNHC

31933 right jugal in lateral view; **L**. *Panolopus costatus* UF 59382 right jugal in lateral view; **M**. *Anniella pulchra* FMNH 130479 left jugal in lateral view; **N**. *Ophisaurus ventralis* TxVP M-8585 left jugal in lateral view; **O**. *Gerrhonotus infernalis* TxVP M-13441 left jugal in lateral and medial views; **P**. *Barisia imbricata* TNHC 76984 right jugal in lateral view. Scale bars = 1 mm. **Abbreviations**: Po.pr, postorbital process; qj.pr, quadratojugal process; so.pr, suborbital process.

posterior margin is straight with a posterolateral parietal articulation facet. There is a postfrontal facet along the posterolateral edge. The anteroventral portion of the crista cranii is broken, but the crest is well-developed, tall, and anteroposteriorly long.

*Identification*. The fossil shares with many anguimorphs and most scincids co-ossified osteoderms [23], well-developed and ventrally directed cristae cranii (subolfactory processes of [120]), and an unfused frontal [23]. The sculpting on the dorsal surface in scincids is more vermiculate compared to the fossil and to many extant anguimorphs which have a pitted texture, such as *Ophisaurus* and *Elgaria* [45] (Fig 39). *Plestiodon* differ in having exceptionally slender cristae cranii [47], and *Scincella* and *Mabuya* differ in having a fused frontal [36]. Examined *Plestiodon* also differ from the fossil and anguines, except for *Anguis fragilis* [114], in having a broader and less pointed anterior portion of the unfused frontal. Frontals are assigned to Anguimorpha. Among NA anguimorphs, unfused frontals occur in Anguinae, Diploglossinae, *Heloderma*, and *Anniella* [24]. *Heloderma* and *Anniella* differ in having the cristae cranii curve to meet at the midline, forming an enclosed olfactory canal [24]. In the fossil and other anguines except for *Anguis fragilis* [114], there is a pointed anterior portion of the unfused frontal. The anterior portion of the frontal is broader in diploglossines [24]. Frontals are assigned to Anguinae based on the presence of an unfused frontal and a pointed anterior portion of the frontal.

**Parietal.** *Description*. TxVP 41229–28153 serves as the basis for our description and is a parietal missing only the ends of the postparietal processes (Fig 37D). The parietal table is rectangular and is covered dorsally in co-ossified osteoderms with a pitted texture. The interparietal sulcus separates the triangular interparietal shield from the lateral shields (*sensu* [114]). The interparietal shield reaches the posterior smooth region of the parietal table. The anterior edge is straight with small frontal tabs and interlacing articulation facets for the frontal. The posterior edge between the postparietal processes is characterized by two small depressions (nuchal fossae) separated by a small ridge. The postparietal processes are broad and flat at the bases. The ventrolateral crests are low, positioned along the lateral margins, and border the cerebral vault. There is a low ridge anterior to the pit for the processus ascendens. The ventrolateral crests curve medially onto the postparietal processes, and together with a ventrolateral ridge on the postparietal process, define distinct depressions. There is a large parietal foramen within the interparietal shield.

*Identification*. The fossil shares with some anguimorphs and scincids co-ossified osteoderms with dorsal sculpting [23], a parietal foramen enclosed by the parietal [23], and ventrally projecting parietal crests or processes [23, 24, 77] (Fig 40). Scincids differ in having long posterior projections (median extensions of [24]) on the posterior edge of the parietal table between the postparietal processes [24, 27] (see also Fig 8 of [68]). Scincids further differ from the fossil in having distinct ventrolateral crests that include long, thin, ventral projections [24, 47] and the postparietal processes of examined scincids are more separated relative to the fossil. Parietals are assigned to Anguimorpha on that basis. *Heloderma* differs in lacking a parietal foramen [23] and *Anniella* differs in having the ventrally projecting parietal crests developed into extensive sheets of bone [24]. Xenosaurids differ in having heavily sculptured dorsal roofing bones with many bumpy, dome-like co-ossified osteoderms [19, 108, 121]. Parietals are assigned to Anguidae.

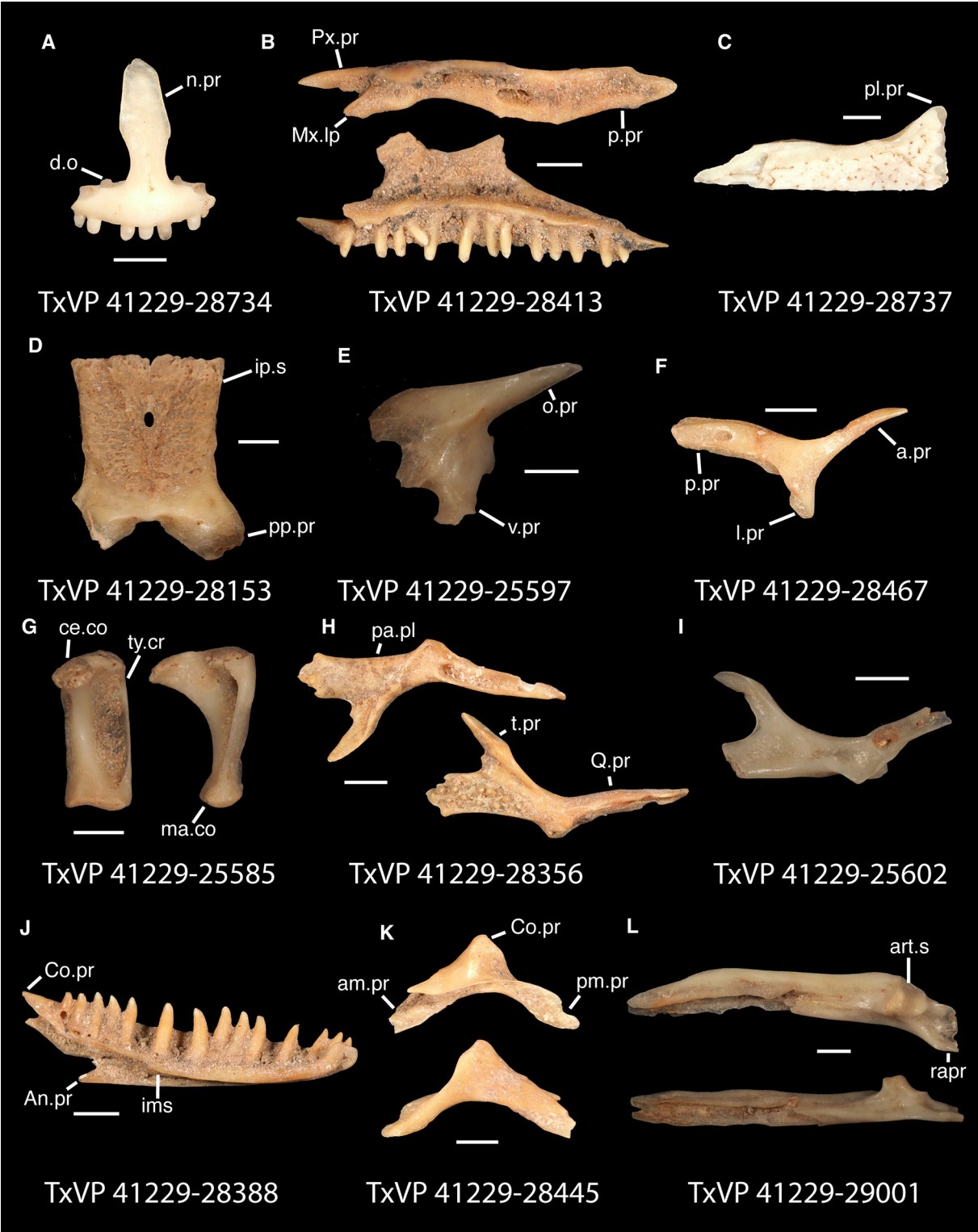

**Fig 37.** Fossil anguines, A–L: *Ophisaurus*. **A**. TxVP 41229–28734 Anterior view of premaxilla; **B**. TxVP 41229–28413 Dorsal and medial view of right maxilla; **C**. TxVP 41229–28737 Dorsal view of frontal; **D**. TxVP 41229–28153 Dorsal view of parietal; **E**. TxVP 41229–25597 Lateral view of prefrontal; **F**. TxVP 41229–28467 Dorsal view of postfrontal; **G**. TxVP 41229–25585 Posterior and lateral view of quadrate; **H**. TxVP 41229–28356 Dorsal and ventral view of pterygoid; **I**. TxVP 41229–25602 Dorsal view of pterygoid; **J**. TxVP 41229–28388 Medial view of dentary; **K**. TxVP 41229–28445 Lateral and medial view of coronoid; L. TxVP 41229–29001 Dorsal and medial view of compound bone. Scale

bars = 1 mm. **Abbreviations**: am.pr, anteromedial process; An.pr, angular process; a.pr, anterior process; art.s, articular surface; ce.co, cephalic condyle; Co.pr, coronoid process; d.o, dorsal ossification; ims, intramadibular septum; ip.s, interparietal shield; l.pr, lateral process; ma.co, mandibular condyle; Mx.lp, maxillary lappet; n.pr, nasal process; o.pr, orbital process; pa.pl, palatal plate; pl.pr, posterolateral process; pm.pr, posteromedial process; pp.pr, postparietal process; p.pr, posterior process; Px.pr, premaxillary process; Q.pr, quadrate process; rapr, retroarticular process; t.pr, transverse process; v.pr, ventral process.

The lateral margins of the parietal table are generally more concave in gerrhonotines and diploglossines, whereas in the fossil and in anguines, the lateral margins are straighter [24]. Furthermore, examined gerrhonotines and diploglossines usually have a larger smooth area on the posterior portion of the parietal table and a more rounded posterior terminus of the inter- parietal shield (see also Fig 9 of [55]). The fossil shares with examined *Ophisaurus* a small pos- terior smooth area on the parietal table and a pointed posterior terminus of the interparietal shield (see also Figs 2–4 of [122]). Fossils are assigned to Anguinae.

**Prefrontal.** *Description*. TxVP 41229–25597 is a left prefrontal (Fig 37E). It is triradiate with a long and pointed orbital process, a short ventral process, and an anterior sheet. The anterior sheet is slightly broken and has a broad articulation facet for the facial process of the maxilla. There is a small ridge on the lateral surface near the base of the orbital process. The ventral process is missing the posteroventral tip but is narrow and squared off. There is a dis- tinct notch for the lacrimal foramen, and the ventral process forms the posterior border of the foramen. Dorsal to the lacrimal foramen notch is a small overhanging lamina. Medially, the boundary of the olfactory chamber is a smooth, rounded, and concave surface. Dorsal to the olfactory chamber is a shallow groove for articulation with the frontal. The orbitonasal flange is broad with a distinct medial projection for articulation with the palatine.

*Identification*. The fossil differs from many snakes in lacking a lacrimal duct that is largely or fully enclosed within the prefrontal [27]. The fossil differs from NA pleurodontans in lack- ing a lateral prefrontal boss [20, 23, 27, 52] (Fig 41), lacking a strong lateral canthal ridge (reported in *Anolis* and *Polychrus* [20]), lacking a supraorbital spine (present in *Phrynosoma* and *Corytophanes* [29]), and lacking a thin, crescent shape with a distinct laterally projecting lamina (present in examined phrynosomatines). NA teiids, gymnophthalmids, and alopoglos- sids, differ in having a distinct laterally projecting lamina (lacrimal flange of [62]) with a dis- tinct articulation facet for the facial process of the maxilla [37, 53]. Examined scincids differ in having a relatively shorter orbital process and, excluding *Scincella*, a more elongate, oblong anterior process [47]. Xantusiids differ in having the lacrimal fused to the prefrontal [46] with a suborbital foramen nearly or entirely enclosed within the prefrontal, and in having a distinct vertical articulation ridge or flange (e.g., in *Xantusia riversiana*) that articulates with the max- illa. Examined *Coleonyx* differ in having an orbitonasal flange that extends farther medially. *Sphaerodactylus roosevelti* has a smaller notch for the lacrimal foramen [104]. Based on differ- ences with other NA lizards, the fossil is identified to Anguimorpha. *Heloderma* differs in hav- ing a much broader orbital process and the prefrontal of *Anniella* is much smaller than the fossil. Diploglossines and gerrhonotines differ in having an orbitonasal flange that does not extend as far medially as in the fossil. The fossil is assigned to Anguinae.

**Postfrontal.** *Description*. TxVP 41229–28467 is a right postfrontal (Fig 37F). It is triradi- ate, delicate bone with a thin, elongate anterior process, a small, rounded lateral process, and an elongate posterior process. There is a jugal articulation facet on the lateral process and a postorbital articulation facet along the lateral margin of the posterior process. The medial mar- gin is widely curved to clasp the frontal and parietal. There is a large dorsal foramen on the posterior process.

*Identification*. Snakes lack a postfrontal [45] and most NA pleurodontans differ from the fossils in either lacking a postfrontal or having a relatively small postfrontal that lacks a facet

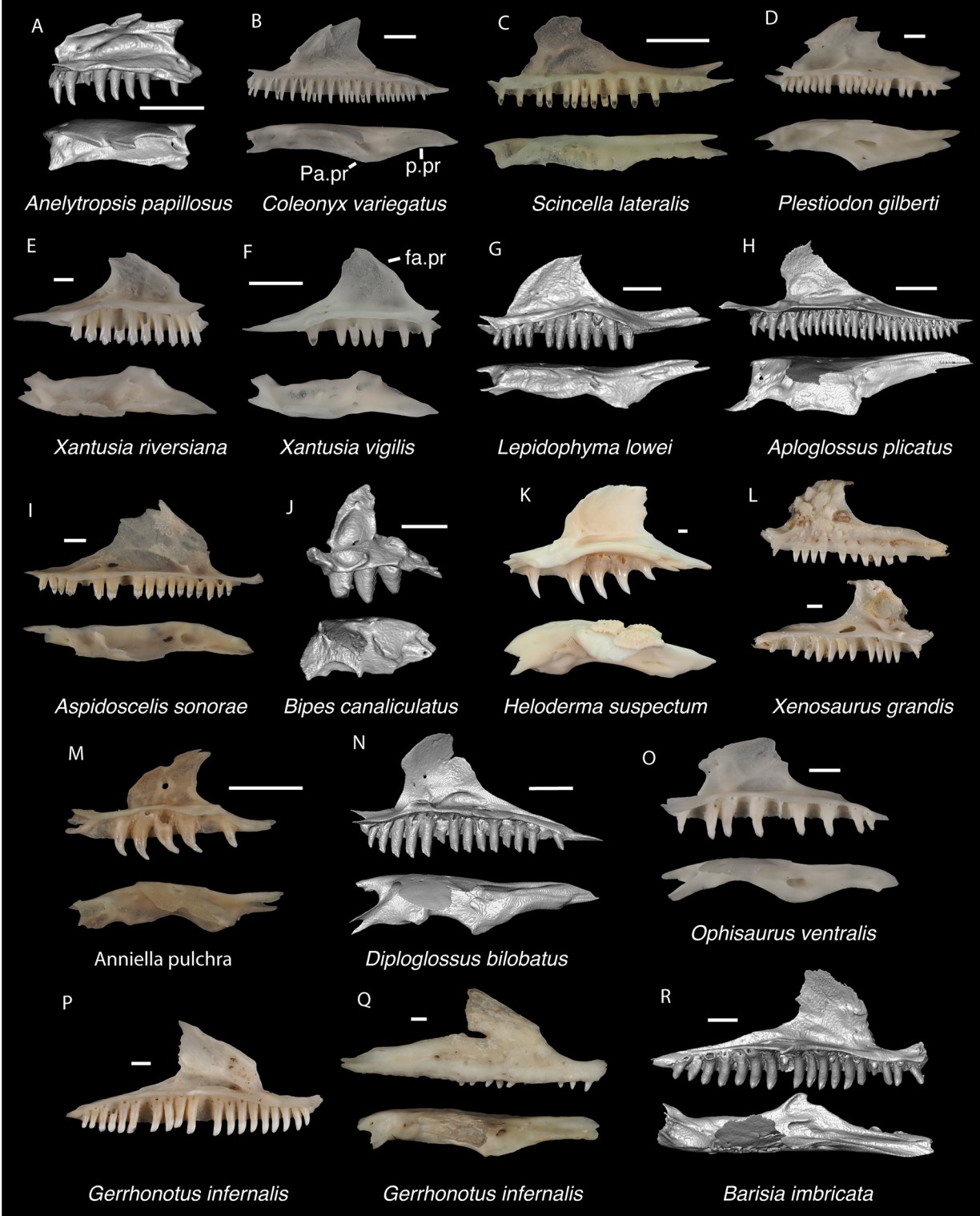

**Fig 38. Non-pleurodontan maxillae.** Maxillae in lateral, medial, and dorsal views–**A**. *Anelytropsis papillosus* UF 86708 right maxilla in medial and dorsal views; **B**. *Coleonyx variegatus* TxVP M-12109 right maxilla in medial and dorsal views; **C**. *Scincella lateralis* TxVP M-4489 right maxilla in medial and dorsal views; **D**. *Plestiodon gilberti* TxVP M-8587 right maxilla in medial and dorsal views; **E**. *Xantusia riversiana* TxVP M-8505 left maxilla in medial and dorsal views; **F**. *Xantusia vigilis* TxVP M-12130 left maxilla in medial and dorsal views; **G**. *Lepidophyma lowei* LACM 143367 right maxilla in medial and dorsal views; **H**. *Aploglossus plicatus* TNHC 34481 right maxilla in medial and dorsal views; **I**.

*Aspidoscelis sonorae* TxVP M-15670 left maxilla in medial and dorsal views; **J**. *Bipes canaliculatus* CAS 134753 right maxilla in medial and dorsal views; **K**. *Heloderma suspectum* TxVP M-9001 right maxilla in medial and dorsal views; **L**. *Xenosaurus grandis* TxVP M-8960 right maxilla in lateral and medial views; **M**. *Anniella pulchra* TxVP M-8678 right maxilla in medial and dorsal views; **N**. *Diploglossus bilobatus* TNHC 31933 right maxilla in medial and dorsal views; **O**. *Ophisaurus ventralis* TxVP M-8585 right maxilla in medial and dorsal views; **P**. *Gerrhonotus infernalis* TxVP M-13441 left maxilla in medial view; **Q**. *Gerrhonotus infernalis* TxVP M-13440 right maxilla in lateral and dorsal views; **R**. *Barisia imbricata* TNHC 76984 left maxilla in medial and dorsal views. Scale bars = 1 mm. **Abbreviations**: fa.pr, facial process; Pa.pr, palatine process; p.pr, posterior process.

for clasping the frontal parietal articulation [24, 29]. Some iguanids (e.g., *Sauromalus ater* TNHC 18483) have a comparatively larger postfrontal that does clasp the frontal parietal articulation; however, the posterior process is shorter compared to the fossil. *Xantusia*, *Xenosaurus*, and NA teiids differ in having a fused postorbitofrontal [45] (Fig 42). *Coleonyx variegatus* and *C. brevis* differ in lacking a lateral process [54]. *Coleonyx elegans* and *C. mitratus* reportedly have a lateral projection [54], but similar to *Sphaerodactylus roosevelti* [104] and *Phyllodactylus baurii*, the lateral process is much shorter than in the fossil. Examined *Plestiodon* differ in lacking a distinct facet on the lateral projection for articulation with the jugal, and *Scincella* and *Mabuya* differ in having a much longer posterior process compared to the anterior process (see also Fig 8 of [68]). Some gymnophthalmids and alopoglossids have a separate, triradiate postorbital [24]; however, to our knowledge, none have the large dorsal foramen on the posterior process that is present in the fossil and several anguimorphs [55, 114]. Based on these differences, fossils were assigned to Anguimorpha. *Anniella* differs in lacking a lateral process [84, 123] and *Heloderma* differs in having a broad postorbitofrontal [24, 123]. Fossils were assigned to Anguidae. Some diploglossines differ in having a fused postorbitofrontal [24]. Examined *Diploglossus* with a separate postfrontal differ from the fossil in having a wider posterior process. Examined gerrhonotines differ in having the angle between the anterior and posterior processes closer to 90 degrees (see also Fig 17 of [55]). Fossils are assigned to Anguinae.

**Quadrate.**  *Description*. TxVP 41229–25585 is a right quadrate (Fig 37G). The bone is thin and parallel-sided. There is no pterygoid lappet. There is a moderately developed medial crest that is directed anteriorly. The conch is deep and narrow. The cephalic condyle projects posteriorly without extensive dorsal ossification. There is an anteriorly expanded dorsal tubercle. There is a quadrate foramen on the anteroventral surface and a foramen medial to the central column.

*Identification*. The fossil shares with geckos, *Scincella*, xantusiids, alopoglossids, some pleurodontans, and anguimorphs except for *Heloderma* the absence of a distinct pterygoid lappet [23, 24] (Fig 43). Examined *Scincella* and alopoglossids differ in having a curved tympanic crest. Examined *Xantusia riversiana* and *Lepidophyma lowei* differ in having a slightly more curved tympanic crest and examined *Xantusia vigilis* have a much more slender quadrate (see also Figs 16–17 of [46]). Examined *Coleonyx* differ in having a more narrow quadrate that slightly narrows ventrally and has a notched dorsolateral margin [38]. *Sphaerodactylus roosevelti* differs in having a more narrow quadrate with a curved tympanic crest [104]. Most examined NA pleurodontans differ in having the dorsal portion much wider compared to the articular surface. In *Anolis* and *Uma*, the lateral margins are parallel but examined *Anolis* differ in having a distinct boss at the ventromedial margin of the quadrate and *Uma* differ in having a laterally slanted cephalic condyle and central column. Fossils were assigned to Anguimorpha.

*Xenosaurus* differs from the fossils in having a wider quadrate with a more shallow lateral conch [19]. *Anniella* differs in having a thin quadrate in posterior view that is widened in medial and lateral views [84]. Fossils were assigned to Anguidae. Examined gerrhonotines (see

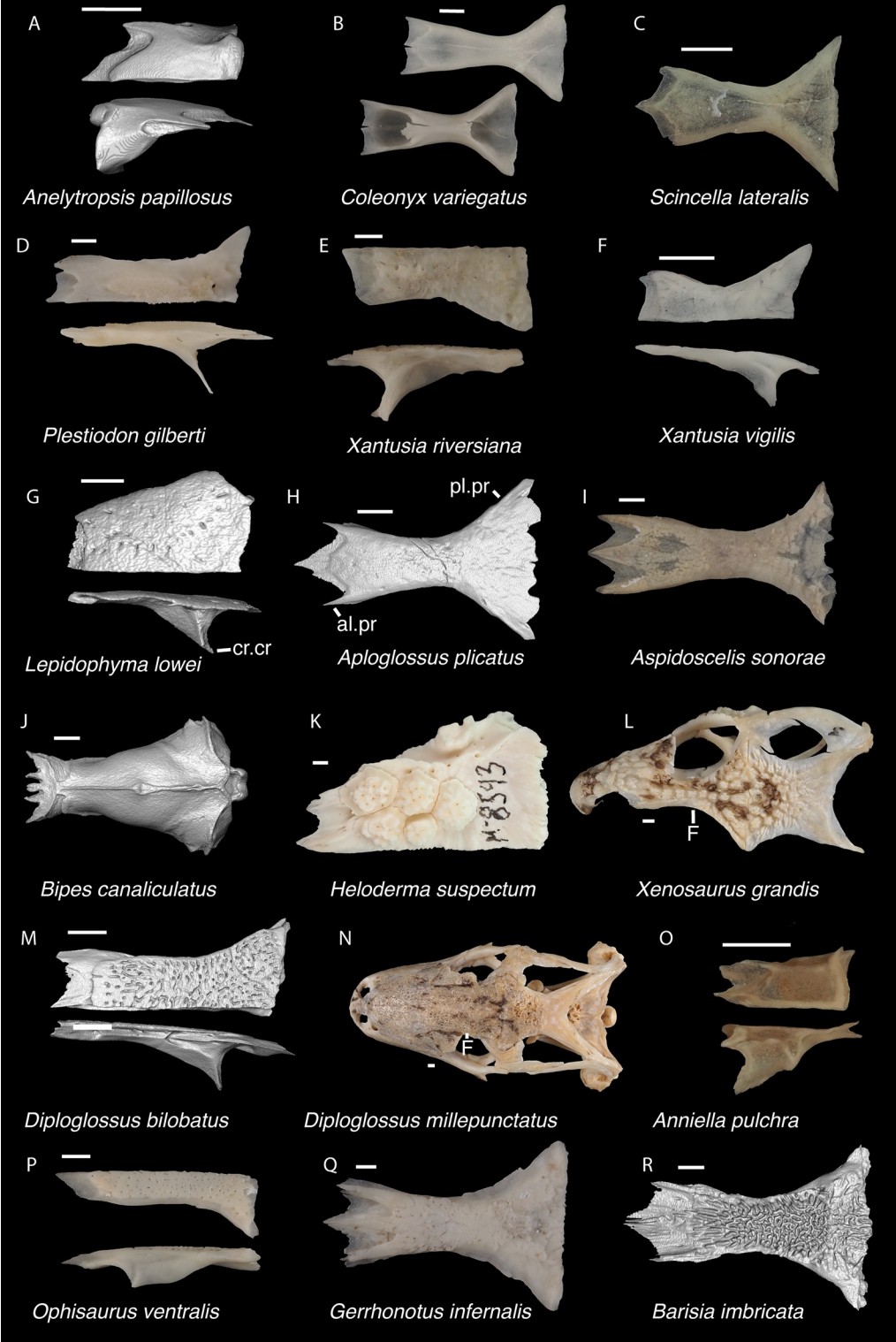

**Fig 39. Non-pleurodontan frontals.** Frontals in dorsal, ventral, and lateral views–**A**. *Anelytropsis papillosus* UF 86708 right frontal in dorsal and lateral views; **B**. *Coleonyx variegatus* TxVP M-12109 frontal in dorsal and ventral views; **C**. *Scincella lateralis* TxVP M-4489 frontal in dorsal view; **D**. *Plestiodon gilberti* TxVP M-8587 right frontal in dorsal and lateral views; **E**. *Xantusia riversiana* TxVP M-8505 left frontal in dorsal and lateral views; **F**. *Xantusia vigilis* TxVP M-12130 right frontal in dorsal and lateral views; **G**. *Lepidophyma lowei* LACM 143367 right frontal in dorsal and lateral

views; **H**. *Aploglossus plicatus* TNHC 34481 frontal in dorsal view; **I**. *Aspidoscelis sonorae* TxVP M-15670 frontal in dorsal view; **J**. *Bipes canaliculatus* CAS 134753 frontoparietal in dorsal view; **K**. *Heloderma suspectum* TxVP M-8593 right frontal in dorsal view; **L**. *Xenosaurus grandis* TxVP M-8960 partial skull in dorsal view; **M**. *Diploglossus bilobatus* TNHC 31933 right frontal in dorsal and lateral views; **N**. *Diploglossus millepunctatus* TxVP M-9010 skull in dorsal view; **O**. *Anniella pulchra* TxVP M-8678 right frontal in dorsal and lateral views; **P**. *Ophisaurus ventralis* TxVP M-8585 left frontal in dorsal and lateral views; **Q**. *Gerrhonotus infernalis* TxVP M-13441 frontal in dorsal view; **R**. *Barisia imbricata* TNHC 76984 frontal in dorsal view. Scale bars = 1 mm. **Abbreviations**: al.pr, anterolateral process; cr.cr, cristae cranii; F, frontal; pl.pr, posterolateral process.

Fig 19 of [55]) and diploglossines (see also Fig 2 of [76]) differ from the fossil and examined *Ophisaurus* in having a more laterally extensive pterygoid lamina and a more curved tympanic crest. Fossils are assigned to Anguinae.

**Pterygoid.** *Description.* TxVP 41229–28356 is a left pterygoid that is missing the distal end of the palatine process (Fig 37H). The palatine process is narrow with an anterior palatine facet. The transverse process is pointed and extends anterolaterally. The transverse process bears a ridge on the dorsal surface for insertion of the superficial pseudotemporal muscle and a developed ectopterygoid facet. There is a large ridge on the ventral surface for insertion of the pterygomandibular muscle, and on the dorsal surface there is a small, deep fossa columella. The epipterygoid crest is anteromedial to the fossa columella. There is a long postepipterygoid groove (*sensu* [124]) on the quadrate process. The quadrate process is elongated, and the medial surface has a shallow groove that serves for insertion of the pterygoideus muscle. There is a small notch in the quadrate process that is likely a result of taphonomic damage. There is a small medial projection at the floor of the basipterygoid fossa and a broad patch of 19 pterygoid tooth positions with 13 pterygoid teeth present. TxVP 41229–25602 is missing the distal ends of the palatine and quadrate processes (Fig 37I) and does not differ substantively from TxVP 41229–28356.

*Identification.* Fossils share with some anguimorphs and pleurodontans a medial projection at the floor of the basipterygoid fossa [24, 52]. Examined NA pleurodontans, except *Petrosaurus*, differ in having the transverse process oriented more medially (Fig 44). Examined *Petrosaurus* differ in lacking pterygoid teeth [44] and lacking a long postepiterygoid groove on the quadrate process. Fossils are assigned to Anguimorpha.

*Xenosaurus* and *Anniella* lack pterygoid teeth [24]. *Heloderma* differs in having only a single row of pterygoid teeth located on an elevated ridge [24]. Thus, fossils are assigned to Anguidae. The presence and number of teeth are variable in gerrhonotines [55, 111] and *Ophisaurus* [60], but diploglossines lack pterygoid teeth [24, 123]. The epipterygoid crest in examined anguines is more distinct and projects farther medially compared to examined gerrhonotines [55, 114]. The palatal process is narrower in examined *Ophisaurus* compared to gerrhonotines and the space between the transverse process and the palatal plate (suborbital incisure of [114]) is wider in examined gerrhonotines and diploglossines compared to *Ophisaurus*, thus fossils are assigned to Anguinae.

**Dentary.** *Description.* TxVP 41229–28388 serves as the basis for our description and is a left dentary with 18 tooth positions and 14 teeth present (Fig 37J). Teeth are unicuspid, but the crowns are slightly eroded. There are three posterolateral processes, including a pointed coronoid process, a long surangular process, and a short angular process. Medially, a minute surangular spine (*sensu* [109]) is visible. The Meckelian canal is open ventrally. The intramandibular septum has a free posteroventral margin. The dental shelf is narrow, and there is a posteriorly projecting splenial spine (*sensu* [109]). There are seven nutrient foramina arranged in a row on the lateral surface. TxVP 41229–27759 differs in lacking an intramandibular septum with a free posteroventral margin and having a relatively longer surangular spine.

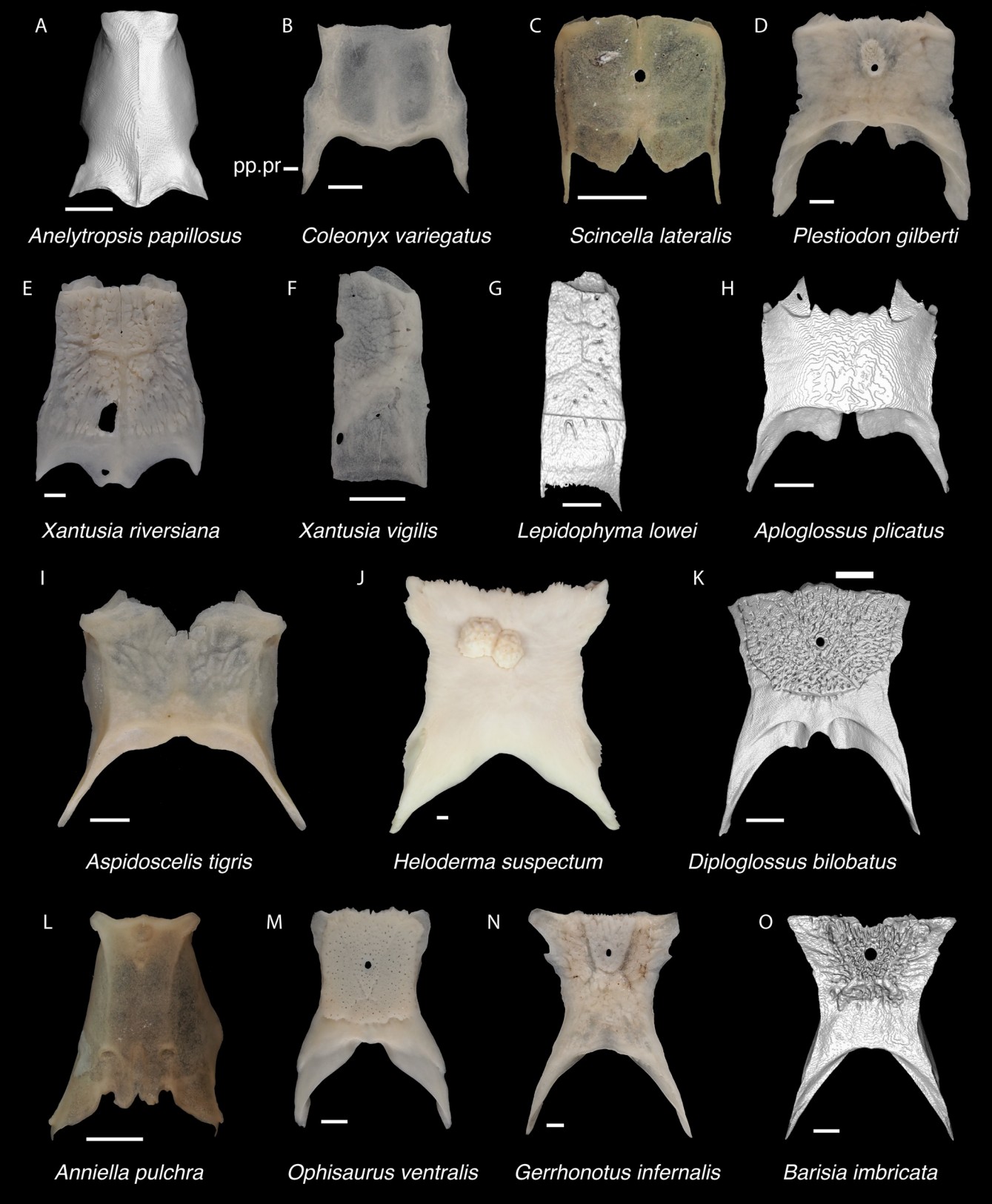

pp.pr

*Anelytropsis papillosus*     *Coleonyx variegatus*     *Scincella lateralis*     *Plestiodon gilberti*

*Xantusia riversiana*     *Xantusia vigilis*     *Lepidophyma lowei*     *Aploglossus plicatus*

*Aspidoscelis tigris*     *Heloderma suspectum*     *Diploglossus bilobatus*

*Anniella pulchra*     *Ophisaurus ventralis*     *Gerrhonotus infernalis*     *Barisia imbricata*

**Fig 40. Non-pleurodontan parietals.** Parietals in dorsal view–**A**. *Anelytropsis papillosus* UF 86708; **B**. *Coleonyx variegatus* TxVP M-12109; **C**. *Scincella lateralis* TxVP M-4489; **D**. *Plestiodon gilberti* TxVP M-8587; **E**. *Xantusia riversiana* TxVP M-8505; **F**. *Xantusia vigilis* TxVP M-12130 right parietal; **G**. *Lepidophyma lowei* LACM 143367 right parietal; **H**. *Aploglossus plicatus* TNHC 34481; **I**. *Aspidoscelis tigris* TxVP M-15667; **J**. *Heloderma suspectum* TxVP M-8593; **K**. *Diploglossus bilobatus* TNHC 31933; **L**. *Anniella pulchra* TxVP M-8678; **M**. *Ophisaurus ventralis* TxVP M-8585; **N**. *Gerrhonotus infernalis* TxVP M-13441; **O**. *Barisia imbricata* TNHC 76984. Scale bars = 1 mm. **Abbreviations**: pp.pr, postparietal process.

*Identification*. Dentaries are assigned to Anguimorpha based on the presence of a discrete surangular process [108, 125] and a posteriorly projecting splenial spine [45, 107] (Fig 34). Dentaries are assigned to Anguidae based on the presence of an intramandibular septum with a free posteroventral margin although that morphology is absent in some anguines [109–111] and is absent in TxVP 41229–27759. Fossils differ from other NA anguimorphs except for *Ophisaurus* in having a surangular spine [109]. A surangular spine was reported in some diploglossines [76, 112]; however, that feature is likely a posterior extension on the intramandibular septum described in other anguids [77, 109]. Furthermore, diploglossines differ from the fossils in lacking a splenial spine and instead have the anterior inferior alveolar foramen completely within the splenial. Fossils are assigned to Anguinae.

**Coronoid.** *Description*. TxVP 41229–28445 is a left coronoid (Fig 37K). The coronoid process is short and rounded and the anteromedial process is elongated but is missing the distal tip. The posteromedial process is directed posteriorly and has a notch on the posterodorsal end. There is a short medial crest that extends from the coronoid process and diminishes on the posteromedial process. There is a large, anteriorly projecting lateral process. The anteromedial process has a medial splenial facet that, together with an anteriorly projecting lateral process, forms a narrow articulation facet for the coronoid process of the dentary. There is no lateral crest and the facet for the dorsal articulation with the surangular is narrow.

*Identification*. The fossil coronoid differs from snakes in having distinct anteromedial and anterolateral processes that serve to clasp the dentary [23]. The fossil is further differentiated from snakes in having an expansive ventral articulation surface that is curved to articulate with both the dorsal and lateral surfaces of the surangular [27]. The fossil coronoid differs from xantusiids and some pleurodontans in having an anterolateral process [23]. The fossil differs from many NA pleurodontans in having a distinct anterolateral process (see discussion in Crotaphytidae section above). The fossil differs from remaining NA pleurodontans (iguanids, *Enyalioides heterolepis*, *Anolis*, and *Leiocephalus*) in having a more posteriorly oriented posteromedial process. The fossil differs from examined *Aspidoscelis* and *Ameiva* in lacking a deeply notched posterior edge forming dorsal and ventral rami [53] (Fig 32). Furthermore, *Aspidoscelis*, *Ameiva*, and *Pholidoscelis* differ in having a lateral crest running from the apex of the coronoid process anteroventrally onto the anterolateral process [2, 53]. The fossil differs from alopoglossids in having a low coronoid process [37], but the fossil shares with several gymnophthalmids a low coronoid process and widely divergent anteromedial and posteromedial processes [24, 103]. The fossil differs from examined gymnophthalmids with a low coronoid process in having a relatively broader posteromedial process (see Figs 7–9 of [103] and Fig 2 of [126]). Examined *Coleonyx variegatus*, *Coleonyx brevis*, *Sphaerodactylus roosevelti*, and *Thecadactylus rapicauda* differ from the fossil in having a thinner posteromedial process [54, 104, 105]. The posteromedial process is slightly wider in *Coleonyx elegans* and *Coleonyx mitratus* compared to other species [54] but examined *Coleonyx* including *Coleonyx elegans* differ in having a pronounced lateral crest on the coronoid process (see also Fig 7 of [54]). The lateral crest on the coronoid process is absent in the fossil, but in examined *Plestiodon* and *Scincella*, the crest is pronounced and obliquely oriented. Furthermore, the anterolateral process is more anteriorly directed in the fossil but more ventrally directed in *Plestiodon* (see also Fig 4 of [47]). Based on differences from other NA lizards, the fossil coronoid is referable to

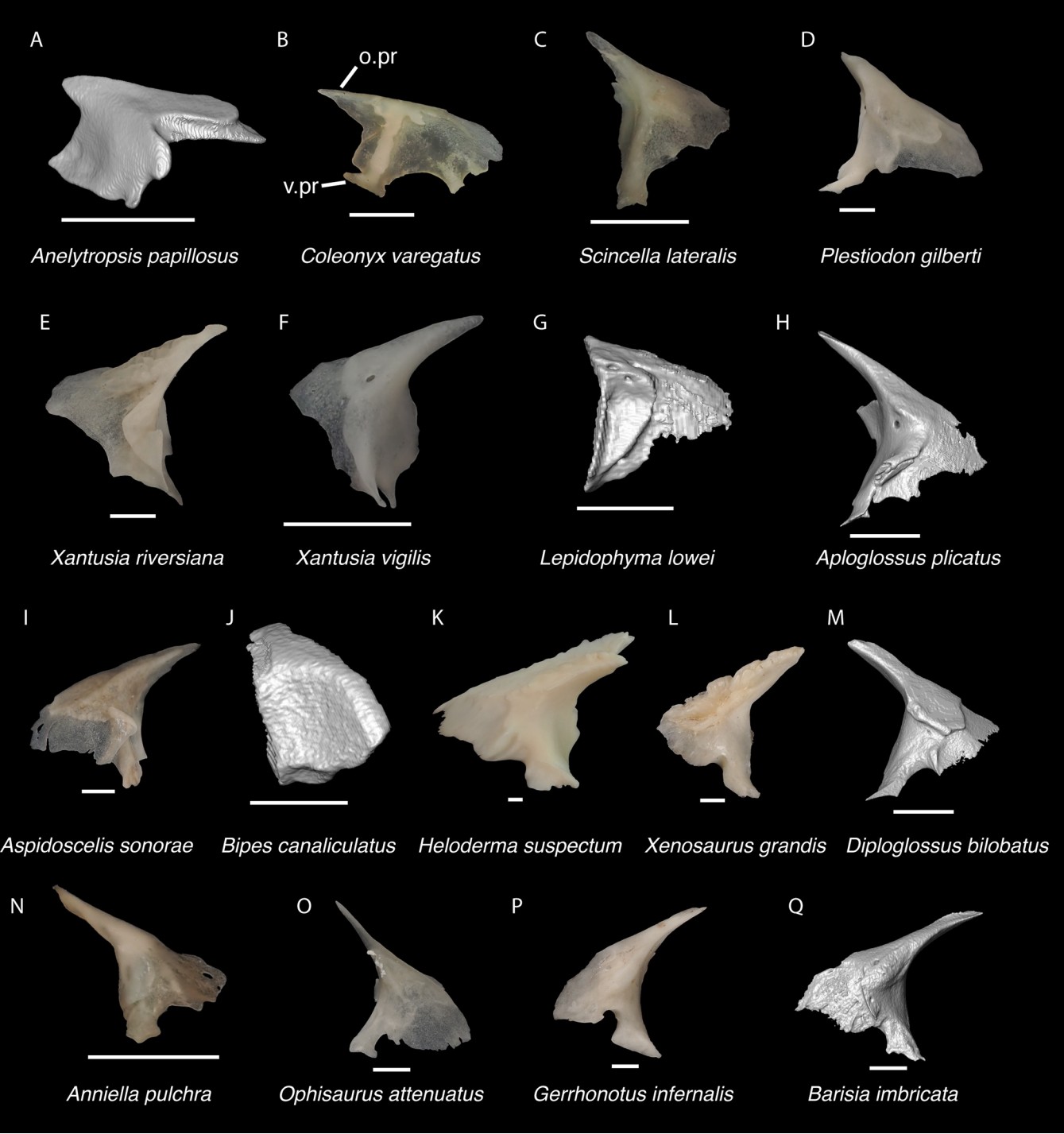

**Fig 41. Non-pleurodontan prefrontals.** Prefrontals in lateral view–**A**. *Anelytropsis papillosus* UF 86708 right prefrontal; **B**. *Coleonyx variegatus* TxVP M-13892 right prefrontal; **C**. *Scincella lateralis* TxVP M-4489 right prefrontal; **D**. *Plestiodon gilberti* TxVP M-8587 right prefrontal; **E**. *Xantusia riversiana* TxVP M-8505 left prefrontal; **F**. *Xantusia vigilis* TxVP M-12130 left prefrontal; **G**. *Lepidophyma lowei* LACM 143367 right prefrontal; **H**. *Aploglossus plicatus* TNHC 34481 right prefrontal; **I**. *Aspidoscelis sonorae* TxVP M-15670 left prefrontal; **J**. *Bipes canaliculatus* CAS 134753 right prefrontal; **K**. *Heloderma suspectum* TxVP M-9001 left prefrontal; **L**. *Xenosaurus grandis* TxVP M-8960 left prefrontal; **M**. *Diploglossus bilobatus* TNHC 31933 right prefrontal; **N**. *Anniella pulchra* TxVP M-8678 right prefrontal; **O**. *Ophisaurus attenuatus* TxVP M-8979 right prefrontal; **P**. *Gerrhonotus infernalis* TxVP M-13441 left prefrontal; **Q**. *Barisia imbricata* TNHC 76984 left prefrontal. Scale bars = 1 mm. **Abbreviations**: o.pr, orbital process; v.pr, ventral process.

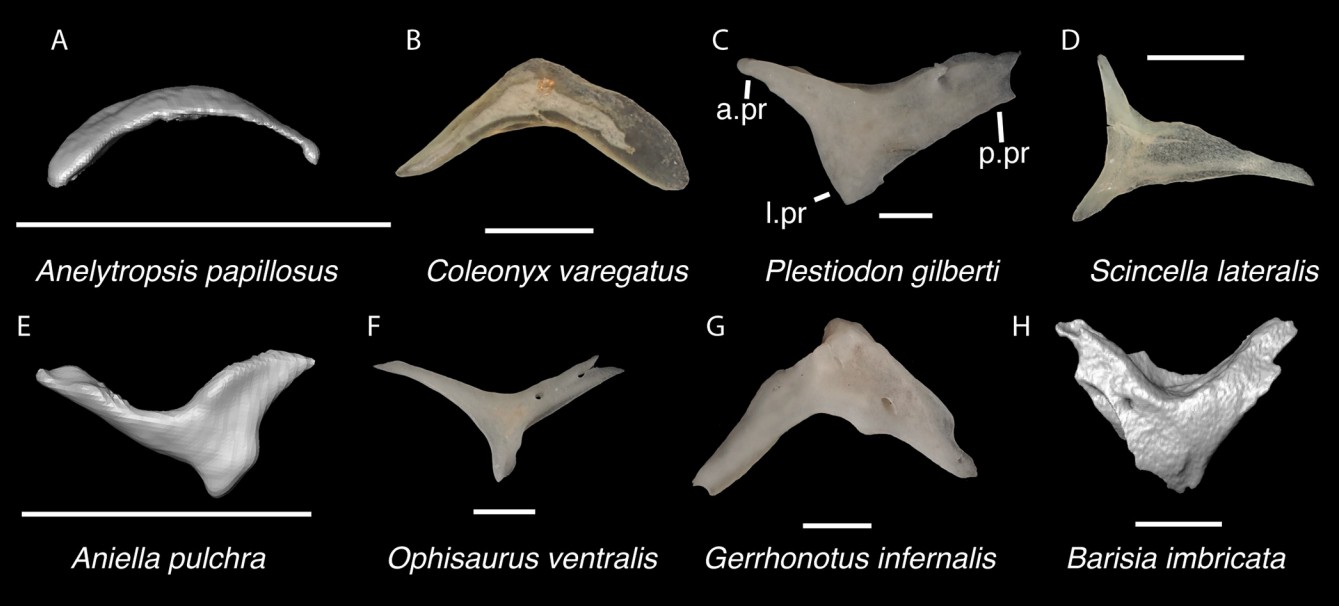

**Fig 42. Non-pleurodontan postfrontals.** Postfrontals in dorsal view with anterior to the left–**A**. *Anelytropsis papillosus* UF 86708 right postfrontal; **B**. *Coleonyx varegatus* TxVP M-13892 right postfrontal; **C**. *Plestiodon gilberti* TxVP M-8587 left postfrontal; **D**. *Scincella lateralis* TxVP M-4489 left postfrontal in dorsal view; **E**. *Aniella pulchra* FMNH 130479 left postfrontal; **F**. *Ophisaurus ventralis* TxVP M-8585 left postfrontal; **G**. *Gerrhonotus infernalis* TxVP M-13441 right postfrontal; **H**. *Barisia imbricata* TNHC 76984 left postfrontal. Scale bars = 1 mm. **Abbreviations**: a.pr, anterior process; l.pr, lateral process; p.pr, posterior process.

Anguimorpha. *Xenosaurus*, except for *Xenosaurus rackhami*, differ from the fossil in having a foramen on the anterolateral process [19, 24]. *Heloderma* differs in having a dorsoventrally expanded anteromedial process [24]. *Anniella* differs in having a facet for the coronoid process of the dentary extending on the anterior face of the coronoid process [24]. The fossil is assigned to Anguidae. The coronoid process of examined gerrhonotines and diploglossines is generally taller and more distinct than that of the fossil and of some *Ophisaurus*. On that basis, the fossil was identified to Anguinae.

**Compound bone.** *Description*. TxVP 41229–29001 is a left compound bone missing the anterior portion of the prearticular and posterior tip of the retroarticular process (Fig 37L). The adductor fossa is narrow. The retroarticular process is broadened and medially oriented. The dorsal surface of the retroarticular process is depressed. The surangular is short and the dorsal margin is rounded. There is a distinct squared-off tubercle anterior to the articular surface. There is a ventral angular articulation facet and an anterolateral coronoid facet. There is a foramen on the surangular just posterior to the adductor fossa. On the lateral surface there are two anterior surangular foramina and one posterior surangular foramen. There is also a dorsal foramen just anterior to the medial process.

*Identification*. The fossil compound bone shares with anguimorphs, geckos, and scincids a medially directed and broadened retroarticular process [23]. Geckos differ from the fossil in having a distinct notch on the medial margin of the retroarticular process [23]. Scincids differ in having a tubercle or flange on the medial margin of the retroarticular process [23], but that feature was not obvious in all examined specimens, particularly in *Scincella*. Examined scincids differ in having a narrower and taller tubercle anterior to the articular surface and in having an adductor fossa that extends farther posteriorly. Examined *Scincella* differ in having a comparatively elongated and more rectangular retroarticular process. Based on differences from other NA lizards, the fossil is referable to Anguimorpha.

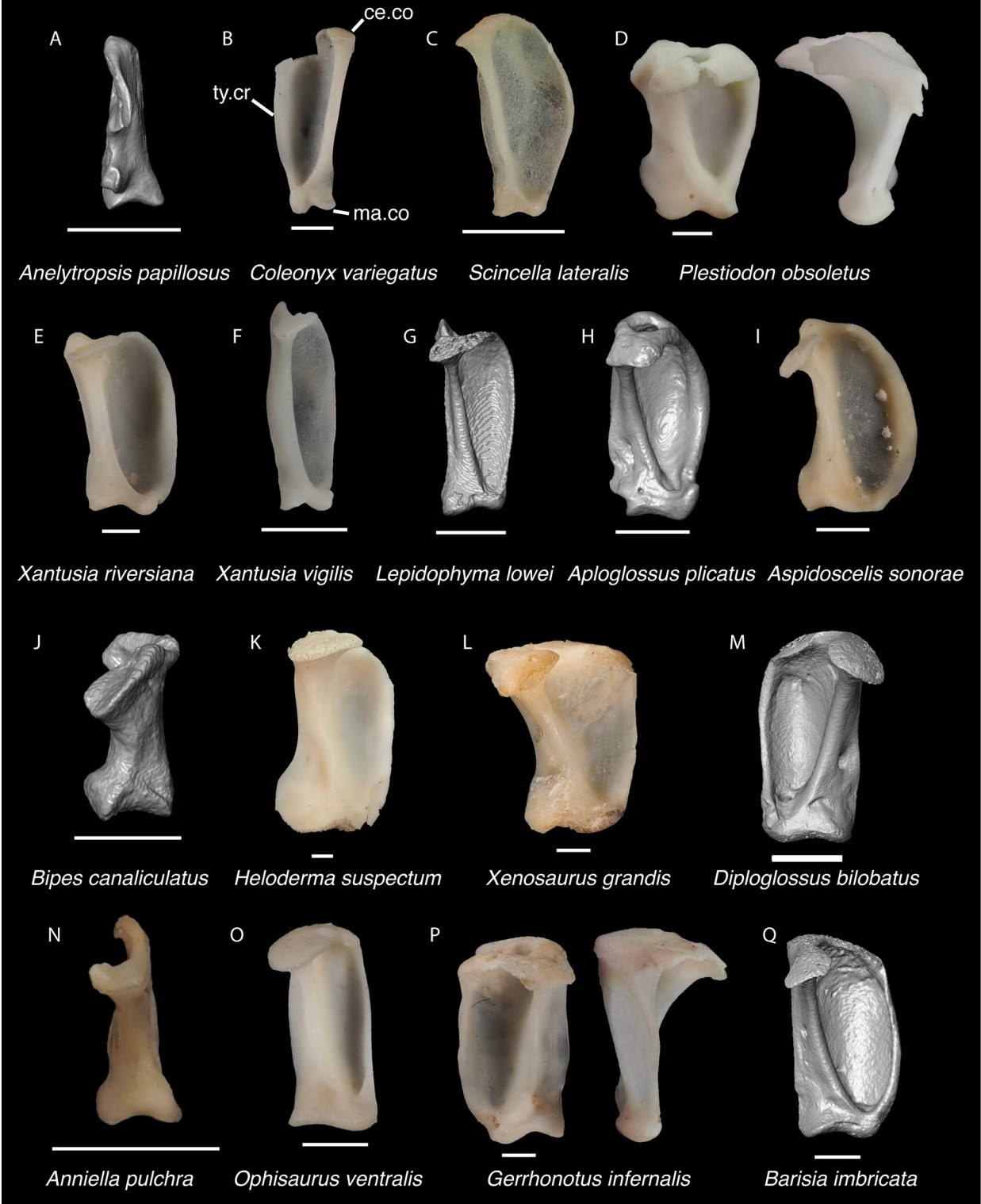

**Fig 43. Non-pleurodontan quadrates.** Quadrates in posterior and lateral views–**A**. *Anelytropsis papillosus* UF 86708 right quadrate in posterior view; **B**. *Coleonyx variegatus* TxVP M-12109 left quadrate in posterior view; **C**. *Scincella lateralis* TxVP M-4489 right quadrate in posterior view; **D**. *Plestiodon obsoletus* TxVP M-8574 right quadrate in posterior and lateral views; **E**. *Xantusia riversiana* TxVP M-8505 right quadrate in posterior view; **F**. *Xantusia vigilis* TxVP M-12130 right quadrate in posterior view; **G**. *Lepidophyma lowei* LACM 143367 right quadrate in posterior view; **H**. *Aploglossus plicatus* TNHC 34481 right quadrate in posterior view; **I**. *Aspidoscelis sonorae* TxVP M-15670 right quadrate in

posterior view; **J**. *Bipes canaliculatus* CAS 134753 right quadrate in posterior view; **K**. *Heloderma suspectum* TxVP M-8593 right quadrate in posterior view; **L**. *Xenosaurus grandis* TxVP M-8960 right quadrate in posterior view; **M**. *Diploglossus bilobatus* TNHC 31933 left quadrate in posterior view; **N**. *Anniella pulchra* TxVP M-8678 right quadrate in posterior view; **O**. *Ophisaurus ventralis* TxVP M-8585 right quadrate in posterior view; **P**. *Gerrhonotus infernalis* TxVP M-13441 left quadrate in posterior and lateral views; **Q**. *Barisia imbricata* TNHC 76984 right quadrate in posterior view. Scale bars = 1 mm. **Abbreviations**: ce.co, cephalic condyle; ma.co, mandibular condyle; ty.cr, tympanic crest.

Among anguimorphs, *Xenosaurus* differs in having a deep subcoronoid fossa [19]. Examined *Anniella* differ in having a more retroarticular process that slants medially, and *Heloderma* have a more slender retroarticular process [24]. The fossil is referable to Anguidae. Among anguids, the fossil differs from many examined gerrhonotines and diploglossines in having a short adductor fossa with a distinctly separate posteromedial surangular foramen. Some examined gerrhonotines also have a short adductor fossa with a distinctly separate posteromedial surangular foramen but differ in having a more mediolaterally expanded retroarticular process. On this basis, the fossil is referred to Anguinae.

## Scincidae Gray, 1825 [102]

Referred specimens: Compound bone, 41229–26992 left, 41229–26998 right.

**Compound bone.**   *Description*. TxVP 41229–26992 is a right compound bone missing only the anterior portion of the prearticular (Fig 45B). The adductor fossa is anteroposteriorly long. The retroarticular process is narrow and elongated posteriorly. There is a medial ridge that extends from the posterior margin of the articular surface to the medial edge of the retroarticular process. The medial edge of the retroarticular process bears a small tubercle. The dorsal and ventral surfaces of the retroarticular process are characterized by slight depressions. There is a tall, rounded tubercle anterior to the articular surface. The anterodorsal margin of the surangular has a dorsally expanded crest for articulation with the coronoid. There is a ventral angular articulation facet and a V-shaped facet on the anterolateral surface of the surangular for articulation with the dentary. There are both anterior and posterior surangular foramina. There is also a dorsal foramen just anterior to the dorsal tubercle and a foramen posterior to the articular surface. TxVP 41229–26998 does not differ substantially (Fig 45C).

*Identification*. Fossils share with scincids, anguimorphs, and geckos a medially directed retroarticular process [23]. Some snakes have a medially directed retroarticular process but differ from the fossils in having relatively narrower process that lacks a concavity on the dorsal surface [23]. Fossils differ from geckos in lacking a notch on the medial margin of the retroarticular process [23]. Fossils differ from anguimorphs and geckos in having a medial tubercle on the retroarticular process, although a small tubercle was found in a few anguimorph taxa (e.g., *Diploglossus millepunctatus* TxVP M-9010). Fossils differ from anguimorphs in having a V-shaped facet on the anterolateral surface of the surangular (see also Fig 10E of [23]) which is present on all examined skinks except specimens of *Plestiodon gilberti*. Fossils also differ from anguimorphs in having a narrower tubercle anterior to the articular surface. Examined anguimorphs tend to have a shorter and more squared-off tubercle. Fossils were assigned to Scincidae on this basis. Fossils share with examined *Scincella*, mabuyines [116], and *Plestiodon tetragrammus* a relatively narrow retroarticular process. We refrain from making a more refined identification pending examination of additional skeletal material.

## Sphenomorphinae Welch, 1982 [127]

### *Scincella* Mittleman, 1950 [128]

Referred specimens: Frontal, 41229–26870.

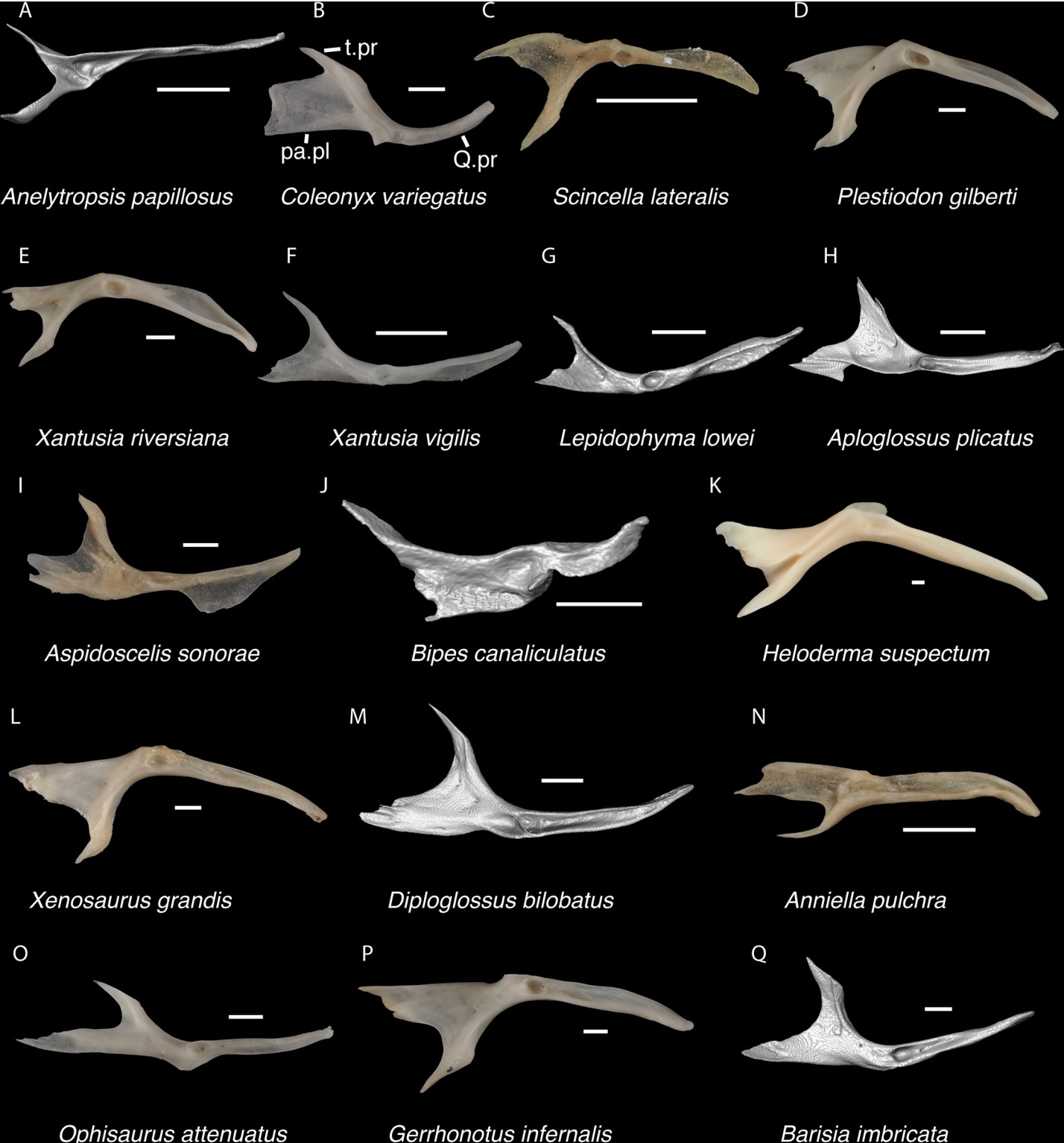

**Fig 44. Non-pleurodontan pterygoids.** Pterygoids in dorsal view–**A**. *Anelytropsis papillosus* UF 86708 left pterygoid; **B**. *Coleonyx variegatus* TxVP M-12109 right pterygoid; **C**. *Scincella lateralis* TxVP M-4489 left pterygoid; **D**. *Plestiodon gilberti* TxVP M-8587 left pterygoid; **E**. *Xantusia riversiana* TxVP M-8505 left pterygoid; **F**. *Xantusia vigilis* TxVP M-12130 right pterygoid; **G**. *Lepidophyma lowei* LACM 143367 right pterygoid; **H**. *Aploglossus plicatus* TNHC 34481 right pterygoid; **I**. *Aspidoscelis sonorae* TxVP M-15670 right pterygoid; **J**. *Bipes canaliculatus* CAS 134753 right pterygoid; **K**. *Heloderma suspectum* TxVP M-9001 left pterygoid; **L**. *Xenosaurus grandis* TxVP M-8960 left pterygoid; **M**. *Diploglossus bilobatus* TNHC 31933 right pterygoid; **N**. *Anniella pulchra* TxVP M-8678 left pterygoid; **O**. *Ophisaurus*

*attenuatus* TxVP M-8979 right pterygoid; **P**. *Gerrhonotus infernalis* TxVP M-13441 left pterygoid; **Q**. *Barisia imbricata* TNHC 76984 right pterygoid. Scale bars = 1 mm.
**Abbreviations**: pa.pl, palatal plate; Q.pr, quadrate process; t.pr, transverse process.

**Frontal.**   *Description*. TxVP 41229–26870 is a frontal (Fig 45A). Sculpturing is restricted to the posterodorsal surface of the bone. There are anterolateral facets for the prefrontal, and the anterior face of the frontal is defined by two short dorsal nasal facets formed by two anterolateral processes and an anteromedial process. The interorbital margins are waisted, and the posterolateral processes extend laterally. The posterior edge is slightly wavy and has interlacing articulation facets for the parietal. There are small postfrontal facets laterally on the posterolateral processes. The cristae cranii are low and diverge posteriorly along the lateral margins of the ventral surface. There is a broad ridge on the ventral surface near the interorbital margin creating separate anterior and posterior depressions.

*Identification*. The fossil frontal shares with Pleurodonta, Gymnophthalmoidea, Mabuyinae and *Scincella* a fused frontal with reduced cristae cranii [23, 36]. Examined teiids differ in having a frontal that is relatively more elongate relative to the width and gymnophthalmids and alopoglossids differ in having frontal tabs [23, 37]. Examined NA pleurodontans differ from the fossil in lacking a distinct anteromedial process that extends far anterior relative to the anterolateral processes. Furthermore, among examined pleurodontans, the cristae cranii converge to a greater degree compared to the fossil. The fossil can be attributed to Scincidae. Examined *Mabuya* differ from *Scincella* in having more dermal sculpting on the dorsal surface and a relatively wider anterior portion. The fossil is referred to *Scincella* on that basis.

## Scincinae Gray, 1825 [102]

Illustrated specimens referenced in the text: Compound bone, 41229–28053 right, 41229–28060 right, 41229–28649 left; Coronoid, 41229–28062 right; Dentary, 41229–27353 right; Frontal, 41229–26008 right; Jugal, 41229–29020 right, 41229–27750 left; Maxilla, 41229–27663 right; Parietal, 41229–27317; Prefrontal, 41229–27716 left, 41229–29162 right; Pterygoid, 41229–27181 left, 41229–28300 right; Quadrate, 41229–28867, 41229–25904 left; See S3 Table for complete list of specimens assigned to Scincinae.

**Maxilla.**   *Description*. TxVP 41229–27663 serves as the basis for our description (Fig 45D). TxVP 41229–27663 is a right maxilla with 19 tooth positions. Teeth are unicuspid with medial striations. The facial process is tall, broad, and gently curves dorsomedially. The medial margin of the facial process has a distinct nasal facet, and the anterior margin has a small projection. The premaxillary process is short and bifurcated with a longer lateral projection and a shorter medial lappet. The crista transversalis is low in height and trends medially from the facial process and defines the posterior border of a shallow depression on the premaxillary process. There is a broad palatal shelf with a triangular palatine process. The medial surface of the facial process bears a distinct ridge that extends anteriorly onto the palatal shelf and defines the posterior border of a deep medial recess on the anteromedial surface of the facial process. The lateral wall of the posterior orbital process is tall, and the posterior end of the process is slightly bifurcated. The dorsal surface of the postorbital process has an elongate, shallow jugal groove. There is an opening for the superior alveolar canal anterior to the facial process. There are two superior alveolar foramina on the palatal shelf medial to the palatine process, 13 lateral nutrient foramina dorsal to the tooth row, and two foramina anterodorsally on the facial process.

*Identification*. Fossil maxillae share with some scincids and anguimorphs unicuspid teeth with striated crowns [52, 115]. Fossils differ from examined anguimorphs, except some *Diploglossus* [76, 112], in having a large notch at the end of the posterior orbital process. Examined anguids differ from the fossils in having a comparatively shorter lateral lamina along the

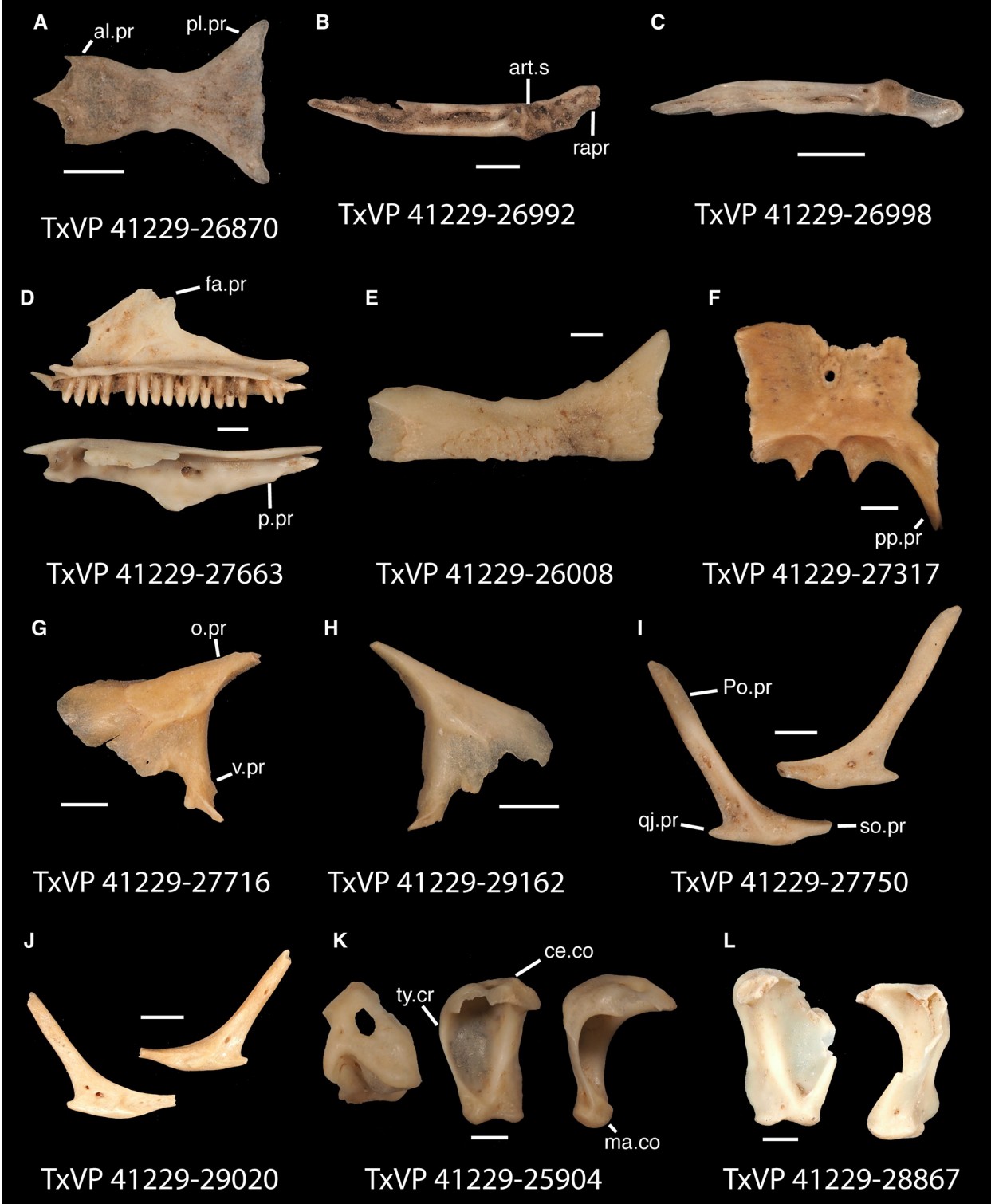

**Fig 45.** Fossil scincids, A: *Scincella*, B–C: Scincidae, D–L: Scincinae. **A**. TxVP 41229–26870 Dorsal view of frontal; **B**. TxVP 41229–26992 Dorsal view of left compound bone; **C**. TxVP 41229–26998 Dorsal view of right compound bone; **D**. TxVP 41229–27663 Medial and dorsal view of right maxilla; **E**. TxVP 41229–26008 Dorsal of frontal; **F**. TxVP 41229–27317 Dorsal view of parietal; **G**. TxVP 41229–27716 Lateral view of left prefrontal; **H**. TxVP 41229–29162 Lateral view of right prefrontal; **I**. TxVP 41229–27750 Medial and lateral view of left jugal; **J**. TxVP 41229–29020 Lateral and medial view of right jugal; **K**. TxVP 41229–25904 Dorsal, posterior, and lateral view of left quadrate; **L**. TxVP 41229–28867 Posterior

and lateral view of right quadrate. Scale bars = 1 mm. **Abbreviations**: al.pr, anterolateral process; art.s, articular surface; ce.co, cephalic condyle; fa. pr, facial process; ma.co, mandibular condyle; o.pr, orbital process; pl.pr, posterolateral process; Po.pr, postorbital process; pp.pr, postparietal process; p.pr, posterior process; qj.pr, quadratojugal process; rapr, retroarticular process; so.pr, suborbital process; ty.cr, tympanic crest; v.pr, ventral process.

posterior orbital process and a crista transversalis that trends anteromedially instead of medially from the facial process [52]. Fossils can be further differentiated from the NA anguimorphs *Anniella* (6–7 tooth positions) and *Heloderma* (6–10 tooth positions) in having more tooth positions [24, 123] and lacking sharp recurved teeth [24]. *Anniella* lack a crista transversalis [52] and *Xenosaurus* have fused osteoderms on the lateral surface of the facial process [19]. Fossils are assigned to Scincidae based on those differences with other NA lizards.

Examined *Scincella* and at least some mabuyine skinks [116] differ from the fossils in lacking striations on the crowns [117], having a crista transversalis that trends anteromedially from the facial process, and having a much narrower palatal shelf posterior to the palatal process. Fossils are assigned to Scincinae on that basis. We refrain from making a more refined identification pending examination of additional skeletal material of scincine lizards.

**Frontal.** *Description*. TxVP 41229–26008 serves as the basis for our description and is an unfused right frontal (Fig 45E). There is a small area of co-ossified osteoderms on the dorsal surface. There is an anterolateral prefrontal facet and an anterodorsal nasal facet. The anterior face of the bone is flat and has a well-developed anterolateral process. The interorbital margin is only slightly narrower than the anterior end. The posterolateral process gently curves laterally and projects posterolaterally past the posteromedial edge. The posterior edge has a shallow parietal articulation facet. There is a postfrontal facet along the posterolateral edge. The ventral portion of the crista cranii (subolfactory processes of [120]) is broken, but the crest is well-developed, tall, and anteroposteriorly long. Ventrally, there is a series of small transverse ridges and corresponding grooves medial to the crista cranii.

*Identification*. The fossil shares with anguimorphs and scincids co-ossified osteoderms with dorsal sculpturing [23], well-developed and ventrally directed cristae cranii, and an unfused frontal [23]. Osteoderm sculpturing has a more pitted texture in many anguimorphs [35, 45]. Examined *Ophisaurus* and other anguines, except for *Anguis fragilis* [114], also differ from the fossil in having a pointed anterior portion of the unfused frontal. Fossils are assigned to Scincidae on that basis.

*Scincella* and mabuyine skinks are excluded because they both have a fused frontal [36]. In one examined extant specimen, *Plestiodon skilitonianus* TxVP M-8498, the frontals are separate but are held together by fused osteoderms. Fossils can be assigned to Scincinae on that basis.

**Parietal.** *Description*.The exemplar TxVP 41229–27317 is missing the anterolateral corner and ends of the postparietal processes (Fig 45F). The parietal table is rectangular and smooth with scattered small foramina. The anterior edge is straight with small frontal tabs. The posterior edge between the postparietal processes has two well-developed nuchal fossae separated by a small ridge, as well as two long posterior projections that create a medial notch. The right postparietal process is broad and flat at its base. There is a flat muscle attachment surface lateral to the ventrolateral crests. The ventrolateral crests are low and are broken at the ends. There is a large ventral swelling at the pit for the processus ascendens. The posterior projections and a ventrolateral ridge on the postparietal process define small ventral depressions. There is a large parietal foramen within the middle of the parietal table.

*Identification*. Parietals are assigned to Scincomorpha based on the presence of long posterior projections (median extensions of [24]) on the posterior edge of the parietal table between the postparietal processes [24, 27]. Xantusiids, except for *Cricosaura*, some *Lepidophyma* (e.g.,

*L. gaigeae*), and some *Xantusia riversiana*, differ in having a paired parietal [24, 46, 129]. *Cricosaura*, *Lepidophyma*, and *Xantusia riversiana* differ in lacking long postparietal processes [24, 46]. Posterior projections on the posterior edge of the parietal table are found in some anguid lizards (see Fig 9 of [55]), but the projections in anguids are much smaller than those found on the fossils and scincids. Fossils are assigned to Scincidae on that basis. Examined *Scincella* and mabuyine skinks have less developed adductor crests compared to that in the fossils [68] and lack long, thin, ventral projections, the base of which is preserved in fossils. Fossils are assigned to Scincinae on that basis.

**Prefrontal.** *Description.* TxVP 41229–27716 is a left prefrontal (Fig 45G). It is triradiate with a short orbital process, a short ventral process, and an elongate anterior facial sheet. The anterior facial sheet has a broad articulation facet for the facial process of the maxilla, which extends dorsally on the prefrontal. There is a small ridge on the lateral surface near the base of the orbital process. The ventral process is pointed, projects posteriorly, and has a small lateral ridge. There is a distinct notch for the lacrimal foramen, with the ventral process forming the posterior border. Medially, there is a smooth, rounded, and concave wall of the olfactory chamber. Dorsal to the olfactory chamber and medially on the orbital process there is a deep groove for articulation with the frontal. The orbitonasal flange is broad and slightly broken on the medial margin. There is a small foramen anterodorsal to the notch for the lacrimal foramen. TxVP 41229–29162 differs from TxVP 41229–27716 in having a small overhanging lamina of bone dorsal to the notch for the lacrimal foramen and in having a notch in the medial margin of the orbitonasal flange (Fig 45H).

*Identification.* The fossil differs from NA pleurodontans in lacking a lateral prefrontal boss [20, 23, 27, 52], lacking a strong lateral canthal ridge (reported in *Anolis* and *Polychrus*; [20]), lacking a supraorbital spine (present in *Phrynosoma* and *Corytophanes*; [29]), and lacking a thin, crescent-shaped prefrontal with a distinct laterally projecting lamina (present in examined phrynosomatines). NA teiids, gymnophthalmids, and alopoglossids differ from the fossil in having a laterally projecting lamina (lacrimal flange of [62]) with a distinct articulation facet for the facial process of the maxilla [53]. Xantusiids are excluded because the lacrimal is fused to the prefrontal [46], the suborbital foramen is nearly or entirely enclosed within the prefrontal, and there is a vertical articulation ridge (most *Xantusia*) or flange (e.g., *X. riversiana*) that articulates with the maxilla. Examined *Coleonyx* differ in having an orbitonasal flange that extends farther medially compared to the fossil. *Sphaerodactylus roosevelti* has a smaller notch for the lacrimal foramen [104]. Examined anguimorphs differ in lacking an oblong, elongate anterior process. Based on differences with other NA lizards, the fossil is identified to Scincidae. The prefrontal of examined *Scincella* lack a posteriorly projected ventral process, have a less elongate anterior process, and are much smaller than the fossil. Mabuyines differ in having a distinct notch on the medial margin of the orbital process for articulation with the frontal [130]. The fossil is assigned to Scincinae on that basis. The fossil, *Plestiodon gilberti*, and *P. multivirgatus* share a relatively short posterodorsal process, but the process is long in examined *P. obsoletus*.

**Jugal.** *Description.* TxVP 41229–27750 is a right jugal (Fig 45I). There is a distinct maxillary facet on the lateral and ventral surface of the suborbital process. The suborbital process thins anteriorly. There is a distinct, pointed quadratojugal process. The postorbital process is long, posterodorsally oriented, and has a shallow postorbital facet on the anteromedial surface. There is a medial ridge that is located on the midline of the suborbital process and anteriorly on the postorbital process. The medial ridge defines the anterior border of a small depression on the medial surface at the base of the quadratojugal process and the base of the postorbital process. There are two medial and two lateral foramina at the inflection point, as well as one medial foramen on the postorbital process. TxVP 41229–29020 (Fig 45J) differs from TxVP

41229–27750 in having a round medial boss near the inflection point and in only having one medial foramen near the inflection point.

*Identification.* The fossils share with other NA lizards except for dibamids, geckos, and some pleurodontans an angulated jugal [45]. Fossils differ from NA pleurodontans in having a quadratojugal process with a dorsally oriented postorbital process [29, 82]. Gymnophthalmoids differ in having a distinct medial ectopterygoid process, which is also sometimes present, albeit comparatively smaller, in gymnophthalmids and alopoglossids [24]. Examined xantusiids differ in lacking a quadratojugal process and in having a short suborbital process [24]. *Xantusia riversiana* and *Lepidophyma* have an anteroposteriorly widened postorbital process [46]. Among NA anguimorphs, *Anniella* differs in having a reduced jugal [84] and *Heloderma* differs in having the postorbital and suborbital processes form a right angle [24]. *Xenosaurus* has co-ossified osteoderms or sculpturing on the lateral surface of the jugal [19]. The fossil differs from anguids in having a medial ridge located midline on the suborbital process and anteriorly on the postorbital process [50]. Furthermore, fossils differ from examined anguids in having a distinct notch on the ventral margin of the suborbital process. Examined *Scincella* and *Mabuya* differ in having a jugal that is more smoothly curved with a relatively thinner and shorter suborbital process [130]. Fossils are assigned to Scincinae on that basis.

**Quadrate.** *Description.* TxVP 41229–25904 is a left quadrate (Fig 45K). The central column narrows and curves laterally at its base. There is a minute pterygoid lappet and a well-developed, anteromedially directed medial crest. The conch is deep and gradually slants laterally from the central column. The cephalic condyle projects far posteriorly and there is extensive ossification dorsally enclosing the squamosal foramen, along with an anteriorly expanded dorsal tuber (*sensu* [114]). There is a foramen medial to the central column and three foramina on the anterior surface. TxVP 41229–28867 differs in having a large pterygoid lappet and a foramen on the posterior surface dorsal to the mandibular condyle (Fig 45L).

*Identification.* The fossils share a distinct pterygoid lappet with scincids, teiids, gymnophthalmids, some pleurodontans, *Heloderma*, *Xenosaurus*, and some gerrhonotines [23, 24, 55]. Examined NA pleurodontans differ in having a smaller pterygoid lappet if present. Furthermore, many NA pleurodontans differ in having a quadrate that, in posterior view, is much wider along the dorsal surface compared to the mandibular condyle. In *Anolis* and *Uma*, the lateral margins are parallel; however, examined *Anolis* differ in having a distinct boss at the ventromedial margin of the quadrate, and *Uma* differ in having a laterally slanted cephalic condyle and central column. *Xenosaurus* differ in having a shallower lateral conch [19] and examined *Heloderma* differ in having a dorsal head with more distinct anterior projection. The central column in the fossils is curved posteriorly to a greater degree than that of examined gerrhonotines. Gymnophthalmids and examined teiids differ in having a more curved tympanic crest [24]. Fossils were assigned to Scincidae on this basis. Examined *Scincella* differ in having a relatively narrow quadrate in anterior and posterior view and examined *Mabuya* differ in having less extensive ossification dorsally on the cephalic condyle. Fossils are assigned to Scincinae on that basis.

**Pterygoid.** *Description.* TxVP 41229–28300 is a right pterygoid (Fig 46B). The palatine process is broad with an anterior palatine facet. The transverse process has a distinct posterolateral corner and a triangular and anterolaterally pointed distal end. The transverse process has a shallow ectopterygoid facet. Ventrally, the transverse process has a large ridge that slightly curls anteriorly for insertion of the pterygomandibular muscle, but there is no dorsal ridge on the transverse process. On the palatal plate there is a dorsal ridge that trends anteriorly and forms the lateral border for a medial depression. There is a deep fossa columella and an epipterygoid crest anteromedial to the fossa columella. There is no postepipterygoid groove (*sensu* [124]) on the quadrate process. The quadrate process is elongated, and the medial

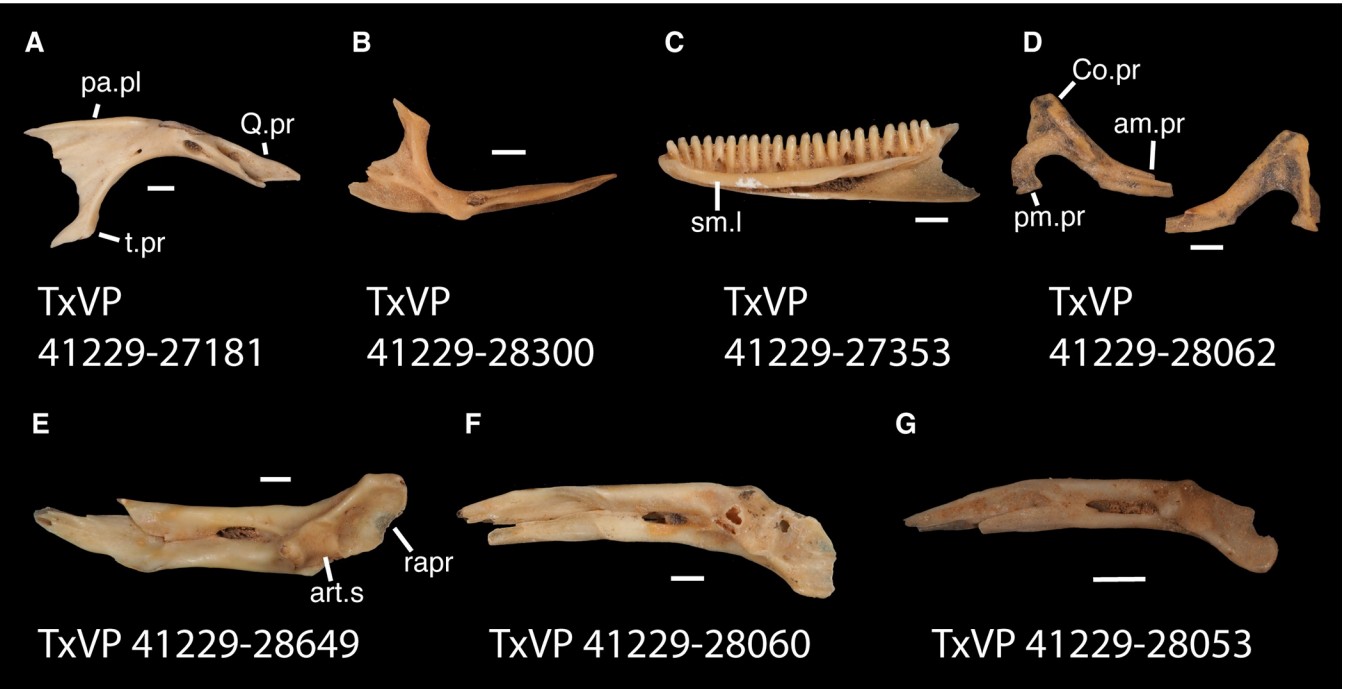

**Fig 46.** Fossil scincids, A–G: Scincinae. **A**. TxVP 41229–27181 Dorsal view of left pterygoid; **B**. TxVP 41229–28300 Dorsal view of right pterygoid; **C**. TxVP 41229–27353 Medial view of right dentary; **D**. TxVP 41229–28062 Lateral and medial view of right coronoid; **E**. TxVP 41229–28649 Dorsal view of left compound bone; **F**. TxVP 41229–28060 Dorsal view of right compound bone; **G**. TxVP 41229–28053 Dorsal view of right compound bone. Scale bars = 1 mm. **Abbreviations**: am.pr, anteromedial process; art.s, articular surface; Co.pr, coronoid process; pa.pl, palatal plate; pm.pr, posteromedial process; Q.pr, quadrate process; rapr, retroarticular process; sm.l, suprameckelian lip; t.pr, transverse process.

surface has a shallow groove that serves for insertion of the pterygoideus muscle. The ventro-lateral surface of the quadrate process bears a distinct groove. The medial margin of the ptery-goid, including the quadrate process and the palatal plate, is curved laterally. There is no medial projection at the floor of the basipterygoid fossa. There is a small patch of seven ptery-goid tooth positions containing four teeth and a dorsal foramen on the palatal plate. TxVP 41229–27181 is missing the dorsal ridge and the epipterygoid crest (Fig 46A).

*Identification.* Fossils share the presence of pterygoid teeth with scincids, teiids, some NA gymnophthalmids and alopoglossids (e.g., *Gymnophthalmus speciosus* and *Alopoglossus plica-tus*), some pleurodontans, and some anguimorphs [23, 24]. Examined NA pleurodontans, except *Petrosaurus*, differ in having the transverse process oriented more medially. *Petrosaurus* also lack pterygoid teeth [44]. Examined NA teiids differ in having an extensive, medially directed lamina of bone on the quadrate process [131]. *Ptychoglossus* differs in having a poster-omedial process along the medial edge of the palatal plate [37] and *Gymnophthalmus* differs in having a more slender palatal plate [24]. Examined anguimorphs differ in having a postepip-terygoid groove on the quadrate process [24]. *Xenosaurus* and *Anniella* can be further differen-tiated in lacking pterygoid teeth [24] and *Heloderma* differs in having a single row of pterygoid teeth located on an elevated ridge [24]. The medial edge of the pterygoid is more strongly curved in the fossils compared to examined anguids. Fossils were assigned to Scincidae on that basis. Examined *Scincella* and mabuyines differ in lacking pterygoid teeth [132]. Fossils were assigned to Scincinae on that basis.

**Dentary.** *Description.* TxVP 41229–27353 is a right dentary with 22 tooth positions and serves as the basis for our description (Fig 46C). Teeth are unicuspid with medial striations.

There is a pointed coronoid process, no surangular process, and an angular process that is broken posteriorly. The Meckelian groove is open ventrally, and the suprameckelian lip is tall. The intramandibular septum does not extend to the distal teeth. There is a deep dental gutter. There are four nutrient foramina arranged in a row on the lateral surface of the bone.

*Identification.* Fossils share with Scincomorpha and Anguimorpha unicuspid teeth with striated crowns [52, 115]. Anguimorphs differ from the fossils in having a posteriorly extended intramandibular septum near the posterior end of tooth row [23, 133]. Xantusiids differ in having a fused spleniodentary [24, 46, 63]. Fossils were assigned to Scincidae on this basis. Mabuyines have a closed or fused Meckelian groove [96, 116]. *Scincella* differs in lacking striations on the crowns and in having a closed but unfused Meckelian groove [117]. Fossils were assigned to Scincinae on that basis.

**Coronoid.** *Description.* TxVP 41229–28062 is a right coronoid and serves as the basis for our description (Fig 46D). The coronoid process is tall, rounded and slopes anteroventrally. The anteromedial process is elongated with a medial splenial facet. The posteromedial process is oriented ventrally and has a small posterior notch. There is a medial crest that extends from the coronoid process and extends onto the posteromedial process. There is a large, anteroventrally projecting lateral process with a dorsal dentary articulation facet. There is an anteroventrally oriented lateral crest that extends along the anterolateral process. There is a small articulation surface for the surangular, located ventral to the anterolateral process.

*Identification.* The fossil coronoid differs from snakes in having distinct anteromedial and anterolateral processes that serve to clasp the dentary [23]. The fossil is further differentiated from snakes in having an expansive ventral articulation surface that is curved to articulate with both the dorsal and lateral surfaces of the surangular [27]. The fossil coronoid differs from xantusiids and some pleurodontans in having an anterolateral process [23]. The fossil differs from examined NA pleurodontans with an anterolateral process (see Crotaphytidae section above) in having a lower and more rounded coronoid process. The fossil differs from examined *Aspidoscelis* and *Ameiva* in lacking a deeply notched posterior edge forming dorsal and ventral rami [53]. The fossil differs from gymnophthalmids and alopoglossids in having the posteromedial process oriented ventrally [24, 37, 103]. Examined geckos differ from the fossils in having an anterolateral process that is not directed as far ventrally [54, 104, 105]. Examined NA anguimorphs also differ in having an anterolateral process that does not project as far ventrally as in the fossils. Fossils can be assigned to Scincidae. Examined *Scincella* have a more posteriorly oriented posteromedial process and examined mabuyines have a taller and more pointed coronoid process [116]. Fossils are assigned to Scincinae on that basis.

**Compound bone.** *Description.* TxVP 41229–28649 is a left compound bone that is missing the anterior portion of the prearticular (Fig 46E). The adductor fossa is short. The retroarticular process is broad and medially oriented. There is a medial ridge that extends from the posterior margin of the articular surface to a medial tubercle on the retroarticular process. The dorsal surface of the retroarticular process is a semicircular depression, and the ventral surface is smooth. There is a tall, rounded tubercle anterior to the articular surface. The anterodorsal margin of the surangular has a dorsally expanded crest where it articulates with the coronoid. There is a ventral angular articulation facet and an anterolateral coronoid facet. There are anterior and posterior surangular foramina. There is also a dorsal foramen just anterior to the medial process, and a foramen posterior to the articular surface. TxVP 41229–28060 differs in having a low ridge lateral to the medial ridge on the retroarticular process (Fig 46F). TxVP 41229–28053 differs in lacking a medial boss and lacking an anterodorsal crest on the surangular (Fig 46G).

*Identification.* Fossils share a medially directed and broadened retroarticular process with scincids, anguimorphs, and geckos [23]. Fossils differ from geckos in lacking a notch on the

medial margin of the retroarticular process [23]. Fossils, except for TxVP 41229–28053, differ from anguimorphs and geckos in having a medial tubercle on the retroarticular process, although a small tubercle was found in a few anguimorph taxa as well (e.g., *Diploglossus mille-punctatus* TxVP M-9010). Fossils differ from anguimorphs in having a narrower tubercle anterior to the articular surface. Examined anguimorphs tend to have a shorter and more squared-off tubercle. In examined *Scincella* and mabuyines, [116], the retroarticular process is narrower compared to the fossils. The retroarticular process is also narrow in examined *Plestiodon tetragrammus*. Fossils were assigned to Scincinae on that basis. Based on the shape of the retroarticular process (and given the good possibility that the fossils represent *Plestiodon*), the compound bones likely represent a species other than *Plestiodon tetragrammus*.

## Teiidae Gray, 1827 [134]

### Teiinae Gray, 1827 [134]

Unless a specific apomorphy is provided, fossils were identified to Teiinae on the basis that Teiinae is the only subfamily within Teiidae that is known to inhabit North America during the Quaternary.

Illustrated specimens referenced in the text: Compound bone, 41229–27339 right; Coronoid, 41229–27699 left; Dentary, 41229–27358 right, 41229–27389 right, 41229–28245 right; Frontal, 41229–27360, 41229–27397; Maxilla, 41229–27345 left; Parietal, 41229–27148; Postorbitofrontal, 41229–27534 left; Prefrontal, 41229–27252 left; Premaxilla, 41229–27594; Pterygoid, 41229–27524 left; See S3 Table for complete list of specimens assigned to Teiinae.

**Premaxilla.** *Description*. TxVP 41229–27594 is a premaxilla with eight tooth positions (Fig 47A). Teeth are unicuspid. The rostral surface of the premaxilla is rounded. The nasal process is slightly broken posterolaterally, but it is evident that it is strongly curved posteriorly and tapers to a point. The process is slightly widened near the midpoint, and ventrally, there is a low keel. On the alveolar plate, there are lateral maxillary facets with small lateral notches. Posteriorly, the palatal plate is strongly incised and v-shaped. There is no incisive process. There are two bilateral ethmoidal foramina posterior to the base of the nasal process, but no anterior foramen.

*Identification*. The fossil is assigned to Gymnophthalmoidea based on the absence of an incisive process [37, 45, 83] and based on the absence of a large median tooth [52, 135]. Alopoglossids differ in having between 10 to 14 tooth positions [37]. Many gymnophthalmids differ in having a laterally flared nasal process [24, 37]. *Gymnophthalmus speciosus* has parallel margins of the nasal process, but that species differs from the fossil in having a nasal process with a more rectangular posterior end [24, 78]. With over 280 species of gymnophthalmid lizards, there is still much to learn about patterns of osteological variation in this group and so we tentatively assign the fossil to Teiidae.

**Maxilla.** *Description*. TxVP 41229–27345 is a left maxilla that serves as the basis for our description (Fig 47B). There are 20 tooth positions. The distal-most tooth is weakly tricuspid with a minute distal cusp, and the mesial teeth, except for the mesial-most tooth, are bicuspid with a smaller mesial cusp. There are substantial deposits of cementum at the tooth bases. The facial process is tall and broad, and faces vertically, except for the anterodorsal apex, which curves dorsomedially. The dorsomedial margin of the facial process has a distinct nasal facet. The premaxillary process is long and bifurcated with a short lateral projection and a long anteriorly directed medial lappet. The crista transversalis is low and diminishes on the premaxillary process. There is a narrow palatal shelf without a distinct palatine process. The medial surface of the facial process bears a distinct ridge with an overhanging lamina that extends anteriorly onto the palatal shelf and defines the posterior border of a medial recess on the anteromedial

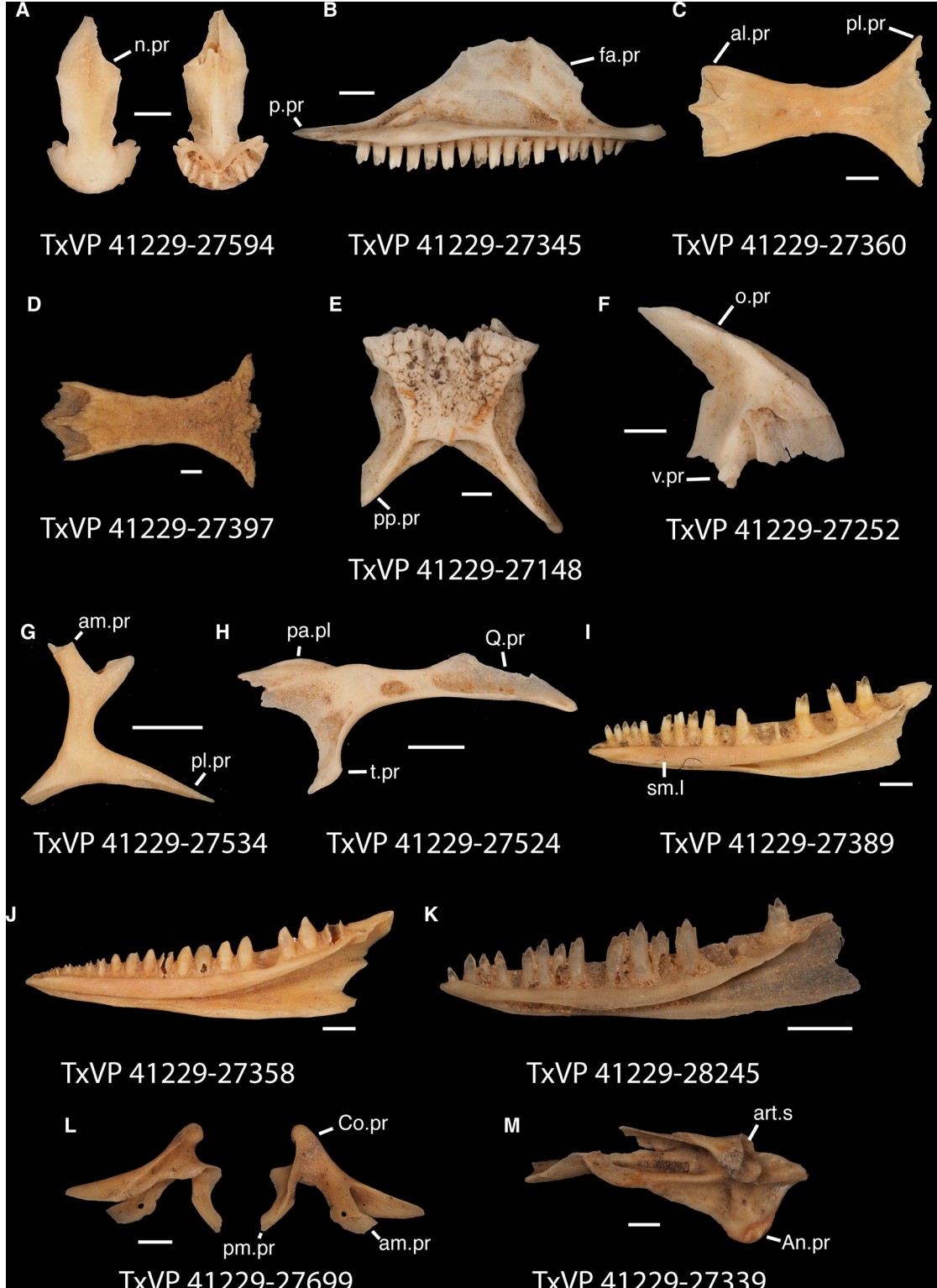

**Fig 47.** Fossil teiids, A–M: Teiinae. **A.** TxVP 41229–27594 Dorsal and ventral view of premaxilla; **B.** TxVP 41229–27345 Medial view of left maxilla; **C.** TxVP 41229–27360 Dorsal view of frontal; **D.** TxVP 41229–27393 Dorsal view of frontal; **E.** TxVP 41229–27148 Dorsal view of parietal; **F.** TxVP 41229–27252 Lateral view of right prefrontal; **G.** TxVP 41229–27534 Dorsal view of postorbitofrontal; **H.** TxVP 41229–27524 Dorsal view of left pterygoid; **I.** TxVP 41229–27389 Medial view of right dentary; **J.** TxVP 41229–27358 Medial view of right dentary; **K.** TxVP 41229–28245 Medial view of right dentary; **L.** TxVP 41229–27699 Lateral and

medial view of left coronoid; **M**. TxVP 41229–27339 Dorsal view of right compound bone. Scale bars = 1 mm. **Abbreviations**: al.pr, anterolateral process; am.pr, anteromedial process; An.pr, angular process; art.s, articular surface; Co.pr, coronoid process; fa.pr, facial process; n.pr, nasal process; o.pr, orbital process; pa.pl, palatal plate; pl.pr, posterolateral process; pm.pr, posteromedial process; pp.pr, postparietal process; p.pr, posterior process; Q.pr, quadrate process; sm.l, suprameckelian lip; t.pr, transverse process; v.pr, ventral process.

surface of the facial process. The lateral wall of the posterior orbital process is tall, and the dorsal surface of the postorbital process has an elongate, shallow jugal groove and a small ectopterygoid facet. There is an opening for the superior alveolar canal positioned anterior to the facial process and medial to the crista transversalis. There is one superior alveolar foramen on the palatal shelf lateral to the palatine process and posterior to the facial process. The facial process is slightly concave laterally. There are seven lateral nutrient foramina dorsal to the tooth row, and several foramina scattered on the lateral surface of the facial process.

*Identification.* Maxillae are assigned to Gymnophthalmoidea based on the presence of asymmetrically bicuspid distal teeth with a smaller anterior cusp that is anteriorly directed [24, 64, 93]. Gymnophthalmids and alopoglossids differ in generally having a relatively short facial process anteroposteriorly compared to the fossils [24, 37]. Furthermore, gymnophthalmids and alopoglossids lack large cementum deposits at the base of the teeth [23, 37, 74]. Maxillae are assigned to Teiidae on that basis.

**Frontal.** *Description.* TxVP 41229–27360 and TxVP 41229–27397 serve as the basis for our description (Fig 47C and 47D). Anteriorly, there are lateral prefrontal facets that extend to the ventral surface, and two dorsal nasal facets defined by shorter anterolateral processes separated by a longer anteromedial process. The interorbital margins are waisted and the posterolateral processes curve laterally such that the posterior end is slightly wider than the anterior end. The posterior edge is undulating and has narrow parietal facets. There are small postorbitofrontal facets laterally and ventrally on the posterolateral processes. There is a small depression on the posterodorsal portion of the frontal. The cristae cranii are low, approach one another in the interorbital region, and bound a groove for attachment of the solium supraseptale. The cristae cranii diverge posteriorly and extend along the lateral margins of the ventral surface. TxVP 41229–27397 differs in having sculpturing on the posterior half of the dorsal surface.

*Identification.* The fossils share a fused frontal with reduced cristae cranii with Pleurodonta, Gymnophthalmoidea, Mabuyinae and *Scincella* [23, 36]. Examined Mabuyinae (see Fig 8 of [68]) and *Scincella* differ in having a frontal that is comparatively shorter relative to the width. Examined pleurodontans differ in having more strongly constricted interorbital margins [23] with a relatively wider posterior margin [24]. Juvenile teiids have more constricted interorbital margins compared to adults, but the posterior margin is not widened as in pleurodontans [2]. Gymnophthalmids and alopoglossids differ in having frontal lappets on the posterior margin [23, 24, 37]. Fossils are assigned to Teiidae based on those differences with other NA lizards.

**Parietal.** *Description.* TxVP 41229–27148 is a parietal with extensive sculpturing on the dorsal surface and serves as the basis for our description (Fig 47E). The anterior edge has frontal tabs. The adductor crests do not meet posteriorly, giving the parietal table a trapezoidal appearance. The ventrolateral crests are tall and are visible in dorsal view. The anterolateral processes curve laterally and have lateral facets for articulation with the postorbitofrontal. The posterior edge between the postparietal processes is characterized by two distinct nuchal fossae. The postparietal process has a dorsal crest that slants laterally. The ventral surface has a deep depression (cerebral vault). There is a deep pit for the processus ascendens along the posteroventral edge. Flanges are present at the bases of the postparietal processes.

*Identification.* Parietals share with gymnophthalmoids and pleurodontans parietal lappets and a parietal foramen that is not fully enclosed by the parietal [23]. Pleurodontans differ in

lacking long descending parietal crests (parietal downgrowths of [23]). Gymnophthalmids and alopoglossids differ from the fossils in having shorter postparietal processes, in lacking dorsal sculpturing, and in lacking a contribution of the adductor muscles dorsally on the parietal [24, 37]. Fossils are assigned to Teiidae on that basis.

**Prefrontal.** *Description*. TxVP 41229–27252 is a right prefrontal that serves as the basis for our description (Fig 47F). It is triradiate with a long and pointed orbital process, a short ventral process, and an anterior sheet. The anterior sheet has a broad, distinct articulation facet for the facial process of the maxilla and a lateral facet for the lacrimal, formed by a laterally extending flange (lacrimal flange of [62]). There is a small ridge on the dorsolateral surface near the base of the orbital process. The ventral process is thin with a slightly widened distal end. There is a distinct notch for the lacrimal foramen, and the ventral process forms the posterior and ventral border of the foramen. Medially, the boundary of the olfactory chamber is a smooth, rounded, and concave surface. Dorsal to the olfactory chamber there is a broad groove for articulation with the frontal. The orbitonasal flange extends far medially with a notch dorsally.

*Identification*. The fossil shares with gymnophthalmoids a distinct lacrimal flange [62] with an articulation facet for the facial process of the maxilla [53]. Examined gymnophthalmids and alopoglossids differ in having a less extensive anterior process (facial sheet of [37]. The fossil is assigned to Teiidae on that basis.

**Postorbitofrontal.** *Description*. TxVP 41229–27534 is a left postorbitofrontal missing the distal end of the anteromedial process (Fig 47G). The bone is slender and quadraradiate. The anteromedial process is slightly longer than the posteromedial processes. Between the medial processes, there is a facet for articulation with the frontal and parietal. The lateral margin is curved, and the posterolateral process is longer than the anterolateral process. The anterolateral process has a lateral facet for the jugal, and the posterolateral process has a lateral facet for articulation with the squamosal. There is a small ventral foramen on the central shaft of the bone.

*Identification*. The postfrontal and postorbital are fused into a single postorbitofrontal in some gymnophthalmoids, xantusiids, some *Xenosaurus*, and some anguids [23, 24] (Fig 48). The fossil is quadraradiate as in some teiids and some *Xenosaurus*. The postorbitofrontal of *Xenosaurus* differs in having dorsal sculpturing and being much broader. The postfrontal and postorbital are separate in many gymnophthalmids [24]. The postorbitofrontal is triradiate in apologlossids in which the postfrontal and postorbital are fused [37]. The fossil is assigned to Teiidae on that basis.

**Pterygoid.** *Description*. TxVP 41229–27524 is a left pterygoid (Fig 47H). The palatine process is thin with an anterior palatine facet. The transverse process projects laterally from the palatal plate and has a well-developed posterolateral corner and a triangular and anterolaterally pointed distal end. The transverse process has a deep ectopterygoid facet. The transverse process has no ridge on the ventral surface for insertion of the pterygomandibular muscle, but there is a dorsal ridge for insertion of the superficial pseudotemporal muscle. The quadrate process is elongated and has a tall, thin lateral ridge and medially directed lamina of bone. On the quadrate process, there is a deep fossa columella, but no postepipterygoid groove. There is no medial projection at the floor of the basipterygoid fossa. There is a ridge on the ventral surface of the palatal plate with four empty tooth positions.

*Identification*. Fossils share with gymnophthalmoids the presence of pterygoid teeth [23] and an extensive, medially directed lamina of bone on the quadrate process [53, 131]. Some gymnophthalmids and alopoglossids have a medially directed lamina of bone on the quadrate process (e.g., *Gymnophthalmus* and *Alopoglossus*), but the lamina does not extend as far medially as in the fossils [24, 37]. *Calyptommatus* differs from the fossils in having the medial

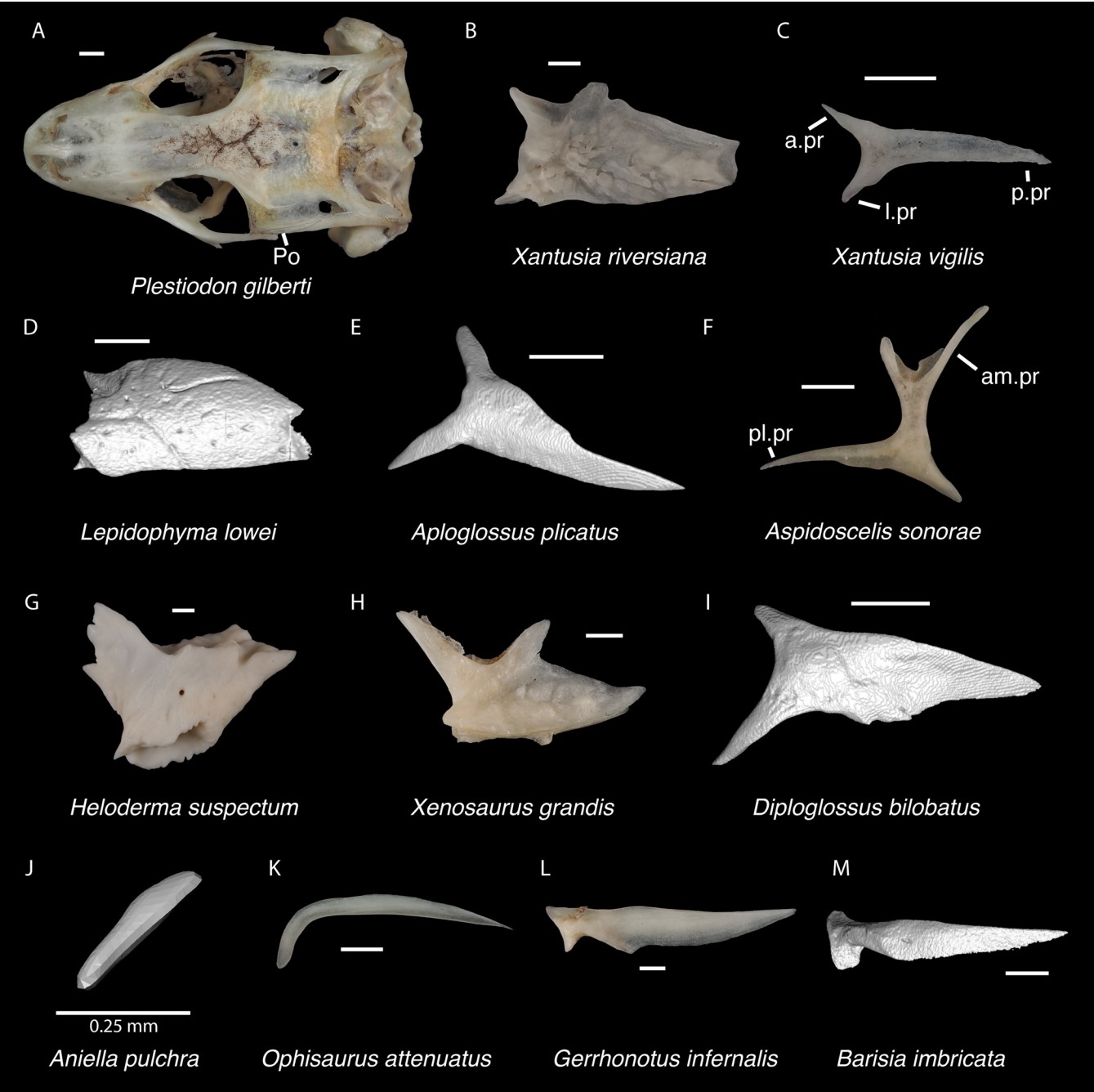

**Fig 48. Non-pleurodontan postorbitals.** Postorbitals and postorbitofrontals in dorsal and lateral views–**A**. *Plestiodon gilberti* TxVP M-15662 skull in dorsal view; **B**. *Xantusia riversiana* TxVP M-8505 right postorbitofrontal in dorsal view; **C**. *Xantusia vigilis* TxVP M-12130 left postorbitofrontal in dorsal view; **D**. *Lepidophyma lowei* LACM 143367 right postorbitofrontal in dorsal view; **E**. *Aploglossus plicatus* TNHC 34481 right postorbitofrontal in dorsal view; **F**. *Aspidoscelis sonorae* TxVP M-15670 left postorbitofrontal in dorsal view; **G**. *Heloderma suspectum* TxVP M-9001 left postorbital in dorsal view; **H**. *Xenosaurus grandis* TxVP M-8960 left postorbitofrontal in dorsal view; **I**. *Diploglossus bilobatus* TNHC 31933 right postorbitofrontal in dorsal view; **J**. *Anniella pulchra* FMNH 130479 right postorbital in lateral view; **K**. *Ophisaurus attenuatus* TNHC 98712 left postorbital in lateral view; **L**. *Gerrhonotus infernalis* TxVP M-13441 right postorbital in dorsal view; **M**. *Barisia imbricata* TNHC 76984 right postorbital in dorsal view. Scale bars = 1 mm unless otherwise noted. **Abbreviations**: am.pr, anteromedial process; a.pr, anterior process; l.pr, lateral process; pl.pr, posterolateral process; Po, postorbital; p.pr, posterior process.

lamina on the quadrate process much more expanded [103, 126]. *Alopoglossus* is further differentiated in having a posteromedial process along the medial edge of the pterygoid [37]. The fossils are assigned to Teiidae on that basis.

**Dentary.** *Description.* Morphotype A: TxVP 41229–27389 serves as the basis for our description (Fig 47I). TxVP 41229–27389 is a right dentary with 21 tooth positions. Teeth, except the three mesial-most teeth, are bicuspid with a smaller anterior cusp. There are substantial deposits of cementum at the tooth bases. The coronoid process is dorsally pointed. The angular process is broken and there is no surangular process. Anteriorly, the Meckelian groove opens ventrally, and near the middle of the tooth row the groove is briefly closed ventrally by the infra- and suprameckelian lips. Posteriorly, the Meckelian groove is tall. The suprameckelian lip is tall anteriorly. The intramandibular septum does not extend to the distal teeth. There is a narrow dental shelf. There is a distinct coronoid facet on the posterolateral surface, and there are five nutrient foramina arranged in a row on the lateral surface.

Morphotype B: TxVP 41229–28245 and TxVP 41229–27358 serve as the basis for our description (Fig 47J and 47K). TxVP 41229–28245 differs from morphotype A in having some tricuspid distal teeth while the crowns of TxVP 41229–27358 are eroded. Both TxVP 41229–28245 and TxVP 41229–27358 differ from morphotype A in having an open Meckelian groove for its entire length.

*Identification.* Fossils share with gymnophthalmoids the presence of asymmetrically bicuspid distal teeth [37, 64]. Gymnophthalmids and alopoglossids differ from the fossils in lacking large amounts of cementum deposits at the base of teeth [23, 64, 74]. Examined gymnophthalmids and alopoglossids also differ in having the coronoid process terminate far anterior relative to the surangular and angular processes (see also Fig 7 of [37]. Fossils are assigned to Teiidae on that basis. The fossils differ from Tupinambinae in lacking a large incision between the coronoid and angular processes. Fossils are assigned to Teiinae on that basis. Among NA teiines, there is a tendency for an increased number of tricuspid distal teeth in *Ameiva*, *Holcosus*, some *Cnemidophorus*, some *Pholidoscelis*, and some *Aspidoscelis* [2, 64, 94, 136]. The suprameckelian lip is relatively shorter in examined *Ameiva* compared to the fossils arguing against assignment of fossils to that genus. The relative height of the suprameckelian lip of the fossils most closely resembles that seen in *Aspidoscelis* and *Pholidoscelis* [2, 64]. Based on the described morphological differences it is likely that fossils of morphotypes A and B represent at least two distinct species, but we refrain from making a more refined identification pending examination of additional skeletal material of teiine lizards.

**Coronoid.** *Description.* TxVP 41229–27699 is a left coronoid and serves as the basis for our description (Fig 47L). The coronoid process is tall and rounded, and slopes anteroventrally. There is a thin shelf of bone posterior to the coronoid process to articulate dorsally with the surangular. The anteromedial process is elongated with a medial splenial facet. The posteromedial process is directed posteroventrally and has a wide notch (emargination of the adductor fossa *sensu* [24]) along the posterior margin resulting in a distinct surangular process (*sensu* [24]). There is a medial crest that extends from the coronoid process onto the posteromedial process. There is a large, anteriorly projecting lateral process with a dorsal dentary articulation facet. There is an anteroventrally oriented lateral crest that extends along the dorsal margin of the anterolateral process. There is a deep and narrow ventral groove for articulation with the surangular. There is a foramen on the anteromedial process.

*Identification.* The fossil coronoid differs from snakes in having distinct anteromedial and anterolateral processes that serve to clasp the dentary [23]. The fossil is further differentiated from snakes in having an expansive ventral articulation surface that is curved to articulate with both the dorsal and lateral surfaces of the surangular [27]. The fossil coronoid differs from xantusiids and some pleurodontans in having an anterolateral process [23]. The fossil differs from

examined NA pleurodontans with an anterolateral process (see above) in having a more posteriorly deflected posteromedial process. Examined scincids and anguimorphs, except for *Heloderma* and *Ophisaurus attenuatus*, have a lateral process that does not extend as far anteriorly. *Heloderma* differs in having a shorter coronoid process with a vertically oriental lateral crest [24], and examined *Ophisaurus* have a much smaller emargination of the adductor fossa along the posteromedial process if present at all. Examined geckos also lack a wide notch along the posterior margin of the posteromedial process [54, 104, 105]. Examined apologlossids and gymnophthalmids, except *Vanzosaura rubricauda* [137], differ in lacking a distinct surangular process [24, 37, 62, 103]. Examined gymnophthalmids and alopoglossids can be further distinguished in lacking a distinct lateral crest that extends along the anterolateral process [24, 37]. Fossils are assigned to Teiidae on that basis.

**Compound bone.** *Description*. TxVP 41229–27339 is a right compound bone (Fig 47M). The anterior portion of the prearticular and much of the surangular are missing. The adductor fossa is widely open dorsally. The retroarticular process is narrow. There is a medially oriented and broad angular process with a low ridge. A sheet of bone connects the retroarticular, and angular processes and another sheet connects the angular process to the anterior prearticular portion of the compound bone. There is a medial crest (tympanic crest of [49]) that extends along the retroarticular process. There is a distinct coronoid articulation facet on the medial surface that extends ventral to the adductor fossa. There is an anteroventrally oriented crest on the lateral surface. There is a foramen on the angular process and a posterior surangular foramen on the lateral surface.

*Identification*. The fossil shares with gymnophthalmoids the presence of a distinct angular process and a widely opened adductor fossa [23]. Examined alopoglossids and gymnophthalmids, except *Calyptommatus* [126] and *Bachia* [138], differ in having a wider and more rounded retroarticular process [24, 37, 62, 103]. *Calyptommatus* and *Bachia* differ in having a smaller angular process [126, 138] and *Calyptommatus* is further excluded based on its entirely South American distribution in the modern biota. Fossils are assigned to Teiidae based on these differences.

## Results

Our fossil identifications from Hall's Cave resulted in a minimum of 11 lizard taxa, including five lizard taxa previously unknown from Hall's Cave (Table 1). In most cases, we did not replicate the taxonomic precision of previous fossil identifications. Our apomorphic framework permitted less precise taxonomic identifications most of the time. We recovered 17 separate lizard skull elements represented as fossils from Hall's Cave. For most skeletal elements, identification could only be made to the family level; however, some elements (e.g., the premaxilla, maxilla, dentary, and jugal) permitted identification to the subfamily or genus level. Many of the skeletal elements that we recovered as fossils were relatively more robust skull bones (e.g., the frontal, parietal, dentary, maxilla, and compound bone), though we also recovered some smaller and more fragile elements (e.g., premaxilla, postfrontal, postorbital, jugal, and squamosal).

Phrynosomatidae is the family with the most identified distinct taxa. This diversity in the fossil assemblage is recapitulated in the modern-day diversity, with phrynosomatids being the most taxonomically diverse lizard family in the region. Most of the identified fossil lizard taxa inhabit the area around Hall's Cave today, except for lizards within the *Phrynosoma douglasii* species complex (Table 2). Anguines (specifically *Ophisaurus attenuatus*) are reported to occur in Kerr County today; however, an examination of records of *Ophisaurus attenuates* [139] revealed only a single specimen (United States National Museum: USNM 32826) in the county

**Table 1. Identified fossil lizard taxa represented in Hall's Cave from this study compared to previous studies.**

| Family | Previously identified lizard taxa by Parmley [18] | Previously identified lizard taxa by Toomey [10] | Updated faunal list from this study |
|---|---|---|---|
| Crotaphytidae | | *Crotaphytus* sp. | Crotaphytidae |
| Phrynosomatidae | | unid. Sceloporinae | Sceloporinae |
| | | | *Urosaurus* |
| | | | Callisaurini |
| | *Phrynosoma* cf. *P. cornutum* | *Phrynosoma cornutum* | *Phrynosoma cornutum* |
| | | *Phrynosoma douglasii* | *Phrynosoma douglasii* species complex |
| | | *Phrynosoma modestum*? | *Phrynosoma* sp. and *Phrynosoma douglasii* species complex |
| Anguidae | *Gerrhonotus liocephalus* | | *Gerrhonotus* |
| | | *Ophisaurus attenuatus* | *Ophisaurus* |
| Scincidae | | | *Scincella* |
| | | | Scincinae |
| Teiidae | | | Teiinae |

which was collected from three miles north of Kerrville in 1897. The next closest records of *Ophisaurus* are over 100 km east in Blanco and Hayes counties. There were a few extant lizard taxa that had records within 100 km of Hall's Cave that we did not detect as fossils, including *Coleonyx brevis*, *Anolis carolinensis*, and *Phrynosoma modestum*. Possible *Phrynosoma modestum* fossils were previously reported from Hall's Cave [10], yet, when we reexamined these fossils, we found evidence arguing against assignment to *Phrynosoma modestum* (see Systematic Paleontology above).

## Discussion

An apomorphy-based fossil identification framework provides a replicable basis for employing the fossil record to understand the past. Such a solid foundation for fossil identifications is of paramount importance for conducting larger evolutionary and ecological analyses using those data. Our study provides apomorphy-based identifications of fossil lizards from Hall's Cave,

**Table 2. Identified fossil lizard taxa represented in Hall's Cave from this study compared to native extant species within 100 km of Hall's Cave.**

| Family | Paleofauna at Hall's Cave identified in this study | Native extant species within 100 km of Hall's Cave [139] |
|---|---|---|
| Eublepharidae | — | *Coleonyx brevis* |
| Anolidae | — | *Anolis carolinensis* |
| Crotaphytidae | Crotaphytidae | *Crotaphytus collaris* |
| Phrynosomatidae | Sceloporinae | *Sceloporus olivaceous, S. poinsettii, S. variabilis, S. consobrinus, Uta stansburiana* |
| | *Urosaurus* | *Urosaurus ornatus* |
| | Callisaurini | *Cophosaurus texanus, Holbrookia lacerata, H. propinqua* |
| | *Phrynosoma cornutum* | *Phrynosoma cornutum* |
| | *Phrynosoma douglasii* species complex | None |
| | — | *Phrynosoma modestum* |
| Anguidae | *Gerrhonotus* | *Gerrhonotus infernalis* |
| | *Ophisaurus* | *Ophisaurus attenuatus* |
| Scincidae | *Scincella* | *Scincella lateralis* |
| | Scincinae | *Plestiodon obsoletus, P. tetragrammus* |
| Teiidae | Teiinae | *Aspidoscelis gularis, A. marmoratus, A. inornatus, A. exsanguis, A sexlineatus* |

and we add five new lizard taxa to the known diversity of the cave fauna. This work increases understanding of the past herpetofaunal diversity on the Edwards Plateau and sets the stage for further analyses examining past responses to environmental change. Additionally, because much of the previous work on Hall's Cave focused on mammals [10, 16] and plants [17], our work on the fossil lizards adds a new dimension towards examining dynamics of the larger ecosystem in the region through time.

Many of our apomorphic identifications are at higher taxonomic levels (i.e., at the family or genus level) compared to previous efforts to identify fossil lizards from this locality, most of which provided species-level identifications [10, 18]. Fossil identifications at the genus or family level were also achieved from other Pleistocene sites when using an apomorphic identification framework [90]. Less taxonomically specific fossil identifications may make taxon-based paleoecological reconstructions less precise [90]. Nevertheless, it is important to recognize the limitations of fossil data and acknowledge that morphology is not always able to provide robust species-level fossil identifications without making assumptions about changes in geographic distributions—assumptions that even we made in this apomorphy-driven study (i.e., excluding taxa not presently found in North America). Identifying and accepting the limitations of our data will lead to more robust and well-supported evolutionary and paleoecological reconstructions [9]. Identification of fossils using apomorphies generally results in identifications at higher taxonomic levels relative to identifications made from a phenetic framework or based on restricted comparative samples. However, alternative, or supplemental methods can be used to refine fossil identifications further. For example, quantitative morphological methods like geometric morphometrics have become more frequently applied towards fossil identification (e.g., [140]). In some cases, fossil identifications may be hindered by substantial intraspecific morphological variation that overwhelms any interspecific signal or by breakage and disarticulation that occur through taphonomy. Paleogenomic (including paleoproteomic) methods are useful for identifying fossils, broken and intact, from more recent geological times, and often can provide species-level identification of fossil remains [141–143]. However, preservation of organic molecules differs between geographic regions and time periods, and paleogenomic data are not always recoverable from fossils, even very recent ones [144]. Despite their limitations, morphology-based identification methods are applicable across the vastness of geologic time and continue to be an important approach for interpreting fossil remains.

Although less taxonomically specific fossil identifications using apomorphies can make paleoecological and paleoenvironmental interpretation less precise, it is still possible to glean insights from these identifications (e.g., [35, 90, 145, 146]). For example, here we used apomorphies to provide additional support for the presence of an extirpated clade of short-horned lizards (*Phrynosoma douglasii* species complex) on the Edwards Plateau during the late Pleistocene to the early Holocene. The closest living species of horned lizards in the *Phrynosoma douglasii* species complex to Hall's Cave is *P. hernandesi* found 435 km west in Jefferson County, Texas. There is a single isolated preserved specimen of *P. hernandesi* (MCZ R-8216) that was possibly collected 285 km northwest in Mitchell County, Texas; however, the specimen's metadata records uncertainty in that locality information. Toomey [10] postulated that the extirpation of lizards in the *P. douglasii* species complex near Hall's Cave may have been driven by complex ecological interactions, including climatic changes and potential changes in resource availability. Additional study, perhaps using stable isotope data, will shed more light on the cause of this extirpation.

There are a few extant lizard taxa (e.g., *Coleonyx brevis*, *Anolis carolinensis*) that live within 100 km of Hall's Cave today that we did not detect as fossils. The absence of these taxa from Hall's Cave may represent a true absence of these taxa from the area around Hall's Cave.

Alternatively, preservation or accumulation biases, particularly against the small and delicate bones of smaller taxa, like *Coleonyx brevis*, may also explain these absences. Although we found evidence contradicting the previous identifications of some fossils to *Phrynosoma modestum*, examination of additional material may confirm that taxon in the Hall's Cave sequence. The absence of *Anolis* from the Hall's Cave sequence is interesting because some skull elements of *Anolis* (e.g., dentary, parietal, frontal, maxilla) are relatively robust, easily recognizable, and commonly preserved as fossils in other Quaternary deposits [147]. Although *Anolis carolinensis* is widespread in Texas today, including around Hall's Cave, there is a striking lack of Pleistocene fossil anoles in Texas [5, 148]. Fossil *Anolis* are known from continental North America since at least the late Oligocene of Florida [149] and *Anolis carolinensis* was estimated to have experienced range expansion out of Florida during the Pleistocene [150]. Analysis of DNA sequence data indicated an eastward expansion of the range of *Anolis carolinensis* out of Florida beginning around 300,000 years ago [151]. A lack of fossil *Anolis* from the Pleistocene of Texas may indicate that they did not reach Texas until more recently. Further study of that dispersal would benefit from increased sampling of fossil sites, especially in east Texas. Although our study includes fossils from only a single locality, it preserves evidence that for at least some lizard taxa, stability was not the rule during parts of the Quaternary. Additional study of other fossil localities across NA is necessary to form a more complete synthesis of herpetofaunal dynamics during the Quaternary.

A main goal of this work is to facilitate the identification of North American lizard fossils in a phylogenetically explicit context. Our hope is that researchers familiar with the lizard skeletal system will find our synthesis of previously reported and new potential apomorphies useful for identifying fossil lizard remains and will spur new investigation into patterns of morphological variation within the lizard skeletal system. Our figures are intended to showcase cranial osteological variation in North American lizards with diverse specimen and taxon sampling. Although we obtained a broad comparative sample, we were unable to obtain comparative specimens for all North American lizard families, genera, and species, especially those for which CT or dry-skeletal material is scarce or non-existent. There is still a paucity of information in the published literature on morphological variation in many vertebrate clades, including lizards [24, 152]. An incomplete understanding of patterns of variation hinders our ability to make well-substantiated claims using the fossil record. Modern comparative morphological data provide a basis by which to interpret the fossil record [9], particularly for fossils from the more recent past (i.e., the Quaternary). Continued investigations into patterns of morphological variation in modern taxa are necessary for understanding patterns encountered in the fossil record [9, 152], including interspecific variation and intraspecific variation (e.g., sexual dimorphism, ontogenetic variation, etc.).

Many of the fossils examined here were found in vials of fossils labeled 'scrap bone.' These 'scrap' fossils were often elements other than the commonly described upper and lower tooth bearing elements that are commonly described in the Quaternary paleoherpetological literature [9]. It is important that these additional elements are recognized because, as we have shown here, there are apomorphies on various skeletal elements that are as useful or more useful for fossil identification [20, 29]. Increased recognition of lizard remains can lead to novel insights and discoveries related to NA lizards as well as a more holistic view of ancient faunal assemblages. We therefore specifically worked to include images of mainly disarticulated skeletal elements, as would likely be encountered in the fossil record, from a diverse set of NA lizards. This will be especially useful for researchers who do not specialize on lizards, and it is our hope that the images we provide here lead to an increased recognition and identification of lizard remains in fossil deposits.

## Supporting information

**S1 Table. Comparative specimens used in this study and their associated metadata.**
(XLSX)

**S2 Table. Global-scale apomorphies taken from the existing literature and interpreted using a phylogenetic framework following Burbrink et al. [2020].**
(DOCX)

**S3 Table. Complete list of identified fossil specimens.**
(XLSX)

## Acknowledgments

We thank Matt Brown and Chris Sagebiel from the UT Vertebrate Paleontology collections for their help in accessing fossil and modern specimens used in this study. We thank Christopher J. Bell for helpful comments and discussions that benefitted this research. We thank Alexandra Boville for helpful comments and discussions as well as help with specimen documentation and curation. We thank Tianyi Xu for providing some photographs of comparative skeletal specimens. We thank Randall Nydam and Mark Powers for their comments and suggestions that helped us improve the manuscript.

## Author Contributions

**Conceptualization:** David T. Ledesma, Melissa E. Kemp.

**Data curation:** David T. Ledesma, John J. Jacisin, III, Antonio Meza.

**Investigation:** David T. Ledesma, Simon G. Scarpetta, John J. Jacisin, III.

**Methodology:** David T. Ledesma, Simon G. Scarpetta, John J. Jacisin, III.

**Project administration:** David T. Ledesma.

**Supervision:** Melissa E. Kemp.

**Visualization:** David T. Ledesma, Antonio Meza.

**Writing – original draft:** David T. Ledesma, Simon G. Scarpetta, John J. Jacisin, III, Melissa E. Kemp.

**Writing – review & editing:** David T. Ledesma, Simon G. Scarpetta, John J. Jacisin, III, Antonio Meza, Melissa E. Kemp.

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
