## [Decision Letter · Decision Letter 0]

16 Jan 2024

PONE-D-23-35401Identification of Late Pleistocene and Holocene fossil lizards from Hall’s Cave (Kerr County, Texas) and a primer on morphological variation in North American lizard skullsPLOS ONE

Dear Dr. Ledesma,

Thank you for submitting your manuscript to PLOS ONE. After careful consideration, we feel that it has merit but does not fully meet PLOS ONE’s publication criteria as it currently stands. Therefore, we invite you to submit a revised version of the manuscript that addresses the points raised during the review process.

Both of the reviewers recognised the importance of this manuscript and the amount of work that went into the paper. Both, however, also recommended major revisions prior to publication, including: inclusion of snakes in the dataset (or a clear explanation on why they are excluded); applying caution to evolutionary statements regarding acquisitions, reversals, etc.; caution with regards to referrals based on the size of a feature; revising the organisation of the figures so that the reader can move through them logically; alternative views and more thorough labelling of elements (both reviewers commented on this); some anatomical errors; and inconsistencies in the formatting of citations at the end of the manuscript. Please see the reviewers' comments and annotated versions of the manuscript for more details.

We look forward to receiving your revised manuscript.

Kind regards,

Laura Beatriz Porro, Ph.D.

Academic Editor

PLOS ONE

2. In your manuscript, please provide additional information regarding the specimens used in your study. Ensure that you have reported human remain specimen numbers and complete repository information, including museum name and geographic location.

For more information on PLOS ONE's requirements for paleontology and archeology research, see https://journals.plos.org/plosone/s/submission-guidelines#loc-paleontology-and-archaeology-research.

Reviewers' comments:

Reviewer's Responses to Questions

**Comments to the Author**

1. Is the manuscript technically sound, and do the data support the conclusions?

Reviewer #1: Partly

Reviewer #2: Partly

2. Has the statistical analysis been performed appropriately and rigorously? 

Reviewer #1: N/A

Reviewer #2: N/A

3. Have the authors made all data underlying the findings in their manuscript fully available?

Reviewer #1: Yes

Reviewer #2: No

4. Is the manuscript presented in an intelligible fashion and written in standard English?

Reviewer #1: Yes

Reviewer #2: Yes

5. Review Comments to the Author

Reviewer #1: The manuscript titled "Identification of Late Pleistocene and Holocene fossil lizards from Hall’s Cave (Kerr

County, Texas) and a primer on morphological variation in North American lizard skulls" is a very useful guide to non-snake lizards of the Hall's Cave locality in Texas. The authors have undergone a tremendous effort to bring exemplar images of relevant elements in comparative plates for the convenience of the reader. I think that this manuscript should be published once several suggestions are considered, and hopefully incorporated to some capacity.

The first of these suggestions is as much philosophical as it is at the root of the methodology and purpose of this manuscript. Throughout the manuscript the authors comment on the benefits of understanding "lizard" diversity and how changes in this diversity could greatly affect our views on changing population dynamics, specifically in localized extirpations and emigrations. However, snakes are not mentioned at all despite being deeply nested in Squamata and therefore is a "lizard". Two dozen or so snake species have currently been identified living in or around the region today so the omission of snake diversity within the scope of lizards is puzzling and leaves a large gap in understanding of lizard diversity through recent time (Pleistocene to present). If the fossils are simply not present due to a preservational bias, this needs to be mentioned. If there are snake fossils from this locality, they should be included - or clear justification for their omission should be provided.

The second suggestion is regarding the phylogenetic framework for apomorphy assessments. In the table S2, evolutionary context is given for apomorphy appearance, reversals, independent acquisitions, etc. If these evolutionary claims are based off the phylogeny you have provided, that should be explicitly stated. This, however, comes with the caveat that these statements could be very different when using a different tree topology (e.g., Simoes et al., 2018). The phylogeny provided also does not include snakes, which are deeply nested in the Squamate clade. I recommend removing evolutionary statements such as number of aquisitions and/or losses, or reversals. These do not matter for diagnosing elements to a group using apomorphy based assessments and only serves to confuse the associations.

The figures are beautiful and specimens are clearly imaged. However, labels are only provided on one figure and do not always reflect the apomorphic states you are trying to discuss. I think it would be beneficial to include labels on other figures, especially those of significant importance for diagnosing an element to one group or another. With the amount of jumping around between figures, this would save from having to also go back to the only two labeled figures and S2 all the time. To the same point, showing multiple aspects of the elements would be a huge benefit as well. There were many times I would have made a more precise, or different general referral based on the images you have provided (e.g., Ophisaurus and fossil anguine elements). Based on the single aspect and general morphology I found a number of elements near spot on for Ophisaurus, but the same was not said in the description. I think showing multiple aspects as well as labelling the apomorphic states would help alleviate false identifications made by a reader's limited view.

The descriptions are done very well! Some concluding sentences in description paragraphs could be more specific to avoid ambiguity. Additionally, referral based on size of a feature (e.g., parietal horns in phrynosomatids) seem poor features to use for genus or species referrals without a relative statement or quantification of these differences. As presented I am unconvinced of some of the more specific referrals within the phrynosomatids and additional data with provided measurements or relative statemens that can be clearly demonstrated would greatly improve the support for these referrals.

Addressing the suggestions above will greatly improve the quality and utility of this manuscript. I encourage the authors to consider them and make efforts to incorporate any and all of them that they have the capacity to do so. I understand there may be difficulties in addressing them all, but the effort to do so will undoubtedly reflect well on the next draft!

Reviewer #2: I was very excited to see this study as it represents an important improvement in the methodology of assessing the systematic identification and taxonomy of Pleistocene/Holocene lizards represented by isolated elements. To see a rigorous representation of the methods long championed by Dr. Chris Bell was rewarding to read. That said, this paper need substantial improvements to be ready for publication.

I want to organize my observations and concerns as follows:

1. Organization of figures—the order and reference to the figures in the manuscript is very confusing. The reader is forced to move back-and-forth amongst several figures rather than moving through them in a more logical, linear fashion. I found this to be very frustrating and often needed to set the manuscript aside out of this frustration. I believe most readers will choose to not spend the time and effort to decode this lack of organization and will therefore not get the benefits of the method and analysis that could otherwise be available. I have noted this—to the point of ad nauseum (for which I do not apologize, but I do understand that it may as equally as frustrating to the authors)—throughout the manuscript.

2. Illustrated support of the methodology—The authors are presenting what they claim is an improved methodology of identifying fossil lizard taxa (why no snakes, which are just snake-lizards?) and that this method is based on the rigorous use of apomorphy-based comparisons with known extant taxa (interesting…and a bit disappointing that no attempt was made to compare to other fossil specimens from other fossil faunas). As such I was expecting an equally robust demonstration of the anatomy through figures and labelling of features. Instead, I was frustrated by the regular lack of alternative views of elements to support described features (noted often in my notes in the text). Additionally, as good as Figures 1–3 are as reference figures with comprehensive labeling of features, there are some disappointing omissions of regularly cited features (nuchal fossae, pterygoid lappet) and alternate views of elements with labeled features such as any ventral features of the frontal or parietal, lateral views of the dentary and maxilla. Additionally, the figures of fossils and representative comparative taxa (more on these below) should be labeled whenever a feature not in the Figs 1–3 reference figures are used (e.g., the “splenial spine of the dentary”). The figures of specimens represent the core data that we publish as paleontologists and if we argue for our hypotheses based on our data, then we must properly and completely illustrate that data. In the age of electronic publications, we should be able to accommodate robust and complete figures.

3. Reference/comparative anatomy images—At the risk of having egg on my face regarding the previous sentence I want to discuss what at times felt like the overuse of comparative figures. While useful and supportive of the apomorphic comparative identifications (what we used to just call comparative anatomy although here without—understandably—the differential aspect of a differential diagnosis), the comparative images greatly lengthen the manuscript. If the addition of these figures is considered essential to the methodology (and I would agree that they are) I wonder if there is a better balance? Fewer elements (though I appreciate the comprehensive nature of the images and I do believe they add to the overall value of the manuscript as an osteological resource)? Move them to the SI (though that brings back the concern of the flow of images relative to the references in the text)? In short, I bring this up based on my editorial concerns, but I do feel that these images are important and add value to the manuscript beyond the methodology and updating of the fauna.

4. Anatomy errors—Thankfully these are few and the authors have done a very good job in managing the descriptions of complex anatomy. However, the superior alveolar foramen of the maxilla CANNOT be medial to the palatine process as that is the most medial feature of the maxilla. The foramen is lateral to the palatine process and the palatine is medial to the palatine process. The maxilla (derivative 1a of arch 1) conducts the maxillary neurvascular bundle consisting of the V2 nerve, maxillary artery, and maxillary vein(s). This system of passages and contents is knows as the superior alveolar system and all reference to openings (posterior superior alveolar foramen lateral to the palatine process, anterior superior alveolar foramen typically just posterior the premaxillary process, nutrient foramina along the lateral surface) all must be identified specifically and accurately. The same is true for the distribution in the dentary/mandible. The V3 nerve and mandibular/inferior alveolar AV are the equivalent distribution systems of derivative 1b of arch 1. As such all identification must be specific and accurate as part of the inferior alveolar system.

5. Anatomical absences—I am curious as to why no vertebrae or limb elements were described or figured. Is that to be another project? Additionally, as mentioned above, there must have been substantial numbers of snake vertebrae and at least some snake cranial elements. Where are they in a paper on squamates?

6. Citations at end of manuscript—In the interest of time, I did not fully check all the references, but I did note that the authors did not use a single formatting strategy for all citations at the end of the manuscript and I encourage them to follow the PlosOne formatting directions for citations.

This manuscript clearly represents an admirable, if not herculean, amount of work and I very much hope to see it come to publication as I believe that—when properly revised—it will represent a new and much needed standard for managing Pleistocene/Holocene fossils. This return to comparative anatomy reminds us that at the core of our science is the morphology of our specimens and that regardless of whatever statistical models or systematic algorithms come and go, the anatomy of the specimens will always be the data at the foundation of what we learn.

6. PLOS authors have the option to publish the peer review history of their article (what does this mean?). If published, this will include your full peer review and any attached files.

Reviewer #1: No

Reviewer #2: **Yes: **Randall Nydam

---

## [Author Response · Author response to Decision Letter 0]

2 May 2024

Reviewer #1

Comment 1) The first of these suggestions is as much philosophical as it is at the root of the methodology and purpose of this manuscript. Throughout the manuscript the authors comment on the benefits of understanding "lizard" diversity and how changes in this diversity could greatly affect our views on changing population dynamics, specifically in localized extirpations and emigrations. However, snakes are not mentioned at all despite being deeply nested in Squamata and therefore is a "lizard". Two dozen or so snake species have currently been identified living in or around the region today so the omission of snake diversity within the scope of lizards is puzzling and leaves a large gap in understanding of lizard diversity through recent time (Pleistocene to present). If the fossils are simply not present due to a preservational bias, this needs to be mentioned. If there are snake fossils from this locality, they should be included - or clear justification for their omission should be provided.

Response 1) While we acknowledge that it would be interesting and beneficial to examine snakes from Hall's Cave to understand squamate diversity, we disagree that not including snakes precludes us from examining lizard diversity from this site. The term "lizard" describes a paraphyletic group and does not include snakes, whereas the clade Squamata includes lizards and snakes (de Queiroz and Gauthier 2020). Snakes have been described from Hall's Cave (Parmley 1988) and we have added commentary in the manuscripts making this clear. Like many (if not most) squamate paleontologists, we found it useful in this paper to examine lizards separately, even though "lizards" are a grade-and snakes represent a highly diverse, and both ecologically and phenotypically divergent, clade within Squamata. More importantly, an examination of snakes from this site would require an equal, if not lengthier, description and photographic documentation of fossil and modern snakes. Given the already great length of this manuscript, we consider this to be far outside of the scope of the current study. 

Comment 2) The second suggestion is regarding the phylogenetic framework for apomorphy assessments. In the table S2, evolutionary context is given for apomorphy appearance, reversals, independent acquisitions, etc. If these evolutionary claims are based off the phylogeny you have provided, that should be explicitly stated. This, however, comes with the caveat that these statements could be very different when using a different tree topology (e.g., Simoes et al., 2018). The phylogeny provided also does not include snakes, which are deeply nested in the Squamate clade. I recommend removing evolutionary statements such as number of aquisitions and/or losses, or reversals. These do not matter for diagnosing elements to a group using apomorphy based assessments and only serves to confuse the associations.

Response 1) Our phylogenetic framework follows that of Burbrink et al. (2020) for apomorphies taken from the literature that we mention in the main text and in table S2. We have added additional text in the methods section and the caption of table S2 reinforcing this. This phylogeny does include all extant squamates and evolutionary hypotheses we make about global-scale apomorphies are based on this inclusive topology. For table S2, we specify that these are global-scale apomorphies taken from the existing literature meaning that these apomorphies do take into account hypothesized phylogenetic relationships among all squamates. In some cases where we are assessing apomorphies within a particular squamate clade (noted in parentheses in table S2 for each apomorphy) it is not necessary to examine the evolutionary history of those characters across all squamates. Hypotheses on character evolution are necessary for developing a framework for apomorphy-based identifications, and if there are several potential apomorphy scenarios then these should be stated explicitly. Although details on acquisitions and/or losses, or reversals are not always referenced when diagnosing fossils, that information is inseparable from the practice of making identifications using apomorphies. Hypotheses regarding character evolution are therefore always present, whether explicitly stated or not for apomorphy identifications. We feel it is important to include hypotheses about character evolution in order to be explicit about our thinking regarding character evolution. 

Comment 3) The figures are beautiful and specimens are clearly imaged. However, labels are only provided on one figure and do not always reflect the apomorphic states you are trying to discuss. I think it would be beneficial to include labels on other figures, especially those of significant importance for diagnosing an element to one group or another. With the amount of jumping around between figures, this would save from having to also go back to the only two labeled figures and S2 all the time. To the same point, showing multiple aspects of the elements would be a huge benefit as well. There were many times I would have made a more precise, or different general referral based on the images you have provided (e.g., Ophisaurus and fossil anguine elements). Based on the single aspect and general morphology I found a number of elements near spot on for Ophisaurus, but the same was not said in the description. I think showing multiple aspects as well as labelling the apomorphic states would help alleviate false identifications made by a reader's limited view.

Response 1) It was our intention to provide a couple of reference figures with labeled anatomical features so that the panels of skeletal elements did not become unintelligible with labels and lines. That said, we do acknowledge that adding some labels to the figures will save readers time without having to always refer to the reference figures. We added labels for relevant anatomical features on a few specimens in each figure panel. 

Response 2) To the second point on increasing the number of illustrated orientations of elements, we agree that this would be informative, but it would require doubling or tripling the number of (already numerous) figures and ultimately is not necessary for this study. To your specific point that some fossils resemble Ophisaurus, fossils were indeed identified to Ophisaurus, but we stated that we only made these identification on the basis of modern biogeography (Ophisaurus is the only extant anguine in North America). We could identify no apomorphies that could be used to differentiate those elements from all other anguines, and images from multiple orientations of the elements would not further illuminate this point. Furthermore, the reader will always have a limited view of a 3D object presented on a 2D medium, but the reader is guided by both the figures as well as the in text descriptions and corresponding identifications. If you have other identification discrepancies in mind please let us know. 

Comment 4) The descriptions are done very well! Some concluding sentences in description paragraphs could be more specific to avoid ambiguity. Additionally, referral based on size of a feature (e.g., parietal horns in phrynosomatids) seem poor features to use for genus or species referrals without a relative statement or quantification of these differences. As presented I am unconvinced of some of the more specific referrals within the phrynosomatids and additional data with provided measurements or relative statemens that can be clearly demonstrated would greatly improve the support for these referrals.

Response 1) We have changed the wording of concluding sentences to make them more specific and reemphasize the features that permit identification. We have also changed our wording on referrals based on size. Our referrals were not based on absolute size, but rather on relative size. The parietal horns of species of Phrynosoma, for example, are differentiated by their size relative to the size of the parietal table. Our figured specimens of Phrynosoma cornutum, Phrynosoma modestum, and Phrynosoma douglasii show the range of the differences in relative proportions of the horn length to the size of the parietal table. If you have specific identification comments on phrynosomatids besides the parietal horns of Phrynosoma (a highly diagnostic and apomorphic feature of that genus) then please let us know.

Lines 98-102: We updated the figure 1 caption stating that it shows the elements recovered as fossils and examined in the text highlighted in purple.

Lines 106-109: We have included text mentioning that snake fossils have been recovered and described from Hall's Cave. We also added text mentioning that treatment of snake fossils using an apomorphy-based fossil identification framework awaits an equally lengthy discussion in future work. 

Lines 125-129: In many cases, our global-scale apomorphy based identifications allowed us to identify fossils to clades with NA and non-NA components. In these cases, we geographically restricted our identifications on the continental scale. For example, in some cases, we could identify fossils using global-scale apomorphies to Gymnophthalmoidea, however, we could not identify any apomorphies that would exclude all lacertids. In these cases, we excluded lacertids because none live in North America natively, and so we instead focused on identifying tentative apomorphies or morphological differences that allowed us to differentiate between teiids, gymnophthalmids, and alopoglossids. Our cladogram helps to convey this methodology and we have updated that caption to state that it represents a "Cladogram, pruned to only include clades that contain North American lizards, showing framework used for evaluating tentative apomorphies reported in this study based on a continentally restricted dataset." 

 The cladogram does not suggest that we did not consider snakes in our fossil identifications. We have added text stating that "We employed an apomorphy-based fossil identification framework using previously published global-scale apomorphies for squamates. Although we do not treat snake fossils here, we note that..." "...global-scale apomorphies have been evaluated in a framework that includes snakes and all fossil lizard elements described here contain apomorphic states that exclude snakes." 

Lines 132-135: We have added the following text further illuminating our methodology: "We employed an apomorphy-based fossil identification framework using previously published global-scale apomorphies for squamates... In some cases, our global-scale apomorphy based identifications allowed us to identify fossils to clades with NA and non-NA components. In these cases, we geographically restricted our identifications on the continental scale." Any instance of failure to identify extirpated clades would be an instance of an extirpation event on a continental scale. In the late Pleistocene to Holocene these types of events are exceedingly unlikely. 

 With regards to how a geographically restricted dataset would affect our apomorphic framework, we note that "We supplement our global-scale apomorphic identification framework with new tentative apomorphies and morphological differences among taxa we report here based on our comparative dataset largely restricted to NA lizard taxa..." and "We note that newly reported apomorphies must be examined in non-NA taxa before being used in a global-scale apomorphic diagnosis." Further, we note that "We were not able to examine every NA lizard species or clade and therefore take a conservative approach to fossil identifications." 

Line 141: We have added labels for relevant anatomical features on at least one specimen in each figure panel. 

Line 196: We have rearranged the lettering in the panel to reflect the order mentioned in the text. 

Lines 210-216: We have added labels for relevant anatomical features on at least one specimen in each figure panel. 

Lines 255-257: We have added labels for relevant anatomical features on at least one specimen in each figure panel. 

Lines 276-279: See our above response about not increasing the number of illustrated orientations of elements (Reviewer #1-Comment #3-Response #2).

Lines 298-299: We added a note that the posterior orbital process of the maxilla is also known as the suborbital process, which are used interchangeably in Evans (2008). Also, we have added labels for relevant anatomical features on at least one specimen in each figure panel. 

Lines 310-311: We have added labels for this feature.

Line 317: We changed "wide posterior orbital process" to "medially widened posterior orbital process."

Line 321: Even though this comment is placed on a figure panel of modern specimens we think this is in reference to the figures with fossils. We have added text in the figure captions on figures with fossils noting fossil assignment. 

Lines 347-349: We have added a reference supporting our observations among teiids. We refrained from referencing too many figures that were more relevant later in the manuscript so that all the figures would not all appear together at the beginning of the manuscript. 

Line 411: We have changed the wording of concluding sentences to make them more specific and reemphasize the features that permit identification. 

Lines 439-448: We feel that having the specimen names in the captions as well as the figures is helpful for quickly matching and validating the specimens in the figures. It would also help readers if the species names are in the figure captions if they wished to do a quick text search for particular species. Such a search would be more difficult if the species names were only in the captions. Also, we don't feel that there is really a need to shorten the figure captions as it would not save that much space at the cost of reducing the information contained in them. 

Lines 523-524: We changed the sentence wording to "Based on these differences with other NA pleurodontans listed above, fossils were assigned to Crotaphytidae."

Lines 639-640: We changed the sentence wording to "Based on these differences with other NA pleurodontans listed above, fossils were assigned to Crotaphytidae."

Lines 1855-1858: Absolute horn size is variable, and what we were actually referring to was horn size relative to the size of the parietal table. In some species the horns project relatively tall compared to the size of the parietal table, while in other species, the horns barely project at all compared to the size of the table. We have changed the wording on this section to make it clear that we are actually referring to the relative size of the horns. 

Line 1933: We have changed the wording on this section to make it clear that we are actually referring to the relative size of the horns. 

Line 1977: We have specified the specimen letter within the figure. 

Lines 2298-2303: The fossils were assigned to Anguinae and then to Ophisaurus with the statement on Line 2068 that "Fossils assigned to Anguinae are identified as Ophisaurus on the basis that Ophisaurus is the only genus within Anguinae known to inhabit North America during the Quaternary." Therefore, your assessment is correct that the fossils elements are most comparable to Ophisaurus. However, adding panels of additional orientation is unnecessary because we clearly lay out the basis for our identification and our descriptions lay out a detailed anatomical description.

Lines 2646-2648: Yes, in Figure 36 only one specimen of Gerrhonotus has a notch on the posterior edge of the facial process. This is not an artifact of cropping, this is a feature that varies intraspecifically within Gerrhonotus infernalis. We have added additional text clearly explaining this variation and we have labeled the notch on the figures. 

Lines 3262-3263. See our response to Comment 1 of Reviewer #1 regarding our justification for our decision to exclude snakes. 

Line 3291: We choose to not use the term "non-snake lizards" because the term "lizard" explicitly describes a paraphyletic group and therefore does not include snakes while the clade Squamata includes lizards and snakes (de Queiroz and Gauthier 2020). 

Lines 3298-3302: Se

---

## [Decision Letter · Decision Letter 1]

11 Jun 2024

PONE-D-23-35401R1Identification of Late Pleistocene and Holocene fossil lizards from Hall’s Cave (Kerr County, Texas) and a primer on morphological variation in North American lizard skullsPLOS ONE

Dear Dr. Ledesma,

Thank you for submitting a revised version of your manuscript to PLOS ONE. After being reviewed a second time, we feel this article is nearly ready for publication with the addition of a few very minor revisions. Therefore, we invite you to submit a revised version of the manuscript that addresses the points raised during the review process. Both original reviewers agreed to review the revised version of the manuscript and both said it was greatly improved over the original version. Moreover, both reviewers recognised the tremendous amount of work involved in this paper and acknowledged it will be a significant contribution to the field. One reviewer raised a few very minor points they would like to see addressed, please see Reviewer 1's comments below.

We look forward to receiving your revised manuscript.

Kind regards,

Laura Beatriz Porro, Ph.D.

Academic Editor

PLOS ONE

Journal Requirements:

Reviewers' comments:

Reviewer's Responses to Questions

**Comments to the Author**

1. If the authors have adequately addressed your comments raised in a previous round of review and you feel that this manuscript is now acceptable for publication, you may indicate that here to bypass the “Comments to the Author” section, enter your conflict of interest statement in the “Confidential to Editor” section, and submit your "Accept" recommendation.

Reviewer #1: (No Response)

Reviewer #2: All comments have been addressed

2. Is the manuscript technically sound, and do the data support the conclusions?

Reviewer #1: Yes

Reviewer #2: Yes

3. Has the statistical analysis been performed appropriately and rigorously? 

Reviewer #1: N/A

Reviewer #2: N/A

4. Have the authors made all data underlying the findings in their manuscript fully available?

Reviewer #1: Yes

Reviewer #2: Yes

5. Is the manuscript presented in an intelligible fashion and written in standard English?

Reviewer #1: Yes

Reviewer #2: Yes

6. Review Comments to the Author

Reviewer #1: The authors have significantly improved the readability and flow of the manuscript. It follows a far more logical pattern and is easier to use the figures for comparison to the relevant text. Some previous reviewer comments were circumvented for the sake of time and brevity, but have been addressed given more relative statements that are clearer and easier to conceptualize for the reader (e.g., squamosal horns in phrynosomatids). I see the paper as ready to publish with a few additional additions for clarity and specificity.

The authors are using cladistic methods while trying to maintain traditional groupings (i.e., squamate apomorphies with a paraphyletic "lizard" concept, distinct from snakes). I see issues with this method as apomorphies are determined from cladistic groupings, and due to snakes being deeply nested within squamates, the classical definition of "lizards" and snakes becomes problematic. I would like to see further explanation of how this is being addressed. The phylogeny being presented excludes snakes, but is stated to be derived from a study that includes all squamates. The apomorphies thus constructed from the provided phylogeny are somewhat artificial without the consideration of potential character overlap with snakes. This can be further exemplified when assessing apomorphies of closely related clades, such as anguids that may share apomorphic states with snakes. Expansion in the methods of how the authors are accounting for this, or mention of the potential shortcomings of their methods in light of snakes deeply nested position within squamates should be included.

Additionally, when features are mentioned in the text that are key to identifying fossils to a living clade, those features should be labelled in one or more of the figures. I noticed several features such as the palatal shelf of the premaxilla, and the pterygoid facet of the quadrate were not labelled in any figure, to name a few. Review of key features and addition of appropriate labels should be done as well.

With these suggestions addressed, I think the manuscript will be ready for publication! It is a wonderful addition to squamate literature resources and will no doubt be highly cited. The authors have done a tremendous amount of work!

Mark Powers

Reviewer #2: While I still feel that the image order is a bit confusing, the manuscript overall is in very good shape and ready for publication. I appreciate the corrections to the anatomical errors. Hopefully this paper will set a new standard for comparative anatomy to be applied rigorously to Pleistocene and sub-fossil faunas.

7. PLOS authors have the option to publish the peer review history of their article (what does this mean?). If published, this will include your full peer review and any attached files.

Reviewer #1: **Yes: **Mark Powers

Reviewer #2: **Yes: **Randall Nydam

---

## [Author Response · Author response to Decision Letter 1]

15 Jul 2024

Reviewer #1

Comment 1. The authors are using cladistic methods while trying to maintain traditional groupings (i.e., squamate apomorphies with a paraphyletic "lizard" concept, distinct from snakes). I see issues with this method as apomorphies are determined from cladistic groupings, and due to snakes being deeply nested within squamates, the classical definition of "lizards" and snakes becomes problematic. I would like to see further explanation of how this is being addressed. The phylogeny being presented excludes snakes, but is stated to be derived from a study that includes all squamates. The apomorphies thus constructed from the provided phylogeny are somewhat artificial without the consideration of potential character overlap with snakes. This can be further exemplified when assessing apomorphies of closely related clades, such as anguids that may share apomorphic states with snakes. Expansion in the methods of how the authors are accounting for this, or mention of the potential shortcomings of their methods in light of snakes deeply nested position within squamates should be included.

Response 1. Based on this comment we think there may be a misunderstanding of our methodology. To be clear- our apomorphic identifications explicitly account for snakes, even though we did not describe any fossil snakes. None of the apomorphies that we used from the literature or developed here diagnose “lizards,” they diagnose specific clades within Squamata that are universally called lizards (e.g., pleurodontan iguanians, teiids, gerrhonotines, etc.). Our decision to treat only fossil lizards and not snakes is a practical one and does not reflect the systematic framework or apomorphic identification methodology. As stated previously we decided to limit this paper to lizards to allow a tractable manuscript length, and because snakes are usually treated apart from other squamates in herpetological studies of both extant and extinct taxa (even though “lizards” is a paraphyletic group). This extends to the cladogram that was previously presented; it was pruned to only include the taxa being studied, and did not indicate that only taxa in the cladogram were included in the apomorphic diagnoses. In sum, in this study we take apomorphies from the literature that have been evaluated for all squamates (including snakes and lizards)-when we identify fossils to particular clades using these apomorphies we are explicitly taking snakes into account.

We have revised our wording in our methods section to clarify our methodology (lines 137 - 144). We also updated the cladogram that we present in the manuscript to include squamates globally while highlighting the North American lizard clades to better convey our identification framework.

Comment 2. Additionally, when features are mentioned in the text that are key to identifying fossils to a living clade, those features should be labelled in one or more of the figures. I noticed several features such as the palatal shelf of the premaxilla, and the pterygoid facet of the quadrate were not labelled in any figure, to name a few. Review of key features and addition of appropriate labels should be done as well.

 Response 1. We have added labels in the figures for the palatal shelf of the premaxilla and the pterygoid facet of the quadrate. We have also reviewed the manuscript to ensure that those key morphological features are labeled. We also note that because we state in our methods that our anatomical terminology follows that of Evans (2008) “The skull of lizards and Tuatara”, readers unfamiliar with terminology used in our manuscript can reference that comprehensive work as well as the literature we reference throughout the manuscript for clarification.

Reviewer #2

We thank the reviewer for their kind words regarding our study. We do hope that our research will assist paleontologists working on Quaternary fossil sites with squamate remains and spark new interest among researchers into patterns of osteological variation in extant squamates.

---

## [Editor Report · Decision Letter 2]

30 Jul 2024

Identification of Late Pleistocene and Holocene fossil lizards from Hall’s Cave (Kerr County, Texas) and a primer on morphological variation in North American lizard skulls

PONE-D-23-35401R2

Dear Dr. Ledesma,

We’re pleased to inform you that your manuscript has been judged scientifically suitable for publication and will be formally accepted for publication once it meets all outstanding technical requirements.

Kind regards,

Laura Beatriz Porro, Ph.D.

Academic Editor

PLOS ONE
---

## [Editor Report · Acceptance letter]

5 Aug 2024

PONE-D-23-35401R2 

PLOS ONE

Dear Dr. Ledesma, 

I'm pleased to inform you that your manuscript has been deemed suitable for publication in PLOS ONE. Congratulations! Your manuscript is now being handed over to our production team.

Kind regards, 

on behalf of

Dr. Laura Beatriz Porro 

Academic Editor

PLOS ONE